  SciPost Phys. Lect. Notes 34 (2021)

# A somewhat random walk through nuclear and particle physics

**Thomas D. Cohen⋆ and Nicholas R. Poniatowski†**

Department of Physics, University of Maryland, College Park

⋆ cohen@umd.edu, † nponiatowski@g.harvard.edu

## Preface

These notes are an outgrowth of an advanced undergraduate course on nuclear and particle physics, which I first taught in the Spring Semester of 2018 at the University of Maryland. This is an elective course and the curriculum often fluctuates depending on the instructor.

There are many ways such a course could be taught. These range from a purely qualitative description of experimental phenomena organized in a historical manner to formal treatments in terms of quantum field theory. Alternatively, one could emphasize experimental methods and results. A course at this level could be a survey covering a wide swath of material rather superficially, or one that focuses deeply on a few topics. A key challenge in designing such a class for undergraduates is that the natural language of the subject is quantum field theory, a subject typically encountered in the second year of graduate school.

The course that I designed and taught was intended as an introduction to various aspects of particle and nuclear physics. The class emphasized the role of symmetry. The basic philosophy was to try to get students to grasp how nuclear and particle physicists think about the underlying physics. One tactic was to emphasize topics that were in some sense simple so that students could understand what is happening. In some cases this was to choose topics that could be understood via extremely simple physical reasoning, such as the semi-empirical mass formula in nuclear physics and its connection to a liquid drop picture. In other cases it was to introduce many of the fundamental physics ideas using relatively sophisticated mathematical tools. Thus, the course introduces many of the ideas of quantum field theory. However, to make this accessible to undergraduates, this is often done in as a simplified context to bring out the underlying ideas. Thus, for example the Higgs mechanism is discussed in terms of an Abelian model, rather than the full electroweak theory. The emphasis of the course is largely, but not entirely, theoretical in orientation. The goal is for students to develop an understanding of many the underlying issues in a relatively sophisticated way.

The subjects of nuclear and particle physics are vast, and within the community there is no agreed upon standard list of topics that an undergraduate class must cover. I tried to find an appropriate mix of topics with immediate experimental relevance such as the use of electron scattering to measure form factors and map out charge distributions and more theoretical issues such as Goldstone's theorem. The collection of topics that ended up in the course might best be described as "A Somewhat Random Walk Through Nuclear and Particle Physics."

While there are a number of undergraduate textbooks aimed at nuclear and/or particle physics, for a variety of reasons none of them were suitable for the kind of course that I thought best for students at this level. Since there was no book which was really suitable for the course, I produced a set of hand written lecture notes which I distributed to the students. They were a poor substitute for a book. Apart from the hand written quality of those notes, to call them "very terse" was a gross understatement: they had equations, very few words, and extremely limited explanations.

The notes produced here are quite different. They are a more-or-less self-contained document. They represent a collaboration between an undergraduate student, Nick Poniatowski and me. Nick's role in producing these notes is quite remarkable and cannot be overstated.

Nick served as the teaching assistant for the course the second time it was taught (Spring 2019). This was remarkable given that, at the time Nick was an undergraduate junior who had not taken the course, had never studied most of the topics in the course and whose research interest was (and remains) in experimental condensed matter physics. *A priori* it seems kind of crazy that a student with his background was allowed to TA such a class under these circumstances. However, when asked Nick me if he could serve as an undergrad TA, I agreed. I knew Nick's talents well, having supervised Nick in an independent study on quantum field theory the previous semester and was certain that he could do it.

As a TA, Nick offered to typeset my handwritten notes to aid the students. This was already above and beyond the call of duty, but I was pleased for the help. At the time, I assumed he was merely going to transcribe the notes. However Nick did much more than this. Instead of the word or two that I had in my original notes, Nick had full, thoughtful and clear explanations.

It is said of the famous series of books by Landau and Lifshitz, "Not a word of Landau and not a thought of Lifshitz." Given my description of the way these notes were put together, one might think that these notes would be "Not a word of Cohen and not a thought of Poniatowski." Indeed, you might think, "how could it be otherwise?" given that Nick wrote these as an undergraduate (mostly as a Junior) working in experimental condensed matter physics. However, that would be grossly unfair to Nick. In point of fact, during the semester when I was teaching when Nick felt that my lectures lack sufficient background, he added supplementary material to the notes. Thus for example, in the section on gauge theories Nick added an entire section: "A Quick and Dirty Group Theory Primer", to which I contributed nothing, similarly the discussion of symmetry breaking in the context of the Ising and Heisenberg models of magnetism was entirely his. Overall Nick was the driving force behind the project to create a set of useful notes that can serve as an undergraduate level introduction to this subject.

We have not included references in the notes. The material is on the whole sufficiently well-established that there is no need to credit the original authors. We have included a list of books for further reading so that students can pursue these topics in greater depth.

At the end of these notes are number of problems. Students that wish to get a better sense of the subject are urged to work through these problems.

Finally, it is likely that there are errors in these notes. The authors would greatly appreciate if you would call any of these to our attention so that we can fix them. Please direct any such feedback to `cohen@umd.edu` or `nponiatowski@g.harvard.edu`.

Tom Cohen
Washington, D.C.
June 2020

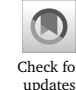

Check for updates

doi:10.21468/SciPostPhysLectNotes.34

# Contents

# 1 Historical Introduction

If one were to ask a contemporary physicist for a simple cartoon overview of our current understanding of fundamental physics they would probably give something like Fig. 1. In that figure the overall description of nature is divided into three areas—matter: the stuff of which things are composed, interactions: the forces that the matter feels, and "the rules of the game": the overarching intellectual structures used to describe the matter and interactions. Things that are well-established are included without question marks in the figure, while more speculative things are labeled with a question mark. Thus, for example "dark matter" has a question mark—while we know that there is dark matter and it constitutes a large fraction of the universe, we do not know what it is.

Now before one panics looking over the complexity of the physical world as described by Fig. 1 and contemplates working through these lecture notes, it is important to realize that these notes are not intended to fully cover the current state of our understanding of fundamental physics. In fact, they address a rather small fraction of the physics in the figure.

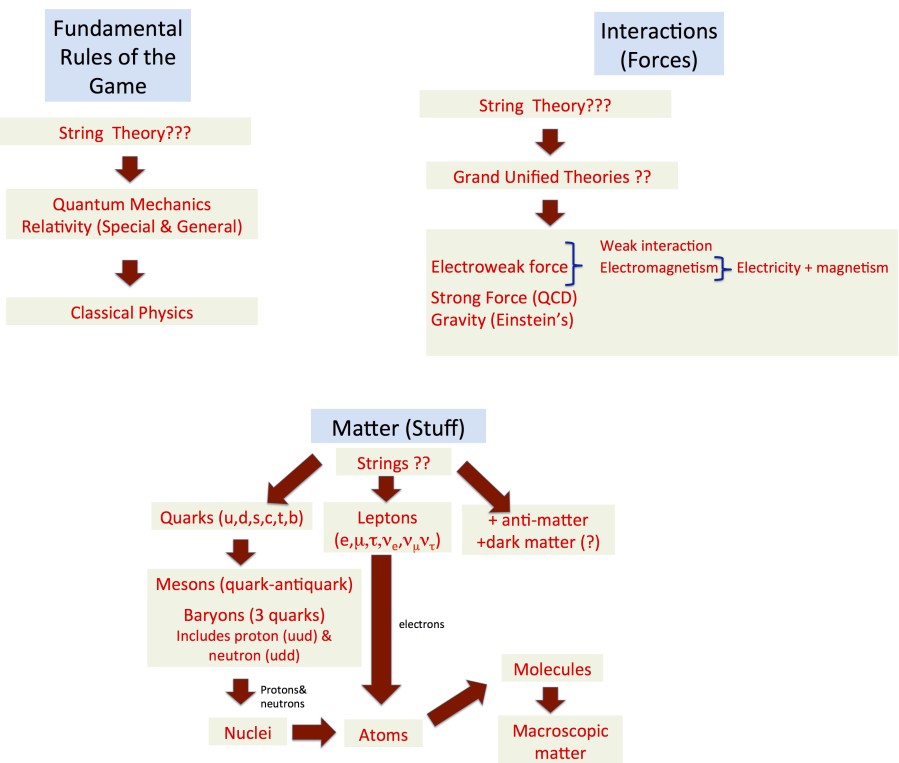

Figure 1: A highly incomplete and cartoon-like overview of the understanding of fundamental physics circa 1895.

Rather, these notes emphasize physics developed over a forty-year span: from some aspects of nuclear physics developed in the mid-1930s to the standard model which was constructed by the mid-1970s. Even then, the notes pretty much over-simplify most of the topics in order to focus on the underlying physical ideas.

Before beginning this scientific journey, it is perhaps instructive to consider briefly how the world's understanding of fundamental physics developed over the forty years prior to the mid-1930s. In fact, that forty year period probably represents the single biggest surge in mankind's understanding of nature at a fundamental level of any comparable period. Consider Fig. 2,

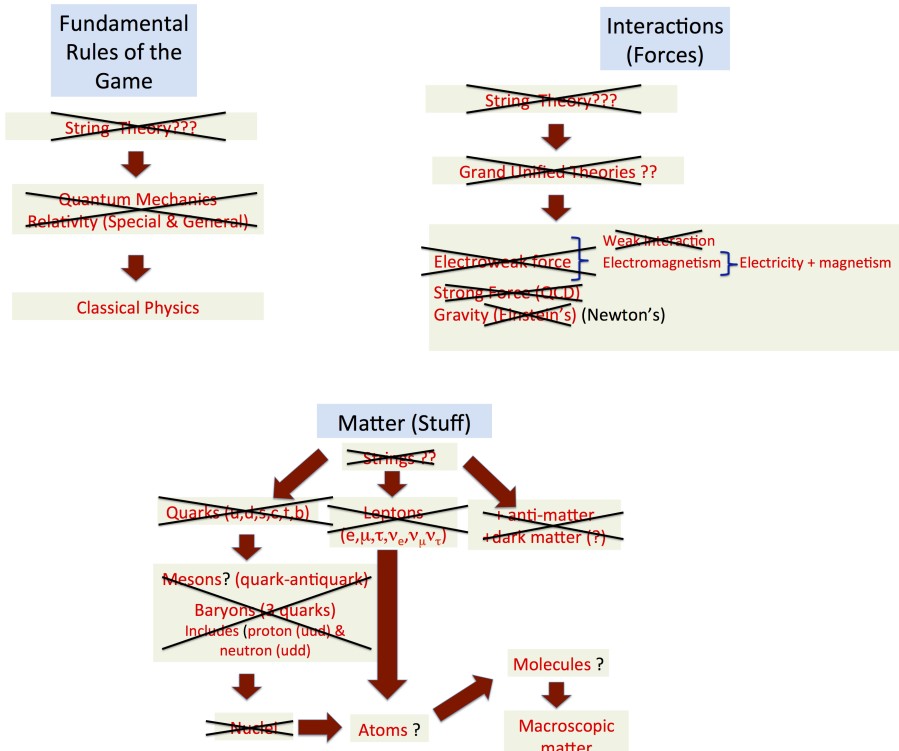

Figure 2: A highly incomplete and cartoon-like overview of our current understanding of fundamental physics.

which revisits fig. 1 but crosses out that which was unknown at the time.

By 1935 many of the crossed out aspects were already well-established including quantum mechanics, special and general relativity, the strong and weak nuclear forces, electrons, protons, neutrons and nuclei, anti-matter and the pion (at least as a conjecture).

If one wishes to skip this historical introduction and jump into the meat of these notes, please free to do so.

In 1895 there was neither nuclear nor particle physics, since neither nuclei nor subatomic particles had yet been discovered.

Indeed, at the time the existence of atoms and molecules remained controversial. While they were useful in describing many aspects of chemistry and the assumption of atoms and molecules allowed the derivation of thermodynamic results from the statistical mechanics of Boltzmann, Maxwell and Gibbs, prominent scientists at the time including Mach (of Mach's principle fame) and Ostwald (a future chemistry Nobel laureate) both doubted the physical existence of atoms at this stage. Indeed, one of the reasons that Einstein's 1905 paper on Brownian motions was so important was that it was one of the final nails in the coffin of resistance to the acceptance of atoms.

Interestingly, the discovery of both subatomic particles and of the nucleus stemmed from the same technical development of the late 19th century: cathode-ray tubes. Many physicists studied what happened when one ran an electric current in evacuated glass vessels which had a large electrostatic potential across them.

J.J. Thompson's 1897 discovery of the first subatomic particle, the electron, was directly tied to cathode ray tubes. By studying the curvature of these rays in magnetic fields Thompson showed that whatever composed cathode rays had a fixed ratio of charge to mass, which

Thompson measured. The simplest explanation for this is that they were composed of a single type of particle with fixed charge and mass. When Millikan subsequently measured the charge of the electron, its mass was determined as well.

The connection of cathode rays to the discovery of the nucleus was far more indirect. At the end of 1895, Röntgen discovered a new type of penetrating radiation— x-rays (or as the German's still call them, Röntgen rays)—while experimenting with a cathode ray tube. These were produced when high voltage cathode rays impinged on a surface. Now we know that x-rays are ordinary electromagnetic radiation at an extremely high frequency. Moreover, its origin is of an atomic nature (inner shell electrons) and has nothing to do with nuclear dynamics.

However, the discovery of X-rays was a true international sensation both in the popular press and among scientists. Virtually any scientists with the resources to study X-rays, did so. One of these was Henri Becquerel. Becquerel was the physics Professor at the Muséum National d'Histoire Naturelle—a job previously held by his father and grandfather. Within a few months of the announcements of Röntgen's discovery, Becquerel conducted a fateful experiment. Röntgen had reported that X-rays cause phosphorescent materials to emit light. Becquerel wondered whether there was an inverse process in which light impinging on a phosphorescent material would emit X-rays and set out to test the idea experimentally.

As it happens, we know this process does not exist. Becquerel, did not. He decided to probe this idea in the following way: expose a phosphorescent screen to bright sunlight and then place it in a sealed envelope with an unexposed photographic plate. Since he knew that X-rays expose film, he hoped to show that the phosphorescent screen would expose the film, indicating X-rays. However, the day he set out to do the experiment was cloudy, he decided to proceed, in any case—presumably to give a comparison to what happened. When the film was developed Becquerel was surprised to discover that it had been exposed. Puzzled by this, he repeated the experiment, this time keeping the phosphorescent material entirely in the dark; again the film was exposed. Becquerel deduced correctly that whatever was exposing the film was spontaneously coming from the phosphorescent material itself.

As it happens, the phosphorescent material used by Becquerel was a uranium salt and Becquerel was able to deduce that some new type of radiation was emanating from the uranium. He had discovered radioactivity.

Thus began an intense period of study of radioactivity. At the time, it was not known that radioactive decays came from nuclei; indeed the existence of the nucleus would not be deduced for another 17 years. But early researchers, led by the husband and wife team of Pierre Curie and Marie Skłodowska Curie were able to determine quite a lot about radioactivity.

It was realized that different types of chemical elements had characteristic radioactive decays. Each type of decay was associated with a characteristic half-life. Radioactive properties were used to deduce the existence of new elements—the first two, radium and polonium were discovered by the Curies. It was soon realized that some chemical elements had distinct decays with different half-lives, so that isotopes of elements which were virtually identical chemically but never-the-less distinct.

It was also recognized there were distinct types of radioactive decays: $\alpha$ decays which were deflected by electromagnetic fields and easily stopped by matter and which were mono-energetic, $\beta$ decays which were also deflected by electromagnetic fields, were far more penetrating and had a continuous energy spectra, and $\gamma$ radiation which was associated with some $\alpha$ and $\beta$ decays, were not deflected by electromagnetic fields, were highly penetrating and mono-energetic. We now understand that $\alpha$ decays involve the emission of $^4$He nuclei and are associated with the strong nuclear force while $\beta$ decays involved the emission of an electron and were associated with an entirely distinct force–the weak nuclear force. Thus, two of nature's forces which had gone unnoticed throughout human history were uncovered within

a few years of each other. It turns out that $\gamma$ radiation involves the emission of an ordinary photons, but they are far more energetic than ones coming from atomic processes.

While these studies of radioactivity were taking place there were two revolutionary theoretical developments. The earliest ideas of quantum physics were formulated by Planck in 1900 to help resolve paradoxes in the statistical mechanics of black-body radiation and five years later by Einstein to explain the photoelectric effect. That same year Einstein proposed special relativity.

On the experimental side, Rutherford then began a series of truly revolutionary studies. He demonstrated that when when radioactive decays occur, chemical elements change from one type to another. When he won the Nobel prize in chemistry for this work he note that he "had dealt with many different transformations with various time-periods, but the quickest he had met was his own transformation from a physicist to a chemist." This work helped clarify what is going when a radioactive decay occurs. The system (which Rutherford subsequently showed to be a nucleus) is characterized by two positive integers $Z$, the electric charge in units of $e$ and $A$ which is to pretty good approximation proportional to the mass. When an $\alpha$ decay occurs $Z$ decreases by 2 while $A$ decreases by four; in contrast in a beta decay, $Z$ increases by one while $A$ remains the same.

Rutherford had a brilliant insight: instead of passively studying matter by looking at how some types of matter emit radioactive particles, one could use radioactivity as a probe of matter. He designed an experiment carried out at the University of Manchester by a postdoctoral scientist Geiger (of Geiger counter fame) and an undergraduate Marsden, in which $\alpha$ radiation impinged on a thin gold foil, and the angle of their deflection was measured. At the time, the prevailing atomic model was Thompson's "plum pudding" model in which the electrons (the "plums") were contained in a diffuse positively charged "pudding". (Ironically this almost the exact opposite of what actually happens in which a diffuse quantum mechanical cloud of electrons surround a compact nucleus.)

Since the electrons were much lighter than the $\alpha$, Rutherford expected very small deflections. He was shocked when Geiger and Marsden found deflections at all angles including at back angles. Rutherford was dumbfounded: "It was quite the most incredible event that has ever happened to me in my life. It was almost as incredible as if you fired a 15-inch shell at a piece of tissue paper and it came back and hit you." During the next two years Rutherford analyzed the data. He eventually realized that that the differential cross-section observed by Geiger and Marsden was that of scattering off of a Coulomb potential. This analysis is quite impressive—while the calculation of the differential cross-section for classical scattering from a $1/r$ potential is now a standard undergraduate exercise, the concept of differential cross-section did not exist when Rutherford began his analysis; he needed to invent it to proceed.

Rutherford's analysis indicated that there was a small, heavy charged core at the center of the atom—a nucleus. Rutherford quickly realized that the mass of the atom was almost entirely contained in the nucleus. The atomic number Z which determined the chemical properties was given by the charge of the nucleus. Rutherford rapidly postulated that the nucleus of the hydrogen atom was a charge unity particle with $A = 1$: the proton.

First Bohr and subsequently Schrödinger, Heisenberg, Dirac and others took the quantum ideas of Planck and Einstein and constructed a viable and quantitatively accurate description of the atom.

Rutherford's experiment gave very little information about the nucleus other than its existence and an upper bound on its size. The differential cross-section was that of a Coulomb potential, and a spherically symmetric charge distribution looks like a point charge at the center when one is outside the distribution. Thus, the results of the gold-foil experiment meant that the few MeV $\alpha$ particles did not have the energy to get close enough to the nucleus to penetrate it.

The question of how one could probe the dynamics of the nucleus itself was critical. Clearly, nuclei had their own internal dynamics: $\gamma$ radiation, the emission of photons from excited nuclear states, was analogous to the emission of photons from excited atomic states that Bohr had described. Rutherford again had an important insight: Since the Coulomb repulsion of gold was too strong for an $\alpha$ particle to penetrate the nucleus, if one shot $\alpha$ particles at much lighter nuclei, they might well be able to penetrate. Rutherford conducted the following experiment: Direct $\alpha$ particles onto a gas of nitrogen in a container. While the analysis took some time, it was ultimately shown that collisions emitted a proton and left behind an isotope of oxygen—Rutherford had discovered nuclear reactions.

Unfortunately, this technique of inducing nuclear reactions was restricted to light nuclei. In order to get charged particles inside of heavier nuclei, one needed more energetic beams than those produced by natural radioactive decays. This motivated early attempts to develop machines that could accelerate particles to high energies. The most significant of these was the invention of the cyclotron by Ernest Lawrence. The key to this device is resonance: rather than giving particles a single large kick, give them many coherent small ones. This was possible because non-relativistic particles in a magnetic field have orbits with a natural frequency that depends on the field strength, the charge and the mass of the particle but not its energy. Thus, if the particle were to cycle in a magnetic field and be driven by a radio frequency electric field tuned to the cyclotron frequency all of the kicks from the RF field would be coherent and the energy would grow. Ultimately, Lawrence's "atom-smashers" grew to be quite large and expensive—it was the beginning of "big science".

Another way to learn about nuclei was developed: instead of using large energies to study them, use high precision. The charge to mass ratio of ions could be determined by measuring how they bent in magnetic fields. This technique—mass spectroscopy—was the same technique by which Thompson discovered the electron. Francis Aston developed the mass spectrometer into a precision instrument. Since the charges of the ions were known (they were some multiple of the charge of the electron) he was able to measure their masses and to do so quite accurately. He found that the masses were not exactly proportional to the mass number $A$. These small discrepancies were related to the underlying masses of the constituents of the nucleus and, through Einstein's mass-energy relation to the binding energy of the nuclear force. The data on nuclear masses was sufficiently accurate that the binding energies themselves could be be determined quite well. This in turn give significant information about the nature of the strong nuclear force.

As the 1930's dawned much of our modern understanding was in place. Relativity and quantum mechanics were well-established and early attempts to develop quantum field theories were ongoing–although they were afflicted by theoretical problems which would not be tamed until after the second world war. Much had been learned about strong interactions and the nuclear world although there were large gaps.

Weak interactions responsible for $\beta$ decays remained very mysterious. A critical problem is that the outgoing electrons had a continuous spectrum. One possibility seriously considered at the time was that energy was simply not conserved. Wolfgang Pauli made a critical suggestion that ultimately turned out to be correct, namely that the missing energy was carried by a very light neutral particle (which Pauli dubbed a "neutron"—as neutrons, the partners of proton, had yet to be discovered; Fermi renamed them "neutrinos," Italian for "little neutral ones.")

Pauli's suggestion was one of the more remarkable scientific communications of the 20th century. Rather than publish this seminal idea in a peer-reviewed journal, he communicated it in in a very flippant letter to a scientific meeting on radioactivity in Tübingen that he did not attend.

The letter begins "Dear radioactive ladies and gentlemen." He notes the problem, suggests neutrinos as a way out and then writes "But so far I do not dare to publish anything about

this idea, and trustfully turn first to you, dear radioactive people." He goes on to say "I admit that my remedy may seem almost improbable because one probably would have seen those neutrons, if they exist, for a long time. But nothing ventured, nothing gained, and the seriousness of the situation, due to the continuous structure of the beta spectrum, is illuminated by a remark of my honored predecessor, Mr Debye, who told me recently in Bruxelles: 'Oh, It's better not to think about this at all, like new taxes.' Therefore one should seriously discuss every way of rescue. Thus, dear radioactive people, scrutinize and judge." Toward the end of the letter he explains that "Unfortunately, I cannot personally appear in Tübingen since I am indispensable here in Zürich because of a ball on the night from December 6 to 7."

A critical discovery about the nature of fundamental physics was made by Carl Anderson. When studying cosmic rays in the early 1930s he found tracks that bent in a magnetic field as though it had the same charge-to-mass ratio as the electron but with the opposite charge. He had discovered the positron–the anti-particle of the electron. Interestingly, Dirac's relativistic quantum treatment (which these notes discuss later on) which was formulated a few years prior to Anderson's experimental observation predicts the existence of positrons. However, the prediction was so radical at the time that Dirac basically did not believe his own prediction and, until the discovery of the positron, tried in vain to come up with a sensible way to interpret the positron as a proton.

One interesting sociological fact about the particle physics community is that while it is now commonplace for particle theorists to postulate new particles—a decent particle theorist should be able to propose six new particles before breakfast—through the 1920s it was hard for physicists to even consider the possibility of new particles. At the time only three particles believed to be fundamental were known: the proton (which we now know to be composite), the electron and the photon. Proposing a new particle at the time was a truly radical step. Thus we see Dirac's unwillingness to accept the implications of his own equation and Pauli's very apologetic and tentative proposal for neutrinos.

As noted above, at the time Pauli introduced what we now call the neutrino, the neutron had not been discovered. At the time, it was generally believed that a nucleus was composed of $A$ protons and $A-Z$ electrons to yield the correct charge and mass. The existence of electrons in the nucleus seemed to make sense in that $\beta$ decays emitted electrons from the nucleus. There were known to be problems with such a picture. In the first place it is unclear how it would fit with a continuous spectrum for $\beta$ emission and how Pauli's neutrinos would fit such a picture. Also the energetics were problematic since the uncertainty principle would indicate that electrons confined to a region as small as a nucleus should have very large kinetic energies.

The neutron was discovered soon there after. Fredric and Irene Joliot-Curie in Paris (Irene was the daughter of Pierre and Marie Curie) followed Rutherford's approach of bombarding light nuclei with $\alpha$ particles. They studied the reaction of $\alpha$ impinging on $^9$Be. A nuclear reaction occurred in which a neutral particle was emitted. The Joliot-Curies assumed that it was a $\gamma$ ray—a photon. Photons were the only known neutral particles at the time, and as noted, at the time postulating new particles was virtually unthinkable to most physicists. However the "$\gamma s$" seen by the Joliot-Curies were highly problematic—when directed on paraffin (a good source of hydrogen) they knocked out protons—as one might expect from $\gamma s$ but—the protons were of very high energies; given the need to conserve energy and momentum when the neutral particle knocks out a proton, one need implausibly energetic $\gamma s$ to emerge from the initial reaction. The Joliot-Curies noted this oddity and considered it a puzzle, but never made the intellectual leap of considering the possibility of a new particle.

Chadwick, a Physicist at Cambridge's Cavendish Lab and one of the many nuclear scientists trained by Rutherford, was not averse to the possibility that what was observed by the Joliot-Curies was a new particle. In part this was because given the difficulties of nuclear modeling at the time, Rutherford had previously speculated about the possibility of a neutron. In a series

of experiments Chadwick demonstrated that the electric neutral emissions observed when $\alpha$ particles impinged on $^9$Be were massive and had a mass nearly identical to that of the proton. Eventually it was shown to be just slightly more massive than the proton—approximately .1% heavier. Ultimately the near degeneracy of the proton and neutron masses gave rise to the understanding that the nuclear force had an underlying approximate symmetry. Going forward in these notes we will use the word "nucleon" to refer to either a proton or neutron when there is no need to specify which one.

With the discovery of the neutron the basic constituents of the nucleus were known. The nucleus was composed of $Z$ protons and $A - Z$ neutrons, bound together by the strong nuclear force. Thus $A$ is the total number of nucleons.

With the discovery of the neutron, Fermi was able to construct a quantum field theoretic description of $\beta$ decay based on the existence of the neutron and Pauli's idea of a neutrino. In Fermi's theory, while there are no electrons or neutrinos in the nucleus, there is a process in which the neutron becomes a proton and the process itself creates an electron and an antineutrino. While Fermi's theory ultimately turned out to be incomplete and not fully consistent mathematically, it played an important role in the development of the standard model and illustrates some key ideas. We will encounter it later in a later chapter of these notes.

Another key idea tied to quantum field theory was developed by the mid-1930s. Mass spectroscopy had determined the binding energies of a great many nuclei and this allowed for the understanding of some key features of the strong nuclear force. As we discuss in the following section, the systematic of nuclear binding energies implied that the force between nucleons must be short-ranged—unlike the long-ranged Coulomb force that is that binds electrons to nuclei in atoms. Yukawa realized that a short-ranged interaction naturally arises in a field-theory if the interaction arises from the virtual exchange of a massive particle with the range of the potential being inversely proportional to the mass of the exchanged particle. The long-ranged nature of the Coulomb force is associated with the fact that it arises due to the virtual exchange of massless photons. Thus, the short-ranged nature of the strong force between nucleons is associated with the exchange of massive particles that we now call mesons. We will discuss the Yukawa theory in some detail later in these notes.

The discovery of the neutron lead to an enormous advance in the study of nuclear physics. As noted above, prior to this discover it was very difficult to study nuclear dynamics experimentally. Electrically charged probes such as $\alpha$ particles from radioactive decays could only penetrate small nuclei due to the strong Coulomb repulsion. While this problem was ultimately overcome by cyclotrons, which were able to produce high-energy beams, these were very expensive to build and operate. Neutrons, which were easy to produce by shooting $\alpha$ particles on light nuclei such as $^9$Be, did not suffer from Coulomb repulsion and hence could be used to probe nuclei large and small.

Fermi, more than any other physicist, seized this opportunity. He was appointed to the newly created chair of theoretical physics in Rome, while still in his mid twenties. From this perch, ostensibly in theoretical physics, he led a group of young scientists that began a remarkable and systematic experimental program that studied neutron induced reactions on nuclei throughout the periodic table. This group made numerous discoveries. One of the most remarkable ones was that the cross-sections for neutron-induced reactions increased dramatically when neutrons were slowed down by elastic scattering against the protons in a medium such as paraffin.

At first this behavior seemed highly counter-intuitive. Naively, if one simply views the nucleus as a collection of protons and neutrons, it seems natural that the higher energy an incident neutron has, the more likely it is to have the oomph to knock out a nuclear constituent and induce a reaction. Bohr advocated a key idea that might explain this apparently counter-intuitive behavior. In the next section we will discuss some aspects of this idea which involves

thinking about nuclei as being analogous to drops of liquid. While this picture is clearly not the entire story, it provides a remarkably simple way to understand some basic features of nuclear physics which apply to a wide-array of nuclei.

# 2 The Liquid Drop Model

Historically, the nucleus was viewed as nothing more than the sum of its constituents, the protons and neutrons. In light of this view, the results of the slow neutron experiments were baffling: why should slow neutrons be more effective at starting reactions than fast ones? Intuitively, one would expect precisely the opposite, that a fast neutron would be more effective at knocking a proton or neutron out of a nucleus and starting off a reaction.

Some light was shed on the issue in the mid 1930s by Niels Bohr, who shifted focus to the interactions which held the nucleus together. Although he didn't actually know any of the details of the forces at play in nuclear dynamics, he was able to realize that whatever force was at work must be capable of efficiently sharing energy between many nucleons, and enabling them to act collectively.

As the title of the section suggests, the key analogy is to a drop of liquid. In a water droplet, there are some kinds of forces holding the water molecules together, as well as *surface tension*, which makes it energetically favorable to minimize the surface area of the drop, pulling it into a spherical shape.

Applying this picture to a nucleus, we can imagine that when a neutron is absorbed it spreads its energy into the "liquid" made up of the nucleons, causing the liquid to be heated up. Over time, the nucleus can dissipate this energy by emitting particles, and eventually return to its ground state. One can then imagine that a low energy particle incident on the nucleus would be more efficient at starting a reaction since it won't just knock off a single nucleon right away, but rather allow its energy to spread throughout the nucleus. As simple as this liquid drop model may sound, it was ultimately the paradigm by which nuclear fission was understood.

## 2.1 Basic Nuclear Energetics

In the context of Bohr's liquid drop model, it is reasonable to suggest that there is a natural density for nuclear matter. While this isn't precisely true for any nucleus (quantum effects play a role), it's a good approximation that roughly fits the trend.

If the density of the nuclear "liquid" is constant, the volume will be proportional to the atomic weight, $A$ (recall this is the total number of neutrons and protons, $A = Z + N$). Since the volume goes like the radius cubed, we then expect the radius to be proportional to $A^{1/3}$. If we take this model seriously, we can imagine several contributions to the binding energy of the nucleus,

▶ Since each nucleon will contribute some binding energy, the total binding energy should be proportional to $A$.

▶ However, just like a liquid drop has surface tension, we also expect it to be energetically costly to maintain a large surface area. Since we've already established the radius goes like $A^{1/3}$, we expect the binding energy from surface tension to be proportional to the surface area, and hence scale like $A^{2/3}$.

▶ The Coulomb force will act to push the nucleus apart, so keeping it together will cost some energy. Although this is a small effect for small nuclei (about 1% of the binding energy), it becomes important for larger ones. We know the electric potential energy goes like the charge squared over the radius, so we expect a contribution $\sim Z^2/A^{1/3}$.

▶ The nucleus "wants" to have an equal number of protons and neutrons, so in the simplest case a proton-neutron asymmetry will come with an energy cost $\sim (Z - N)^2$, or equivalently $(2Z - A)^2$. The fact that nuclei with fixed $A$ will tend to energetically favor

configurations with an equal number of protons and neutrons (all else being the same) can be understood as due in part to the Pauli principle; the detailed dynamics of the strong nuclear force also plays a role. In any case, it is an empirical fact.

Putting all of these pieces together, we arrive at the Bethe-Von Weizsäcker semi-empirical mass formula, where the nuclear mass is given by

$$M = ZM_p + (A-Z)M_N - \frac{BE}{c^2}, \tag{1}$$

where $M_p$ and $M_N$ are the proton and neutron masses respectively, and $BE$ is the binding energy,

$$BE = a_V A - a_S A^{2/3} - a_{\text{elec}} \frac{Z^2}{A^{1/3}} - a_A \frac{(2Z-A)^2}{A}. \tag{2}$$

This equation simply sums up the considerations we discussed above: the first term is due to the fixed nuclear energy density of nuclear matter, the second accounts for the surface energy, the third for the electric energy, and the fourth for the proton-neutron asymmetry. The coefficients of each term are determined by fitting to experimentally measured masses, and are roughly

$$
\begin{aligned}
a_V &\approx 15.8 \text{ MeV} \\
a_S &\approx 17.8 \text{ MeV} \\
a_{\text{elec}} &\approx .711 \text{ MeV} \\
a_A &\approx 23.7 \text{ MeV}.
\end{aligned}
\tag{3}
$$

By rewriting this as the binding energy per nucleon, $BE/A$, we can quantify the stability of nuclei,

$$\frac{BE}{A} = a_V - a_S A^{-1/3} - a_{\text{elec}} f_p^2 A^{2/3} - a_A (2f_p - 1)^2, \tag{4}$$

where we have defined

$$f_p = \frac{Z}{A} \tag{5}$$

as the fraction of protons to total nucleons. This form of the equation makes it easy to see that if we can ignore the electric energy, the only remaining term dependent on $f_p$ is the final one, which for a nucleus with fixed $A$ is minimized when $f_p = 1/2$, i.e. when we have an equal number of protons and neutrons.

We have already mentioned that the electric term is unimportant for small nuclei, so we expect that for small nuclei the most stable configuration (largest binding energy) has an equal number of protons and neutrons. However, as $A$ increases and the nucleus gets larger, the proton fraction for the most stable configuration will decrease as the Coulomb force becomes more important, making the nucleus neutron rich. This is shown in Figure 3, and a comparison between the semi-empirical mass formula and experimental data is shown in Fig. 4.

The most stable nucleus is that of $^{56}$Fe, which has 8.8 MeV of binding energy per nucleon.[1] This is the maximum binding energy per nucleon, and allows us to neatly divide nuclei into those larger or smaller than $^{56}$Fe. Smaller nuclei will gain binding energy from getting pushed together and fusing, emitting the extra energy via mass and kinetic energy. To achieve fusion, one must start with sufficient energy to overcome the Coulomb repulsion encountered when the nucleus is compressed, which typically only occurs in very hot, "thermonuclear" environments, such as the interior of a star or during the explosion of a hydrogen bomb. The

---

[1]Notice that this is much smaller than the $\sim 16$ MeV contribution from the volume term in the semi-empirical mass formula. The other terms are important!

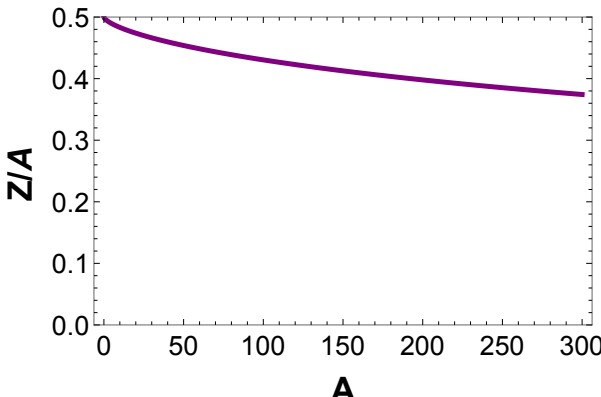

Figure 3: Fraction of protons in most stable configuration, given by semi-empirical mass formula

central challenge of creating a safe and controlled fusion reactor is maintaining a sufficient temperature and density for fusion reactions to take place.

On the other hand, nuclei much larger than $^{56}$Fe gain binding energy by breaking apart into smaller nuclei in a process known as fission. Typically the nucleus is locally stable, so it needs some "encouragement" to undergo fission. One commonly used means of encouragement is hitting it with an external particle. Fission can also occur spontaneously due to quantum mechanical tunneling, but it happens at a sufficiently low rate that we wouldn't recommend trying to start an energy company based on it.

### Nucleosynthesis

It is generally believed that protons and neutrons were created after the big bang, as the universe cooled. Since the universe was still hot and dense, nuclear fusion could easily occur, and left over neutrons could $\beta$ decay into protons. Models of this process suggest that after the big bang, the universe contained deuterium ($^2$H), tritium ($^3$H), $^3$He, $^4$He, and a small amount of Li. These predictions are in line with astronomical measurements, but a big bang nucleosynthesis offers no explanation of where heavier elements come from. These elements are synthesized in stars, which are powered by nuclear fusion. This is essentially an equilibrium process, of which we have a solid understanding; we expect that any step in the fusion process will result in an increase of the binding energy. But, this means that fusion in stars can only account for the creation of elements up to $^{56}$Fe!

Elements heavier than $^{56}$Fe must come about via some complicated non-equilibrium process in the presence of many excess neutrons. For a long time, the popular view was that creation of these elements occurred in supernova explosions, but the recent LIGO observation of colliding neutron stars showed that heavy nuclei were synthesized in the process. This means that some (or perhaps most, or all) heavy nuclei are forged in neutron star collisions.

The semi-empirical mass formula also implies that there is a maximum size for nuclei. All of the terms scale as $A$ to some power (using appropriate variables), and the fastest growing term in the binding energy per nucleon is the electric term,

$$\frac{BE}{A} \sim -a_{\text{elec}} f_p^2 A^{2/3} . \tag{6}$$

The minus sign tells us the interaction is repulsive, since $f_p^2$ and $A$ are positive. Further, $f_p^2$

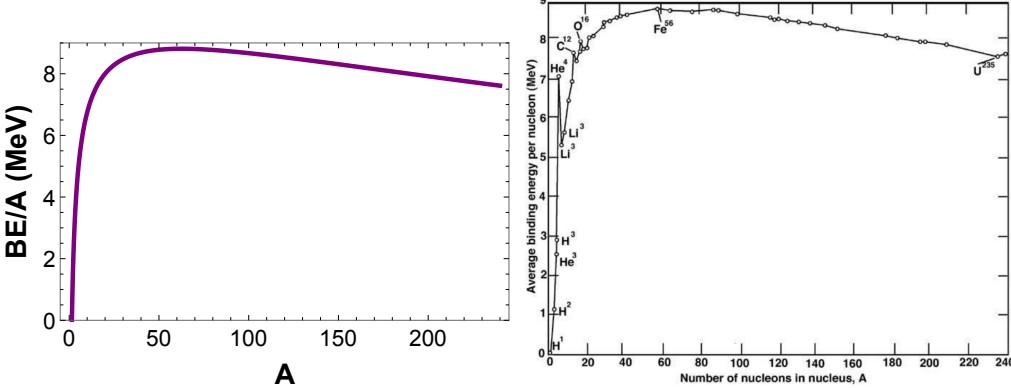

Figure 4: Left: theoretical binding energy per nucleon for most stable configuration, from semi-empirical mass formula; Right: Actual data (courtesy of Xiangdong Ji), note the remarkable agreement, despite the simplicity of the model!

is generically nonzero because of the symmetry term in (4). The numerical coefficient $a_{\text{elec}}$ is small, causing the term to be unimportant, except when $A$ is large. In fact, since this term has the fastest $A$ dependence it dominates at sufficiently large $A$. Since we've established this term is repulsive, this could feasibly lead to a bound on $A$, and hence the size of the nucleus.

In addition to the electric force, which is fairly weak, there is also the strong nuclear force. Which, as its name suggests, is strong, but is extremely short-ranged. Meanwhile, the electric force is effectively infinite ranged. Since this course is largely theoretical in emphasis, we can make things simpler by turning off the Coulomb force and asking what happens. (It turns out that experimentalist friends are somehow unwilling to do this for us in the lab). In such a world, the volume term dominates, so even with the surface energy term, there is still nothing precluding infinite nuclear matter.

This infinite nuclear matter is a theoretical substance that gets to the heart of the semi-empirical mass formula, in that it would then suggest a natural density for nuclear matter. In fact, experiment suggests that such a natural density exists. Extrapolating from the density of real nuclei, the density of infinite nuclear matter is estimated to be

$$\rho_N \sim .16 \quad \text{fm}^{-3}, \tag{7}$$

which is to say that there are .16 nucleons per cubic femtometer (1 fm = $10^{-15}$ m).

---

## Improving the Semi-empirical Mass Formula

There are several corrections we could add to the semi-empirical mass formula to improve its accuracy. First of all, we could add an "even-odd" correction, reflecting that fact that nucleons "want" to bind into pairs, and thus there is an energy cost for odd numbers of nucleons.

We could also add a "shell correction," by considering each nucleon to sit in an effective potential due to all the other nucleons, and additionally subjecting them to the Pauli Principle. Filled shells are most stable, and occur when the number of protons $Z$ or neutrons $N = A - Z$ are equal to $2, 8, 20, 28, 50, 82$, or $126$. The first three levels can be found by considering a simple spherical potential, and the subsequent ones can be found by taking into account spin-orbit coupling. The last (126) is only possible for neutrons, since we have yet to find an element with $Z \geq 126$.

Another class of effect one could include is the contribution of higher order terms in $(Z-N)^2$. Of course, it must be an even function, but we need more data on neutron-rich

# 3 Measuring Nuclear Density

Having established that the density of nuclei is something interesting we'd like to learn more about, the question becomes how we measure it. Ideally, we get a very accurate scale and a very tiny meterstick, but unfortunately things aren't quite so easy.

Instead, the trick is to shoot something at the nucleus that has simple and well understood interactions. Then we can hopefully infer the nuclear density from how the particle scatters. The ideal candidate is electron scattering, since we know its interactions are electromagnetic and thus it will only couple to the electrically charged constituents of the nucleus – the protons. We also understand the form of the interaction quite well, whether it be the Coulomb interaction of non-relativistic quantum mechanics or photon exchange in Quantum Electrodynamics (QED), to be discussed later in the course. Further, as we will see shortly the scattering is weak, enabling tractable calculations using either the Born approximation in non-relativistic quantum mechanics, or single photon exchange in QED.

In what follows, and for the rest of the course, we will typically work with natural units where $\hbar = c = 1$.

> **Natural Units**
>
> A brief word about natural units is in order. Conventionally, one measures distances and times with different units. However, we do not have to. Since we know the speed of light, we can specify a spatial distance by specifying a time and consider the distance to be the how far light would travel in that time. You are undoubtedly familiar with the notion of a light-year. This is of course not the same as an ordinary year with 1/3 fewer calories; rather it is the distance that light travels in a year. Thus, light travels 1 light-year per year. Now the innovation of natural units is simply to recast light-years as years. In that case the speed of light is unity—with no dimensions. One can similarly recast $\hbar$ to be unity so that energies and inverse times have the same units. One neat thing about doing this is that it greatly simplifies dimensional analysis.
>
> While we generally will use natural units, in the nuclear domain it is not uncommon to measure distances in fm (which corresponds to $10^{-15}$ m and stands for either Fermi or femtometer) while measure energies in MeV. To convert between them one uses $\hbar c \approx 197$ MeV-fm

To resolve distances of order the nuclear size, $R \sim 1$ fm, the uncertainty principle $\Delta x \Delta p \gtrsim 1$ (in units with $\hbar = 1$) and tells us that we need a momentum transfer $q = |\mathbf{p}_f - \mathbf{p}_i|$ of at least $2\pi/R$. To get a momentum transfer of this size, we need an initial momentum $\mathbf{p}_i$ of the same order. Then, the maximum momentum transfer for elastic scattering is $q_{\max} = 2|\mathbf{p}_i|$, and we need an initial momentum of roughly

$$|\mathbf{p}_i| \gtrsim \frac{\pi}{1 \text{ fm}} \sim 600 \text{ MeV}. \tag{8}$$

Thus, we need ultra-relativistic energy scales to probe the nuclear charge distribution. However, for the sake of simplicity we will ignore this and pretend that non-relativistic quantum mechanics is adequate to address this situation. We'll come back to this problem later in the course—after we have introduced the Dirac equation and treat the issue relativistically.

But, for the time being, let's continue on non-relativistically. Suppose we have a charge distribution $\rho(\mathbf{r})$, such that

$$\int \mathrm{d}^3 r \, \rho(\mathbf{r}) = Z \,, \tag{9}$$

where we have chosen to measure the electric charge in units of $e$. The potential seen by an electron is just the Coulomb potential,

$$V(\mathbf{r}) = \int \mathrm{d}^3 r' \, \frac{-e^2}{|\mathbf{r} - \mathbf{r}'|} \, \rho(\mathbf{r}') \,, \tag{10}$$

where we have once again made our equations simpler by choosing units with $1/4\pi\varepsilon_0 = 1$. Hopefully, you've already met the fine structure constant,

$$\alpha = \frac{1}{4\pi\varepsilon_0} \frac{e^2}{\hbar c} \approx \frac{1}{137} \,, \tag{11}$$

which in our choice of units is simply $\alpha = e^2$. Since $V(\mathbf{r}) \sim e^2 \sim \alpha$ and $\alpha$ is a small number, the potential, and hence the scattering off of it, is weak. This allows us to use the *Born approximation*, which is valid for weak scattering, in that it basically assumes that the particle only interacts with the potential once.

When we treat scattering in quantum mechanics, our goal is to calculate the *scattering amplitude*, $f(\theta, \phi)$, from which we can calculate the differential cross section,

$$\frac{\mathrm{d}\sigma}{\mathrm{d}\Omega} = |f(\theta, \phi)|^2 \,, \tag{12}$$

which, roughly speaking, is the ratio of particles scattered in a given direction to the number of incoming particles. In the Born Approximation the scattering amplitude is proportional to the Fourier transform of the potential,

$$f(\theta) = -\frac{m}{2\pi} \int \mathrm{d}^3 r' \, \mathrm{e}^{-i(\mathbf{p}_f - \mathbf{p}_i) \cdot \mathbf{r}'} \, V(\mathbf{r}') \,. \tag{13}$$

For later convenience, we will define the scattering angle and momentum transfer as

$$\frac{\mathbf{p}_f \cdot \mathbf{p}_i}{|\mathbf{p}|^2} = \cos\theta \,, \qquad \mathbf{q} = \mathbf{p}_f - \mathbf{p}_i \tag{14}$$

Looking back at our potential (10), we notice that it is a convolution of $1/r$ and $\rho(\mathbf{r})$.

---

**Reminder: Convolution Theorem**

A convolution $h(\mathbf{r})$ of two functions $f$ and $g$ is of the form

$$h(\mathbf{r}) = \int \mathrm{d}^3 r' \, f(\mathbf{r} - \mathbf{r}') \, g(\mathbf{r}') \,, \tag{15}$$

and we say $h$ is $f$ convolved with $g$. There is a useful theorem, called the *Convolution Theorem*, which says that the Fourier transform of a convolution is the product of the Fourier transforms. That is, for $h(\mathbf{r})$ as defined above, its Fourier transform $\tilde{h}(\mathbf{q})$ is

$$\tilde{h}(\mathbf{q}) = \int \mathrm{d}^3 r \, \mathrm{e}^{-i\mathbf{q}\cdot\mathbf{r}} \, h(\mathbf{r}) = \int \mathrm{d}^3 r \, \mathrm{e}^{-i\mathbf{q}\cdot\mathbf{r}} \int \mathrm{d}^3 r' \, f(\mathbf{r} - \mathbf{r}') \, g(r') = \tilde{f}(\mathbf{q}) \, \tilde{g}(\mathbf{q}) \,, \tag{16}$$

where

$$\tilde{f}(\mathbf{q}) = \int d^3r \, e^{-i\mathbf{q}\cdot\mathbf{r}} f(\mathbf{r}),$$

$$\tilde{g}(\mathbf{q}) = \int d^3r \, e^{-i\mathbf{q}\cdot\mathbf{r}} g(\mathbf{r}),$$

(17)

are the Fourier transforms of $f(\mathbf{r})$ and $g(\mathbf{r})$.

Using the Convolution theorem we can write the scattering amplitude as the product of the Fourier transforms of the charge density and the $1/r$ potential,

$$f(\theta) = \underbrace{\left( \int d^3r \, e^{-i\mathbf{q}\cdot\mathbf{r}} \rho(\mathbf{r}) \right)}_{\equiv g_E(q^2)} \left( \frac{me^2}{2\pi} \int d^3r \, \frac{e^{-i\mathbf{q}\cdot\mathbf{r}}}{r} \right).$$

(18)

The first term is simply the Fourier transform of the nuclear charge distribution, which is called the *electric form factor*, $g_E(q^2)$. We write it as a function of $q^2$ for a spherically symmetric distribution. The second term is the Born approximation for scattering off of a point charge of charge $+e$. This can be evaluated exactly, and gives the well known Rutherford formula.

Now, suppose our experimental friends measure the scattering cross section, as a function of $q = p\cos\theta$. By dividing out the theoretically calculable cross section for a point charge, we can determine the electric form factor,

$$\frac{\left(\frac{d\sigma}{d\Omega}\right)_{\text{exp}}}{\left(\frac{d\sigma}{d\Omega}\right)_{\substack{\text{point} \\ \text{charge}}}} = \left| \int d^3r \, e^{-i\mathbf{q}\cdot\mathbf{r}} \rho(\mathbf{r}) \right|^2 = |g_E(q^2)|^2,$$

(19)

from which we may perform an inverse Fourier transform to find the nuclear charge distribution[2]

$$\rho(\mathbf{r}) = \int \frac{d^3q}{(2\pi)^3} \sqrt{\frac{\left(\frac{d\sigma}{d\Omega}\right)_{\text{exp}}}{\left(\frac{d\sigma}{d\Omega}\right)_{\substack{\text{point} \\ \text{charge}}}}} \, e^{i\mathbf{q}\cdot\mathbf{r}} = \int \frac{d^3q}{(2\pi)^3} \, g_E(q^2) \, e^{i\mathbf{q}\cdot\mathbf{r}}.$$

(20)

So, the punchline is that electron scattering allows us to map the nuclear charge distribution! However, before we get too proud of ourselves, we have to remember that we did this calculation within the Born approximation, and we really should figure out how to do this relativistically. Also, we only considered elastic scattering, and inelastic scattering opens up an entirely different set of information. However, the fact remains that within these limits the density can still be extracted, and if we extrapolate to infinite nuclear matter we will find $\rho_N \sim .16$ fm$^{-3}$, as advertised above.

However, you've been swindled (get used to it, it'll happen a lot throughout these notes)! In addition to doing this problem non-relativistically, we also treated it as an electron scattering off of a static potential, when in fact, real scattering is a two body problem, where the nucleus moves as well. We can do this more carefully by using the center of mass variable

$$R = \frac{m_e r_e + m_n r_n}{m_e + m_n},$$

(21)

with $m_e$ and $r_e$ being the mass and coordinate of the electron, and $m_n$ and $r_n$ being those of the nucleon. We also introduce the relative coordinate $r = r_e - r_n$ and reduced mass

---

[2]The only ambiguity here is in the sign of the square root

$\mu = m_e m_n / (m_e + m_n)$, so we can write down the Hamiltonian

$$H = \frac{p^2}{2\mu} + V(r),\tag{22}$$

where $p$ is the momentum conjugate to $R$. We then repeat our analysis in the center of mass frame where $\mathbf{p} = 0$, and will still find that

$$\rho(\mathbf{r}) = \int \frac{\mathrm{d}^3 q}{(2\pi)^3}\, g_E(q^2)\, \mathrm{e}^{i\mathbf{q}\cdot\mathbf{r}},\tag{23}$$

where $g_E(q^2)$ is calculated from the cross sections in the center of mass frame,[3]

$$|g_E(q^2)|^2 = \frac{\left(\frac{\mathrm{d}\sigma}{\mathrm{d}\Omega}\right)^{\mathrm{CM}}_{\mathrm{exp}}}{\left(\frac{\mathrm{d}\sigma}{\mathrm{d}\Omega}\right)^{\mathrm{CM}}_{\substack{\mathrm{point}\\\mathrm{charge}}}}.\tag{24}$$

Our limitation to elastic scattering also poses problems, since real scattering can be inelastic, and the nucleus can break up when hit in a process such as (here, $D$ represents a deuteron)

$$e^- + D \to e^- + p + n.\tag{25}$$

Clearly, this can't be described by a two body potential! So, only the elastic part of the scattering gives us the form factor. Adding on a comedic number of superscripts, this means

$$|g_E(q^2)|^2 = \frac{\left(\frac{\mathrm{d}\sigma}{\mathrm{d}\Omega}\right)^{\mathrm{CM,\ elastic}}_{\mathrm{exp}}}{\left(\frac{\mathrm{d}\sigma}{\mathrm{d}\Omega}\right)^{\mathrm{CM}}_{\substack{\mathrm{point}\\\mathrm{charge}}}}.\tag{26}$$

Finally, even though our derivation was non-relativistic, the end result holds if we calculate the point charge cross section taking into account relativistic effects (assuming there is no spin involved, which slightly complicates things). The form factor then gets another superscript,

$$|g_E(q^2)|^2 = \frac{\left(\frac{\mathrm{d}\sigma}{\mathrm{d}\Omega}\right)^{\mathrm{CM,\ elastic}}_{\mathrm{exp}}}{\left(\frac{\mathrm{d}\sigma}{\mathrm{d}\Omega}\right)^{\mathrm{CM,\ rel.}}_{\substack{\mathrm{point}\\\mathrm{charge}}}}.\tag{27}$$

Finally, it is worth introducing some notation that you will inevitably encounter in the literature. Namely, the form factor is usually expressed as

$$g_E(q^2) = \langle \mathbf{p} + \mathbf{q} | \hat{\rho}(\mathbf{0}) | \mathbf{p} \rangle.\tag{28}$$

Here, $|\mathbf{p}\rangle$ and $|\mathbf{p} + \mathbf{q}\rangle$ are momentum eigenstates, which is the natural basis to use for a scattering problem. After all, scattering experiments basically amount to inserting some particle in a well-defined momentum state, allowing something complicated to happen, and then eventually measure the outgoing particles which have settled into a new momentum state. The $\rho(\mathbf{0})$ operator in the middle is the charge density operator evaluated at $\mathbf{r} = \mathbf{0}$, which we pick solely for convenience. To see why we can always do so, recall that the momentum operator is the generator of spatial translations, i.e. we may translate an operator by $\mathbf{r}$ if we act with

$$\hat{\rho}(\mathbf{0}) = \mathrm{e}^{-i\hat{\mathbf{p}}\cdot\mathbf{r}}\, \hat{\rho}(\mathbf{r})\, \mathrm{e}^{i\hat{\mathbf{p}}\cdot\mathbf{r}}.\tag{29}$$

---

[3]Even though the experimental cross section is of course measured in the lab frame, we can reconstruct it in the center of mass frame.

If we put this relation into (28), we have

$$
\begin{aligned}
g_E(q^2) &= \langle \mathbf{p} + \mathbf{q} | e^{-i\hat{\mathbf{p}} \cdot \mathbf{r}} \hat{\rho}(\mathbf{r}) e^{i\hat{\mathbf{p}} \cdot \mathbf{r}} | \mathbf{p} \rangle \\
&= e^{-i\mathbf{q} \cdot \mathbf{r}} \langle \mathbf{p} + \mathbf{q} | \hat{\rho}(\mathbf{r}) | \mathbf{p} \rangle .
\end{aligned}
\tag{30}
$$

We can then evaluate at $\mathbf{r} = \mathbf{0}$ to get the form factor, so without loss of generality we can always simply use

$$
g_E(q^2) = \langle \mathbf{p} + \mathbf{q} | \hat{\rho}(\mathbf{0}) | \mathbf{p} \rangle .
\tag{31}
$$

# 4 Modeling the Nucleus

Now, we'd like to consider how to model the nucleus. This is a very rich subject and one that has seen significant advances in recent years. One could easily construct an entire semester-long course on this subject, and still not do it justice. Here we are only going to consider a handful of very simple models that capture some key aspects.

The simplest approach is to simply treat nuclear matter as a finite region of *stuff*, and describe that stuff as the idealized infinite nuclear matter we discussed in the section on the liquid drop model. Then, the game simply becomes understanding the nature of infinite nuclear matter.

Although the nucleons in a given piece of nuclear matter will create an overall attractive potential for other nuclei, to good approximation the potential will be constant within the interior of the nucleus. This justifies treating the nucleons within the nuclear matter as a non-interacting Fermi gas. Since the nucleons are non-interacting, the single-particle Schrödinger equation for a given nucleon is simply

$$-\frac{1}{2m}\nabla^2\psi = \varepsilon\psi\,,\tag{32}$$

where $\varepsilon$ is the single particle energy eigenvalue. We're already familiar with the solution to this equation,

$$\psi \sim e^{i\mathbf{k}\cdot\mathbf{r}}\,,\qquad \varepsilon_{\mathbf{k}} = \frac{\mathbf{k}^2}{2m}\,.\tag{33}$$

If the nucleons were bosons, they would all occupy the lowest energy level $\mathbf{k} = 0$, but they are fermions and are restricted by the Pauli principle. Instead, the nucleons fill the available states starting from the lowest energy levels, until all of the nucleons are accounted for. The energy of the most energetic nucleons is the *Fermi energy*, $\varepsilon_F = \mathbf{k}_F^2/2m$, where the corresponding momentum $\mathbf{k}_F$ is called the *Fermi momentum*.

If we imagine putting the system in a giant box of size $L$, the momentum will be quantized as[4]

$$\mathbf{k} = \frac{2\pi n_x}{L}\hat{\mathbf{x}} + \frac{2\pi n_y}{L}\hat{\mathbf{y}} + \frac{2\pi n_z}{L}\hat{\mathbf{z}}\,,\tag{34}$$

for $n_x, n_y, n_z \in \mathbb{Z}$, so the allowed momenta form a lattice in momentum space with spacing $2\pi/L$, and hence the volume of a state in momentum space is

$$(\Delta k)^3 = \left(\frac{2\pi}{L}\right)^3\,.\tag{35}$$

To count the total number of nucleons in the system, we can just add up the occupancy of each state. However, for a system of many nucleons we can replace the sum over levels with an integral over momenta, divided by the volume per state in $k$-space. That is,

$$N = 4\left(\frac{L}{2\pi}\right)^3\int \mathrm{d}^3p\,\Theta(k_F^2 - p^2)\,,\tag{36}$$

where the factor of four accounts for the fact that we have two different kinds of nucleons (protons and neutrons), each with two spin states for a given energy level.[5] The step function

---

[4]Here, to make our lives easier, we are using periodic boundary conditions. This means that we identify $\psi(x) = \psi(x + L)$, and similarly for $y$ and $z$.

[5]Recall that the Pauli Principle only applies to identical fermions: two fermions of different species may occupy the same state

tells us that we should only integrate up to the Fermi-momentum. We can now divide both sides by the volume of the box $L^3$ to find an expression for the nuclear density,[6]

$$\rho_N = 4 \int \frac{\mathrm{d}^3 p}{(2\pi)^3} \, \Theta(k_F^2 - p^2). \tag{37}$$

---

**Reminder: Step functions and Delta functions**

Recall that the step function (sometimes called the Heaviside function) is defined as

$$\Theta(x) = \begin{cases} 0 & x < 0 \\ 1 & x > 0 \end{cases}. \tag{38}$$

In the context of (37), it tells us that the integrand is 1 if $p^2 < k_F^2$, i.e. the state is filled, and the integrand is 0 for $p^2 > k_F^2$, i.e. the state is empty.

While we're at it, let's take this opportunity to also remind ourselves of the delta function $\delta(x)$, which is infinite when its argument is zero, and zero everywhere else. By definition, the delta function integrates to one so long as its "spike" is within the range of integration,

$$\int_{-\infty}^{\infty} \mathrm{d}x \, \delta(x) = 1. \tag{39}$$

This means that if we integrate it against a function $f(x)$, the delta function picks out its value at one point,

$$\int_{-\infty}^{\infty} \mathrm{d}x \, f(x)\delta(x - a) = f(a). \tag{40}$$

We can also represent a delta function as

$$\delta(x - a) = \frac{1}{2\pi} \int \mathrm{d}k \, e^{ik(x-a)}. \tag{41}$$

To round out our collection of delta function facts, $\delta(ax) = \frac{1}{|a|}\delta(x)$, or more generally if we put any function $f(x)$ inside the argument of a delta function, we have

$$\delta(f(x)) = \sum_i \frac{\delta(x - x_i)}{|f'(x_i)|}, \tag{42}$$

where $x_i$ are the zeroes of $f(x)$ and $f'(x) = \partial_x f$. We can also define the derivative of a delta function inside an integral by integrating by parts,

$$\int \mathrm{d}x \, f(x)\partial_x \delta(x - a) = -\int \mathrm{d}x \, \delta(x - a)\partial_x f(x) = -\partial_x f(a). \tag{43}$$

Finally, the delta function is the derivative of the step function,

$$\partial_x \Theta(x) = \delta(x). \tag{44}$$

---

Since the single-particle energy depends only on $|\mathbf{k}|$, the momentum space distribution is

---

[6]Here we can safely take the limit $L \to \infty$, where $N$ will go to infinity with it, but the density will remain fixed.

spherically symmetric, allowing us to evaluate the integral in spherical coordinates,

$$\rho_N = 4 \frac{4\pi}{8\pi^3} \int_0^{k_F} \mathrm{d}p \, p^2 = \frac{2}{3\pi^2} k_F^3 \,. \tag{45}$$

This expression can then be used to estimate several bulk properties of nuclei.

A less crude approach is to use a shell model, where each nucleon experiences an effective potential due to all of the others. This effective interaction can be included in the Hamiltonian, from which one can find the allowed energy levels, which form shells that are filled up by the nucleons. The details of the model can then be appropriately tweaked to reproduce the experimental data.

In the traditional potential approach, one deduces the potential between two nucleons from phase shift analysis of scattering data, which looks something like Fig. 5. The potential is repulsive at short ranges, attractive and intermediate distances, and has a tail well described by single pion exchange.

However, this potential model is only valid for low energy scattering, since at higher energies mesons can be produced and scattering can otherwise be inelastic. There are several further challenges in using this approach to model many-body systems such as nuclei.

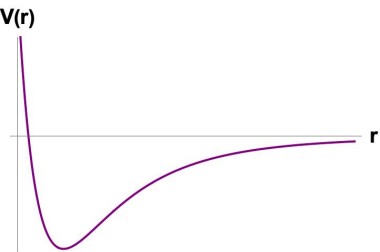

Figure 5: Effective potential

In many cases, a simple two-body potential is insufficient. A more accurate calculation requires three- or four-body potentials. Unfortunately, there is no superposition principle for many-body potentials, and a three-body potential generally cannot be decomposed as

$$V(\mathbf{r}_1, \mathbf{r}_2, \mathbf{r}_3) \neq V(\mathbf{r}_1 - \mathbf{r}_2) + V(\mathbf{r}_1 - \mathbf{r}_3) + V(\mathbf{r}_2 - \mathbf{r}_3) \,. \tag{46}$$

That is to say, it depends on the positions of all three bodies, not just the relative coordinates between pairs. In this respect, it is fundamentally different from the Coulomb potential, which is a two-body potential and satisfies superposition. So, although one can fix $V(\mathbf{r}_1 - \mathbf{r}_2)$ from experimental scattering data, it is much more challenging to constrain the three particle interaction, even armed with scattering data for systems containing three nucleons.

The other major problem is our inability to solve the Schrödinger equation for large systems. Few body systems can in many cases be solved nearly exactly using numerical methods (at least for the ground state), but larger systems are far more challenging. In some cases, their solutions may be approximated by variational methods or some kind of Green's function Monte Carlo, but even these approximations are only reliable for a system with relatively few particles.

# 5 Review of Special Relativity

The language in which nuclear and particle physics is written is that of quantum field theory (QFT). In turn, QFT is the marriage of quantum mechanics and special relativity. Hopefully, after two semesters of courses you have a decent memory of quantum mechanics, but just to ensure everyone is on the same page, its worth briefly recapping special relativity, which may not be as fresh in everyone's memory. This review also has the added benefit of clearly establishing the notation that we will use throughout the rest of the course.

We label the coordinates of an event by a *four-vector* in spacetime,

$$x^\mu = \begin{pmatrix} t \\ x \\ y \\ z \end{pmatrix}, \tag{47}$$

where the index $\mu = 0, 1, 2, 3$ tells us which component of the four-vector we're talking about, e.g. $x^0 = t, x^1 = x, x^2 = y$, and $x^3 = z$. Notice that the index is raised ("upstairs"), and we call such objects *contravariant*.

We also have the metric tensor,[7]

$$g_{\mu\nu} = \begin{pmatrix} 1 & 0 & 0 & 0 \\ 0 & -1 & 0 & 0 \\ 0 & 0 & -1 & 0 \\ 0 & 0 & 0 & -1 \end{pmatrix}, \tag{48}$$

which lets us turn contravariant vectors into *covariant* vectors, which have lowered ("downstairs") indices,

$$x_\mu = \sum_{\nu=0}^3 g_{\mu\nu} x^\nu \equiv g_{\mu\nu} x^\nu. \tag{49}$$

In the second equality we have inroduced the *Einstein summation convention*, where we agree that everytime we see an index appear twice in a term–once raised and once lowered– that we will sum over it. This saves us the time and energy of having to write lots of summation signs, so much so that this is often (semi-) jokingly referred to as Einstein's greatest contribution to physics.

Indices that are repeated, and thus summed over (or, "contracted," if you're fancy) are called *dummy indices*, reflecting the fact that their names don't matter. That is,

$$g_{\mu\nu} x^\nu = g_{\mu\xi} x^\xi = g_{\mu\clubsuit} x^\clubsuit = g_{\mu\,\mathrm{apple}} x^{\mathrm{apple}} = g_{\mu\,\mathrm{banana}} x^{\mathrm{banana}}. \tag{50}$$

This is the discrete version of the probably familiar fact that

$$\int \mathrm{d}x\, f(x) = \int \mathrm{d}\xi\, f(\xi) = \int \mathrm{d}(\mathrm{tomato})\, f(\mathrm{tomato}). \tag{51}$$

However, the names of uncontracted or *free* indices do matter: they must match on both sides of an equation. It's not hard to see that if the contravariant vector is given by (47), then the associated covariant vector is

$$x_\mu = \begin{pmatrix} t, & -x, & -y, & -z \end{pmatrix}. \tag{52}$$

---

[7]There are different conventions for the metric, so this may be different from what you've used before! This convention is the standard in the particle physics literature, whereas the convention $\tilde{g}_{\mu\nu} = \mathrm{diag}(-1, 1, 1, 1)$ is the standard in gravitational physics and string theory.

We can also use the inverse metric $g^{\mu\nu}$, which happens to be the same as $g_{\mu\nu}$, to raise indices and turn a covariant vector into a contravariant vector,

$$x^{\mu} = g^{\mu\nu} x_{\nu}. \tag{53}$$

---

### Reminder: Indices

For the purposes of this box, let's just stay in familiar $\mathbb{R}^3$. We typically represent vectors as $\mathbf{v} = v_x \hat{x} + v_y \hat{y} + v_z \hat{z}$. If we take the completely superficial step of renaming $\hat{x} \mapsto \hat{\mathbf{e}}_1$, $\hat{y} \mapsto \hat{\mathbf{e}}_2$, $\hat{z} \mapsto \hat{\mathbf{e}}_3$, and similarly for the components, we can write this more succinctly as

$$\mathbf{v} = v_1 \hat{\mathbf{e}}_1 + v_2 \hat{\mathbf{e}}_2 + v_3 \hat{\mathbf{e}}_3 = \sum_{a=1}^{3} v_a \hat{\mathbf{e}}_a. \tag{54}$$

If we choose orthonormal basis vectors, so $\hat{\mathbf{e}}_a \cdot \hat{\mathbf{e}}_b = \delta_{ab}$, i.e. the product is one if $a = b$ and zero if $a \neq b$, then the dot product of two vectors can be written

$$\mathbf{v} \cdot \mathbf{w} = \left( \sum_{a=1}^{3} v_a \hat{\mathbf{e}}_a \right) \left( \sum_{b=1}^{3} w_b \hat{\mathbf{e}}_b \right) = \sum_{a=1}^{3} \sum_{b=1}^{3} v_a w_b (\hat{\mathbf{e}}_a \cdot \hat{\mathbf{e}}_b) = \sum_{a=1}^{3} \sum_{b=1}^{3} v_a w_b \delta_{ab} = \sum_{a=1}^{3} v_a w_a. \tag{55}$$

A smart guy named Einstein realized that we don't actually have to write all of the summation signs, since whenever we sum over an index it appears twice. So, if we just agree that any time an index is repeated we know to sum over it, we can stop writing the summation signs, so the dot product is just $\mathbf{v} \cdot \mathbf{w} = v_a w_a$, with the sum implicit. Now, suppose we have a matrix,

$$\mathbf{M} = \begin{pmatrix} M_{11} & M_{12} & M_{13} \\ M_{21} & M_{22} & M_{23} \\ M_{31} & M_{32} & M_{33} \end{pmatrix}. \tag{56}$$

If we use the normal "row-column" method of matrix multiplication, we can calculate how it acts on $\mathbf{v}$,

$$\mathbf{Mv} = \begin{pmatrix} M_{11} & M_{12} & M_{13} \\ M_{21} & M_{22} & M_{23} \\ M_{31} & M_{32} & M_{33} \end{pmatrix} \begin{pmatrix} v_1 \\ v_2 \\ v_3 \end{pmatrix} = \begin{pmatrix} M_{11} v_1 + M_{12} v_2 + M_{13} v_3 \\ M_{21} v_1 + M_{22} v_2 + M_{23} v_3 \\ M_{31} v_1 + M_{32} v_2 + M_{33} v_3 \end{pmatrix}. \tag{57}$$

You can convince yourself that this is equivalent to the index expression

$$(\mathbf{Mv})_a = \sum_{b=1}^{3} M_{ab} v_b \equiv M_{ab} v_b, \tag{58}$$

where $(\mathbf{Mv})_a$ represents the $a^{th}$ component of the vector $\mathbf{Mv}$. Notice the index $a$ is free– it tells us which component of $\mathbf{Mv}$ we're talking about, whereas the index $b$ is contracted (summed over). You can explicitly calculate the terms in this sum and confirm it agrees with "row-column" matrix multiplication.

---

Given a covariant vector and a contravariant vector, we can form the *scalar product*,

$$s = x_{\mu} x^{\mu} = t^2 - x^2 - y^2 - z^2, \tag{59}$$

which is the moral equivalent of the dot product in normal three-dimensional space. We now define the *Lorentz transformations* as the set of all linear transformations which preserve the

scalar product, $s$. We can represent a Lorentz transformation as a matrix, $\Lambda^\mu{}_\nu$. The funny index structure (one up, one down) is chosen to reflect that when we contract it with a contravariant vector, we get back a contravariant vector,

$$\bar{x}^\mu = \Lambda^\mu{}_\nu x^\nu. \tag{60}$$

There is a useful analogy here. Recall that in normal $\mathbb{R}^3$ we define a rotation matrix R as a linear transformation that preserves the norm of a three-vector $\mathbf{v}$. That is, if $\bar{\mathbf{v}} = R\mathbf{v}$, then R is a rotation matrix if $\bar{\mathbf{v}} \cdot \bar{\mathbf{v}} = \mathbf{v} \cdot \mathbf{v}$. This specifies a condition on R that lets us determine if it is a rotation matrix,[8]

$$
\begin{aligned}
\bar{\mathbf{v}} \cdot \bar{\mathbf{v}} &= \mathbf{v} \cdot \mathbf{v}, \\
\mathbf{v}^T R^T R \mathbf{v} &= \mathbf{v}^T \mathbf{v}, \\
R^T R &= \mathbb{1}.
\end{aligned}
\tag{61}
$$

The same is true in special relativity. A Lorentz transformation is defined to be a linear transformation which preserves the scalar product, and we can determine a condition on $\Lambda^\mu{}_\nu$ to check if it is a Lorentz transformation, just as in the three-dimensional case. We'll do this in both normal matrix notation and index notation at the same time to help you get a hang of index gymnastics,

$$
\begin{array}{l|l}
x \cdot gx = \bar{x} \cdot g\bar{x}, & x_\mu x^\mu = \bar{x}_\mu \bar{x}^\mu, \\
x \cdot gx = (\Lambda x) \cdot g\Lambda x, & g_{\alpha\beta} x^\alpha x^\beta = g_{\mu\nu} \bar{x}^\mu \bar{x}^\nu, \\
x^T gx = x^T(\Lambda^T g\Lambda)x, & g_{\alpha\beta} x^\alpha x^\beta = g_{\mu\nu} \Lambda^\mu{}_\alpha x^\alpha \Lambda^\nu{}_\beta x^\beta, \\
g = \Lambda^T g\Lambda. & g_{\alpha\beta} = g_{\mu\nu} \Lambda^\mu{}_\alpha \Lambda^\nu{}_\beta.
\end{array}
\tag{62}
$$

Making contact with physics, we can state the Principle of Relativity as the requirement that the laws of physics be invariant under Lorentz transformations. Since we know from earlier classes that Lorentz transformations take us from one intertial frame to another, this is just a different way of stating a familiar law.

Now, we need to expand our vocabulary: $x^\mu$ is not the only four-vector in town! In fact, we *define* a four-vector to be anything that transforms under a Lorentz transformation as

$$\bar{A}^\mu = \Lambda^\mu{}_\nu A^\nu. \tag{63}$$

In general, a column of four random numbers is *not* a four-vector, it has to follow this very particular transformation law. On the other hand, if we find two four-vectors $A^\mu$ and $B^\mu$, we can take their scalar product $A^\mu B_\mu$ and get a *Lorentz scalar*, which is a quantity that is the same in all frames.

One four-vector that warrants a brief discussion is the momentum four-vector,

$$p^\mu = \begin{pmatrix} E \\ p_x \\ p_y \\ p_z \end{pmatrix}, \tag{64}$$

whose scalar product is the mass squared,

$$m^2 = p^\mu p_\mu = E^2 - p^2. \tag{65}$$

---

[8]We get to the last line by requiring that this condition hold for any vector $\mathbf{v}$.

To derive these properties, let's consider a moving particle of mass $m$. We define the proper time $\tau$ to be the time measured by a clock moving alongside the particle (i.e. the time as measured in the particle's rest frame). The difference in proper time between two events is

$$\Delta\tau = \sqrt{(t_2 - t_1)^2 - (\mathbf{x}_2 - \mathbf{x}_1)^2}\,, \tag{66}$$

which is manifestly Lorentz invariant. Now, let us parameterize the trajectory of the particle using the proper time,

$$x^\mu(\tau) = \begin{pmatrix} t(\tau) \\ x(\tau) \\ y(\tau) \\ z(\tau) \end{pmatrix}\,, \tag{67}$$

where we have an event at each value of $\tau$. If we differentiate this with respect to the proper time, we will end up with another four-vector

$$u^\mu \equiv \frac{\partial x^\mu}{\partial \tau}\,. \tag{68}$$

If $u^\mu$ is a four-vector, then $u^\mu u_\mu$ is a scalar and must be the same in all frames. This means that we can evaluate the scalar product in whatever frame is most convenient, and the result will hold in every other frame. Let's choose the rest frame of the particle, where $t = \tau$ and

$$\frac{\partial t}{\partial \tau} = 1\,, \qquad \frac{\partial \mathbf{x}}{\partial \tau} = 0\,. \tag{69}$$

Thus, $u^\mu u_\mu = 1$ in the rest frame, and every other frame. You can check that the most general four-vector for which $u^\mu u_\mu = 1$ is

$$u^\mu = \begin{pmatrix} \gamma \\ \gamma v_x \\ \gamma v_y \\ \gamma v_z \end{pmatrix}\,, \qquad \gamma = \frac{1}{\sqrt{1 - v^2}}\,. \tag{70}$$

Since this looks a velocity, it makes sense to multiply it by the mass to get the momentum,

$$p^\mu = mu^\mu = \begin{pmatrix} m\gamma \\ m\gamma v_x \\ m\gamma v_y \\ m\gamma v_z \end{pmatrix} = \begin{pmatrix} E \\ p_x \\ p_y \\ p_z \end{pmatrix}\,. \tag{71}$$

Then, $E^2 - p^2 = p^\mu p_\mu = m^2 u_\mu u^\mu = m^2$, as advertised.

Another important object is the derivative operator:

$$\partial_\mu \equiv \frac{\partial}{\partial x^\mu} = \left( \frac{\partial}{\partial t}, \frac{\partial}{\partial x}, \frac{\partial}{\partial y}, \frac{\partial}{\partial z} \right)\,. \tag{72}$$

Notice that we've defined this derivative operator with a lowered index, implying that it behaves like a covariant object. We can see intuitively why this should be the case by considering a Lorentz-scalar field, $s(x)$, whose value at a given spacetime point should be the same in any reference frame. Consider the value of this field at two nearby points, $s'$ and $s$. Their difference is $\Delta s = s' - s$, which for small separations we can expand as

$$\Delta s = \frac{\partial s}{\partial x^0}\Delta x^0 + \frac{\partial s}{\partial x^1}\Delta x^1 + \frac{\partial s}{\partial x^2}\Delta x^2 + \frac{\partial s}{\partial x^3}\Delta x^3 = \frac{\partial s}{\partial x^\mu}\Delta x^\mu\,, \tag{73}$$

where $\Delta x^\mu$ is the distance between the two points. $\Delta x^\mu$ is clearly a contravariant vector, and we've already stated that $\Delta s$ must be a scalar. The only way for this to be satisfied is if the derivative $\partial_\mu s$ is a covariant vector, implying $\partial_\mu$ itself is covariant. In particular, take note that the contraction of the derivative operator and a Lorentz vector field has all plus signs:

$$\partial_\mu J^\mu = \frac{\partial J^0}{\partial x^0} + \frac{\partial J^1}{\partial x^1} + \frac{\partial J^2}{\partial x^2} + \frac{\partial J^3}{\partial x^3}. \tag{74}$$

This is sometimes called the divergence, for obvious reasons. We can also raise the index on the derivative operator using the metric,

$$\partial^\mu = g^{\mu\nu}\partial_\nu = \begin{pmatrix} \frac{\partial}{\partial x^0} \\ -\frac{\partial}{\partial x^1} \\ -\frac{\partial}{\partial x^2} \\ -\frac{\partial}{\partial x^3} \end{pmatrix}. \tag{75}$$

Finally, we'll introduce a Lorentz tensor. Just like we defined a vector by its transformation law (and a scalar by the fact it does not transform), we define a tensor as something that transforms with two copies of the Lorentz transformation,

$$\bar{G}^{\mu\nu} = \Lambda^\mu{}_\alpha \Lambda^\nu{}_\beta G^{\alpha\beta}. \tag{76}$$

Notice that each index transforms like a vector. One of the most important examples of a tensor is the electromagnetic field strength,

$$F_{\mu\nu} = \partial_\mu A_\nu - \partial_\nu A_\mu \tag{77}$$

which is constructed from the four-potential $A_\mu = (\Phi, -\mathbf{A})$ where $\Phi$ and $\mathbf{A}$ are the scalar and vector potentials. If you evaluate each component of the above and compare to the definitions of the electric and magnetic fields,

$$\begin{aligned} \mathbf{E} &= -\nabla\Phi - \partial_t \mathbf{A}, \\ \mathbf{B} &= \nabla \times \mathbf{A}, \end{aligned} \tag{78}$$

you can easily show that

$$\begin{aligned} F_{01} &= E_x, & F_{02} &= E_y, & F_{03} &= E_z, \\ F_{32} &= B_x, & F_{13} &= B_y, & F_{21} &= B_z. \end{aligned} \tag{79}$$

In light of this, we can organize the components of the field strength into an array of numbers that is *not* a matrix,

$$F_{\mu\nu} = \begin{pmatrix} 0 & E_x & E_y & E_z \\ -E_x & 0 & -B_z & B_y \\ -E_y & B_z & 0 & -B_x \\ -E_z & -B_y & B_x & 0 \end{pmatrix}, \qquad F^{\mu\nu} = \begin{pmatrix} 0 & -E_x & -E_y & -E_z \\ E_x & 0 & -B_z & B_y \\ E_y & B_z & 0 & -B_x \\ E_z & -B_y & B_x & 0 \end{pmatrix}. \tag{80}$$

We can contract both indices of $F_{\mu\nu}$ with itself to get a Lorentz scalar,

$$F_{\mu\nu}F^{\mu\nu} = 2(\mathbf{E}^2 - \mathbf{B}^2). \tag{81}$$

We'll use this fact in section 9 to formulate electromagnetism in the Lagrangian formalism.

# 6  The Yukawa Potential

The semi-empirical mass formula that we discussed in section 2 is only sensible if the nuclear force is short-ranged. However, simply saying a dimensionful quantity like a distance is "short" isn't good enough, we need some other scale with which to compare it. In our case, the important length scale is the typical size of the nucleus, which is typically a few fm. So the range of the nuclear force should be shorter than the typical size of the nucleus, but it also can't be too short: if its shorter than the typical size of a nucleon (proton or neutron) then things stop making sense. We now have an upper and a lower bound for the range of the force, but the question still remains as to what the range of the force actually is, and more importantly *why* it has the range that it does.[9]

In 1935, Yukawa arrived at an insightful answer to this question: he posited that the nuclear force is mediated by the virtual exchange of massive bosons, and the mass of the boson sets the range of the force. This idea was nothing short of brilliant; at the time people did not simply invent particles out of thin air, so to do so was an act of genius. The idea is still relevant today, in fact if one were to observe a mysterious short ranged force not accounted for by the standard model the first thing any theorist would try is a new particle with a mass commensurate with the force's range. Of course, like many discoveries in particle physics, Yukawa's original model, while getting at a fundamental truth, was not quite correct in detail.

Now, we've said a few times that the mass of the particle sets the range of the force, but it may not be clear how this is so. Since we're interested in process that are both quantum mechanical and relativistic, the two constants $\hbar$ and $c$ are in the game (they've been hiding so far, since we set them equal to one). If we now have a mass $m$, we can use these constants to write down a length scale,

$$\frac{1}{R} \sim \frac{mc}{\hbar} \,, \tag{82}$$

where $R$ is taken to be the range of the interaction (after all, it's the only length scale we have). Returning to civilized units (those with $\hbar = 1$ and $c = 1$), this is simply written $R \sim m^{-1}$. We previously figured that the range of the interaction should be about a femtometer, so the mass of the mediating boson should be

$$m \sim \frac{1}{1 \text{ fm}} \sim 200 \text{ MeV} \,. \tag{83}$$

It turns out that the particle Yukawa was looking for is the pion, which actually comes in three kinds: the charged $\pi^+$ and $\pi^-$ which are antiparticles of one another, and the neutral $\pi^0$ which is its own antiparticle. The masses of these particles are

$$m_{\pi^\pm} \approx 139.6 \text{ MeV} \,, \quad m_{\pi^0} \approx 135 \text{ MeV} \,, \tag{84}$$

which are both reasonably close to the rough estimate of 200 MeV, so our simple reasoning about scales was fairly predictive. However, it is not only the pions that mediate the nuclear force; there are many other particles, generally called *mesons*, which act in this capacity. One should also note that these particles are not fundamental, they're made out of quarks and gluons.

We also said that the exchange of these particles is "virtual," which is worth briefly explaining. The notion of a virtual particle only really makes sense in the context of perturbation theory, which we'll learn much more about in section 9.5. As we'll see, the basic idea is that over the course of some physical process, the system can be in an "intermediate state" and

---

[9]In other words, we are looking for another characteristic scale in the problem that sets the range for the nuclear force

include particles that don't appear in the initial or final states. A virtual particle is a particle that exists only in such an intermediate state, and isn't actually observable: it just comes and goes as part of an interaction between other entities.

Although these mesons appear only as virtual particles in the nuclear force, they also exist as real, observable particles. However, the heavier mesons are not stable; they all decay into pions via the strong interaction. The pions themselves also decay via the weak interaction for the $\pi^\pm$ and via the electromagnetic interaction for the $\pi^0$, with lifetimes of $2.6 \times 10^{-8}$ s and $8.4 \times 10^{-17}$ s respectively. These might seem very short, but they are in fact much longer than the typical timescale in strong interactions involving hadrons–which is about $10^{-24}$ s; we will discuss hadrons a bit later in these notes. The timescales relevant to nuclear physics are often a bit longer than these typical hadronic scales, but are still very much shorter than the pion lifetime. The net result being that the pions "look" stable in hadronic and nuclear interactions.

The final aspect of Yukawa's idea that we need to explain is the notion of a particle mediating a force. To do so, let's go back to E&M: we have an electromagnetic field, which in the quantum picture can be thought of as a swarm of virtual photons (which are bosons) in the same state. Charged particles can interact with the one another via the electromagnetic field by exchanging these virtual photons with one another. This photon exchange gives rise to the familiar Coulomb potential between charged particles,

$$V(\mathbf{r}) = \frac{e^2}{4\pi r} \,. \tag{85}$$

The same picture holds for the nuclear force, with mesons playing the role of the photons. The key difference is that the mesons are massive whereas photons are massless. This means that the potential we get is not the Coulomb potential, but instead the *Yukawa potential*,

$$V(\mathbf{r}) = \frac{g^2 \, e^{-mr}}{r} \,, \tag{86}$$

where $g^2$ is the square of the coupling constant, analogous to the $e^2/4\pi$ in the Coulomb potential, and $m$ is the mass of the particle. This potential decays exponentially with $mr$, so the characteristic range of the interaction is indeed $m^{-1}$. It is often said the the nucleons are a "source" for the meson field. We can understand what this means by again appealing to E&M, where the sources of the electromagnetic potential $\Phi$ are charges and currents, $J^\mu$. These sources (charges and currents) can then interact with one another via the electromagnetic potential.

In E&M it is particularly useful to consider physics in the absence of sources, i.e. solutions to the Maxwell equations in vacuum. We know quite well that these solutions are electromagnetic waves which satisfy the wave equation,

$$(\partial_t^2 - \nabla^2)\phi = 0\,. \tag{87}$$

We've written this equation for some generic massless bosonic field $\phi$ rather than $\mathbf{E}$, $\mathbf{B}$, or $A_\mu$ to avoid the messy complications that arise in E&M due to gauge invariance, which will not be an issue for the present discussion. Notice that we can use the fancy relativistic language from the previous section to write $\partial_t^2 - \nabla^2 = \partial_\mu \partial^\mu$, and the wave equation as simply[10]

$$\partial_\mu \partial^\mu \phi = 0\,. \tag{88}$$

---

[10]The operator $\partial_\mu \partial^\mu$ is called the *D'Alembertian* operator and plays the role of the Laplacian in four space-time dimensions. Because mathematicians often write the Laplacian as a triangle $\triangle$ (since it is a three-dimensional derivative), it is common to see the D'Alembertian written as a box $\Box$ in the literature, in which case the wave equation is $\Box \phi = 0$. A less common (but aesthetically superior) notation is $\partial^2 \phi = 0$. To avoid confusion, in this course we'll always just write out $\partial_\mu \partial^\mu$.

We'd now like to figure out what the equation of motion is for the meson field in the absence of sources. The trick to do this quickly is to use quantum mechanical ideas: we know that as operators we can replace

$$E \to i\partial_t, \qquad \mathbf{p} \to -i\nabla. \tag{89}$$

In case you're not familiar with the first relationship, it's nothing more than the Schödinger equation, which says that

$$i\partial_t \psi = H\psi. \tag{90}$$

The Hamiltonian is the energy operator, so we can relate the operator on the left-hand side, $i\partial_t$, to the energy $E$. Next, we recall that we live in a relativistic world, so the energy and momentum of a massive particle are related by (remember $c = 1$)

$$E^2 = \mathbf{p}^2 + m^2, \tag{91}$$

which we can write as $\mathbf{p}^2 - E^2 + m^2 = 0$. Swapping out $E$ and $\mathbf{p}$ with (89) this becomes

$$(-i\nabla)^2 - (i\partial_t)^2 + m^2 = 0 \implies \partial_t^2 - \nabla^2 + m^2 = 0 \implies \partial_\mu\partial^\mu + m^2 = 0. \tag{92}$$

Of course, for this to make any sense we should act with this operator on a function. Doing so results in the *Klein-Gordon* equation,

$$(\partial_\mu\partial^\mu + m^2)\phi = 0. \tag{93}$$

You can check that the solutions to this equation are propagating plane waves,

$$\phi(\mathbf{x}, t) = A e^{i(\mathbf{k}\cdot\mathbf{x} - \omega t)}, \tag{94}$$

which satisfy the dispersion relation

$$\omega^2 = \mathbf{k}^2 + m^2. \tag{95}$$

In quantum mechanics $E = \hbar\omega$ and $\mathbf{p} = \hbar\mathbf{k}$, so this dispersion is equivalent to the relativistic energy-momentum relation (91). In light of this, you can think of the Klein-Gordon equation as the moral equivalent of the wave equation for massive particles.

This is all well and good, but we went down this rabbit hole to understand the Yukawa potential, and a potential is the energy for a *static* object. This means that the solutions we care about are time-independent, i.e. $\partial_t^2\phi = 0$. In this case the Klein-Gordon equation simplifies to

$$(-\nabla^2 + m^2)\phi = 0. \tag{96}$$

This equation has lots of solutions, but for the time being we'll only be interested in spherically symmetric solutions (that is, solutions that depend only on $r = |\mathbf{r}|$). It turns out that there are two solutions of this form,

$$\phi \sim \frac{e^{-mr}}{r}, \qquad \phi \sim \frac{e^{mr}}{r}. \tag{97}$$

The second option diverges as $r \to \infty$ which is unphysical, so we should throw it away. Don't worry about how we found these solutions, but feel free to plug them into (96) and check that they work.

If we now stick a source (particle 1) at position $\mathbf{r}_1$ the resultant meson field will be

$$\phi_1(\mathbf{r}) = g \frac{e^{-m|\mathbf{r}-\mathbf{r}_1|}}{|\mathbf{r}-\mathbf{r}_1|}, \tag{98}$$

where $g$ is the strength of the coupling to the source. If we add a second particle at position $\mathbf{r}_2$, the potential particle 2 feels due to particle 1 is

$$V(\mathbf{r}_1, \mathbf{r}_2) = -g^2 \frac{e^{-m|\mathbf{r}_1 - \mathbf{r}_2|}}{|\mathbf{r}_1 - \mathbf{r}_2|}, \tag{99}$$

which is precisely the Yukawa potential. The overall minus sign means the potential is attractive, and comes about for subtle reasons that we won't worry about in this course.

### Discovery of the Pion

Although Yukawa originally proposed mesons to explain the short range of the nuclear force in the 1930's, it's actual application is chiefly to the pions, which were discovered in 1947 in a cosmic ray collision. This was a remarkable discovery but also a source of confusion, considering another heavy(ish) particle was discovered a few years earlier in 1936 by Anderson and Neddermeyer in a separate cosmic ray experiment: the muon. The muon has a mass of 105 MeV, which was within the range expected for the pion, however its not a meson at all. Rather, it is a lepton (which is a kind of fermion which plays no role in nuclear interactions) that is more-or-less a heavier version of the electron. The discovery of the muon was completely unexpected, in that it wasn't needed to explain any previously observed phenomena, and is the subject of I.I. Rabi's famous quip of "who ordered that?"

It took a while to disentangle muons from pions, but once things were straightened out it was clear that the pions interacted very strongly with nuclei. The force cause by exchange of physical pions (the one pion exchange potential, or OPEP) was found to fall off like a Yukawa force, with a mass $m_\pi$. However, the coupling to nucleons is slightly convoluted, owing to the fact that the pions have negative parity and couple to the nucleon's spin in a rather intricate way. We'll discuss this issue later in the course in section 10.5.

# 7 The Dirac Equation

At this point we know how to do quantum mechanics, and we know how to do special relativity. The question now becomes how we can put the two together. It turns out that this is not so easy. Our story starts in 1928, when Dirac sought to combine relativity and quantum mechanics in such a way as to maintain the probabilistic interpretation of the single-particle wavefunction.

It turns out that his approach was misguided, and the Dirac equation as it was originally envisioned doesn't really make sense. However, when interpreted in the context of quantum field theory (QFT), the Dirac equation is the basis for the correct theory of a fundamental spin 1/2 particle. The Dirac equation also predicted the existence of anti-matter, and automagically gives us the correct $g$-factor for the electron and spin-orbit coupling in the presence of an electromagnetic field. Despite its rocky start, the Dirac equation was an extraordinarily profound accomplishment. It's also aesthetically quite pretty. Here it is:

$$(i\gamma^\mu \partial_\mu - m)\psi = 0 \,. \tag{100}$$

We'll spend the rest of this section learning where this equation comes from, and what it means.

## 7.1 All is Not Well with Relativistic Wave Equations

Let's remember our old friend the Schrödinger equation,

$$i\,\partial_t \psi = \left(-\frac{1}{2m}\nabla^2 + V(\mathbf{r})\right)\psi \,. \tag{101}$$

We define the probability density as the squared modulus of the wavefunction,

$$\rho = \psi^\star \psi \tag{102}$$

and the probability current as[11]

$$\mathbf{J} = -\frac{i}{2m}\left(\psi^\star \nabla \psi - (\nabla \psi^\star)\psi\right). \tag{103}$$

It's then trivial to show that any wavefunction that satisfies the Schrödinger equation also satisfies

$$\frac{\partial \rho}{\partial t} = -\nabla \cdot \mathbf{J}\,, \tag{104}$$

which is a *continuity equation*, and tells us that probability is locally conserved. If we consider the time rate of change of the the probability to find the particle within a region $\mathcal{V}$, we have

$$\frac{\partial P}{\partial t} = \int_{\mathcal{V}} \mathrm{d}^3 r\, \frac{\partial \rho}{\partial t} = -\int_{\mathcal{V}} \mathrm{d}^3 r\, \nabla \cdot \mathbf{J} = -\int_{\partial \mathcal{V}} \mathrm{d}\mathbf{a} \cdot \mathbf{J}\,, \tag{105}$$

where to get to the last equality we used the divergence theorem to turn the volume integral over $\mathcal{V}$ into a surface integral over its boundary $\partial \mathcal{V}$. This has a simple physical interpretation: the only change in the probability contained in a region is the outward probability flux through the surface enclosing it. Further, we can take the region $\mathcal{V}$ to be all of space, with the boundary at spatial infinity. Since any normalizable wavefunction must vanish as $\mathbf{r} \to \infty$, the current $\mathbf{J}$ must also go to zero at infinity. Thus,

$$\frac{\partial P_{\text{total}}}{\partial t} = 0\,, \tag{106}$$

---

[11]This kind of antisymmetrized derivative shows up a lot, and you will often see the shorthand $\psi^\star \overleftrightarrow{\nabla} \psi \equiv \psi^\star \nabla \psi - (\nabla \psi^\star)\psi$.

so the total probability to find the particle is globally conserved.

Now, let's try this for the Klein-Gordon equation, which we recall from section 6 is

$$(\partial_\mu \partial^\mu + m^2)\psi = 0. \tag{107}$$

It turns out this equation also has a conserved current, which we might be able to interpret as a probability current. Since the Klein-Gordon equation is in some sense relativistic, the conserved current is a four-vector,

$$J^\mu = \begin{pmatrix} \rho \\ J_x \\ J_y \\ J_z \end{pmatrix}. \tag{108}$$

A continuity equation is then written as

$$\partial_\mu J^\mu = \partial_t \rho + \partial_x J_x + \partial_y J_y + \partial_z J_z = \partial_t \rho + \nabla \cdot \mathbf{J} = 0. \tag{109}$$

In this notation, its easy to see that the current

$$J^\mu = \frac{i}{2}\big(\psi^\star \partial^\mu \psi - (\partial^\mu \psi^\star)\psi\big) \tag{110}$$

is conserved: we just differentiate

$$
\begin{aligned}
\partial_\mu J^\mu &= \frac{i}{2}\partial_\mu \psi^\star \partial_\mu \psi + \frac{i}{2}\psi^\star \partial_\mu \partial^\mu \psi - \frac{i}{2}\partial^\mu \psi^\star \partial_\mu \psi - \frac{i}{2}(\partial_\mu \partial^\mu \psi^\star)\psi \\
&= \frac{i}{2}\Big[\psi^\star \underbrace{\partial^\mu \partial_\mu \psi}_{-m^2\psi} - \underbrace{(\partial_\mu \partial^\mu \psi^\star)}_{-m^2\psi^\star}\psi\Big] \\
&= \frac{i}{2}\big[-m^2\psi^\star\psi + m^2\psi^\star\psi\big] \\
&= 0.
\end{aligned}
\tag{111}
$$

In the second line we used the fact that $\psi$ obeys the Klein-Gordon equation, $\partial_\mu \partial^\mu \psi = -m^2\psi$ and its complex conjugate $\partial_\mu \partial^\mu \psi^\star = -m^2\psi^\star$. So, we've found a conserved current! Given the similarity to the probability current in the Schrödinger case, there is hope this could represent a probability current. However, these hopes are dashed by considering the candidate probability density,

$$\rho = J^0 = \frac{i}{2}\big(\psi^\star \partial_t \psi - (\partial_t \psi^\star)\psi\big) \neq \psi^\star\psi. \tag{112}$$

Problematically, $\rho$ is not positive definite, so we could have negative probabilities—which clearly violates the notion of probability. Moreover, there are real solutions to the Klein-Gordan equation, in which case $\rho = 0$ everywhere, always. These two issues kill any chance of $\rho$ functioning as a single particle probability density, so $\psi$ can't be interpreted as a single particle wavefunction á la Schrödinger. In 1934 Pauli and Weisskopf pointed out that this isn't actually a problem if one thinks a little differently, but back in 1928 this perspective was unknown and Dirac was determined to write down a relativistic wave equation with a single-particle probabilistic interpretation. Now, we'll consider how he did it.

## 7.2 Motivating the Dirac Equation

In the most dramatic of fashions, Dirac supposedly had the epiphany of how to fix the Klein-Gordon equation while staring into a fire. He realized the problem was that the Klein-Gordon equation was second order in time, and to admit a probabilistic interpretation he needed an

equation first order in time. Then to have any hope of Lorentz invariance the equation must be first order in space as well, so as not to treat space and time on different footing.

The idea is to essentially take the square root of the Klein-Gordan equation: all we need to do is factorize $\partial_\mu \partial^\mu$ into first order pieces, and things should work out. Lorentz invariance requires that the first order equation must have the differential operator appear contracted with a four-vector, i.e. in a term like $\gamma^\mu \partial_\mu$.

Now, let's suppose that we can find a four-vector $\gamma^\mu$ such that

$$(\gamma^\mu \partial_\mu)(\gamma^\nu \partial_\nu) = \partial_\mu \partial^\mu . \tag{113}$$

If we can, then we can simply factor the Klein-Gordon equation as

$$(-\partial_\mu \partial^\mu - m^2)\psi = (i\gamma_\nu \partial^\nu + m)(i\gamma^\mu \partial_\mu - m)\psi = 0 . \tag{114}$$

Then any solution to

$$(i\gamma^\mu \partial_\mu - m)\psi = 0 , \tag{115}$$

which we've already seen is the Dirac equation, will automatically also be a solution to the Klein-Gordon equation, and ensure we have the desirable relativistic dispersion $E^2 = p^2 + m^2$. Now, the problem has been reduced to just finding the right $\gamma^\mu$.

Notice that we can trivially rewrite $\gamma^\mu \gamma^\nu = \frac{1}{2}(\gamma^\mu \gamma^\nu + \gamma^\nu \gamma^\mu)$.[12] Also remember that we're looking for $\gamma^\mu$ such that $\gamma^\mu \partial_\mu \gamma^\nu \partial_\nu = \partial_\mu \partial^\mu$, where we can write $\partial_\mu \partial^\mu = g^{\mu\nu} \partial_\mu \partial_\nu$. Putting these pieces together, we want

$$\frac{1}{2}(\gamma^\mu \gamma^\nu + \gamma^\nu \gamma^\mu)\partial_\mu \partial_\nu = g^{\mu\nu} \partial_\mu \partial_\nu , \tag{116}$$

or,

$$\gamma^\mu \gamma^\nu + \gamma^\nu \gamma^\mu = 2g^{\mu\nu} . \tag{117}$$

Unfortunately, as you can convince yourself, there is no ordinary four-vector that can satisfy this equation. However, all hope is not lost: what if we let $\gamma^\mu$ be a four-vector of matrices? This idea isn't as outlandish as you might think. In fact, you're already very familiar with a vector of matrices, namely the Pauli matrices which are often packaged into a vector,

$$\boldsymbol{\sigma} = \begin{pmatrix} \sigma_x \\ \sigma_y \\ \sigma_z \end{pmatrix} , \tag{118}$$

and happen to satisfy $\sigma_i \sigma_j + \sigma_j \sigma_i = 2\delta_{ij}$, which is reminiscent of (117) above. It turns out that there do exist sets of matrices that satisfy our desired condition, in fact there are an infinite number of them! Any set of matrices $\gamma^\mu$ which satisfy (117) are said to form a *Clifford algebra*, and we can choose any convenient set of them we wish. You can convince yourself that the smallest matrices that can form such an algebra are 4×4. In this class, we'll use the convention

$$\gamma^0 = \begin{pmatrix} 1 & 0 & 0 & 0 \\ 0 & 1 & 0 & 0 \\ 0 & 0 & -1 & 0 \\ 0 & 0 & 0 & -1 \end{pmatrix} , \qquad \gamma^1 = \begin{pmatrix} 0 & 0 & 0 & 1 \\ 0 & 0 & 1 & 0 \\ 0 & -1 & 0 & 0 \\ -1 & 0 & 0 & 0 \end{pmatrix} ,$$

$$\gamma^2 = \begin{pmatrix} 0 & 0 & 0 & -i \\ 0 & 0 & i & 0 \\ 0 & i & 0 & 0 \\ -i & 0 & 0 & 0 \end{pmatrix} , \qquad \gamma^3 = \begin{pmatrix} 0 & 0 & 1 & 0 \\ 0 & 0 & 0 & -1 \\ -1 & 0 & 0 & 0 \\ 0 & 1 & 0 & 0 \end{pmatrix} . \tag{119}$$

---

[12]Note that is only true if $\gamma^\mu$ takes scalar values. If (as we will see in a moment), $\gamma^\mu$ is matrix-valued this is generally *not* true.

In this notation, these are pretty hard to remember. However, recalling the definition of the Pauli matrices, (labeling $x \to 1, y \to 2, z \to 3$)

$$\sigma^1 = \begin{pmatrix} 0 & 1 \\ 1 & 0 \end{pmatrix}, \quad \sigma^2 = \begin{pmatrix} 0 & -i \\ i & 0 \end{pmatrix}, \quad \sigma^3 = \begin{pmatrix} 1 & 0 \\ 0 & -1 \end{pmatrix}, \tag{120}$$

we can write the $\gamma$ matrices as 2×2 matrices of 2×2 matrices. That is,

$$\gamma^0 = \begin{pmatrix} \mathbb{1} & 0 \\ 0 & -\mathbb{1} \end{pmatrix}, \quad \gamma^1 = \begin{pmatrix} 0 & \sigma^1 \\ -\sigma^1 & 0 \end{pmatrix}, \quad \gamma^2 = \begin{pmatrix} 0 & \sigma^2 \\ -\sigma^2 & 0 \end{pmatrix}, \quad \gamma^3 = \begin{pmatrix} 0 & \sigma^3 \\ -\sigma^3 & 0 \end{pmatrix}. \tag{121}$$

Or, in even slicker notation,

$$\gamma^0 = \begin{pmatrix} \mathbb{1} & 0 \\ 0 & -\mathbb{1} \end{pmatrix}, \quad \gamma^i = \begin{pmatrix} 0 & \sigma^i \\ -\sigma^i & 0 \end{pmatrix}, \tag{122}$$

where Roman indices always run over just the spatial components, $i = 1, 2, 3$. In this form its straightforward to check that this choice of matrices satisfies (117):

$$(\gamma^0)^2 = \mathbb{1}, \quad (\gamma^i)^2 = -\mathbb{1}, \quad \gamma^\mu \gamma^\nu + \gamma^\nu \gamma^\mu = 0, \quad \text{for } \mu \neq \nu. \tag{123}$$

So, we have found the equation we were looking for! But what does it mean?

## 7.3 Solutions to the Dirac Equation

Before going any further, we're going to introduce the *Feynman slash notation*, where we denote contraction with the $\gamma$ matrices by a slash through the contracted four-vector, i.e. $\slashed{\partial} \equiv \gamma^\mu \partial_\mu$. The $\gamma$ matrices show up almost everywhere, so this shorthand will save us a lot of writing. The Dirac equation is then written

$$(i\slashed{\partial} - m)\psi = 0. \tag{124}$$

Using our representation of the $\gamma$ matrices from the previous section, we see this is a compact way of writing the matrix equation

$$\left[ i \begin{pmatrix} \mathbb{1} & 0 \\ 0 & -\mathbb{1} \end{pmatrix} \partial_t + i \begin{pmatrix} 0 & \sigma^x \\ -\sigma^x & 0 \end{pmatrix} \partial_x + i \begin{pmatrix} 0 & \sigma^y \\ -\sigma^y & 0 \end{pmatrix} \partial_y + i \begin{pmatrix} 0 & \sigma^z \\ -\sigma^z & 0 \end{pmatrix} \partial_z - m \begin{pmatrix} \mathbb{1} & 0 \\ 0 & \mathbb{1} \end{pmatrix} \right] \psi = 0. \tag{125}$$

For this equation to make sense, we see $\psi$ must be a four component object, which we call a *Dirac spinor*. Since we've expressed the $\gamma$ matrices in terms of 2×2 matrices, it is natural to also split this four-component spinor into to its upper and lower components $U$ and $L$, each of which is itself a two-component spinor,

$$\psi = \begin{pmatrix} U \\ L \end{pmatrix}. \tag{126}$$

Carrying out the matrix multiplication, we get two equations in terms of the upper and lower components,

$$\begin{aligned} i\partial_t U + i\boldsymbol{\sigma} \cdot \nabla L - mU &= 0, \\ -i\partial_t L - i\boldsymbol{\sigma} \cdot \nabla U - mL &= 0. \end{aligned} \tag{127}$$

The two-component spinors $U$ and $L$ can each be written in terms of the spinors

$$\chi_\uparrow = \begin{pmatrix} 1 \\ 0 \end{pmatrix}, \qquad \chi_\downarrow = \begin{pmatrix} 0 \\ 1 \end{pmatrix}, \tag{128}$$

which form a basis for two-component spinors. We can use these, and a plane wave with dispersion (remember $\hbar = 1$),

$$\psi \sim e^{i(\mathbf{k}\cdot\mathbf{x}-\omega t)}, \qquad \mathbf{p} = \mathbf{k}, \qquad \omega = E = \sqrt{\mathbf{p}^2 + m^2}, \tag{129}$$

to construct solutions to the Dirac equation:

$$\psi_\uparrow^+(\mathbf{x},t) = \sqrt{\frac{E+m}{2m}} \begin{pmatrix} \chi_\uparrow \\ \frac{\sigma\cdot\mathbf{p}}{E+m}\chi_\uparrow \end{pmatrix} e^{i(\mathbf{p}\cdot\mathbf{x}-Et)},$$

$$\tag{130}$$

$$\psi_\downarrow^+(\mathbf{x},t) = \sqrt{\frac{E+m}{2m}} \begin{pmatrix} \chi_\downarrow \\ \frac{\sigma\cdot\mathbf{p}}{E+m}\chi_\downarrow \end{pmatrix} e^{i(\mathbf{p}\cdot\mathbf{x}-Et)},$$

where again each component of $\psi$ is itself a two-component spinor, and the prefactor out front is just a convenient normalization. You're invited to check that these are in fact solutions to the Dirac equation by plugging them in. Notice that for a given momentum $\mathbf{p}$ we have two linearly independent solutions, suggestively named $\psi_\uparrow^+$ and $\psi_\downarrow^+$. As the subscripts suggest, these are in fact the two states of a spin 1/2 particle. Also note that in the limit $p \ll m$, the lower components go to zero and the upper components give us back the familiar two component spinors of non-relativistic quantum mechanics (with the components corresponding to spin up and spin down).

We can take linear combinations of these two solutions to construct more general spinors, or even combine solutions of different momenta into wavepackets. However, we musn't forget that the Dirac equation is a set of four coupled differential equations, and thus we expect *four* solutions for a given momentum. Where are the other two?

Luckily, the two remaining solutions aren't hard to find, but they do end up posing a whole new set of issues of interpretation. They are given by

$$\psi_\uparrow^-(\mathbf{x},t) = \sqrt{\frac{|E|+m}{2m}} \begin{pmatrix} \frac{\sigma\cdot\mathbf{p}}{E+m}\chi_\uparrow \\ \chi_\uparrow \end{pmatrix} e^{i(\mathbf{p}\cdot\mathbf{x}-Et)},$$

$$\tag{131}$$

$$\psi_\downarrow^-(\mathbf{x},t) = \sqrt{\frac{|E|+m}{2m}} \begin{pmatrix} \frac{\sigma\cdot\mathbf{p}}{E+m}\chi_\downarrow \\ \chi_\downarrow \end{pmatrix} e^{i(\mathbf{p}\cdot\mathbf{x}-Et)},$$

where

$$\mathbf{p} = \mathbf{k}, \qquad E = -\sqrt{\mathbf{p}^2 + m^2}, \tag{132}$$

which means these solutions have negative energy. What do negative energies even mean in this context? The situation seems to be very bad!

To convince ourselves that this isn't as bad as it looks, let's first remember that the Dirac field represents a fermion, since we saw above that it describes a spin 1/2 particle. Crucially, this means that it satisfies the Pauli principle and we may have at most one particle per energy level. The state with the lowest energy is called the ground state, and in the case of fundamental physics has another special name: the vacuum!

So, if negative energy states exist, the lowest-energy physical state (the vacuum) will have all of them filled.[13] This picture of the vacuum being comprised of a sea of filled negative

---

[13]This makes the vacuum appear to have a negative energy, but this problem can be easily fixed by closing our eyes and repeating the mantra "physics only measures energy differences." This is actually a pretty effective solution, as long as you don't ask about gravity!

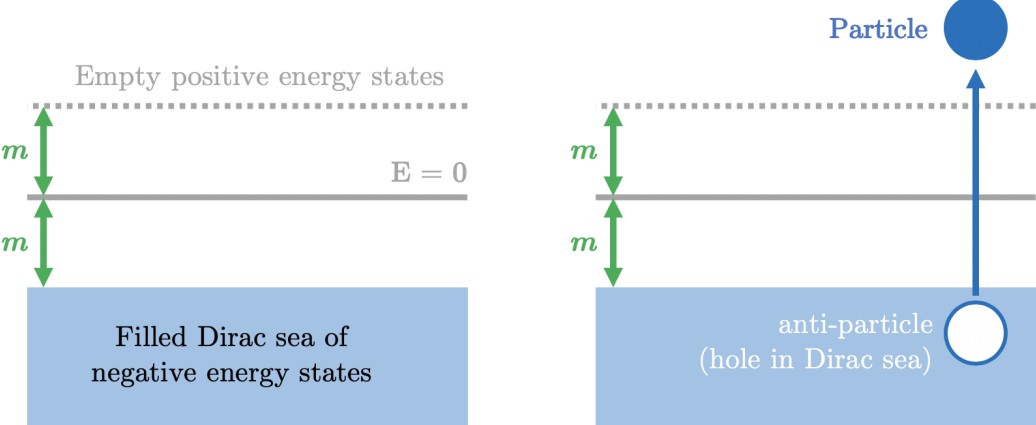

Figure 6: Left: the vacuum, with positive energy states empty and negative energy states fully occupied; Right: creation of a particle-antiparticle pair from the vacuum

energy states is called the *Dirac sea* (illustrated in Fig. 6), and although this is certainly not in line with the modern understanding of physics via quantum field theory,[14] it nonetheless enabled Dirac to predict the existence of anti-matter by the following observation: suppose we remove a state of energy $-E$ and momentum $\mathbf{p}$ from the Dirac sea. This increases the energy of the vacuum by $E$, and changes its momentum by $-\mathbf{p}$. So, this "hole" in the Dirac sea looks just like a particle with energy $+E$ and momentum $-\mathbf{p}$! So, we interpret it as an anti-particle. Finally, note that if some interaction were to knock a filled negative energy state into a previously unoccupied positive energy level, we would get a particle where the positive energy state was filled, and a hole (anti-particle) from where the negative energy state was vacated. Thus, particle-hole pairs can be pulled from the vacuum!

## 7.4 Properties of the Dirac Equation

Just like its convenient to define the Hermitian conjugate $\psi^\dagger$ when working with complex vectors, it is useful to define the *Dirac conjugate*,

$$\bar{\psi} = \psi^\dagger \gamma^0 \,. \tag{133}$$

For reasons we won't go into in these notes, it turns out that products like $\bar{\psi}\psi$ are Lorentz invariant, while those like $\psi^\dagger\psi$ are not. Basically, it all comes down to different representations of the Lorentz group and their properties, and you are mercifully being spared from having to learn about the details. It's not too hard to show that any solution of the Dirac equation will also satisfy

$$\bar{\psi}(-i\overleftarrow{\slashed{\partial}} - m) = 0 \,, \tag{134}$$

where the arrow on top of the derivative indicates that it acts to the left. This might seem odd at first, but it saves us from having to write annoying expressions like $-i\partial_\mu\bar{\psi}\gamma^\mu$, which is what the first term above represents more compactly.

---

[14]That is, the notion that the universe contains a sea of occupied negative energy states is completely archaic. Nonetheless, we have discussed this picture since it is physically straightforward, historically important, and represents a fruitful analogy with condensed matter physics, and the notion of a Fermi sea in many-electron systems. This interpretation is not essential to what follows.

**Proof**

We'll get this fact for free in the next section when we derive the Dirac equation from a Lagrangian, but to get some practice working with the $\gamma$ matrices we'll also prove it here. We'll start by taking the Hermitian conjugate of the Dirac equation,

$$0 = \left( (i\gamma^\mu \partial_\mu - m)\psi \right)^\dagger = \psi^\dagger (-i\gamma_\mu^\dagger \overleftarrow{\partial}^\mu - m) \tag{135}$$

Since the left-hand side is zero, we can multiply the equation by anything we like. Let's multiply by $\gamma^0$ on the right,

$$\psi^\dagger (-i\gamma_\mu^\dagger \overleftarrow{\partial}^\mu - m)\gamma^0 = 0. \tag{136}$$

We may now use the following properties of the $\gamma$ matrices,

$$(\gamma^0)^\dagger = \gamma^0, \qquad (\gamma^i)^\dagger = -\gamma^i, \tag{137}$$

to trivially write $\gamma_\mu^\dagger \gamma^0 = \gamma^0\gamma^0 = \gamma^0\gamma_\mu$ for $\mu = 0$. If $\mu \neq 0$, the same holds: $\gamma_i^\dagger \gamma^0 = -\gamma^i\gamma^0 = \gamma^0\gamma^i$, where we've used the defining property (117). Thus, $\gamma_\mu^\dagger \gamma^0 = \gamma^0\gamma_\mu^\dagger$, so we can bring the $\gamma^0$ on the right of (136) next to the $\psi^\dagger$ and have

$$0 = \psi^\dagger \gamma^0 (-i\overleftarrow{\slashed{\partial}} - m) = \bar{\psi}(-i\overleftarrow{\slashed{\partial}} - m), \tag{138}$$

as advertised.

Coming back to Dirac's original goal of a relativistic wave equation with a sensible probabilistic interpretation, we do indeed have a conserved current,

$$J^\mu = \bar{\psi}\gamma^\mu\psi. \tag{139}$$

Provided $\psi$ satisfies the Dirac equation, we can show this current is conserved (i.e. $\partial_\mu J^\mu = 0$) by differentiating

$$\begin{aligned}
\partial_\mu J^\mu &= \partial_\mu(\bar{\psi}\gamma^\mu\psi) \\
&= \bar{\psi}\overleftarrow{\slashed{\partial}}\psi + \bar{\psi}\slashed{\partial}\psi \\
&= -i\Big( \underbrace{\bar{\psi}(i\overleftarrow{\slashed{\partial}})}_{-m\bar{\psi}}\psi + \bar{\psi}\underbrace{(i\slashed{\partial})\psi}_{m\psi} \Big) \\
&= -i\big( -m\bar{\psi}\psi + m\bar{\psi}\psi \big) \\
&= 0.
\end{aligned} \tag{140}$$

The probability density is the zeroth component of the current,

$$\rho = J^0 = \bar{\psi}\gamma^0\psi = \psi^\dagger \underbrace{\gamma^0\gamma^0}_{\mathbb{1}}\psi = \psi^\dagger\psi, \tag{141}$$

which is exactly what Dirac wanted! We'll see later that although we don't interpret this as the single-particle probability function that Dirac intended it to be, the conserved charge associated with this density,

$$Q = \int d^3r\, J^0 = \int d^3r\, \psi^\dagger\psi, \tag{142}$$

is a statement of the conservation of fermion number (electrons minus positrons).

Finally, we can introduce a cousin of the $\gamma$ matrices,

$$\gamma^5 = i\gamma^0\gamma^1\gamma^2\gamma^3 = \begin{pmatrix} 0 & \mathbb{1} \\ \mathbb{1} & 0 \end{pmatrix}, \tag{143}$$

which is a Lorentz scalar (although it has negative parity, which will be discussed later in the course). You can show that $\gamma^\mu\gamma^5 = -\gamma^5\gamma^\mu$ for all of the $\gamma$ matrices. Using this new object, we can construct the *axial current*

$$J_5^\mu = \bar{\psi}\gamma^\mu\gamma^5\psi. \tag{144}$$

We can differentiate it to see if it is conserved, and find

$$\begin{aligned}
\partial_\mu J_5^\mu &= \partial_\mu\left(\bar{\psi}\gamma^\mu\gamma^5\psi\right) \\
&= \bar{\psi}\overleftarrow{\partial}\gamma^5\psi + \bar{\psi}\underbrace{\gamma^\mu\gamma^5}_{-\gamma^5\gamma^\mu}\partial_\mu\psi \\
&= \underbrace{\bar{\psi}\overleftarrow{\partial}}_{im\psi}\gamma^5\psi - \bar{\psi}\gamma^5\underbrace{\partial\psi}_{-im\psi} \\
&= im\left(\bar{\psi}\gamma^5\psi + \bar{\psi}\gamma^5\psi\right) \\
&= 2im\bar{\psi}\gamma^5\psi.
\end{aligned} \tag{145}$$

So $\partial_\mu J_5^\mu \neq 0$, and thus in general axial current is not conserved. However, suppose we have a massless particle. If $m = 0$ then the RHS above will vanish and the axial current *will* be conserved! One could be skeptical as to why this matters, since there aren't actually any massless spin 1/2 particles in nature to which this relation would apply. However, there are *approximately* massless spin 1/2 particles, i.e. particles for which the mass is much smaller than the typical momentum of the system.

For example, in Quantum Chromodynamics (QCD), the up and down quarks have masses $m_u \approx 2.3$ MeV and $m_d \approx 2.8$ MeV, which are considerably smaller than typical hadronic mass scales of $\approx 1$ GeV. So, in QCD the axial current is *almost* conserved. In practice, one can treat it as being exactly conserved, and then apply perturbative corrections to reflect that it actually isn't.

### 7.5 The Electron Magnetic Moment

One of the greatest successes of the Dirac equation is that if we couple it to an electromagnetic field in the standard way by replacing $i\partial_\mu \mapsto i\partial_\mu - qA_\mu$ (see box), we automatically find the electron $g$-factor to be 2. As a reminder, the $g$-factor appears in the definition of the magnetic moment,

$$\boldsymbol{\mu} = g\left(\frac{q}{2m}\right)\mathbf{S}, \tag{146}$$

where $q$ is the charge of the particle, $m$ is its mass, and $\mathbf{S}$ is the spin. The algebra required to show this gets pretty messy, but given the historical importance of this calculation, we would be remiss to not include it. In case you'd rather skip the details, the important implication is that according to the Dirac equation, a truly structureless "point particle" with spin 1/2 should have $g = 2$. It turns out a real electron has

$$g = 2.002319304\ldots, \tag{147}$$

which is *almost* in exact agreement with theory. But why isn't it exact? The reason is that the electron isn't truly a point particle; in fact, within QED we can show that the electron

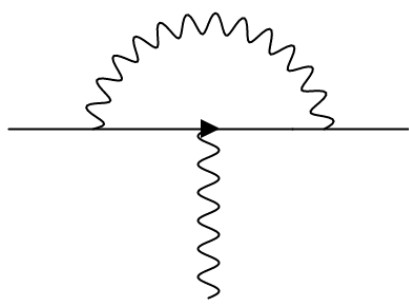

Figure 7: This is our first example of a Feynman diagram, which we will have much more to say about in section 9. The solid line represents the propagation of an electron, and the squiggly line on the bottom represents the externally applied magnetic field we use to probe the system. The arcing squiggly line is a spontaneously emitted and reabsorbed virtual photon.

has structure due to processes where the electron spontaneously emits and absorbs a virtual photon, as shown in Fig. 7. This coupling is of order $\alpha$, so its impact on the $g$-factor is small. On the other hand, a proton, which is also spin 1/2, has a $g$-factor of $g = 5.585$, which is very much not equal to 2. This was one of the earliest pieces of evidence that the proton has internal structure.

## Minimal Coupling

The fact that the replacement $i\partial_\mu \mapsto i\partial_\mu - qA_\mu$ will couple a particle to an electromagnetic field is often taken to be common knowledge, but is rarely ever actually explained. We'll start at the beginning.

As you may know, the classical Lagrangian for a charged particle in an electromagnetic field is

$$L = \frac{1}{2}m\dot{\mathbf{x}}^2 - q\Phi + q\dot{\mathbf{x}}\cdot\mathbf{A}, \tag{148}$$

where $\Phi$ and $\mathbf{A}$ are the scalar and vector potentials. To see that this is the correct Lagrangian we can calculate the Euler-Langrange equations and confirm that the equation of motion this will give us is the Lorentz force law. If you need a refresher on Lagrangian mechanics, see the review in section 9.1.

Things are easier if we consider components $x_i$ instead of vectors, and the Einstein summation convention will remain in effect. The Langrangian is then written

$$L = \frac{1}{2}m\dot{x}_j\dot{x}_j - q\Phi + q\dot{x}_jA_j. \tag{149}$$

We'll need the derivatives

$$\frac{\partial L}{\partial x_i} = -q\partial_i\Phi + \partial_i(\dot{x}_jA_j) = -q\partial_i\Phi + \dot{x}_j\partial_iA_j, \tag{150}$$

$$\frac{\partial L}{\partial \dot{x}_i} = m\dot{x}_i + qA_i, \tag{151}$$

$$\frac{\mathrm{d}}{\mathrm{d}t}\frac{\partial L}{\partial \dot{x}_i} = m\ddot{x}_i + q\frac{\mathrm{d}A_i}{\mathrm{d}t} = m\ddot{x}_i + q\left(\dot{A}_i + \frac{\partial A_i}{\partial x_j}\frac{\partial x_j}{\partial t}\right) = m\ddot{x}_i + q\dot{A}_i + q\dot{x}_j\partial_jA_i, \tag{152}$$

to write down the Euler-Lagrange equations:

$$0 = \frac{d}{dt}\frac{\partial L}{\partial \dot{x}_i} - \frac{\partial L}{\partial x_i},$$
$$m\ddot{x}_i = q(-\partial_i \Phi - \dot{A}_i) + q\dot{x}_j(\partial_i A_j - \partial_j A_i). \tag{153}$$

Using the definition of the electric field, $E_i = -\partial_i \Phi - \dot{A}_i$, we see the first term is the $i^{th}$ component of $\mathbf{F} = q\mathbf{E}$. The second term will turn out to be the $i^{th}$ component of $q\mathbf{v} \times \mathbf{B}$. You can show this by using $\mathbf{B} = \nabla \times \mathbf{A}$, which in terms of components is written $B_i = \varepsilon_{ijk}\partial_j A_k$ and the identity $\varepsilon_{ijk}\varepsilon_{ilm} = \delta_{jl}\delta_{km} - \delta_{jm}\delta_{lk}$. In any case, if you work through the algebra you'll find the Euler-Lagrange equations give us

$$m\ddot{\mathbf{x}} = q\mathbf{E} + q\mathbf{v} \times \mathbf{B}, \tag{154}$$

so the Lagrangian we started with does indeed describe a charged particle in an electromagnetic field. Going back to vector notation, the canonical momentum is

$$\mathbf{p} = \frac{\partial L}{\partial \dot{\mathbf{x}}} = m\dot{\mathbf{x}} + q\mathbf{A}. \tag{155}$$

We can then construct the Hamiltonian via the Legendre transformation,

$$H = \mathbf{p} \cdot \dot{\mathbf{x}} - L = \frac{1}{2m}(\mathbf{p} - q\mathbf{A})^2 + q\Phi. \tag{156}$$

So we see the effect of the field is to shift the canonical momentum $\mathbf{p} \mapsto \mathbf{p} - q\mathbf{A}$. Now, to go from classical to quantum mechanics we must replace the $c$-number momentum with the momentum operator, which has the position space representation $\hat{\mathbf{p}} = -i\nabla$. We then see that in the presence of an electromagnetic field, this operator gets shifted $\hat{\mathbf{p}} = -i\nabla \mapsto -i\nabla - q\mathbf{A}$, or generalizing to the relativistic case,

$$\partial_\mu \mapsto \partial_\mu + iqA_\mu. \tag{157}$$

This procedure is called *minimal coupling*, and is often implemented by introducing the *gauge covariant derivative*, $\mathcal{D}_\mu \equiv \partial_\mu + iqA_\mu$. We'll see much more of this when we discuss gauge theories in section 11.

To find the magnetic moment of the electron, we will need to take the non-relativistic limit of the Dirac equation. As a warmup, we'll illustrate the process by first considering a relativistic boson obeying the Klein-Gordan equation,

$$(\partial_\mu \partial^\mu + m^2)\phi = 0. \tag{158}$$

In the spirit of this section, we will interpret this equation (incorrectly) as a relativistic single-particle wave equation. We know that the solutions to this equation time evolve like $\phi \sim e^{-i\omega t}$, where $\omega = E = \sqrt{p^2 + m^2}$ is the energy of the state. In the non-relativistic limit, $p^2 \ll m^2$, so $E \sim m$ plus small corrections. This motivates us to rewrite $\phi$ as

$$\phi(\mathbf{x}, t) = e^{-imt}\varphi(\mathbf{x}, t), \tag{159}$$

where $\varphi$ is a field whose time dependence is much slower than $e^{-imt}$, and will turn out to be the non-relativistic wavefunction. The time derivative terms in the Klein-Gordan equation are

then

$$
\begin{aligned}
\partial_t^2 \phi &= \partial_t^2\big(\mathrm{e}^{-imt}\varphi\big) \\
&= \partial_t\big(-im\varphi + \dot{\varphi}\big)\mathrm{e}^{-imt} \\
&= \big(-im\dot{\varphi} + \ddot{\varphi}\big)\mathrm{e}^{-imt} - im\big(-im\varphi + \dot{\varphi}\big)\mathrm{e}^{-imt} \\
&= \big(-m^2\varphi - 2im\dot{\varphi} + \ddot{\varphi}\big)\mathrm{e}^{-imt}\,.
\end{aligned}
\tag{160}
$$

Since the time-dependence of $\varphi$ is much slower than $m$, $\ddot{\varphi} \ll m\dot{\varphi}$, and thus we can drop the second time derivative above, leaving us with $\partial_t^2 \phi \approx (-m^2\varphi - 2im\dot{\varphi})\mathrm{e}^{-imt}$. Plugging this into the Klein-Gordan equation, we have

$$
\begin{aligned}
\big(\partial_t^2 - \nabla^2 + m^2\big)\phi &= 0\,, \\
\big(-m^2\varphi - 2im\dot{\varphi} - \nabla^2\varphi + m^2\varphi\big)\mathrm{e}^{-imt} &= 0\,, \\
-2im\dot{\varphi} - \nabla^2\varphi &= 0\,, \\
\implies i\partial_t\varphi &= -\frac{1}{2m}\nabla^2\varphi\,,
\end{aligned}
\tag{161}
$$

which is the Schrödinger equation for a free particle. Now, taking the non-relativistic limit of the Dirac equation is a little more involved, in that we have to deal with the spinorial nature of the equation. Recall from the previous subsections that we can write the four-component Dirac spinor $\psi$ in terms of its positive and negative energy solutions,

$$
\psi = \begin{pmatrix} \psi_+ \\ \psi_- \end{pmatrix},
\tag{162}
$$

each of which are themselves two-component spinors. Recall also that $-i\nabla = \mathbf{p}$, and thus we can write

$$
i\gamma^\mu\partial_\mu = i\gamma^0\partial_0 + i\gamma^j\partial_j = i\gamma^0\partial_0 - \gamma^j p_j\,,
\tag{163}
$$

and the Dirac equations becomes $(i\gamma^0\partial_t - \gamma^j p_j - m)\psi = 0$, which in matrix form says

$$
\left[ i\begin{pmatrix} \mathbb{1} & 0 \\ 0 & -\mathbb{1} \end{pmatrix}\partial_t - \begin{pmatrix} 0 & \boldsymbol{\sigma} \\ -\boldsymbol{\sigma} & 0 \end{pmatrix}\cdot\mathbf{p} - m\begin{pmatrix} \mathbb{1} & 0 \\ 0 & \mathbb{1} \end{pmatrix} \right]\begin{pmatrix} \psi_+ \\ \psi_- \end{pmatrix} = 0\,.
\tag{164}
$$

This gives us two equations in terms of the two-component spinors,

$$
\begin{aligned}
(i\partial_t - m)\psi_+ - \boldsymbol{\sigma}\cdot\mathbf{p}\,\psi_- &= 0\,, \\
(-i\partial_t - m)\psi_- + \boldsymbol{\sigma}\cdot\mathbf{p}\,\psi_+ &= 0\,.
\end{aligned}
\tag{165}
$$

In the non-relativistic limit, we are of course interested in the positive energy solutions rather than the negative energy ones. To get rid of the dependence on $\psi_-$, recall that its time dependence is like $\mathrm{e}^{-i(-E)t} = \mathrm{e}^{iEt}$. Again, in the non-relativistic limit $E \sim m$, and we can replace the time derivative in the second equation by $-i\partial_t\psi_- \approx -m\psi_-$. The second equation is then just an algebraic equation that can be used to write $\psi_-$ in terms of $\psi_+$,

$$
\psi_- = \frac{\boldsymbol{\sigma}\cdot\mathbf{p}}{2m}\,\psi_+\,.
\tag{166}
$$

Putting (166) into the first line of (165), we get a closed equation for $\psi_+$,

$$
(i\partial_t - m)\psi_+ - \frac{(\boldsymbol{\sigma}\cdot\mathbf{p})^2}{2m}\psi_+ = 0\,.
\tag{167}
$$

Now, we play the same game as with the bosonic field, writing $\psi_+ = e^{-imt}\Psi$, were $\Psi$ is the slowly-time-dependent non-relativistic wavefunction. In terms of this wavefunction, the equation above reads

$$i\partial_t\left(e^{-imt}\Psi\right) - me^{-imt}\Psi - \frac{(\boldsymbol{\sigma}\cdot\mathbf{p})^2}{2m}e^{-imt}\Psi = 0\,,$$

$$e^{-imt}\left(m\Psi + i\partial_t\Psi - m\Psi - \frac{(\boldsymbol{\sigma}\cdot\mathbf{p})^2}{2m}\Psi\right) = 0\,, \tag{168}$$

$$\implies i\partial_t\Psi = \frac{(\boldsymbol{\sigma}\cdot\mathbf{p})^2}{2m}\Psi\,.$$

Again, we arrive at a Schrödinger equation, with the non-relativistic Dirac Hamiltonian,

$$H = \frac{(\boldsymbol{\sigma}\cdot\mathbf{p})^2}{2m}\,. \tag{169}$$

> ### The Dirac Equation in Condensed Matter Physics
>
> Having seen that the Schrödinger equation emerges as the low-energy limit of the Dirac equation, we would be remiss to not mention that the Dirac equation can sometimes re-emerge at even lower energy scales. This occurs in a number of condensed matter systems where the periodic potential of the crystal structure can lead to the lowest energy excitations of the system being governed by the Dirac equation, rather than the Schroödinger equation, and give rise to qualitatively new physics. Most notably, these low energy Dirac fermions appear in graphene (a single atomic layer of carbon atoms), $d$-wave superconductors (including high-temperature superconductors), and the surfaces of exotic materials called topological insulators.

To couple the electron to an electromagnetic field, we implement the minimal coupling procedure outlined above, replacing $\mathbf{p} \to \mathbf{P} = \mathbf{p} - q\mathbf{A}$. Specializing to the electron with $q = -e$, the Hamiltonian becomes

$$H = \frac{(\boldsymbol{\sigma}\cdot\mathbf{P})^2}{2m}\,, \qquad \mathbf{P} = \mathbf{p} + e\mathbf{A}\,. \tag{170}$$

At this point, we're done with physics, and all that follows is algebra. Life will be much easier if we use index notation instead of vectors, and make use of the algebraic properties of the Pauli matrices,

$$[\sigma^i, \sigma^j] = 2i\varepsilon^{ijk}\sigma^k\,,$$
$$\{\sigma^i, \sigma^j\} = 2\delta^{ij}\,. \tag{171}$$

We can then write the product of two Pauli matrices as

$$\sigma^i\sigma^j = \frac{1}{2}\left(\{\sigma^i, \sigma^j\} + [\sigma^i, \sigma^j]\right) = \delta^{ij} + i\varepsilon^{ijk}\sigma^k\,. \tag{172}$$

Using this identity, the Hamiltonian is

$$H = \frac{1}{2m}(\sigma_i P_i)(\sigma_j P_j) = \frac{1}{2m}\sigma_i\sigma_j P_i P_j = \frac{1}{2m}(\delta^{ij} + i\varepsilon^{ijk}\sigma^k)P_i P_j = \frac{1}{2m}\left(P_i P_i + i\varepsilon^{ijk}P_i P_j\sigma_k\right)\,. \tag{173}$$

In the second term, $P_i P_j$ is contracted with the completely antisymmetric $\varepsilon^{ijk}$, so only the antisymmetric piece of $P_i P_j$ contributes. This means we can write $\varepsilon^{ijk}P_i P_j = \frac{1}{2}\varepsilon^{ijk}[P_i, P_j]$.[15]

---

[15]You can think about this by using the usual trick: write $P_i P_j = \frac{1}{2}(\{P_i, P_j\} + [P_i, P_j])$. When we contract this with $\varepsilon^{ijk}$ (which is completely antisymmetric), the $\varepsilon^{ijk}\{P_i, P_j\}$ is identically zero. To see this, either expand out all the terms, or notice that if we switch $i \leftrightarrow j$, $\varepsilon_{jik} = -\varepsilon_{ijk}$ while $\{P_j, P_i\} = \{P_i, P_j\}$. Thus, under this change of indices $\varepsilon_{ijk}\{P_i, P_j\} = -\varepsilon_{ijk}\{P_i, P_j\}$ and therefore must be zero. This is just like integrating an even function against an odd one: their opposite behaviors cause them to vanish identically.

So, our task is now to evaluate

$$H = \frac{1}{2m}\left(P_i P_i + \frac{i}{2}\varepsilon^{ijk}[P_i, P_j]\sigma_k\right). \tag{174}$$

The first term will give us the coupling between the orbital angular momentum and the field, so we won't worry about it here. Our interest instead lies in the second term: to evaluate the commutator we work out

$$P_i P_j = (p_i + eA_i)(p_j + eA_j) = p_i p_j + eA_i p_j + e p_i A_j + A_i A_j, $$
$$P_j P_i = (i \leftrightarrow j) = p_j p_i + eA_j p_i + e p_j A_i + A_j A_i. \tag{175}$$

Subtracting the two, we find

$$[P_i, P_j] = e[A_i, p_j] + e[p_i, A_j]. \tag{176}$$

We now change representations to $p_j = -i\partial_j$ to calculate $[A_i, p_j] = -i[A_i, \partial_j]$. Just like in introductory quantum mechanics, it is easiest to compute this commutator by acting with it on a test function,

$$\begin{aligned}-i[A_i, \partial_j]\psi &= -i\big(A_i \partial_j \psi - \partial_j(A_i\psi)\big)\\ &= -i\big(A_i\partial_j\psi - (\partial_j A_i)\psi - A_i\partial_j\psi\big)\\ &= i(\partial_j A_i)\psi.\end{aligned} \tag{177}$$

Peeling off the test function, we have the operator equation for the commutator,

$$[A_i, p_j] = -i[A_i, \partial_j] = i\partial_j A_i. \tag{178}$$

The second commutator we need in (176) is $[p_i, A_j] = -[A_j, p_i]$, which is just the above with $i \leftrightarrow j$, so $[p_i, A_j] = -i\partial_i A_j$. Putting these back into (176) gives

$$[P_i, P_j] = -ie\big(\partial_i A_j - \partial_j A_i\big) = -ieF_{ij}, \tag{179}$$

where in the second equality we've used the definition of the electromagnetic field strength tensor. Using the above relation in the Hamiltonian (174), we have

$$H = \frac{1}{2m}\left(P_i P_i + \frac{e}{2}\varepsilon^{ijk}F_{ij}\sigma_k\right). \tag{180}$$

As you can convince yourself by comparing to the explicit matrix form given in section 4, the magnetic field can be written in terms of the spatial components of the field strength as

$$B_k = \frac{1}{2}\varepsilon_{ijk}F_{ij}, \tag{181}$$

and thus the Hamiltonian is simply

$$H = \frac{1}{2m}(P_i P_i + eB_k\sigma_k) = \frac{(\mathbf{p} + e\mathbf{A})^2}{2m} + \frac{e}{2m}\mathbf{B}\cdot\boldsymbol{\sigma}. \tag{182}$$

In terms of the spin operator $\mathbf{S} = \frac{1}{2}\boldsymbol{\sigma}$, this is

$$H = \frac{(\mathbf{p} + e\mathbf{A})^2}{2m} + 2\underbrace{\left(\frac{e}{2m}\right)}_{\mu_B}\mathbf{S}\cdot\mathbf{B}. \tag{183}$$

Comparing the second term to the canonical interaction term $H_{\text{int}} = g\mu_B \mathbf{S}\cdot\mathbf{B}$, we immediately identify

$$g = 2. \tag{184}$$

And thus we have completed a historically important calculation!

Additionally, if we couple the Dirac equation to a Coulomb potential, we will automatically get the correct spin-orbit coupling term and the short-ranged relativistic correction called the Darwin term, both in good agreement with experimental data. However, this is a computation for another class.

# 8 Electron Scattering Revisited

In section 3, we considered how one could probe the charge distribution of the nucleus via elastic electron scattering. However, we neglected the intrinsically relativistic nature of the problem, since one needs momentum transfers comparable to the mass of the nucleon to achieve adequate resolution. In this section, we return to the problem of electron scattering and sketch its proper relativistic treatment for determining the charge and and current distributions [16] of the nucleus and in nucleons.

The quantity which one seeks to measure in a scattering experiment is the matrix element of the electromagnetic current, $J^\mu$. If we take the simplest model of a nucleon as a point relativistic particle, this matrix element is simply given by the current appearing in the Dirac equation,

$$\langle N', s', \mathbf{p}' | J^\mu | N, s, \mathbf{p} \rangle = \bar{N}(s', \mathbf{p}') \gamma^\mu N(s, \mathbf{p}) \,. \tag{185}$$

Here, $N(s, \mathbf{p})$ is the Dirac spinor for the nucleon and $s$ and $\mathbf{p}$ label its spin and momentum. Of course, the nucleon is not a point particle and has an internal structure, reflected in the dependence of the current matrix element on the momentum transfer $\mathbf{q} = \mathbf{p}' - \mathbf{p}$. The charge and current densities are then given by the appropriate Fourier transforms of the $\mathbf{q}$-dependent current matrix elements. Our goal in this section is to determine some of the possible structures such functions can take.

Note that by virtue of energy and momentum conservation, an elastic scattering process viewed from the center of mass frame will leave the final energies of the electron and nucleon unchanged from their initial values. So, in this frame we have $q_0 = 0$ and $\mathbf{q} \neq 0$, so that $q^2 = q_0^2 - \mathbf{q}^2 < 0$. But, since $q^2$ is a Lorentz scalar, and hence frame-independent, we find that $q^2 < 0$ in any reference frame. At this point, it is conventional to introduce the notation $Q^2 \equiv -q^2$, which is a positive quantity.

By appealing to the symmetries of the nucleon, we can greatly constrain the form of the current matrix element. Since nucleons are governed by Quantum Chromodynamics, they must respect parity and time-reversal invariance (as discussed in later sections), and the only functional forms consistent with these symmetries lead to the matrix element

$$\langle N', s', \mathbf{p}' | J^\mu | N, s, \mathbf{p} \rangle = \bar{N}(s', \mathbf{p}') \left[ F_1(Q^2) \gamma^\mu + F_2(Q^2) \frac{q_\nu \sigma^{\mu\nu}}{2m} \right] N(s, \mathbf{p}) \,, \tag{186}$$

where $F_1(Q^2)$ and $F_2(Q^2)$ are called the Dirac and Pauli form factors. It turns out that both of these functions can be extracted from an electron scattering experiment. However, these form factors do not separate out the electric and magnetic effects, as we know that $J^0$ and $\mathbf{J}$ mix under Lorentz transformations and there is no Lorentz-invariant way to distinguish the charge and current densities. However, one can choose a "natural" frame in which the electric and magnetic contributions can be defined.

The most convenient choice is the so-called "Breit frame" in which the electron's three-momentum is reversed in the scattering process, i.e. $\mathbf{p}' = -\mathbf{p}$ and $\mathbf{q} = -2\mathbf{p}$. This defines a line over which the scattering event occurs (rather than a plane) and allows one to consider the projection of a vector along that line. However, the Breit frame is generally different between different scattering events, i.e. it is a mathematical convenience rather than a physically useful object.

It can be shown that in the Breit frame the electric and magnetic form factors, which are

---

[16]Why is there a current? Answer: it's spinning (unless it is a J=0 state).

the Fourier transforms of the charge and magnetization densities,

$$G_E(Q^2) = \int d^3r \; e^{-i\mathbf{q}\cdot\mathbf{r}} \rho(\mathbf{r}), \tag{187}$$

$$G_M(Q^2) = \int d^3r \; e^{-i\mathbf{q}\cdot\mathbf{r}} \mathbf{M}(\mathbf{r}) \cdot \hat{\mathbf{z}}, \tag{188}$$

(here, $\mathbf{M}(\mathbf{r})$ is the magnetization density and $\hat{\mathbf{z}}$ is the direction of momentum in the Breit frame) are related to the Dirac and Pauli form factors in the following simple manner:

$$G_E(Q^2) = F_1(Q^2) - \tau F_2(Q^2), \qquad \tau \equiv \frac{Q^2}{4m^2}, \tag{189}$$

$$G_M(Q^2) = F_1(Q^2) + F_2(Q^2). \tag{190}$$

We will omit the demonstration that this is the case for reasons of simplicity. We note in passing that these form factors evaluated at $Q^2 = 0$ correspond to familiar quantities: $G_E(0) = Z$ is the charge of the nucleon ($+e$ for the proton and zero for the neutron), and $G_M(0)$ is the $g$-factor, which measured in units of $e/2m_p$ is 5.585 for the proton and $-3.826$ for the neutron.

Having stated the functional form of the electric and magnetic form factors, we can now briefly discuss how they are extracted from electron scattering experiments. The differential scattering cross section for unpolarized electrons and nucleons is,

$$\frac{d\sigma}{d\Omega} = \frac{d\sigma}{d\Omega}\bigg|_{\text{Mott}} \frac{G_E^2(Q^2) + (\tau/\epsilon)G_M^2(Q^2)}{1+\tau}, \tag{191}$$

where $d\sigma/d\Omega|_{\text{Mott}}$ is the Mott cross section for relativistic scattering of point particles, whose form is well-known, and

$$\epsilon = \left[1 + 2(1+\tau)\tan^2(\theta/2)\right]^{-1}. \tag{192}$$

Again we simply state this result without proof for reasons of simplicity.

By measuring with different angles of incidence and energies, one can measure the cross section at fixed $Q^2$ but different values of $\tau/\epsilon$, from which one can separately determine the values of $G_E(Q^2)$ and $G_M(Q^2)$.

In a homework problem at the end of these notes, you can show that the mean radius of a charge distribution is given by the derivative of the electric form factor,

$$\langle r^2 \rangle_E = -6 \frac{\partial^2 G_E(Q^2)}{\partial Q^2}. \tag{193}$$

Extracting the nucleon radius from an electron scattering experiment in this way gives the radius of the proton and neutron to be, respectively,

$$\langle r^2 \rangle_E^p = \left[(.886 \pm 0.12)\,\text{fm}\right]^2, \tag{194}$$

$$\langle r^2 \rangle_E^n = \left[-0.12\,\text{fm}\right]^2. \tag{195}$$

The negative radius of the neutron suggests that the nuclear charge density is positive at short distances and becomes negative further away. But, there is a problem with the proton charge radius: the same quantity can be measured using atomic physics, in which the answer was found to be

$$\langle r^2 \rangle_E^p = \left[(1.8409 \pm 0.0004)\,\text{fm}\right]^2, \tag{196}$$

which does not agree with the result from electron scattering!

This so-called "proton radius puzzle" was a major experimental anomaly. Barring an experimental error, it implies that either there is some subtle QED effect which invalidates the atomic calculation, or, most excitingly,it hints at physics beyond the standard model. Recent measurements suggest, however, that the resolution to this puzzle may well lie in experimental difficulties in the older atomic measurements.

> ### Atomic Measurement of Proton Radius
>
> The basic principle behind the atomic measurement of the proton charge radius is that the Coulomb potential goes like $-e^2/r$ all the way down to $r = 0$. So, if the proton has a finite size, this potential is less negative inside the volume of the proton, and consequently the atomic level is less tightly bound. The magnitude of this effect depends on the size of the proton and the density of the wavefunction near the origin. For this reason, the experiments are carried out using muonic hydrogen, as the muon is 200 times heavier than an electron, and hence the wavefunction has $(200)^3$ times as much weight near the origin as one would find in ordinary hydrogen. Of course, there are many other effects which shift atomic energy levels, but they can all be calculated (up to very small corrections), and one can perform what is believed to be a reliable calculation for the proton charge radius.

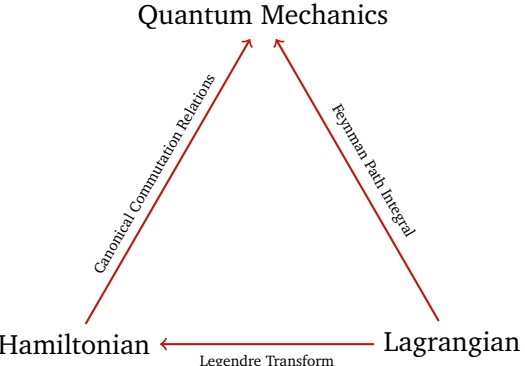

Figure 8: Illustration of the relationship between different formalisms of quantum and classical mechanics.

# 9 Quantum Field Theory for Pedestrians

In this section we will introduce the language of modern particle physics: quantum field theory. First, however, we will briefly discuss *classical* field theory, which we will then quantize to arrive at QFT. Usually one uses the Hamiltonian formulation of classical mechanics to move from classical to quantum physics, but one can also use the Lagrangian formulation to quantize a theory using Feynman's path integral approach (this is illustrated in Fig. 8. Although we won't discuss path integrals in this class, the Lagrangian formalism will still be used to treat classical field theories, so to refresh everyone's memory and establish notation we will briefly review Lagrangian mechanics in the following section.

## 9.1 Review of Classical Mechanics

Suppose we have a system with $N$ degrees of freedom, e.g. the positions of $N/3$ particles in 3 spatial dimensions. We combine all of these degrees of freedom or "generalized coordinates" into an $N$ dimensional vector $\mathbf{q}$ in the $N$-dimensional *configuration space* of the system. If we run the system forward in time, the vector $\mathbf{q}$ will trace out a path in configuration space, which specifies the time evolution of all of the particles in real space. The question of how the entire system changes in time is now encapsulated in what path the system takes through configuration space. So, our goal is to determine the path.

We do so by associating a number, called the *action*, with each possible path through configuration space. For a path which starts at $\mathbf{q}_i$ at time $t_i$ and ends at $\mathbf{q}_f$ at time $t_f$, the action is defined to be

$$S = \int_{t_i}^{t_f} dt \, L(\mathbf{q}(t), \dot{\mathbf{q}}(t)), \tag{197}$$

where $L$ is the *Lagrangian*, which depends on the generalized coordinates $\mathbf{q}$ and their time derivatives $\dot{\mathbf{q}}$. The central axiom of Lagrangian mechanics is the *Principle of Least Action*, which states that the actual path taken by the system is the one for which the action is minimized. So, to determine the path the system takes through configuration space, or equivalently its time evolution, we need to simply minimize $S$ with respect to the path. We can't just set the derivative of $S$ equal to zero, since we are minimizing with respect to a path, not a single variable.

To carry out the minimization (technically, the "variation") we consider some path $\mathbf{q}(t)$ through configuration space, and suppose we deform it slightly by adding a new time dependent function $\delta\mathbf{q}(t)$, so that we have a new path $\mathbf{q}(t) + \delta\mathbf{q}(t)$. To compare apples to apples,

we need to make sure that the new path begins and ends at the same point as the original one, which means the deformation $\delta\mathbf{q}(t)$ must vanish at the endpoints,

$$\delta\mathbf{q}(t_i) = \delta\mathbf{q}(t_f) = 0 . \tag{198}$$

The time derivative of the deformed path is $\dot{\mathbf{q}}(t) + \delta\dot{\mathbf{q}}(t)$. The change in the action as a result of this deformation is then

$$\delta S \equiv S[\mathbf{q} + \delta\mathbf{q}, \dot{\mathbf{q}} + \delta\dot{\mathbf{q}}] - S[\mathbf{q}, \dot{\mathbf{q}}], \tag{199}$$

and the actual path taken by the system is the $\mathbf{q}(t)$ for which $\delta S = 0$. The change in the action is

$$\delta S = \int \mathrm{d}t\, L(\mathbf{q} + \delta\mathbf{q}, \dot{\mathbf{q}} + \delta\dot{\mathbf{q}}) - \int \mathrm{d}t\, L(\mathbf{q}, \dot{\mathbf{q}}) \tag{200}$$

$$\approx \int \mathrm{d}t \left[ L(\mathbf{q}, \dot{\mathbf{q}}) + \sum_{i=1}^{N} \frac{\partial L(\mathbf{q}, \dot{\mathbf{q}})}{\partial q^i} \delta q^i + \sum_{i=1}^{N} \frac{\partial L(\mathbf{q}, \dot{\mathbf{q}})}{\partial \dot{q}^i} \delta\dot{q}^i \right] - \int \mathrm{d}t\, L(\mathbf{q}, \dot{\mathbf{q}}) \tag{201}$$

$$= \int \mathrm{d}t \sum_{i=1}^{N} \left[ \frac{\partial L}{\partial q^i} \delta q^i + \frac{\partial L}{\partial \dot{q}^i} \delta\dot{q}^i \right]. \tag{202}$$

In the second line, we simultaneously used the first order approximation

$$f(x + \delta x) \approx f(x) + \frac{\partial f}{\partial x} \delta x \tag{203}$$

on each argument of the Lagrangian, which is justified since the deformation of the path is assumed to be small. The next step is to integrate the second term(s) in (202) by parts,

$$\int \mathrm{d}t \sum_{i=1}^{N} \frac{\partial L}{\partial \dot{q}^i} \delta\dot{q}^i = \left[ \sum_{i=1}^{N} \frac{\partial L}{\partial \dot{q}^i} \delta q^i \right]_{t_i}^{t_f} - \int \mathrm{d}t \sum_{i=1}^{N} \left( \frac{\mathrm{d}}{\mathrm{d}t} \frac{\partial L}{\partial \dot{q}^i} \right) \delta q^i . \tag{204}$$

The first term is a boundary term, and vanishes since $\delta\mathbf{q}(t_i) = \delta\mathbf{q}(t_f) = 0$. Putting the remaining term back into (202), we have

$$\delta S = \int \mathrm{d}t \sum_{i=1}^{N} \left[ \frac{\partial L}{\partial q^i} - \frac{\mathrm{d}}{\mathrm{d}t} \frac{\partial L}{\partial \dot{q}^i} \right] \delta q^i . \tag{205}$$

To have $\delta S = 0$, the above must be zero for any $\delta\mathbf{q}$, which requires the term in brackets vanish, resulting in the equations of motion

$$\frac{\partial L}{\partial q^i} - \frac{\mathrm{d}}{\mathrm{d}t} \frac{\partial L}{\partial \dot{q}^i} = 0 , \tag{206}$$

called the *Euler-Lagrange equations*. Notice that $i$ is now a free index, so we have one equation for each generalized coordinate. To make contact with Newtonian physics, its easy to show that if we take

$$L = T - V \tag{207}$$

the resulting equations of motion are simply Newton's Law $\mathbf{F} = m\mathbf{a}$.

In the Lagrangian formalism, the *canonical momentum* is defined as

$$p_i = \frac{\partial L}{\partial \dot{q}^i} . \tag{208}$$

For example, if $L = T - V$ where $T = \sum_i \frac{1}{2} m \dot{q}_i^2$, then $p_i = m \dot{q}_i$. However, a more complicated system can have canonical momenta that do not agree with the naive Newtonian definition.

Having found the canonical momenta, we may construct the Hamiltonian $H(\mathbf{q}, \mathbf{p})$, which is a function of the generalized coordinates and the canonical momenta conjugate to them. To perform the change of variables, we use the *Legendre transformation*,

$$H(\mathbf{q}, \mathbf{p}) = \mathbf{p} \cdot \dot{\mathbf{q}} - L(\mathbf{q}, \dot{\mathbf{q}}), \tag{209}$$

where since the Hamiltonian may not depend on $\dot{\mathbf{q}}$, we must invert the canonical momentum to solve for $\dot{\mathbf{q}}$ in terms of $\mathbf{p}$, and use that expression everywhere $\dot{\mathbf{q}}$ appears. For example, if $L = T - V$ and $p_i = m \dot{q}_i$, then $\dot{q}_i = p_i / m$, and the Legendre transformation proceeds as

$$H = \sum_{i=1}^{N} p_i \left( \frac{p_i}{m} \right) - \sum_{i=1}^{N} \frac{1}{2} m \left( \frac{p_i}{m} \right)^2 + V(\mathbf{q}) = \sum_{i=1}^{N} \frac{p_i^2}{2m} + V(\mathbf{q}). \tag{210}$$

Given a classical Hamiltonian, one typically quantizes it by promoting $\mathbf{q}$ and $\mathbf{p}$ to operators on the Hilbert space and imposing the canonical commutation relations,

$$[q_i, p_j] = i \delta_{ij}. \tag{211}$$

In passing, we also note that quantization can be implemented in the Lagrangian formalism via the Feynman path integral, wherein the system need not take the path for which the action is minimized, but rather it may take any path with a probability weighted by $e^{iS}$. Having reminded ourselves of the details of classical particle mechanics, we will now turn to the classical theory of fields.

## 9.2 Classical Field Theory

Simply put, a field theory describes a system whose dynamics is specified by an uncountably infinite number of degrees of freedom. For example, throughout this section we will consider a *scalar field*, $\phi(\mathbf{x}, t)$, which assigns a scalar (number) to each point in spacetime. To fully specify the configuration of this field, you would need to tell me the value it takes at each point $\mathbf{x}$ in space for every time $t$. One can think of $\phi(\mathbf{x}, t)$ as the continuous analogue of the finite dimensional vector $\mathbf{q}$ from the previous section.

Now, given a scalar field $\phi(\mathbf{x}, t)$ we'd like to construct a theory describing it. In point particle quantum mechanics we could equally well use the Hamiltonian or Lagrangian formalism, but for a relativistic field theory the Lagrangian approach is preferable. The key word here is *relativistic*: in any sensible theory the action, which lies at the heart of the Lagrangian formalism, will be Lorentz invariant, and we may thus use it to describe physics in any frame. Since a field theory has degrees of freedom at every point in space, it is traditional to write the Lagrangian $L$ in terms of the *Lagrangian density*, $\mathcal{L}$,

$$L = \int \mathrm{d}^3 r \, \mathcal{L}(\phi, \partial_\mu \phi), \tag{212}$$

where $\mathcal{L}$ depends on the values of the field $\phi$ and its spacetime derivatives $\partial_\mu \phi$ at a particular point in spacetime. The Lagrangian density is used so often that, perhaps confusingly, everyone simply calls it the Lagrangian. In terms of the Lagrangian (density), the action is

$$S = \int \mathrm{d}^4 x \, \mathcal{L}(\phi, \partial_\mu \phi), \tag{213}$$

where the four-dimensional integral measure is shorthand for $\mathrm{d}^4 x = \mathrm{d}t \, \mathrm{d}^3 r$. So, to formulate a relativistic theory, all we have to do is make sure that the action is Lorentz invariant, and we're good to go.

On the other hand, the Hamiltonian is not so easy to work with. Just like the Lagrangian, we can write it in terms of a *Hamiltonian density*,

$$H = \int \mathrm{d}^3 r \, \mathcal{H}. \tag{214}$$

However, the Hamiltonian density is *not* Lorentz invariant, since it is the 00 component of the stress energy tensor $T^{\mu\nu}$, and hence transforms as

$$\mathcal{H} \mapsto \Lambda^0_{\ \alpha} \Lambda^0_{\ \beta} \, T^{\alpha\beta}. \tag{215}$$

Thus, it is substantially easier to work with the Lagrangian to formulate relativistic dynamics.

Now, let's suppose we've figured out the Lagrangian for some field theory, how do we get the equations of motion out of it? Another convenience of the Lagrangian formalism is that the story is essentially the same as in the point particle case: we simply need to minimize the action with respect to variations of the field $\phi(\mathbf{x}, t)$. In what follows we'll work with a scalar field for simplicity, but the generalization to other kinds of fields is straight forward (you just switch the letters!). To carry out the variation, we again consider deforming the field $\phi(\mathbf{x}, t) \mapsto \phi(\mathbf{x}, t) + \delta\phi(\mathbf{x}, t)$. The change in the action $\delta S$ is then

$$
\begin{aligned}
\delta S &= \int \mathrm{d}^4 x \, \mathcal{L}(\phi + \delta\phi, \partial_\mu \phi + \delta\partial_\mu \phi) - \int \mathrm{d}^4 x \, \mathcal{L}(\phi, \partial_\mu \phi) \\
&\approx \int \mathrm{d}^4 x \left[ \mathcal{L}(\phi, \partial_\mu \phi) + \frac{\partial \mathcal{L}}{\partial \phi} \delta\phi + \frac{\partial \mathcal{L}}{\partial(\partial_\mu \phi)} \delta(\partial_\mu \phi) \right] - \int \mathrm{d}^4 x \, \mathcal{L}(\phi, \partial_\mu \phi) \\
&= \int \mathrm{d}^4 x \left[ \frac{\partial \mathcal{L}}{\partial \phi} \delta\phi + \frac{\partial \mathcal{L}}{\partial(\partial_\mu \phi)} \delta(\partial_\mu \phi) \right].
\end{aligned}
\tag{216}
$$

Just as before, we used a linear approximation (203) on all of the arguments of $\mathcal{L}$ in the second line (don't forget the index $\mu$ is repeated, and thus summed over!). We can integrate the second term by parts, and provided $\delta\phi \to 0$ at the boundary (usually taken to be spatial and temporal infinity) the surface term vanishes so

$$\frac{\partial \mathcal{L}}{\partial(\partial_\mu \phi)} \delta(\partial_\mu \phi) = -\left( \partial_\mu \frac{\partial \mathcal{L}}{\partial(\partial_\mu \phi)} \right) \delta\phi, \tag{217}$$

and thus the variation in the action is

$$\delta S = \int \mathrm{d}^4 x \left[ \frac{\partial \mathcal{L}}{\partial \phi} - \partial_\mu \frac{\partial \mathcal{L}}{\partial(\partial_\mu \phi)} \right] \delta\phi. \tag{218}$$

Requiring this to vanish for all $\delta\phi$, we arrive at the four dimensional generalization of the Euler-Lagrange equations,

$$\frac{\partial \mathcal{L}}{\partial \phi} - \partial_\mu \frac{\partial \mathcal{L}}{\partial(\partial_\mu \phi)} = 0. \tag{219}$$

As our first example, we'll consider a real scalar field $\phi(\mathbf{x}, t)$, by which we mean $\phi(\mathbf{x}, t)$ assigns a real number to each point in spacetime, and that real number is a scalar in the technical sense. That is, it is a Lorentz scalar. The Lagrangian we will consider is

$$\mathcal{L} = \frac{1}{2} \partial_\mu \phi \, \partial^\mu \phi - \frac{1}{2} m^2 \phi^2, \tag{220}$$

which is sometimes called the "Klein Gordon Lagrangian," for reasons we are about to see. To find the equations of motion we need to calculate the derivatives with respect to $\phi$ and $\partial_\mu \phi$. The first is trivial,

$$\frac{\partial \mathcal{L}}{\partial \phi} = -m^2 \phi. \tag{221}$$

The second can be computed by writing $\partial_\mu \phi \partial^\mu \phi = g^{\alpha\beta} \partial_\alpha \phi \, \partial_\beta \phi$ and noting that derivatives in different directions are independent variables, which we can formalize by writing

$$\frac{\partial(\partial_\alpha \phi)}{\partial(\partial_\mu \phi)} = \delta^\alpha_\mu \, . \tag{222}$$

The derivative of the Lagrangian with respect to $\partial_\mu \phi$ is then

$$\frac{\partial \mathcal{L}}{\partial(\partial_\mu \phi)} = \frac{\partial \left( \frac{1}{2} g^{\alpha\beta} \partial_\alpha \phi \, \partial_\beta \phi \right)}{\partial(\partial_\mu \phi)} = \frac{1}{2} \left( g^{\mu\alpha} \partial_\alpha \phi + g^{\mu\beta} \partial_\beta \phi \right) = g^{\mu\nu} \partial_\nu \phi = \partial^\mu \phi \, , \tag{223}$$

where to get from the third to the fourth equality we noted that $\alpha$ and $\beta$ were both dummy indices and thus we could combine the two identical terms. In practice, $\partial_\mu \phi \, \partial^\mu \phi \sim (\partial_\mu \phi)^2$, so the derivative with respect to $\partial_\mu \phi$ should be something like $2 \partial_\mu \phi$, which happens to be correct.

Putting these into the Euler-Lagrange equations (219), we find the equations of motion

$$\begin{aligned} \frac{\partial \mathcal{L}}{\partial \phi} - \partial_\mu \frac{\partial \mathcal{L}}{\partial(\partial_\mu \phi)} &= 0 \, , \\ -m^2 \phi - \partial_\mu \partial^\mu \phi &= 0 \, , \\ \implies (\partial^\mu \partial_\mu + m^2) \phi &= 0 \, , \end{aligned} \tag{224}$$

which is the Klein-Gordon equation! So, we have learned that the Klein-Gordon is the *classical* equation of motion for a real scalar field. We play a similar game with the Dirac equation, starting with the Lagrangian

$$\mathcal{L} = \bar{\psi}(i\slashed{\partial} - m)\psi \, , \tag{225}$$

where $\psi$ and $\bar{\psi}$ are to be treated as two independent fields. Since $\partial\mathcal{L}/\partial(\partial_\mu \bar{\psi}) = 0$, the equation of motion for $\bar{\psi}$ is simply the Dirac equation,

$$\frac{\partial \mathcal{L}}{\partial \bar{\psi}} - \partial_\mu \frac{\partial \mathcal{L}}{\partial(\partial_\mu \bar{\psi})} = (i\slashed{\partial} - m)\psi = 0 \, . \tag{226}$$

We can also find the equations of motion for $\psi$,

$$\begin{aligned} \frac{\partial \mathcal{L}}{\partial \psi} - \partial_\mu \frac{\partial \mathcal{L}}{\partial(\partial_\mu \psi)} &= 0 \, , \\ -\bar{\psi} m - \partial_\mu (i \bar{\psi} \gamma^\mu) &= 0 \, , \\ \bar{\psi}(-i\overleftarrow{\slashed{\partial}} - m) &= 0 \, , \end{aligned} \tag{227}$$

which we also met in our discussion of the Dirac equation. Although these equations *look* like single-particle quantum mechanics, we must emphasize that they really are *classical* equations!

To consider something less "quantum-looking," lets consider good old-fashioned electro-magnetism, the most famous classical field theory, from the perspective of the Maxwell Lagrangian

$$\mathcal{L} = -\frac{1}{4} F_{\mu\nu} F^{\mu\nu} - A_\mu J^\mu \, , \tag{228}$$

where the field strength tensor $F_{\mu\nu}$ is defined in terms of the potentials $A_\mu = (\Phi, -\mathbf{A})$, as

$$F_{\mu\nu} = \partial_\mu A_\nu - \partial_\nu A_\mu \, , \tag{229}$$

and the second term in the Lagrangian couples the electromagnetic field to a current source, $J^\mu = (\rho, \mathbf{J})$. The fundamental field here is $A_\mu$, and we will get an Euler-Lagrange equation for each component,

$$\frac{\partial \mathcal{L}}{\partial A_\nu} - \partial_\mu \frac{\partial \mathcal{L}}{\partial(\partial_\mu A_\nu)} = 0. \tag{230}$$

Obviously, $\partial \mathcal{L}/\partial A_\nu = -J^\nu$, but the variation with respect to $\partial_\mu A_\nu$ requires a little care. We'll again use the fact that the components and different derivatives of $A_\mu$ are all independent variables. Here it goes,

$$\begin{aligned}
\frac{\partial(F_{\rho\sigma}F^{\rho\sigma})}{\partial(\partial_\mu A_\nu)} &= F^{\rho\sigma}\frac{\partial F_{\rho\sigma}}{\partial(\partial_\mu A_\nu)} + F_{\rho\sigma}\frac{\partial F^{\rho\sigma}}{\partial(\partial_\mu A_\nu)} \\
&= 2F^{\rho\sigma}\frac{\partial F_{\rho\sigma}}{\partial(\partial_\mu A_\nu)} \\
&= 2F^{\rho\sigma}\left[\frac{\partial(\partial_\rho A_\sigma)}{\partial(\partial_\mu A_\nu)} - \frac{\partial(\partial_\sigma A_\rho)}{\partial(\partial_\mu A_\nu)}\right] \\
&= 2F^{\rho\sigma}\left(\delta^\mu_\rho \delta^\nu_\sigma - \delta^\mu_\sigma \delta^\nu_\rho\right) \\
&= 2(F^{\mu\nu} - F^{\nu\mu}) \\
&= 4F^{\mu\nu}.
\end{aligned} \tag{231}$$

To get to the last line we used the fact that the field strength is antisymmetric, i.e. $F_{\nu\mu} = -F_{\mu\nu}$. So,

$$\frac{\partial \mathcal{L}}{\partial(\partial_\mu A_\nu)} = -F^{\mu\nu}. \tag{232}$$

Putting this into (230), we get the equations of motion

$$\begin{aligned}
\frac{\partial \mathcal{L}}{\partial A_\nu} - \partial_\mu \frac{\partial \mathcal{L}}{\partial(\partial_\mu A_\nu)} &= 0, \\
-J^\nu - \partial_\mu(-F^{\mu\nu}) &= 0 \\
\implies \partial_\mu F^{\mu\nu} &= J^\nu.
\end{aligned} \tag{233}$$

It turns out, this is nothing more than Maxwell's equations, albeit in an elegant notation. First of all, notice that $\nu$ is a free index, so the above is actually four equations. Let's first consider the $\nu = 0$ equation, which reads

$$\begin{aligned}
\partial_0 F^{00} + \partial_1 F^{10} + \partial_2 F^{20} + \partial_3 F^{30} &= j^0 = \rho, \\
\partial_x E_x + \partial_y E_y + \partial_z E_z &= \rho, \\
\nabla \cdot \mathbf{E} &= \rho,
\end{aligned} \tag{234}$$

which is Gauss's law. For your convenience, recall the components of the field strength are

$$F^{\mu\nu} = \begin{pmatrix} 0 & -E_x & -E_y & -E_z \\ E_x & 0 & -B_z & B_y \\ E_y & B_z & 0 & -B_x \\ E_z & -B_y & B_x & 0 \end{pmatrix}. \tag{235}$$

Next, let's consider the $\nu = 1$ equation:

$$\begin{aligned}
\partial_0 F^{01} + \partial_1 F^{11} + \partial_2 F^{21} + \partial_3 F^{31} &= j^1 = j_x, \\
-\partial_t E_x + \partial_y B_z - \partial_z B_y &= j_x, \\
(\nabla \times \mathbf{B})_x &= j_x + \partial_t E_x,
\end{aligned} \tag{236}$$

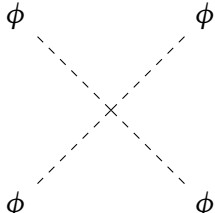

Figure 9: Feynman diagram representing the $\phi^4$ interaction

which is the $x$ component of Ampere's Law. As you're invited to confirm, the $\nu = 2$ and $\nu = 3$ equations give us the $y$ and $z$ components of Ampere's law, respectively. You may ask where the other two Maxwell equations went, and the answer is that they've been hiding in front of us all along. The equations $\nabla \cdot \mathbf{B} = 0$ and $\nabla \times \mathbf{E} + \partial_t \mathbf{B} = 0$ are implicit in the definitions of the electric and magnetic fields in terms of the potentials, so these equations are actually already embedded in the structure of $F_{\mu\nu}$.[17]

As a final example before moving onto the big bad world of QFT, we'll consider the unimaginatively named "$\phi^4$ theory," defined by the Lagrangian

$$\mathcal{L} = \frac{1}{2}\partial_\mu \phi \, \partial^\mu \phi - \frac{1}{2}m^2 \phi^2 - \frac{1}{4}\lambda \phi^4 , \tag{237}$$

where $\phi$ is once again a real scalar field and $m$ and $\lambda$ are constants. Although this may look like a minor variation of the (free) Klein-Gordon Lagrangian we previously discussed, the eponymous $\phi^4$ is actually something we haven't met yet; it is an *interaction term*. The equations of motion are easy to find, since we're already experts on the free Klein-Gordon theory,

$$\frac{\partial \mathcal{L}}{\partial \phi} - \partial_\mu \frac{\partial \mathcal{L}}{\partial(\partial_\mu \phi)} = 0 ,$$
$$-m^2 \phi - \lambda \phi^3 - \partial_\mu \partial^\mu \phi = 0 \tag{238}$$
$$\implies (\partial^\mu \partial_\mu + m^2)\phi + \lambda \phi^3 = 0 .$$

The first two terms represent the free propagation of a bosonic field, while the last nonlinear term is interpreted as an interaction between the particles. Any theory lacking such interaction terms is said to be *free*. Because the term in the Lagrangian is $\phi^4$, this interaction term encodes $\phi\phi \to \phi\phi$ scattering, as shown in Fig. 9. We'll learn more about how this works when we discuss Feynman diagrams later on in this section.

## 9.3 Quantizing Canonically: Scalar Fields

Having acquainted ourselves with classical field theory, its time to jump into the deep end and quantize it. We'll follow the normal procedure from particle quantum mechanics: identify the canonical momentum $p_i = \partial L/\partial \dot{q}_i$ and impose the canonical commutation relations $[q_i, p_j] = i\delta_{ij}$. The only difference in field theory is that our set of degrees of freedom is continuous rather than discrete.

Unsurprisingly, the canonical momentum in a field theory is itself a field, defined analogously to the point particle case as

$$\pi(\mathbf{x}, t) = \frac{\partial \mathcal{L}}{\partial \dot{\phi}} . \tag{239}$$

---

[17]You can also be fancy and introduce the "dual field strength," $^\star F^{\mu\nu} = \frac{1}{2}\varepsilon^{\mu\nu\rho\sigma}F_{\rho\sigma}$, and the resulting equation of motion, called the Bianchi identity, $\partial_\mu {}^\star F^{\mu\nu} = 0$, gives the two remaining equations. However, these equations basically come from the structure of $F_{\mu\nu}$ anyway, so this isn't worth worrying too much about.

Note that the fields are time dependent. That means when we promote them to operators in the quantum theory you should think of them like operators in the Heisenberg picture of normal quantum mechanics.

---

### Heisenberg and Schrödinger Pictures of QM

Recall that in quantum mechanics, the things we actually measure are quantum averages, like $\langle \psi | \mathcal{O} | \psi \rangle$, where $|\psi\rangle$ is a state and $\mathcal{O}$ is an operator. Ostensibly, such an average could change in time, and in light of the philosophy that we should only take seriously observable quantities, it shouldn't matter how we decide to implement this time dependence. There are two main "pictures" of how to deal with time evolution. The first is the familiar Schrödinger picture where operators are time-independent and the states evolve via

$$\psi(t) = e^{-iHt}\psi(0). \tag{240}$$

There is also the perhaps unfamiliar Heisenberg representation, where we take the opposite approach: states are time-independent and operators carry the time-dependence, evolving as

$$\mathcal{O}(t) = e^{iHt}\mathcal{O}(0)e^{-iHt}. \tag{241}$$

This picture is typically more natural in QFT, given that we are primarily concerned with objects such as creation and annihilation operators and field operators (to be introduced below) rather than wave functions. We should also note that there is a third common picture, called the interaction representation, where both states and operators time-evolve according to different parts of the Hamiltonian. This is the picture one typically uses to derive the Feynman rules (within the canonical formalism), but we will not need to use it explicitly in these notes.

---

In point particle quantum mechanics, if we impose canonical commutation relations at $t = 0$ its easy to see from the above that they will still hold at a later time $t$,

$$[q_i(t), p_j(t)] = i\delta_{ij}. \tag{242}$$

Notice that both operators are evaluated at the *same* time. Because of this, the above are called *equal time commutation relations*. To quantize our bosonic field theory, we can just replace the discrete operators in the above equation with their continuous field theory counterparts,

$$[\phi(\mathbf{x}, t), \pi(\mathbf{x}', t)] = i\delta^3(\mathbf{x} - \mathbf{x}'). \tag{243}$$

Again note that both operators are evaluated at the same time. We've also upgraded our delta from Kronecker to Dirac in light of the continuous nature of our fields.

Now, to actually compute the energies and momenta of physical states we need to construct the Hamiltonian (density), and in doing so choose a reference frame (and lose covariance). The Hamiltonian (density) is still defined as the Legendre transform of the Lagrangian (density),

$$\mathcal{H}(\phi, \pi) = \dot{\phi}\pi - \mathcal{L}(\phi, \partial_\mu \phi). \tag{244}$$

For example, let's construct the Hamiltonian for our free Klein-Gordon field, where

$$\mathcal{L} = \frac{1}{2}\partial_\mu \phi \, \partial^\mu \phi - \frac{1}{2}m^2\phi^2 = \frac{1}{2}\dot{\phi}^2 - \frac{1}{2}(\nabla\phi)^2 - \frac{1}{2}m^2\phi^2. \tag{245}$$

The canonical momentum is

$$\pi = \frac{\partial \mathcal{L}}{\partial \dot{\phi}} = \dot{\phi}, \tag{246}$$

from which it follows that the Hamiltonian is

$$\mathcal{H} = \pi\dot{\phi} - \mathcal{L} = \frac{1}{2}\pi^2 + \frac{1}{2}(\nabla\phi)^2 + \frac{1}{2}m^2\phi^2. \tag{247}$$

Now, the next thing we might want to do is look at what kinds of physical states we can have. Let's first go back to the classical Klein-Gordon equation, which we saw has plane wave solutions,

$$
\begin{aligned}
\phi &\sim e^{-i(\mathbf{k}\cdot\mathbf{x}-\omega t)} = e^{ik_\mu x^\mu}, \\
\phi &\sim e^{i(\mathbf{k}\cdot\mathbf{x}-\omega t)} = e^{-ik_\mu x^\mu},
\end{aligned}
\tag{248}
$$

with dispersion

$$\omega(\mathbf{k}) = \sqrt{\mathbf{k}^2 + m^2}. \tag{249}$$

We can think of these solutions as the normal modes, which we can superimpose to write a general solution. This means we can decompose the field $\phi(\mathbf{x}, t)$ as

$$\phi(\mathbf{x}, t) = \int \frac{d^3k}{(2\pi)^3} \frac{1}{\sqrt{2\omega(\mathbf{k})}} \left[a(\mathbf{k})e^{ik_\mu x^\mu} + a^\dagger(\mathbf{k})e^{-ik_\mu x^\mu}\right], \tag{250}$$

where classically, the $a(\mathbf{k})$ and $a^\dagger(\mathbf{k})$ are the amplitudes to be in a given mode. These amplitudes are time dependent, but we're usually going to suppress the explicit time dependence when writing them in order to keep the notation compact. The factor of $\omega(\mathbf{k})^{-1/2}$ is just a conventional normalization to clean up later results.

---

### Lorentz Invariant Integral Measures

To motivate the funny factor of $\omega_\mathbf{k}$ in the denominator of our expressions above, let's ask what a Lorentz-invariant integral measure should look like. It should be no surprise that $\int d^4k$ is manifestly Lorentz invariant, but integrating over all possible frequencies and momenta isn't usually what we want. We want to only integrate over states with positive energy ($k^0 > 0$) and which satisfy the dispersion relation $k^0 = \sqrt{\mathbf{k}^2 + m^2} \equiv \omega_\mathbf{k}$. Note that in this expression $k^0$ is an integration variable, while $\omega_\mathbf{k}$ is a fixed function of $\mathbf{k}$. This means the Lorentz invariant measure we really want is

$$\int \frac{d^4k}{(2\pi)^4} \delta^4(k^2 - m^2)\theta(k^0), \tag{251}$$

where $k^2 - m^2 = 0$ is the Lorentz-invariant "on-mass-shell" condition. We can use one of the standard delta function identities to write the above as

$$\int \frac{d^4k}{(2\pi)^4} \left[\frac{\delta(k^0 - \omega_\mathbf{k})}{2k^0} + \frac{\delta(k^0 + \omega_\mathbf{k})}{2k^0}\right] \theta(k^0). \tag{252}$$

The theta function kills the second term, so performing the $k^0$ integral using the delta function, this is simply

$$\int \frac{d^3k}{(2\pi)^3} \int \frac{dk^0}{2\pi} \frac{\delta(k^0 - \omega_\mathbf{k})}{2k^0} = \int \frac{d^3k}{(2\pi)^3} \frac{1}{2\omega_\mathbf{k}}. \tag{253}$$

This is a Lorentz-invariant integral measure. We then choose the square-root in the denominator for our field operators simply as a matter of convention, as it leads to simpler expressions down the road.

We've shown the canonical momentum is just $\dot{\phi}$, and if we take the time derivative of (250) we'll find

$$\pi(\mathbf{x}, t) = -i \int \frac{\mathrm{d}^3 k}{(2\pi)^3} \sqrt{\frac{\omega(\mathbf{k})}{2}} \left[ a(\mathbf{k}) e^{i k_\mu x^\mu} - a^\dagger(\mathbf{k}) e^{-i k_\mu x^\mu} \right]. \tag{254}$$

Now, let's promote $a(\mathbf{k})$ and $a^\dagger(\mathbf{k})$ to operators. If we insist that

$$[a(\mathbf{k}), a^\dagger(\mathbf{k}')] = (2\pi)^3 \delta^3(\mathbf{k} - \mathbf{k}') \tag{255}$$

we can work through the algebra and find that this implies

$$[\phi(\mathbf{x}, t), \pi(\mathbf{x}', t)] = i \delta^3(\mathbf{x} - \mathbf{x}'), \tag{256}$$

which is precisely the canonical commutation relation we wanted! Both the names and commutation relations (255) of the $a$ and $a^\dagger$ operators remind us of the raising and lowering operators from the normal quantum harmonic oscillator. In fact, they actually *are* the raising and lowering operators for a harmonic oscillator, except now we have an infinite number of harmonic oscillators: one for each mode, indexed by the wave-vector $\mathbf{k}$. However, they now come with a new interpretation, and a fancier name. We'll now call $a^\dagger$ and $a$ the *creation and annihilation operators* for the field, which are interpreted as creating and annihilating a particle with momentum $\mathbf{k}$. To further illustrate the similarity with the harmonic oscillator, we can put (250) and (254) into the Hamiltonian (247), and find

$$H = \int \frac{\mathrm{d}^3 k}{(2\pi)^3} \frac{1}{2} \omega(\mathbf{k}) \left[ a(\mathbf{k}) a^\dagger(\mathbf{k}) + a^\dagger(\mathbf{k}) a(\mathbf{k}) \right], \qquad \omega(\mathbf{k}) = \sqrt{\mathbf{k}^2 + m^2}. \tag{257}$$

In fact, there's an easier way to see this. Instead of considering a system of infinite volume (which we've been doing implicitly), let's consider putting the system in a box of size $L$ with periodic boundary conditions, just like we did previously when discussing the Fermi gas. We'll start with a finite size box, and then take the $L \to \infty$ limit at the end. The allowed momenta are quantized as

$$\mathbf{k} = \frac{2\pi n_x}{L} \hat{\mathbf{x}} + \frac{2\pi n_y}{L} \hat{\mathbf{y}} + \frac{2\pi n_z}{L} \hat{\mathbf{z}}, \tag{258}$$

for $n_x, n_y, n_z \in \mathbb{Z}$. The integral over momenta then becomes a sum over the allowed modes,

$$\int \frac{\mathrm{d}^3 k}{(2\pi)^3} \to \sum_{n_x} \sum_{n_y} \sum_{n_z}, \tag{259}$$

and we get one set of creation and annihilation operators per mode,

$$\begin{aligned} a(\mathbf{k}) &\to a_{n_x, n_y, n_z}, \\ a^\dagger(\mathbf{k}) &\to a^\dagger_{n_x, n_y, n_z}, \end{aligned} \tag{260}$$

each of which obeys the commutation relations

$$\left[ a_{n_x, n_y, n_z}, a^\dagger_{n'_x, n'_y, n'_z} \right] = \delta_{n_x, n'_x} \delta_{n_y, n'_y} \delta_{n_z, n'_z}, \tag{261}$$

which are exactly those of a harmonic oscillator. The Hamiltonian (257) becomes

$$H = \sum_{n_x} \sum_{n_y} \sum_{n_z} \frac{1}{2} \omega(\mathbf{k}_{n_x, n_y, n_z}) \left( a^\dagger_{n_x, n_y, n_z} a_{n_x, n_y, n_z} + a_{n_x, n_y, n_z} a^\dagger_{n_x, n_y, n_z} \right). \tag{262}$$

To avoid losing our eyesight (and sanity) from all of the subscripts on subscripts, we can rewrite this in terms of the allowed momenta $\mathbf{k}$, as long as we don't forget $\mathbf{k}$ now takes discrete values,

$$H = \sum_{\text{allowed } \mathbf{k}} \frac{1}{2}\omega(\mathbf{k})\left(a_{\mathbf{k}}^{\dagger}a_{\mathbf{k}} + a_{\mathbf{k}}a_{\mathbf{k}}^{\dagger}\right). \tag{263}$$

We can then use the commutator (261) to rewrite the second term as

$$a_{\mathbf{k}}a_{\mathbf{k}}^{\dagger} = a_{\mathbf{k}}^{\dagger}a_{\mathbf{k}} + [a_{\mathbf{k}}, a_{\mathbf{k}}^{\dagger}] = a_{\mathbf{k}}^{\dagger}a_{\mathbf{k}} + 1. \tag{264}$$

Putting this back into the Hamiltonian, we arrive at a familiar result,

$$H = \sum_{\text{allowed } \mathbf{k}} \omega(\mathbf{k})\left(a_{\mathbf{k}}^{\dagger}a_{\mathbf{k}} + \frac{1}{2}\right), \tag{265}$$

which is nothing more than the Hamiltonian for a bunch of harmonic oscillators! Just like in the normal harmonic oscillator, we can define the number operator,

$$n_{\mathbf{k}} = a_{\mathbf{k}}^{\dagger}a_{\mathbf{k}}, \tag{266}$$

which now has the interpretation of counting the number of particles in a given mode $\mathbf{k}$. The Hamiltonian is then

$$H = \sum_{\text{allowed } \mathbf{k}} \omega(\mathbf{k})\left(n_{\mathbf{k}} + \frac{1}{2}\right). \tag{267}$$

This has a nice interpretation. The energy of one particle with momentum $\mathbf{k}$ is $\omega(\mathbf{k})$, and $n_{\mathbf{k}}$ counts how many particles we have with momentum $\mathbf{k}$. So, the Hamiltonian is just the sum of the single particle energies $\omega(\mathbf{k})$ for each particle in the system. This agrees with the free theory representing non-interacting particles, since there are no contributions to the energy from the particles talking to one another. However, notice that when we have no particles in the system, i.e. when $n_{\mathbf{k}} = 0$ for all $\mathbf{k}$, there is still a contribution to the energy; this is just the energy of zero-point motion familiar from the quantum mechanical harmonic ocillator. However, in the present context there are an infinite number of harmonic oscillators, since there is one for each mode. Thus the zero-point motion contribution to the energy is

$$E_{\text{zero-point}} = \frac{1}{2}\sum_{\text{allowed } \mathbf{k}} \omega(\mathbf{k}), \tag{268}$$

which is not only non-zero, but is in fact divergent! However, just as before we can ignore this by chanting "only energy differences matter in physics" until the bad thoughts go away.

Before going back to the infinite volume case with a continuum of modes, let's note that we can write down a set of basis states as

$$|n_{\mathbf{k}_1}, n_{\mathbf{k}_2}, \ldots n_{\mathbf{k}_{\alpha}}, \ldots\rangle, \tag{269}$$

where $n_{\mathbf{k}_{\alpha}}$ is the number of particles in the $\alpha^{th}$ mode. This is called the *occupation number representation*. The energy difference between a state with the mode $\mathbf{k}_{\alpha}$ occupied and the vacuum (no modes occupied) is

$$E_{\mathbf{k}_{\alpha}} - E_{\text{zero-point}} = \sqrt{\mathbf{k}_{\alpha}^2 + m^2}. \tag{270}$$

Usually, we'll just drop the zero-point energy as a matter of convention. This is equivalent to setting the $E = 0$ bar (which are free to place wherever we like) to the zero-point energy.

Now, returning to the infinite volume case with a continuum of modes, we can build states in a similar way. First of all, we define the vacuum as the state with no particles. Formally, we

can define this just like we define the ground state of the harmonic oscillator, i.e. the vacuum is the state for which

$$a(\mathbf{k})|\text{vac}\rangle = 0, \quad \text{for all } \mathbf{k}. \tag{271}$$

To create a particle with momentum-space wavefunction $\phi_a(\mathbf{k})$ or $\phi_b(\mathbf{k})$ we can act on the vacuum with the *field operators*,

$$
\begin{aligned}
\hat{\phi}_a &= \int \frac{d^3 k}{(2\pi)^3}\ \phi_a(\mathbf{k}) a^\dagger(\mathbf{k}), \\
\hat{\phi}_b &= \int \frac{d^3 k}{(2\pi)^3}\ \phi_b(\mathbf{k}) a^\dagger(\mathbf{k}).
\end{aligned}
\tag{272}
$$

Let's interpret what this means. The momentum-space wave function $\phi_a(\mathbf{k})$ tells us "how much" of the state is in a given momentum, that is, its amplitude; $a^\dagger(\mathbf{k})$ creates an excitation in that momentum mode. So, the net effect is that the operator creates a particle which is "smoothly distributed" in momentum space by the momentum space wavefunction $\phi_a(\mathbf{k})$. That is, it gives us a wave-packet. Then, to get the quantum mechanical states, we just act with the field operators on the vacuum,

$$|\phi_a\rangle = \hat{\phi}_a|\text{vac}\rangle, \qquad |\phi_b\rangle = \hat{\phi}_b|\text{vac}\rangle. \tag{273}$$

We can also use a field operator to add a particle to a state which already has a particle in it, i.e. acting with $\hat{\phi}_a$ on $|\phi_b\rangle$ adds a particle with momentum-space wavefunction $\phi_a$ to a state that already has another particle with momentum-space wavefunction $\phi_b$. This is equivalent to acting on the vacuum with both field operators,

$$|\phi_a\phi_b\rangle = \hat{\phi}_a|\phi_b\rangle = \hat{\phi}_a\hat{\phi}_b|\text{vac}\rangle. \tag{274}$$

Now, the key to this game is the commutation relations. We know $[a^\dagger(\mathbf{k}), a^\dagger(\mathbf{k}')] = 0$, i.e. the creation operators always commute with one another. Looking back at (272), this means that the field operators also commute with one another,

$$[\hat{\phi}_a, \hat{\phi}_b] = 0. \tag{275}$$

Coming back to our two-particle state, this means that we can switch the order of the field operators, so

$$|\phi_a\phi_b\rangle = \hat{\phi}_a\hat{\phi}_b|\text{vac}\rangle = \hat{\phi}_b\hat{\phi}_a|\text{vac}\rangle = |\phi_b\phi_a\rangle. \tag{276}$$

So, we have shown that

$$|\phi_a\phi_b\rangle = |\phi_b\phi_a\rangle. \tag{277}$$

which means that if we switch the particles, the state is the same. So, we have shown that the particles described by this scalar field theory are bosons!! Given this tremendous success, it is now time to turn to fermions.

## 9.4 Quantizing Canonically: Fermion Fields

Luckily, the story for fermions is very similar to the one we've just been through, with one major difference: we'll replace all the commutators with anti-commutators. Just as the commutator is defined as

$$[A, B] = AB - BA, \tag{278}$$

we can also define the *anti-commutator*,

$$\{A, B\} = AB + BA. \tag{279}$$

Before getting too far into the details, let's see the effect of this new structure. Let us write the fermion field operators as

$$\hat{\psi}_a = \int \frac{d^3k}{(2\pi)^3}\, \psi_a(\mathbf{k})\, b^\dagger(\mathbf{k}),$$
$$\hat{\psi}_b = \int \frac{d^3k}{(2\pi)^3}\, \psi_b(\mathbf{k})\, b^\dagger(\mathbf{k}), \tag{280}$$

where we have named the fermion creation operators $b^\dagger$ instead of $a^\dagger$. For the bosonic theory we had $[a^\dagger(\mathbf{k}), a^\dagger(\mathbf{k}')] = 0$, and thus $[\hat{\phi}_a, \hat{\phi}_b] = 0$. So, for the fermions let's impose $\{b^\dagger(\mathbf{k}), b^\dagger(\mathbf{k}')\} = 0$ and see what happens. First of all, this implies that

$$\{\hat{\psi}_a, \hat{\psi}_b\} = 0, \tag{281}$$

which via the definition (279) can also be written $\hat{\psi}_b\hat{\psi}_a = -\hat{\psi}_a\hat{\psi}_b$. We'll define the states obtained by acting with the field operators on the vacuum as

$$|\psi_a\rangle = \hat{\psi}_a|\text{vac}\rangle, \qquad |\psi_b\rangle = \hat{\psi}_b|\text{vac}\rangle. \tag{282}$$

We can consider a two particle state,

$$|\psi_a\psi_b\rangle = \hat{\psi}_a\hat{\psi}_b|\text{vac}\rangle, \tag{283}$$

or, acting with the operators in the opposite order and using the anti-commutator (281), we have the state

$$|\psi_b\psi_a\rangle = \hat{\psi}_b\hat{\psi}_a|\text{vac}\rangle = -\hat{\psi}_a\hat{\psi}_b|\text{vac}\rangle = -|\psi_a\psi_b\rangle. \tag{284}$$

That is,

$$|\psi_b\psi_a\rangle = -|\psi_a\psi_b\rangle, \tag{285}$$

which is to say that switching the particles changes the state by a minus sign, or that the state is antisymmetric with respect to interchange. Regardless how we phrase it, this unambiguously tells us our particles are fermions! Specifically, if we take $\hat{\psi}_a = \hat{\psi}_b$, we have $|\psi_a\psi_a\rangle = -|\psi_a\psi_a\rangle$, which is only satisfied if

$$|\psi_a\psi_a\rangle = 0, \tag{286}$$

which means two fermions cannot occupy the same state. This is the Pauli Principle.

Now, onto the details: we'll start from the Dirac Lagrangian, which describes a free fermion field,

$$\mathcal{L} = \bar{\psi}(i\slashed{\partial} - m)\psi. \tag{287}$$

Since we've already been through this process, we'll move a little more quickly this time. The momentum conjugate to $\psi$ is

$$\pi = \frac{\partial \mathcal{L}}{\partial \dot{\psi}} = i\bar{\psi}\gamma^0 = i\psi^\dagger \underbrace{\gamma^0\gamma^0}_{\mathbb{1}} = i\psi^\dagger. \tag{288}$$

The Hamiltonian is

$$\mathcal{H} = \pi\dot{\psi} - \mathcal{L} = \psi^\dagger(-i\boldsymbol{\alpha}\cdot\nabla + \beta m)\psi, \tag{289}$$

where we've defined

$$\alpha_j = \gamma^0\gamma^j, \qquad \beta = \gamma^0, \tag{290}$$

which you will often see used in the literature.

The important point is that any $\psi$ that satisfies the Dirac equation also satisfies the Klein-Gordon equation, as we saw in section 7. This lets us decompose the field $\psi(\mathbf{x},t)$ into normal modes just as we did before, with a few added complications. We'll first have to introduce the basis spinors, $u(\mathbf{p},s)$ and $v(\mathbf{p},s)$, which are solutions to the Dirac equation with positive and negative energies, when combined with a plane wave, so

$$u(\mathbf{p},s)e^{-ip_\mu x^\mu}, \quad \text{and} \quad v(\mathbf{p},s)e^{ip_\mu x^\mu}, \tag{291}$$

are solutions to the Dirac equation, with positive and negative energy respectively. The index $s$ accounts for the spin of the particle, which can be up or down. A general spinor field may then be decomposed as

$$\psi(\mathbf{x},t) = \sum_{s=\uparrow,\downarrow} \int \frac{\mathrm{d}^3 p}{(2\pi)^3} \sqrt{\frac{m}{E(\mathbf{p})}} \left( b(\mathbf{p},s)u(\mathbf{p},s)e^{-ip_\mu x^\mu} + d^\dagger(\mathbf{p},s)v(\mathbf{p},s)e^{ip_\mu x^\mu} \right), \tag{292}$$

where $b(\mathbf{p},s)$ and $d^\dagger(\mathbf{p},s)$ are currently the amplitudes to occupy a given mode, and will shortly be promoted to operators when we quantize the theory. The dagger on $d$ is just a convention we've introduced to make later results work out simply, as is the overall normalization. It's straightforward to find the canonical momentum,

$$\pi(\mathbf{x},t) = i\psi^\dagger = i \sum_{s=\uparrow,\downarrow} \int \frac{\mathrm{d}^3 p}{(2\pi)^3} \sqrt{\frac{m}{E(\mathbf{p})}} \left( b^\dagger(\mathbf{p},s)u(\mathbf{p},s)e^{ip_\mu x^\mu} + d(\mathbf{p},s)v(\mathbf{p},s)e^{-ip_\mu x^\mu} \right). \tag{293}$$

Now, to quantize the theory we promote $b$ and $d$ to operators, and impose the *anti*-commutation relations,

$$\left\{ b(\mathbf{p},s), b(\mathbf{p}',s') \right\} = \left\{ b^\dagger(\mathbf{p},s), b^\dagger(\mathbf{p}',s') \right\} = 0, \tag{294}$$

$$\left\{ d(\mathbf{p},s), d(\mathbf{p}',s') \right\} = \left\{ d^\dagger(\mathbf{p},s), d^\dagger(\mathbf{p}',s') \right\} = 0, \tag{295}$$

$$\left\{ b(\mathbf{p},s), b^\dagger(\mathbf{p}',s') \right\} = \left\{ d(\mathbf{p},s), d^\dagger(\mathbf{p}',s') \right\} = (2\pi)^3 \delta^3(\mathbf{p}-\mathbf{p}')\delta_{s,s'}. \tag{296}$$

You're invited to confirm that if these hold, then the canonical anti-commutation relation

$$\left\{ \psi(\mathbf{x},t), \pi(\mathbf{x}',t) \right\} = i\delta^3(\mathbf{x}-\mathbf{x}') \tag{297}$$

is satisfied. The Hamiltonian can then be written

$$H = \sum_{s=\uparrow,\downarrow} \int \frac{\mathrm{d}^3 p}{(2\pi)^3} E(\mathbf{p}) \left( \underbrace{b^\dagger(\mathbf{p},s)b(\mathbf{p},s)}_{\substack{\text{positive} \\ \text{energy}}} - \underbrace{d(\mathbf{p},s)d^\dagger(\mathbf{p},s)}_{\substack{\text{negative} \\ \text{energy}}} \right). \tag{298}$$

To gain some intuition, we'll once again put our system in a finite box so the momentum is quantized into discrete modes, and the Hamiltonian becomes

$$H = \sum_{s=\uparrow,\downarrow} \sum_{\text{allowed } \mathbf{p}} E(\mathbf{p}) \left( b^\dagger_{\mathbf{p},s} b_{\mathbf{p},s} - d_{\mathbf{p},s} d^\dagger_{\mathbf{p},s} \right), \tag{299}$$

where we've written the momenta as subscripts to remind us they now take discrete values. For a moment, let's drop the spin label and consider a single mode of wave-vector $\mathbf{k}$. If we let $|0\rangle$ be the state with no particles, we can see that

$$b^\dagger_{\mathbf{k}}|0\rangle = |1\rangle \qquad \text{creates a particle in the mode},$$

$$b_{\mathbf{k}}|1\rangle = |0\rangle \qquad \text{annihilates a particle in the mode}.$$

To see that the operators represent fermions, we'll use the anti-commutator $\{b_{\mathbf{k}}^{\dagger}, b_{\mathbf{k}'}^{\dagger}\} = 0$, which means that $b_{\mathbf{k}'}^{\dagger} b_{\mathbf{k}}^{\dagger} = -b_{\mathbf{k}}^{\dagger} b_{\mathbf{k}'}^{\dagger}$. Specifically, if $\mathbf{k} = \mathbf{k}'$ this means $b_{\mathbf{k}}^{\dagger} b_{\mathbf{k}}^{\dagger} = -b_{\mathbf{k}}^{\dagger} b_{\mathbf{k}}^{\dagger}$, which is only possible if $b_{\mathbf{k}}^{\dagger} b_{\mathbf{k}}^{\dagger} = 0$. With this in mind, let's try to add a particle into a mode that's already occupied,

$$b_{\mathbf{k}}^{\dagger}|1\rangle = \underbrace{b_{\mathbf{k}}^{\dagger} b_{\mathbf{k}}^{\dagger}}_{0}|0\rangle = 0. \tag{300}$$

So, we can't put two particles into the same mode, which tells us the particles we are dealing with are fermions. We can also use these relations to show that just as in the bosonic case, the operator $n_{\mathbf{k}} = b_{\mathbf{k}}^{\dagger} b_{\mathbf{k}}$ counts the number of particles in a mode. However, $n_{\mathbf{k}}$ now only has eigenvalues 0 and 1:

$$\begin{aligned} b_{\mathbf{k}}^{\dagger} b_{\mathbf{k}}|0\rangle &= 0|0\rangle, \\ b_{\mathbf{k}}^{\dagger} b_{\mathbf{k}}|1\rangle &= 1|1\rangle. \end{aligned} \tag{301}$$

Now, coming back to our mode decomposition (292), we treated the positive and negative energy states differently. That is, the positive energy states were written with an annihilation operator $b(\mathbf{p}, s)$, whereas the negative energy states were written with a creation operator, $d^{\dagger}(\mathbf{p}, s)$. There isn't any deep reason for this, its simply a matter of convention: the ant-commutation relations don't care what we call our operators. However, writing it this way makes things clearer, since having quantized the theory we will now interpret $d^{\dagger}$ as the creation operator for an anti-particle.

Recalling our discussion of the Dirac sea, we expect all of the negative energy states to be occupied in the vacuum. In the occupation number representation, we can write a general state as

$$|\underbrace{\ldots, n_{-\alpha}, \ldots n_{-2}, n_{-1}}_{\substack{\text{negative} \\ \text{energy} \\ \text{states}}}, \underbrace{n_1, n_2, \ldots n_{\beta}, \ldots}_{\substack{\text{positive} \\ \text{energy} \\ \text{states}}}\rangle. \tag{302}$$

In this notation, the vacuum (with all negative energy states filled), is written

$$|\text{vac}\rangle = |\underbrace{1, 1, 1, \ldots}_{\substack{\text{negative} \\ \text{energy} \\ \text{states}}}, \underbrace{0, 0, 0, \ldots}_{\substack{\text{positive} \\ \text{energy} \\ \text{states}}}\rangle, \tag{303}$$

which means that

$$\begin{aligned} b_{\mathbf{k}}|\text{vac}\rangle &= 0, \\ d_{\mathbf{k}}^{\dagger}|\text{vac}\rangle &= 0, \end{aligned} \tag{304}$$

for all modes. We can explicitly see the existence of the Dirac sea by using the anti-commutator $\{d_{\mathbf{p},s}, d_{\mathbf{p},s}^{\dagger}\} = 1$ (the discrete version of (296)) in the Hamiltonian (299),

$$\begin{aligned} H &= \sum_{s=\uparrow,\downarrow} \sum_{\text{allowed } \mathbf{p}} E(\mathbf{p})\left( b_{\mathbf{p},s}^{\dagger} b_{\mathbf{p},s} + d_{\mathbf{p},s}^{\dagger} d_{\mathbf{p},s} - \underbrace{\{d_{\mathbf{p},s}, d_{\mathbf{p},s}^{\dagger}\}}_{1} \right) \\ &= \sum_{s=\uparrow,\downarrow} \sum_{\text{allowed } \mathbf{p}} \sqrt{\mathbf{p}^2 + m^2}\left( b_{\mathbf{p},s}^{\dagger} b_{\mathbf{p},s} + d_{\mathbf{p},s}^{\dagger} d_{\mathbf{p},s} \right) - \underbrace{\sum_{s=\uparrow,\downarrow} \sum_{\text{allowed } \mathbf{p}} \sqrt{\mathbf{p}^2 + m^2}}_{\substack{\text{negative energy of} \\ \text{filled Dirac sea}}}. \end{aligned} \tag{305}$$

When we have no particles in the system, i.e. the vacuum, the first terms vanish and the energy is given by the last term, which we have identified as the negative energy due to the

filled Dirac sea. Again, this is an infinite quantity, but since it is constant we can appropriately calibrate our energy scales such that we can drop this term. In the event we do have particles in the system, the $b^\dagger_{\mathbf{p},s} b_{\mathbf{p},s}$ term counts the number of particles in each mode, and the $d^\dagger_{\mathbf{p},s} d_{\mathbf{p},s}$ term counts the number of anti-particles in each mode, both of which are multiplied by the single particle energy. This reflects the non-interacting nature of our system of fermions.

## 9.5 Adding Interactions: Perturbation Theory and Feynman Diagrams

So far, we've seen how to solve non-interacting ("free") theories of bosons and fermions, but we haven't said anything about interactions. Unfortunately, once we turn on interactions the theory usually can't be solved. However, all hope is not lost because there is often a reliable way to *approximate* the answer.

In fact, regardless of whether or not we have a good approximation scheme at our disposal, the *structure* of the results for physical processes (scattering, decays, etc.) is well understood. In general, we have something like

$$\text{physical result} = (\text{kinematic factor}) \cdot |\mathcal{M}|^2, \tag{306}$$

where the "kinematic factor" is a frame-dependent factor that is usually obtained from Fermi's Golden rule,[18] and is sometimes called the "phase space factor." We also have the all important $\mathcal{M}$, which is a Lorentz-invariant matrix element between the ingoing and outgoing states. For example, let's suppose we're interested in a decay from particle 1 with (four-) momentum $p_1$ into particles $1', 2', \ldots N'$, each with (four-) momentum $p_{1'}, p_{2'}, \ldots p_{N'}$. The decay rate for this process for a given set of momenta is

$$d\Gamma = |\mathcal{M}|^2 (2\pi)^4 \underbrace{\delta^4(p_1 - p_{1'} - p_{2'} - \cdots - p_{N'})}_{\text{momentum conservation}} \frac{1}{2E_1} \frac{d^3\mathbf{p}_{1'}}{(2\pi)^3 2E_{1'}} \frac{d^3\mathbf{p}_{2'}}{(2\pi)^3 2E_{2'}} \cdots \frac{d^3\mathbf{p}_{N'}}{(2\pi)^3 2E_{N'}}, \tag{307}$$

where $E_n = \sqrt{\mathbf{p}_n^2 + m_n^2}$. We can then find the decay rate by integrating over all of the outgoing momenta,

$$\Gamma = \int d\Gamma = \int |\mathcal{M}|^2 (2\pi)^4 \delta^4(p_1 - p_{1'} - p_{2'} - \cdots - p_{N'}) \frac{1}{2E_1} \frac{d^3\mathbf{p}_{1'}}{(2\pi)^3 2E_{1'}} \frac{d^3\mathbf{p}_{2'}}{(2\pi)^3 2E_{2'}} \cdots \frac{d^3\mathbf{p}_{N'}}{(2\pi)^3 2E_{N'}}. \tag{308}$$

So, assuming that we can look up the appropriate formula for the kinematic factor, the real question becomes how to calculate the matrix element $\mathcal{M}$, and this is where the approximations come in. One of the most widely used approximation schemes is that of *perturbation theory*, which we can use if the interaction term is small. For example, in electrodynamics most interactions are proportional to $e^2$, where $e^2/4\pi = \alpha \approx 1/137$ is a small number, making perturbation theory valid in many cases.

Let's first briefly remember some basic facts about perturbation theory in normal quantum mechanics. The words "perturbation theory" probably make you think of something like this,

$$\sum_n \frac{|\langle n|H_I|\psi\rangle|^2}{E_\psi - E_n}, \tag{309}$$

where just to clarify notation, $\{|n\rangle\}$ are energy eigenstates of the non-interacting Hamiltonian, $H_0|n\rangle = E_n|n\rangle$, $H_I$ is the interaction Hamiltonian, and the full Hamiltonian is $H = H_0 + H_I$. This term has a nice interpretation. The squared matrix element upstairs can be

---

[18]Fermi's Golden rule relates transition rates to the square of the transition matrix element and the density of states. It is derived in standard introductory quantum mechanics textbooks.

written $\langle\psi|H_I|n\rangle\langle n|H_I|\psi\rangle$, which can be thought of as a transition from the original state $|\psi\rangle$ into some other state $|n\rangle$, and then back into $|\psi\rangle$, all mediated by the interaction $H_I$. The system is said to be in a *virtual state* when it is in the intermediate state $|n\rangle$. Note that the energy of the virtual state is different from that of $|\psi\rangle$, so the likelihood of the transition is suppressed by the energy difference, which the virtual state must "borrow" from the vacuum, and then return when it transitions back into $|\psi\rangle$.

However, the state can't borrow momentum from the vacuum, so (presuming the theory preserves momentum and that momentum is a good quantum number for the states) the momentum of the initial and virtual states must be the same. This means that when we sum over the possible intermediate states, we should also integrate over momentum. One can use this sort of calculational scheme (sometimes called "time-ordered" perturbation theory) in field theory, but it turns out to be rather tedious and painful.

A better approach is to note that we are interested in relativistic theories, so a manifestly covariant formalism is ideal. Such a formalism was developed by Feynman, Schwinger, and Tomonaga, and was worth a Nobel prize for them. The idea is to work in 4-momentum space, and integrate over both momentum *and* energy. We then require that interactions conserve both energy and momentum, i.e. that the virtual states have the same energy and momentum as the initial state. However, this requires that the virtual states don't always satisfy the usual relativistic dispersion, $E^2 = \mathbf{p}^2 + m^2$, or $p_\mu p^\mu = m^2$. Because states which satisfy $p_\mu p^\mu = m^2$ trace out a sphere in 4-momentum space, states for which this does not hold are said to be "off the mass shell," or just "off shell."

Doing a calculation in covariant perturbation theory essentially amounts to doing one big Taylor expansion in powers of various small quantities. To organize the terms in this expansion, Feynman realized that the same cast of characters appears in every term, and usually in particular combinations. He then assigned a symbol to each of the usual suspects, which when put together allow us to represent each term in the expansion as a diagram. To work out the details of how and why this works takes a bit of effort, so instead of getting into the nitty-gritty details, we'll just state how Feynman diagrams work.

Roughly speaking, there are two main ingredients in a perturbation theory expansion: propagators and interactions. A *propagator* is exactly what it sounds like, it represents the free (non-interacting) propagation of a particle. Mathematically, it corresponds to the Green's function of whatever operator appears in the non-interacting part of the Lagrangian. For example, the non-interacting part of the Lagrangian for a scalar field is $\partial_\mu \phi \partial^\mu \phi - m^2 = -\phi(\partial_\mu \partial_\mu + m^2)\phi$, where in the second equality we integrated parts.[19] Then, the propagator is the inverse of the operator $\partial_\mu \partial^\mu + m^2$, which our mathematically inclined friends call a *Green function*.[20] Although one can develop perturbation theory in real space, it is *far* simpler to work in momentum space, so the propagators we'll talk about are actually the inverses of the free part of the Lagrangian in momentum space. For a real scalar field, which represents a spin zero boson, we represent the propagator by a dashed line,

$$\text{-------------} \quad = \quad \frac{i}{p^2 - m^2 + i\varepsilon}. \tag{310}$$

The value of the propagator isn't too hard to calculate, if you use the fact that in momentum space $\partial_\mu \mapsto i p_\mu$. In the denominator, $p^2$ without the vector boldface is shorthand for $p_\mu p^\mu$ and $\varepsilon$ is an infinitesimal quantity which among other things *encodes causality*.

---

[19]We can always integrate by parts inside a Lagrangian, since it appears inside an integral in the action. Further, we can almost always drop the boundary term since the fields should vanish at infinity. However, in some theories where topology plays a central role these boundary terms can matter. Such topological effects cannot be captured by Feynam digrams.

[20]The name comes from the mathematician who invented these functions: there is no deep physical or mathematical significance to the color green.

Since the free part of the Lagrangian is different for different particles, the propagators will also be different. As a second example, the propagator for a spin 1/2 fermion is represented by a solid line and has the value (remember that $\not{p} = \gamma^\mu p_\mu$)

$$\underline{\hspace{3cm}} = \frac{i}{\not{p} - m + i\varepsilon} = \frac{i(\not{p} + m)}{p^2 - m^2 + i\varepsilon}. \tag{311}$$

Next, we'll introduce *interactions*, which are determined by the rest of the Lagrangian (the non-non-interacting part). In general, an interaction is a *vertex* in the diagram, where multiple lines (propagators) meet. How many propagators, and for which particles, are allowed in the interaction is determined by the form of the interaction term. This is most easily illustrated by example: suppose we have the interaction[21]

$$\mathcal{L}_{\text{int}} = g\phi\bar{\psi}\psi. \tag{312}$$

Since we have one scalar field and two fermion fields (one barred, and one not), the interaction is between one scalar and two fermion fields. Diagrammatically, we draw a vertex where two fermion propagators (one incoming, one outgoing) and one scalar propagator meet, and assign value $g$ to the vertex, i.e

$$\sim \quad g. \tag{313}$$

As a second example, consider the $\phi^4$ interaction we've already met,

$$\mathcal{L}_{\text{int}} = \frac{\lambda}{4!}\phi^4. \tag{314}$$

This describes an interaction between four scalar fields, and is represented by the diagram

$$\sim \quad \lambda. \tag{315}$$

To calculate a matrix element in perturbation theory, we first decide which parameters we want to expand in, and then to what order we wish to calculate.[22] For a process that begins with a set $A$ of particles and ends with a set $B$ of particles, we then draw all distinct diagrams that have the set $A$ of initial particles, and the set $B$ of final particles, as allowed by the interactions in the theory and up to the desired order. At each interaction, we impose momentum and energy conservation. Any undetermined propagators on the interior of the diagram (i.e. in virtual states) are to be integrated over in 4-momentum space. In contrast, the propagators corresponding to the initial and final states ("the external legs") are not counted. The matrix element is then the sum of all such diagrams.

This is all made much clearer by a few examples. Let's take the $\phi^4$ theory, and compute the scattering amplitude for the process $\phi\phi \to \phi\phi$, where the ingoing particles have four-momenta $p_1$ and $p_2$, and the outgoing particles have momenta $p'_1$ and $p'_2$. Since these particles

---

[21]This is usually called a *Yukawa interaction.*

[22]Its worth noting that there exist techniques perform infinite-order expansions by resumming classes of diagrams. This may sound fancy, but it basically comes down to the geometric series you learned about in calculus class. These infinite-order resummations are essential for studying many-body problems, which arise in condensed matter and nuclear physics.

are on shell, we must have $p_1^2 = p_2^2 = (p_1')^2 = (p_2')^2 = m^2$. Let's assume $\lambda$ is a small parameter such that perturbation theory is valid. Then, to first order in $\lambda$, there is only one diagram we can draw,

$$
\sim \quad \lambda. \tag{316}
$$

Since there are no internal lines and the external legs do not contribute, the matrix element is just the factor of $\lambda$ we get from the interaction. Thus, to first order, $\mathcal{M}$ is momentum independent, and $|\mathcal{M}|^2 = \lambda^2$.

To second order in $\lambda$, we have two diagrams

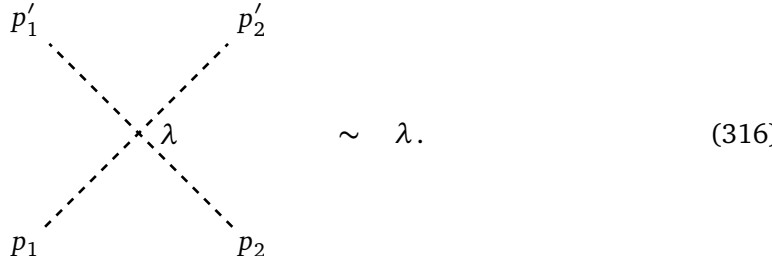

$$
= \int \frac{\mathrm{d}^4 k}{(2\pi)^4} \frac{i^2 \lambda^2}{((p_1+k)^2 - m^2 + i\varepsilon)((p_2-k)^2 - m^2 + i\varepsilon)},
$$

$$\tag{317}$$

$$
= \int \frac{\mathrm{d}^4 k}{(2\pi)^4} \frac{i^2 \lambda^2}{(k^2 - m^2 + i\varepsilon)^2}. \tag{318}
$$

The initial and final momenta are on shell, i.e. $p_1^2 = p_2^2 = (p_1')^2 = (p_2')^2 = m^2$, but the momentum $k$ inside the diagram is unconstrained, and thus is integrated over. Now, at second order the matrix element *is* momentum dependent, as we would expect. Finally, note that even at second order in $\lambda$ the theory includes scattering processes that do not conserve particle number, such as the $\phi\phi \to \phi\phi\phi\phi$ process given by the diagram

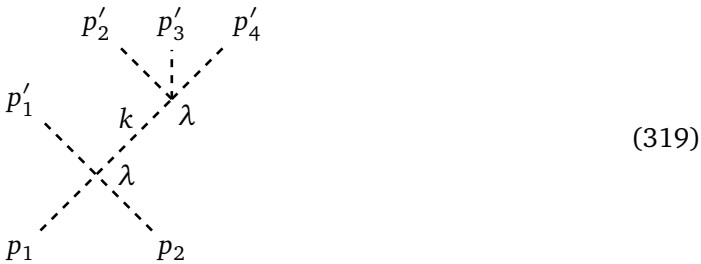

$$\tag{319}$$

This may be surprising, especially considering the classical Lagrangian only included a two-to-two interaction. In principle, these *Feynman rules* are sufficient to calculate the matrix element for any process within perturbation theory. However, it turns out there is a big problem!

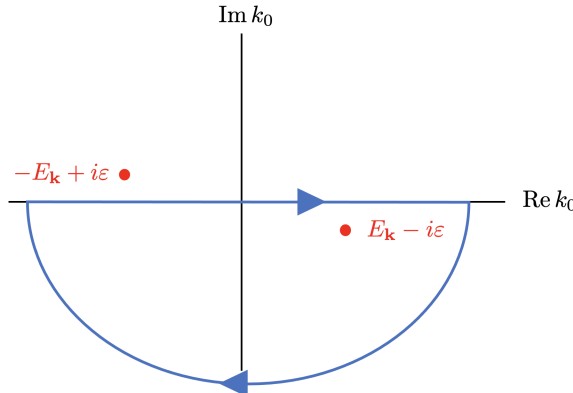

Figure 10: The integration contour in the complex $k_0$ plane, and the poles of the integrand

## 9.6   R is for Renormalization

Let's now consider the "two point" function, which includes perturbative corrections to the mass term. To zeroth order in $\lambda$, it's just the bare $m^2$ from the Lagrangian. To first order, we have the diagram

$$= \lambda \int \frac{\mathrm{d}^4 k}{(2\pi)^4} \frac{i}{k_\mu k^\mu - m^2 + i\varepsilon}. \tag{320}$$

This diagram acts just like a mass term. Note that since the momentum $k$ inside the loop is unconstrained we must integrate it over all possible values. Let's try computing this integral. We can write it as

$$\begin{aligned}
\Delta m^2 &= \lambda \int \frac{\mathrm{d}^3 k}{(2\pi)^3} \frac{\mathrm{d}k_0}{2\pi} \frac{i}{k_0^2 - \underbrace{(\mathbf{k}^2 + m^2)}_{E_\mathbf{k}^2} + i\varepsilon} \\
&= \int \frac{\mathrm{d}^3 k}{(2\pi)^3} \frac{\mathrm{d}k_0}{2\pi} \frac{i}{k_0^2 - E_\mathbf{k}^2 + i\varepsilon} \\
&= \int \frac{\mathrm{d}^3 k}{(2\pi)^3} \frac{\mathrm{d}k_0}{2\pi} \frac{i}{(k_0 - E_\mathbf{k} + i\varepsilon)(k_0 + E_\mathbf{k} - i\varepsilon)}.
\end{aligned} \tag{321}$$

It's now straightforward to do the $k_0$ integral. Since the integral goes to zero like $1/k_0$ as $k_0 \to \infty$, we can replace our integral over the real line with the contour integral through the lower half complex plane shown in Fig. 10. The $i\varepsilon$ factor moves the poles off the contour, and we can easily compute the integral using the Residue theorem,

$$\begin{aligned}
\int \frac{\mathrm{d}k_0}{2\pi} \frac{i}{(k_0 - E_\mathbf{k} + i\varepsilon)(k_0 + E_\mathbf{k} - i\varepsilon)} &= -2\pi i \cdot \text{Residue} \\
&= -2\pi i \frac{1}{2\pi} \frac{i}{k_0 + E_\mathbf{k} + i\varepsilon}\bigg|_{k_0 = E_\mathbf{k} - i\varepsilon} \\
&= \frac{1}{2E_\mathbf{k}}.
\end{aligned} \tag{322}$$

Now we just have to do the integral over the spatial components,

$$
\begin{aligned}
\Delta m^2 &= \lambda \int \frac{\mathrm{d}^3 k}{(2\pi)^3} \frac{1}{2E_{\mathbf{k}}} \\
&= \frac{\lambda}{2} \int \frac{\mathrm{d}^3 k}{(2\pi)^3} \frac{1}{\sqrt{\mathbf{k}^2 + m^2}} \\
&= \frac{\lambda}{2} \int_0^\infty \underbrace{\frac{4\pi \mathbf{k}^2 \mathrm{d}k}{(2\pi)^3}}_{\substack{\text{spherical} \\ \text{coords}}} \frac{1}{\sqrt{\mathbf{k}^2 + m^2}} \\
&= \frac{\lambda}{4\pi^2} \int_0^\infty \mathrm{d}k \, \frac{k^2}{\sqrt{k^2 + m^2}} \\
&= \frac{\lambda}{4\pi^2} \left[ \frac{1}{2} k \sqrt{k^2 + m^2} - \frac{m^2}{2} \log\left(k^2 + \sqrt{k^2 + m^2}\right) \right]_0^\infty \\
&= \infty . \qquad \textcolor{red}{\text{OOPS!!}}
\end{aligned}
\tag{323}
$$

Obviously, infinities are not good! Essentially, they arise due to fluctuations of the large momentum (or equivalently, short wavelength) modes, and are called *UV divergences*. It turns out the problem is not the formalism of covariant perturbation theory, but rather these infinities are a sickness of the particular quantum field theory we're studying. But does this mean we should throw it away? Luckily, not always.

We can salvage our theories using the process of *renormalization*, in which we note that the values of the parameters we put into the Lagrangian may not be the same as their physical values. For example, the mass in the Lagrangian may not actually be the physical mass of the particle that we measure in the lab. In fact, they can be changed by any (even infinite[23]) amount in each order of perturbation theory due to virtual processes. For example, if we call the mass in the Lagrangian $m_0$, the physical mass can be written

$$
m^2 = m_0^2 + \lambda \, (\text{loop contribution}) + \lambda \Delta m^2 .
\tag{324}
$$

The second term is from the divergent diagram we just computed, and the last term is a *counterterm*, which we can choose to cancel the loop diagram when the system is on shell. However, this doesn't mean we've completely thrown away the impact of the loop diagram: the cancellation only occurs when $k^2 = m^2$, so it can still have an effect when the system is off-shell. To first order in $\lambda$ the propagator gets corrected to

$$
\text{-------------} \quad = \quad \frac{i}{k^2 - m_0^2 + \lambda \, \Sigma(k^2) + i\varepsilon} ,
\tag{325}
$$

where the *self energy* $\Sigma(k^2)$ vanishes when $k^2 = m^2$, such that it only contributes when the particle is off-shell.

However, this procedure will not necessarily fix every theory. The theory is *renormalizable* only if all of its infinities can be absorbed into a finite number of parameters that are already in the theory (or should have been). On the other hand, if we need an infinite number of different parameters (and hence an infinite number of terms needed to be renormalized) the theory is *non-renormalizable*, since it is not predictive.

So, we would now very much like to know whether a given theory is renormalizable. A litmus test does exist, but the proof that it works is fairly subtle, so we'll just state the

---

[23]Strictly speaking, it may change by amounts comparable to the "ultraviolet cutoff," which usually ends up getting sent to infinity

result here, but before doing so we need to develop some new vocabulary. Let's recall the fundamental quantity in our quantum field theory is the action,

$$S = \int \mathrm{d}^4 x \, \mathcal{L}. \tag{326}$$

The action has dimensions (the fancy word for units) of energy × time, which are the dimensions of $\hbar$. But, in our choice of units $\hbar = 1$, so the action is dimensionless. One of the nice features of natural units is that the dimension of any quantity can be written as mass to some power. So, instead of having to think about messy things like units, we can characterize the dimension of any quantity by a single number: its "mass dimension," i.e. we say a quantity has dimension $d$ if its units are (mass)$^d$. If we denote the dimension of the action as $[S]$, this means that $[S] = 0$. By definition, $[m] = 1$, and it follows that $[x] = -1$, $[\partial_\mu] = 1$, etc.

Now, turning back to the action, we know $[S] = 0$, and the integral measure has $[\mathrm{d}^4 x] = -4$. This means that for the above expression to make sense, the Lagrangian must have dimension $[\mathcal{L}] = 4$. The Lagrangian is a sum of terms, and we can break each term into two ingredients. First of all, we have an *operator* which is some combination of fields and their derivatives. For example, things like $\phi^2$, $\partial_\mu \phi \partial^\mu \phi$, $\phi \bar{\psi} \psi$, and $\phi^4 (\partial_\mu \phi)^8$ are all operators. The second set of ingredients is the *coupling constants*, which are numerical coefficients that multiply the operators. For example, the term $m^2 \phi^2$ is comprised of the operator $\phi^2$ and coupling constant $m^2$. Or, for $\lambda \phi^4$, the operator is $\phi^4$ and the coupling constant is $\lambda$.

We can now state the criteria which helps determine whether a theory is renormalizable, which turns out to be very simple: *a theory is (perturbatively) renormalizable if the dimension of all operators in the Lagrangian is less than or equal to the dimension of spacetime.*[24] Note that since every term in the Lagrangian must have dimension four (in our world, where the dimension of space-time is four) this condition is equivalent to every coupling constant having a positive (or zero) dimension.

To assess the dimension of a given term, we first need to determine the dimension of the fields. For a free boson the kinetic term is

$$\mathcal{L} \sim \partial_\mu \phi \partial^\mu \phi. \tag{327}$$

We know that $[\partial_\mu] = 1$ and that every term in the Lagrangian must have $[\mathcal{L}] = 4$, thus $[\phi] = (4-2)/2 = 1$. For a fermion, the kinetic term is

$$\mathcal{L} \sim \bar{\psi} i \slashed{\partial} \psi, \tag{328}$$

from which we see that $[\psi] = (4-1)/2 = 3/2$. With this information, we may now find the dimension of any term in the Lagrangian of a theory for boson and fermion fields. It's then easy to see that mass terms such as $m^2 \phi^2$ or $m \bar{\psi} \psi$ are renormalizable, since the operators have dimension 2 and 3 respectively, or equivalently the coupling constant for both terms has positive dimension. Next, let's consider the $\phi^4$ interaction,

$$\mathcal{L} \sim \lambda \phi^4. \tag{329}$$

We see the operator $\phi^4$ has dimension 4, and the coupling constant is dimensionless (dimension zero), and thus the term is renormalizable. On the other hand, if we had a four fermion interaction such as

$$\mathcal{L} \sim G \bar{\psi} \psi \bar{\psi} \psi, \tag{330}$$

---

[24]Strictly speaking, this is a necessary but not always sufficient condition. Particularly, in the presence of strong interactions this rule may not always hold.

the dimension of the operator is 6 and the dimension of the coupling constant is $-2$, so we see this interaction is *not* renormalizable.

This result is important: Fermi's theory of weak interactions is of the four-fermion type. Since it has a neutron decaying into a proton, an electron and an anti-neutrino, it contains a term proportional to $\overline{\psi}_P \psi_N \overline{\psi}_e \psi_\nu$. This implies that the Fermi theory is not renormalizable and hence is not a fundamental theory of nature.

As our last two examples we can consider the Yukawa interaction

$$\mathcal{L} \sim g\phi\bar{\psi}\psi, \tag{331}$$

which has an operator of dimension 4 and dimensionless coupling constant, and is therefore renormalizable, while a term such as

$$\mathcal{L} \sim \kappa\phi^2\bar{\psi}\psi, \tag{332}$$

has an operator with dimension 5 and coupling constant of dimension $-1$, and is not renormalizable.

Renormalizability is a crucial property of any fundamental theory, and in fact was a guiding principle in constructing the standard model. However, even a non-renormalizable theory can be useful: these are "effective field theories" which are valid at low momenta and low order in perturbation theory. For example, we've seen a four fermion interaction is nonrenormalizable, and that the Yukawa coupling is. The diagrams for these processes are

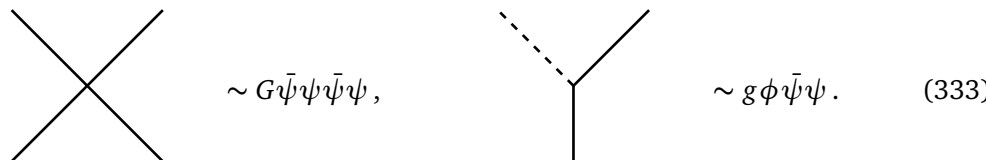

$$\sim G\bar{\psi}\psi\bar{\psi}\psi, \qquad\qquad\qquad \sim g\phi\bar{\psi}\psi. \tag{333}$$

However, a theory with a Yukawa coupling has an effective four-fermion interaction mediated by the boson,

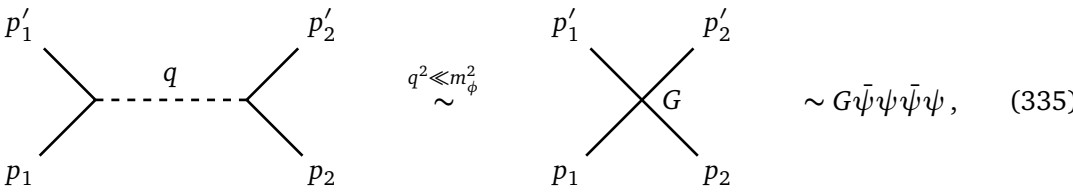

$$\sim \frac{-g^2}{q^2 - m_\phi^2 + i\varepsilon}, \tag{334}$$

where $m_\phi$ is the mass of the boson and $q$ is the momentum transfer, i.e. $p'_1 = p_1 + q$ and $p'_2 = p_2 - q$. If we are only interested in physics at low momenta compared to the boson mass such that $q^2 \ll m_\phi^2$, then to very good approximation this looks like a four-fermion interaction,

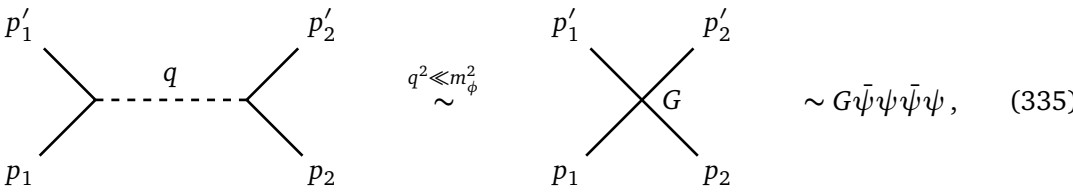

$$\sim G\bar{\psi}\psi\bar{\psi}\psi, \tag{335}$$

with the effective coupling constant given by

$$G = \frac{g^2}{m_\phi^2}, \tag{336}$$

which is the low momentum limit of the initial interaction. One can only see the difference between the two interactions if the system is probed at momentum scales comparable to $m_\phi$, so it serves as a viable effective model at low energies. Effective field theories such as this one have been an invaluable tool in understanding low energy physics, and are useful not only in particle physics; they play a major role in condensed matter as well.

In the context of particle physics this explains why Fermi's theory of weak-interactions can capture much of what is happening even though it is not renormalizable and not a fundamental theory of nature. In fact something very much like the process above happens in the standard model: the weak decay is mediated by the virtual exchange of a heavy boson: the W. There are number of other refinements as well, among them is the fact the fundammental process happens at the level of quarks not nucleons

# 10 Symmetries in Nuclear and Particle Physics

The importance of symmetry in physics is hard to overstate. In this section, we will try to show why. First of all, it is worth clarifying what exactly a symmetry is. Simply put, *a symmetry is an operation which leaves a system invariant*. For example, we can rotate a square by 90° and it will look the same. Since only rotations by a multiple of 90° will leave the square invariant, the symmetry is said to be *discrete*. On the other hand, we can rotate a circle by any amount and it will always look the same, and its symmetry is said to be *continuous*. These are both examples of symmetries of objects in real space, but much of our later discussion will consist of more abstract symmetries that operate on "internal" spaces.

We'll start by considering continuous symmetries in classical point-particle physics before applying them to field theory. We'll then introduce the notion of internal symmetries, and finally discuss two discrete symmetries central to particle physics: parity and time reversal. Symmetry will remain a central theme of the following sections as well. In section 11 we'll discuss the idea of a gauge symmetry, in section 12 the consequence of "breaking" a symmetry, and finally in section 13 we will introduce the Higgs mechanism, which is inseparable from the physics of gauge symmetry.

## 10.1 Continuous Spacetime Symmetries

We'll start by considering some of the most intuitive symmetries: those that act on the space-time in which we live. Specifically, we'll consider *continuous* symmetries in the familiar context of classical particle mechanics. As you may be aware, continuous symmetries are associated with conservation laws.

For example, time translation symmetry is associated with energy conservation. To see this, consider a Lagrangian of generalized coordinates $\mathbf{q}$ and their velocities $\dot{\mathbf{q}}$, that may be explicitly time dependent, $L(\mathbf{q}, \dot{\mathbf{q}}, t)$. From this we may construct the Hamiltonian,

$$H(\mathbf{q}, \mathbf{p}, t) = \sum_i p_i q_i - L(\mathbf{q}, \dot{\mathbf{q}}, t), \qquad p_i = \frac{\partial L}{\partial \dot{q}_i}. \tag{337}$$

If we differentiate the Hamiltonian with respect to time, we have

$$\frac{\mathrm{d}H}{\mathrm{d}t} = \sum_i \left[ \dot{p}_i \dot{q}_i + p_i \ddot{q}_i - \frac{\partial L}{\partial q_i} \dot{q}_i - \underbrace{\frac{\partial L}{\partial \dot{q}_i}}_{p_i} \ddot{q}_i \right] - \frac{\partial L}{\partial t}. \tag{338}$$

The second and last terms cancel, and when the equations of motion are satisfied,

$$\frac{\partial L}{\partial q_i} = \frac{\mathrm{d}}{\mathrm{d}t} \frac{\partial L}{\partial \dot{q}_i} = \dot{p}_i, \tag{339}$$

the third term can be written $-\dot{p}_i \dot{q}_i$, which cancels the first term, leaving us with

$$\frac{\mathrm{d}H}{\mathrm{d}t} = -\frac{\partial L}{\partial t}. \tag{340}$$

If the Lagrangian is not explicitly time dependent then $\partial L / \partial t = 0$, and the system is said to be time translation invariant (since the Lagrangian does not change from one moment of time to the next), and we have

$$\frac{\mathrm{d}H}{\mathrm{d}t} = 0, \tag{341}$$

which is to say that the energy of the system is conserved. Thus, time translation invariance leads to energy conservation. We can also see that spatial translation invariance leads to conservation of momentum by considering a system of $N$ particles, each with coordinates $\mathbf{r}_1, \mathbf{r}_2, \ldots \mathbf{r}_N$, with the Lagrangian $L(\mathbf{r}_1, \mathbf{r}_2, \ldots \mathbf{r}_N, \dot{\mathbf{r}}_1, \dot{\mathbf{r}}_2, \ldots \dot{\mathbf{r}}_N)$. Translational invariance means that we can shift the coordinate of every amount by any constant vector $\mathbf{c}$, and the Lagrangian will be the same, i.e.

$$L(\mathbf{r}_1 + \mathbf{c}, \mathbf{r}_2 + \mathbf{c}, \ldots \mathbf{r}_N + \mathbf{c}, \dot{\mathbf{r}}_1, \dot{\mathbf{r}}_2, \ldots, \dot{\mathbf{r}}_N) = L(\mathbf{r}_1, \mathbf{r}_2, \ldots \mathbf{r}_N, \dot{\mathbf{r}}_1, \dot{\mathbf{r}}_2, \ldots \dot{\mathbf{r}}_N). \tag{342}$$

Physically, this means that nothing changes if we move the whole system by the same amount, which means the potential energy is only a function of the *relative* displacements of the particles, not their absolute position. This implies that

$$\sum_{a=1}^{N} \sum_{i=1}^{3} \frac{\partial L}{\partial q_i^a} c_i = 0, \tag{343}$$

where the index $i$ labels the vector components, and the index $a$ tells us which particle we are talking about. When the equations of motion are satisfied,

$$\frac{\partial L}{\partial q_i^a} = \frac{\mathrm{d}}{\mathrm{d}t} \frac{\partial L}{\partial \dot{q}_i^a} = \frac{\mathrm{d}}{\mathrm{d}t} p_i^a, \tag{344}$$

we can write (343) as

$$\frac{\mathrm{d}}{\mathrm{d}t} \sum_{i=1}^{3} \underbrace{\sum_{a=1}^{N} p_i^a}_{p_i^{\mathrm{tot}}} c_i = \frac{\mathrm{d}}{\mathrm{d}t} \left( \mathbf{p}^{\mathrm{tot}} \cdot \mathbf{c} \right), \tag{345}$$

where in the second equality we identified the total momentum in the $i$ direction, and in the last equality used the definition of the dot product. Since this must be true for any constant vector $\mathbf{c}$, we must have

$$\frac{\mathrm{d}\mathbf{p}^{\mathrm{tot}}}{\mathrm{d}t} = 0, \tag{346}$$

which is to say that the total momentum is conserved. One can go through a similar process to show that rotational symmetry implies the conservation of angular momentum. All of these spacetime symmetries and their associated conservation laws carry over to classical field theory, as well as to point-particle quantum mechanics and QFT.

## 10.2 Internal Symmetries and Noether's Theorem

Next, we will discuss the slightly more abstract *internal symmetries* a theory may possess. Roughly speaking, these are symmetries of the fields themselves, with no reference to the underlying spacetime. As we will now show, continuous internal symmetries also correspond to conservation laws. The relationship between continuous symmetries (spacetime or internal) is formalized by *Noether's theorem*, which states that *for every continuous symmetry of a classical field theory there is an associated conserved current*. Recall that a current $J^\mu$ is conserved if $\partial_\mu J^\mu = 0$, which is the relativistic notation for the more familiar continuity equation $\dot{\rho} + \nabla \cdot \mathbf{J} = 0$.

Noether's theorem turns out to be a powerful tool, and is arguably one of the most important results in modern physics, so we will take the time to derive it, at least for the case of internal symmetries. For the sake of generality, suppose we have a set of $N$ independent fields $\phi_1, \phi_2, \ldots \phi_N$, governed by the Lagrangian

$$\mathcal{L}(\phi_1, \partial_\mu \phi_1; \phi_2, \partial_\mu \phi_2; \ldots; \phi_N, \partial_\mu \phi_N). \tag{347}$$

Further, let us suppose that the Lagrangian is invariant under some infinitesimal transformation of the fields,

$$\phi_n \to \phi_n + \varepsilon \sum_{m=1}^{N} c_{nm}\phi_m , \tag{348}$$

where $\varepsilon$ is an infinitesimal (constant) parameter. By summing over all the fields $\phi_m$ with weights $c_{nm}$ this transformation may mix up our definition of which field is which, i.e. it may take $\phi_1 \to \phi_1 + 7\phi_2 - 42\phi_9$. Alternatively, if $c_{nm} \propto \delta_{nm}$ it could simply change each field individually, i.e. $\phi_n \to \phi_n + 32\,e^{3\pi i/2}\phi_n$. In any case, the change in the field under this transformation is clearly

$$\delta\phi_n = \varepsilon \sum_{m=1}^{N} c_{nm}\phi_m . \tag{349}$$

If the Lagrangian is invariant under this transformation, then the change in the Lagrangian as a result of the transformation is zero. If we call the original Lagrangian $\mathcal{L}_0$ and the transformed Lagrangian $\mathcal{L}_T$ this means that $\delta\mathcal{L} \equiv \mathcal{L}_T - \mathcal{L}_0 = 0$. Since the transformation is infinitesimal, we can use the same tricks we used in deriving the Euler-Lagrange equations in the previous section. That is, we can write the change in the Lagrangian as

$$\delta\mathcal{L} = \sum_{n=1}^{N} \left[ \frac{\partial\mathcal{L}}{\partial\phi_n}\delta\phi_n + \frac{\partial\mathcal{L}}{\partial(\partial_\mu\phi_n)}\delta(\partial_\mu\phi_n) \right] . \tag{350}$$

By linearity, $\delta(\partial_\mu\phi_n) = \partial_\mu(\delta\phi_n)$, and when the equations of motion are satisfied, we may rewrite the first term using

$$\frac{\partial\mathcal{L}}{\partial\phi_n} = \partial_\mu \frac{\partial\mathcal{L}}{\partial(\partial_\mu\phi_n)} , \tag{351}$$

so the change in the Lagrangian becomes

$$\delta\mathcal{L} = \sum_{n=1}^{N} \left[ \left( \partial_\mu \frac{\partial\mathcal{L}}{\partial(\partial_\mu\phi_n)} \right) \delta\phi + \frac{\partial\mathcal{L}}{\partial(\partial_\mu\phi_n)}\partial_\mu(\delta\phi) \right] . \tag{352}$$

However, as you can check using the product rule, this can also be written as a total derivative,

$$\delta\mathcal{L} = \partial_\mu \left( \sum_{n=1}^{N} \frac{\partial\mathcal{L}}{\partial(\partial_\mu\phi_n)}\delta\phi_n \right) . \tag{353}$$

We can now insert our expression for the change in the fields (349), so we have

$$\delta\mathcal{L} = \varepsilon\,\partial_\mu \left( \sum_{n=1}^{N}\sum_{m=1}^{N} \frac{\partial\mathcal{L}}{\partial(\partial_\mu\phi_n)}c_{nm}\phi_m \right) . \tag{354}$$

If the transformation is a symmetry then $\delta\mathcal{L} = 0$ and we have

$$\partial_\mu \left( \sum_{n=1}^{N}\sum_{m=1}^{N} \frac{\partial\mathcal{L}}{\partial(\partial_\mu\phi_n)}c_{nm}\phi_m \right) = 0 , \tag{355}$$

which is a conservation law! If we define the current to be

$$J^\mu = \sum_{n=1}^{N}\sum_{m=1}^{N} \frac{\partial\mathcal{L}}{\partial(\partial_\mu\phi_n)}c_{nm}\phi_m , \tag{356}$$

then (355) says this current is conserved, $\partial_\mu J^\mu = 0$. This is Noether's theorem. Notice that not only does Noether's theorem say that a conserved current exists, but it also tells us precisely

what the conserved current is! You may be skeptical about this proof since we only considered an infinitesimal transformation. In fact, this is where the continuous nature of the symmetry comes in. The wonderful thing about a continuous transformation is that you can compose a finite transformation by performing an infinitesimal transformation many times. For example, to rotate something by a finite angle $\theta$ we can rotate it by an infinitesimal angle $\varepsilon$ many times in succession. Thus, if the theory is invariant under an infinitesimal transformation, it is also invariant under any finite transformation obtained by composing many infinitesimal ones.

To see how this works in practice, let's go through a few examples. First, let's consider a complex scalar $\phi$ obeying the Klein-Gordon Lagrangian,

$$\mathcal{L} = \partial_\mu \phi^\star \, \partial^\mu \phi - m^2 \phi^\star \phi \, . \tag{357}$$

It is invariant under a simultaneous change of the phase of both fields,

$$\phi \to e^{-i\alpha/2} \phi \, , \qquad \phi^\star \to e^{i\alpha/2} \phi^\star \, . \tag{358}$$

This kind of phase transformation is called a $U(1)$ transformation. This is because mathematically, symmetries are described by *groups*, and the group of 1 x 1 unitary matrices is called $U(1)$. But, a $1 \times 1$ unitary matrix is simply a complex number of unit modulus, i.e. a phase factor of the form $e^{i\theta}$.

So, we have found a symmetry of the theory. To find its associated conserved current, we first write the infinitesimal form of this transformation. If $\alpha$ is small, then to first order $e^{-i\alpha/2} \approx 1 - i\alpha/2 + \dots$ and the infinitesimal transformation is

$$\phi \to \phi - i\frac{\alpha}{2}\phi \, , \qquad \phi^\star \to \phi^\star + i\frac{\alpha}{2}\phi^\star \, , \tag{359}$$

from which we can identify

$$\delta\phi = -\frac{i\alpha}{2}\phi \, , \qquad \delta\phi^\star = \frac{i\alpha}{2}\phi^\star \, . \tag{360}$$

From (353) we know the conserved current is

$$J^\mu = \frac{\partial \mathcal{L}}{\partial(\partial_\mu \phi)}\delta\phi + \frac{\partial \mathcal{L}}{\partial(\partial_\mu \phi^\star)}\delta\phi^\star. \tag{361}$$

Note that we treat $\phi$ and $\phi^\star$ as independent fields. This is simply because a complex field has two degrees of freedom, and we may parameterize them however we so choose. We can do this by taking the real and imaginary parts as our independent fields, but we can just as well use $\phi$ and $phi^\star$, which is more convenient in this context. The necessary derivatives are

$$\frac{\partial \mathcal{L}}{\partial(\partial_\mu \phi)} = \partial^\mu \phi^\star \, , \qquad \frac{\partial \mathcal{L}}{\partial(\partial_\mu \phi^\star)} = \partial^\mu \phi \, . \tag{362}$$

Putting (362) and (360) into (361) we find the conserved current to be

$$J^\mu = \frac{i}{2}\big(\phi^\star \partial^\mu \phi - \phi \partial^\mu \phi^\star\big) \, , \tag{363}$$

which you may remember is the same conserved current we discussed in section 7, although now we have derived it from symmetry principles.

Next, consider the Dirac Lagrangian,

$$\mathcal{L} = \bar{\psi}(i\slashed{\partial} - m)\psi \, . \tag{364}$$

It also is invariant under a $U(1)$ transformation,

$$\psi \to e^{-i\alpha}\psi\,, \qquad \bar{\psi} \to e^{i\alpha}\bar{\psi}\,. \tag{365}$$

Just as before, the infinitesimal version of this transformation is

$$\delta\psi = -i\alpha\psi\,, \qquad \delta\bar{\psi} = i\alpha\bar{\psi}\,. \tag{366}$$

We need the derivatives

$$\frac{\partial\mathcal{L}}{\partial(\partial_\mu\psi)} = i\bar{\psi}\gamma^\mu\,, \qquad \frac{\partial\mathcal{L}}{\partial(\partial_\mu\bar{\psi})} = 0\,. \tag{367}$$

Putting these together, the conserved current is

$$J^\mu = \frac{\partial\mathcal{L}}{\partial(\partial_\mu\psi)}\delta\psi = \bar{\psi}\gamma^\mu\psi\,, \tag{368}$$

which is the same current we encountered in our previous discussion of the Dirac equation. Moving on to some more complicated theories, you can repeat the calculation and convince yourself that in the $\phi^4$ theory,

$$\mathcal{L} = \partial_\mu\phi^\star\partial_\mu\phi - m^2\phi^\star\phi - \frac{\lambda}{2}\phi^\star\phi^\star\phi\phi\,, \tag{369}$$

we will have the same conserved current as in the free Klein-Gordon theory,

$$J^\mu = \frac{i}{2}\left(\phi^\star\partial^\mu\phi - \phi\partial^\mu\phi^\star\right)\,, \tag{370}$$

due to the $U(1)$ symmetry (360). We can also have theories with more than one conserved current. For example, consider a theory including a fermion field $\psi$, a complex scalar field $\sigma$, and a real scalar field $\phi$ with the Lagrangian

$$\mathcal{L} = \bar{\psi}(i\slashed{\partial} - m_\psi - g\phi)\psi + \frac{1}{2}\partial_\mu\phi\partial^\mu\phi - \frac{1}{2}m_\phi^2\phi^2 + \partial_\mu\sigma^\star\partial^\mu\sigma - m_\sigma^2\sigma^\star\sigma - \frac{\lambda}{2}\phi^2\sigma^\star\sigma\,. \tag{371}$$

Both the fermion and complex scalar fields are invariant under separate $U(1)$ transformations,

$$\begin{aligned}
\psi \to e^{-i\alpha}\psi\,, &\qquad \bar{\psi} \to e^{i\alpha}\bar{\psi}, \\
\sigma \to e^{-i\beta/2}\sigma\,, &\qquad \sigma^\star \to e^{i\beta/2}\sigma^\star.
\end{aligned} \tag{372}$$

Nothing changes the preceding arguments, and we will find two independent conserved currents,

$$\begin{aligned}
J_\psi^\mu &= \bar{\psi}\gamma^\mu\psi, \\
J_\sigma^\mu &= \frac{i}{2}\left(\sigma^\star\partial^\mu\sigma - \sigma\partial^\mu\sigma^\star\right),
\end{aligned} \tag{373}$$

for which $\partial_\mu J_\psi^\mu = \partial_\mu J_\sigma^\mu = 0$. This is called a $U(1) \times U(1)$ symmetry, since we essentially have two copies of $U(1)$. It turns out that there are more interesting and complex continuous symmetries, but we will hold off discussing them until the next section. For now we will turn to discrete symmetries.

## 10.3 Discrete Symmetries: Parity

Among the most important discrete symmetries are *parity*, *time reversal*, and *charge conjugation*. Although we will not discuss the last of these in these notes, there is an important theorem, the *CPT theorem*, which states that any sensible quantum field theory must be invariant under simultaneous parity, time reversal, and charge conjugation transformations.

Unlike continuous symmetries, there are no conserved currents associated with discrete symmetries. However, there are still conservation laws. That is, if the interactions in a theory preserve parity (as do the electromagnetic and strong interactions) then the initial and final states of the interaction must have the same parity.

Now, to discuss parity specifically we will begin by defining it. A parity transformation is a spatial inversion, which transforms

$$\mathcal{P}: \quad \mathbf{x} \to -\mathbf{x}. \tag{374}$$

You can convince yourself that both the Klein-Gordon and Dirac Lagrangians are invariant under this transformation. Further, if the Lagrangian is invariant under parity, the Hamiltonian will be invariant as well. Formally, the statement that the Hamiltonian is invariant under parity is that $[H, \mathcal{P}] = 0$, where $\mathcal{P}$ is the parity operator. Then, we know from linear algebra that if two operators commute they can be simultaneously diagonalized, and hence share the same eigenvalues. Note that if we invert space with a parity transformation and then invert again, we end up where we started, i.e. acting with parity twice gives the identity operator,

$$\mathcal{P}^2 = \mathbb{1}. \tag{375}$$

Now, suppose we have a state $|\psi\rangle$ which is an eigenvalue of the parity operator,

$$\mathcal{P}|\psi\rangle = \lambda|\psi\rangle. \tag{376}$$

Acting with $\mathcal{P}$ again we have

$$\underbrace{\mathcal{P}\mathcal{P}}_{\mathbb{1}}|\psi\rangle = |\psi\rangle = \lambda^2|\psi\rangle, \tag{377}$$

which means that $\lambda^2 = 1$, so $\mathcal{P}$ has eigenvalues $\lambda = \pm 1$. A parity eigenstate with eigenvalue $+1$ is said to be *even* under parity, while a state with a $-1$ eigenvalue is said to be *odd*.

We know that the Hamiltonian generates the state's time evolution, and thus if $[H, \mathcal{P}] = 0$ (i.e. the dynamics preserve parity) a parity eigenstate will always time evolve into another parity eigenstate. Perhaps surprisingly, a particle, even if it is just sitting at rest, may have an intrinsic parity. It may seem strange that a particle can be odd under parity, but just think back to the hydrogen atom. You may remember that the parity of a hydrogen atom in a state with orbital angular momentum $\ell$ is

$$\mathcal{P} = (-1)^\ell. \tag{378}$$

Now, suppose we have a hydrogen atom in a *p*-wave[25] state (i.e. $\ell = 1$), and further that the spin of the electron and proton add to 1. If the spin and orbital angular momenta add to a singlet configuration (where the total angular momentum $J = 0$), we have a particle with no net angular momentum, but the parity is still $\mathcal{P} = -1$. If we now look at the hydrogen atom from far away (or after a few glasses of your alcoholic beverage of choice) and forget that it is a composite system made of smaller individual particles, it *looks* like you have one big particle with an intrinsic negative parity (since it appears stationary).

As we will now discuss in some detail, parity conservation imposes strong constraints on allowed processes in theories with parity-conserving interactions. Two notable interactions

---

[25]Recall that the $\ell = 0$ state is called *s*-wave, $\ell = 1$ is called *p*-wave, $\ell = 2$ is called *d*-wave, and so on. We'll use this language throughout the rest of the course.

which preserve parity are the electromagnetic and strong forces. In what follows, we will consider several examples of how parity restricts the set of allowed decays mediated by the strong interaction. To help us, we've included a table of some particles and their properties.

| Particle | Spin | Parity | Type | Nucleon | Spin | Parity | Type |
|---|---|---|---|---|---|---|---|
| $\pi\,(\pi^+,\pi^-,\pi^0)$ | 0 | $-$ | meson | proton ($p$) | 1/2 | $+$ | baryon |
| $\rho\,(\rho^+,\rho^-,\rho^0)$ | 1 | $-$ | meson | neutron ($n$) | 1/2 | $+$ | baryon |
| $\omega$ | 1 | $-$ | meson | $\Delta\,(\Delta^{++},\Delta^+,\Delta^0,\Delta^-)$ | 3/2 | $+$ | baryon |
| $\sigma$ or $f_0$ | 0 | $+$ | meson | N(1440) (+,0) | 1/2 | $+$ | baryon |
| $a_0\,(a_0^+,a_0^-,a_0^0)$ | 0 | $+$ | meson | N(1520) (+,0) | 3/2 | $-$ | baryon |
| $a_1\,(a_1^+,a_1^-,a_1^0)$ | 1 | $+$ | meson | N(1535) (+,0) | 1/2 | $-$ | baryon |

The superscript on a particle indicates its electric charge, and the distinction between baryons and mesons is based on their quark content (a baryon is made of three quarks and a meson is made of a quark and an anti-quark). The important point for now is that a meson is a boson while a baryon is a fermion. Now, let's consider some possible decay processes:

▶ $\rho^+ \to \pi^+\pi^0$ : We have three conserved quantities to worry about: charge, angular momentum, and parity. The first is easy: the initial state has $Q = 1$, and the final state has $Q = 1$, so charge is conserved. For angular momentum, the initial state has $j = 1$ from the spin of the $\rho^+$. Since the pions are spinless, in order to get $j = 1$ in the final state, the two must be in a $p$-wave orbital state with $\ell = 1$ and hence $j = 1$. Finally, the initial state has parity $-1$. The final state has several contributions to its parity. Since the pions are intrinsically odd under parity, we get one factor of $-1$ for each. But, they are also in a $p$-wave state with parity $(-1)^\ell = -1$, and the net parity is given by the product:

$$\mathcal{P}_{\text{final}} = (-1)(-1)(-1)^\ell = -1\,. \tag{379}$$

Thus, the final state is odd under parity, and consequently parity is conserved. Altogether, this means that such a decay is allowed, provided it is not forbidden by some other consideration. Also note that the decay is only possible if the resultant pions are in a $p$-wave orbital state.

▶ $a_1^- \to \pi^-\pi^0$: We can immediately see charge is conserved by looking at the superscripts. The initial state has $j = 1$, so just as in the previous examples, the pions must be in a $p$-wave orbital with $\ell = 1$ for angular momentum to be conserved. The initial state has parity $+1$, but the parity of the final state is $(-1)^2$ from the intrinsic parity of the two pions, times the $-1$ parity of the $p$-wave orbital, giving a net parity of $-1$. The parity of the initial and final states are not the same, and thus this interaction is forbidden by parity conservation! It cannot happen!

▶ $\sigma \to \pi^+\pi^-$: Once again, charge is manifestly conserved. The initial state is spinless so $j = 0$, which requires the pions in the final state be in a $s$-wave orbital such that $j = 0$ there as well. The parity of the initial state is $+1$, and the parity of the final state is $(-1)^2$ from the intrinsic parity of the pions times $(-1)^0 = 1$ from the $s$-wave orbital. Thus, the final state has parity $+1$, and parity is conserved. So, the decay is allowed.

▶ $\Delta^{++} \to p\,\pi^+$: The initial and final states both have $Q = 2$, so electric charge is conserved. The initial state has $j = 3/2$, so for angular momentum to be conserved we need $j = 3/2$ in the final state as well. Since the pion is spinless and the proton has spin 1/2, there are two ways we can get a total $j = 3/2$: we can either have the proton and pion in a $p$-wave or $d$-wave orbital.[26]

---

[26] Recall how angular momentum addition works: to add the spin of the proton ($s = 1/2$) to the orbital angular momentum $\ell$, we can get $j = \ell \pm 1/2$. For a $p$-wave, $\ell = 1$ so $j = 1 \pm 1/2 = 3/2$ or $5/2$. We can also get $j = 3/2$ from the $d$-wave with $\ell = 2$ since $j = 2 \pm 1/2 = 3/2$ or $7/2$.

Finally, let's turn to parity. The parity of the initial state is +1, and if the final state is in a $p$-wave configuration we'll have a final parity of +1 from the intrinsic parity of the proton, times $-1$ from the intrinsic parity of the pion, times $(-1)^1 = -1$ from the orbital angular momentum, for a net parity of +1. Thus, for a $p$-wave final state, parity is conserved and the decay is allowed.

On the other hand, if the final state is $d$-wave the final parity is +1 from the proton times $-1$ from the pion times $(-1)^2 = 1$ from the orbital angular momentum, for an end result of $-1$ parity. So, if the final state is $d$-wave parity is *not* conserved and the interaction is forbidden. In conclusion, this decay is *only* possible for a $p$-wave configuration.

## 10.4 Discrete Symmetries: Time Reversal

The other important discrete symmetry that we will discuss in these notes is *time reversal*. Just as parity flips the coordinates of space, time reversal flips the direction of motion. It is the transformation

$$\mathcal{T} : \begin{cases} \mathbf{x} & \to \mathbf{x} \\ \mathbf{p} & \to -\mathbf{p} \\ \mathbf{J} & \to -\mathbf{J} \end{cases} \tag{380}$$

If we want to consider the behavior of more general operators under time reversal, things get somewhat complicated. This is because time reversal is unlike most of the quantum mechanical operators you've met so far in that it is *anti-unitary*. Things are further complicated by the fact that there are multiple different things that people call time reversal operators, and different conventions for everything as well. So, we'll eschew all this complicated business by just focusing on a simple, but very physically relevant, case. Namely, let's suppose we have a state for which the angular momentum is a good quantum number. In addition to the total angular momentum $j$ we also usually work with its projection onto the $z$-axis, which we call $m$. From the rules above, time reversal maps $\mathcal{T} : m \to -m$.

Once again, if we have a time reversal invariant Lagrangian, the Hamiltonian for the system will also respect time reversal. If we consider an energy eigenstate $|\psi; j, m\rangle$ where $\psi(x)$ is the position space wave function, it should then transform under time reversal as

$$\mathcal{T} : |\psi; j, m\rangle \to e^{i\alpha} |\psi; j, -m\rangle, \tag{381}$$

where we've allowed for an extra phase factor, which will depend on the conventions one uses. Just like parity, time reversal invariance constrains the allowed processes within a theory. Here, we will focus on the electric dipole moment of particles, and will show that any particle which is an energy eigenstate of a time reversal invariant Hamiltonian must have an electric dipole moment of *zero*.

First off, recall that the electric dipole moment $\mathbf{d}$ is something like a charge density integrated against a displacement vector, or, for a discrete bunch of charges,

$$\mathbf{d} = \sum q_i \mathbf{x}_i . \tag{382}$$

Obviously, the charge shouldn't transform under time reversal, and by our definition (380) the displacement does not transform either. So, under time reversal $\mathbf{d} \to \mathbf{d}$. Let's now consider the expectation value of the electric dipole moment in an energy eigenstate, $\langle \psi; j, m | \mathbf{d} | \psi; j, m \rangle$. There is a powerful result in quantum mechanics called the Wigner-Eckert theorem, which tells us that the expectation value of the electric dipole moment must be proportional to the expectation value of the angular momentum, i.e.

$$\langle \psi; j, m | \mathbf{d} | \psi; j, m \rangle = h \langle \psi; j, m | \mathbf{J} | \psi; j, m \rangle, \tag{383}$$

where $h$ is an angular momentum independent constant of proportionality. A good heuristic justification for this is to ask what else could it be? The expectation value of $\mathbf{d}$ is a vector, and the only vector that describes the state is $\mathbf{J}$, so we must have $\langle \mathbf{d} \rangle \propto \langle \mathbf{J} \rangle$, simply because there is no other vector operator in the game. Taking the $z$ component of (383) we have

$$\langle \psi; j, m | d_z | \psi; j, m \rangle = h \langle \psi; j, m | J_z | \psi; j, m \rangle = hm, \tag{384}$$

where in the last equality we used $J_z | \psi; j, m \rangle = m | \psi; j, m \rangle$. Now, let's ask what happens under time reversal. The left-hand side should be invariant by our previous considerations of the dipole operator, and by (381) the right-hand side transforms as

$$h \langle \psi; j, m | J_z | \psi; j, m \rangle \to h \langle \psi; j, -m | e^{-i\alpha} J_z e^{i\alpha} | \psi; j, -m \rangle = -hm . \tag{385}$$

Equating (384) and (385), which should hold for a time reversal invariant system, we have $hm = -hm$ which implies that $h = 0$, and thus $\langle \mathbf{d} \rangle = 0$, which is to say that the electric dipole matrix element vanishes for any energy eigenstate of a time reversal invariant theory.

> **Time Reversal Violation in the Standard Model**
>
> We know that there is violation of time reversal symmetry (TRS) in the standard model, and we also know that there must be more of it than is presently understood. For example, without extra TRS breaking cosmology can't explain why there is more matter than anti-matter in the universe. One likely way to detect TRS breaking is to search for small electric dipole moments of neutral particles (why neutral particles? simply because it makes the experiments easier: to measure an electric dipole moment one puts the particle in an electric field and sees how the resonance properties are affected; this is hard to do with charged particles which tend to accelerate in electric fields!), since such a moment violates time reversal due to our discussion above.

## 10.5  Isospin

Now, let's turn back to continuous symmetries. Before our discussion of discrete symmetries, we mentioned that there are more complicated continuous symmetries than just $U(1)$, or multiple copies of it. These are called *non-Abelian* symmetries for reasons that will be explained in the next section. For now, we'll discuss the simplest non-Abelian symmetry group, $SU(2)$. This is the same group associated with the spin of a spin $1/2$ particle, so much of the structure will be familiar but the physical context will be different.

The particular *physical* symmetry we will discuss is isotopic spin, or *isospin*, which is an approximate symmetry of the strong interaction. Roughly speaking, we can think of an exact symmetry as implying that some parameter in the theory is zero (for example, the divergence of a conserved current), but there are many interesting cases where the parameter is not zero, but is still very small compared to other scales in the theory. If this is the case, we say we have an *approximate symmetry*. Operationally, one typically first treats the symmetry as exact, and then adds in the symmetry breaking perturbatively afterward.

The original idea of isospin is due to Heisenberg, who noted that the masses of the proton and neutron are nearly equal; $m_p \approx 938.3$ MeV and $m_n \approx 939.5$ MeV. The binding energy of $^3$He (two protons and one neutron) is 7.72 MeV, which is very close to that of $^3$H (one proton and two neutrons) which is 8.48 MeV. In fact, this is generally true for any small nucleus, where the Coulomb force is unimportant, as we discussed in section 2. Recall that the proton-neutron asymmetry term goes like $\sim (Z - (A - Z))^2$ for a nucleus with $Z$ protons and $A - Z$ neutrons. The point is that switching the protons and neutrons doesn't change the binding energy very much.

Altogether, this could lead one to think that protons and neutrons are kind of the same thing. Specifically, we can think of them as two different states of a single underlying entity, just like up and down spin electrons are two different states of the same fundamental particle. So, roughly speaking we can think of a proton being the "isospin up" state and the neutron as the "isospin down" state. We can combine them into a two-component nucleon,

$$N = \begin{pmatrix} p \\ n \end{pmatrix}, \tag{386}$$

which is identical to how we usually describe the spin state of a spin $1/2$ particle,

$$\psi = \begin{pmatrix} \psi_\uparrow \\ \psi_\downarrow \end{pmatrix}. \tag{387}$$

The considerations above suggest that physics should be approximately symmetric under switching protons and neutrons. The idea of isospin is to further suppose that the strong interaction is invariant under rotating the proton and neutron into one another. By this we mean that acting on the nucleon vector with a $2 \times 2$ special ($\det U = 1$) unitary ($U^\dagger U = 1$) matrix is a symmetry of the theory. In other words, we can transform

$$\begin{pmatrix} p \\ n \end{pmatrix} \to U \begin{pmatrix} p \\ n \end{pmatrix}, \tag{388}$$

and the physics is the same. Essentially, this operation mixes up our definition of what is a proton versus what is a neutron. Mathematically, it is completely analogous to rotating the quantization axis of a spin system. In both cases, the symmetry is a manifestation of the arbitrary convention we pick to distinguish the two different states. As a matter of terminology, the set of $2 \times 2$ special unitary matrices is called $SU(2)$.

We can take the analogy with spin further, since the math is identical. Just like we say a spin $1/2$ particle transforms in the $s = 1/2$ or doublet representation, we can also classify nuclear states into isospin multiplets.[27] Since the nucleons transform just like a spin, it is clearly an isospin doublet with $I = 1/2$, where $I$ is the analogue of the spin magnitude $s$ that tells us which representation of the group the particles transform under.

Other hadrons also form isospin multiplets. For example, the pions have nearly identical masses $m_{\pi^0} \approx 139$ MeV, $m_{\pi^\pm} \approx 135$ MeV, and form a triplet with $I = 1$. The rho mesons with $m_{\rho^\pm} \approx m_{\rho^0} \approx 775$ MeV also form a triplet. The $\omega$ with $m_\omega \approx 738$ MeV has no partners and is a singlet with $I = 0$, while the $\Delta$'s have masses $m_{\Delta^{++}} \approx m_{\Delta^\pm} \approx m_{\Delta^0} \approx 1232$ MeV and form a quartet[28] with $I = 3/2$.

In addition to categorizing the representations under which particles transform, we can borrow another quantum number from the study of spin: the $z$-component of the spin projection. In the context of isospin, we call this $I_3$, or "the third component of isospin." The only reason we call it $I_3$ instead of $I_z$ is because its conventional (and we're fancy). Since the nucleons form a doublet, with the proton playing the role of the spin up state and the neutron playing that of the spin down state, it is evident that the proton has $I_3 = +1/2$ and the neutron has $I_3 = -1/2$.

There is an interesting formula that relates the electric charge and $I_3$ which applies for nuclei and non-strange hadrons[29],

$$Q = \frac{B}{2} + I_3, \tag{389}$$

---

[27]A multiplet is just a different word for the representation. For example, something that transforms in the $s = 1/2$ representation of $SU(2)$ is said to be a doublet, something that transforms in the $s = 1$ representation is a triplet, and so on. Something that transforms in the trivial $s = 0$ representation is said to be a singlet.

[28]By which we mean the four-dimensional representation

[29]However, there is a straightforward generalization that does apply to strange particles

where $B$ is the *baryon number*. For example, we know protons and neutrons are baryons (they're each made from three quarks), so $B = 1$ for both. Then, for the proton this says that $Q = 1/2 + 1/2 = 1$, and for the neutron $Q = 1/2 - 1/2 = 0$, which we know to be true. We can also use this formula in reverse to easily deduce the third component of the isospin for the pions, which are mesons so $B = 0$. For the $\pi^+$, $Q = 1$, so by the above $1 = 0 + I_3$, and thus the $\pi^+$ has $I_3 = 1$. By the same process we can see that the $\pi^0$ has $I_3 = 0$ and the $\pi^-$ has $I_3 = -1$. This relationship was known before the discovery of quarks, and even helped motivate the quark picture.

Speaking of quarks, the fundamental symmetry underlying isospin is the approximate symmetry of rotating up and down quarks (whose masses are both very small) into one another with an $SU(2)$ transformation,

$$\begin{pmatrix} u \\ d \end{pmatrix} \to U \begin{pmatrix} u \\ d \end{pmatrix}. \tag{390}$$

Since any baryon is made of three quarks, a single quark has baryon number $B = 1/3$. It is also known that the up quark has electric charge $Q = 2/3$ and the down quark has electric charge $Q = -1/3$. These facts, combined with defining the up quark to have $I_3 = +1/2$ and the down quark to have $I_3 = -1/2$ imply the quarks satisfy (389). Further, since $B$ and $I_3$ are simple additive quantities[30] this ensures that any particle made out of up and down quarks will also satisfy (389). So, we now understand (roughly) where this equation comes from.

Returning to the mathematical structure of isospin, we have seen that the proton and neutron can be combined into an isospinor $N$ which transforms under the two-dimensional representation of $SU(2)$. As we know from the physics of spin, the important matrices[31] when dealing with $SU(2)$ are the Pauli matrices, which when used in the context of isospin are conventionally denoted as $\tau^i$ rather than $\sigma^i$,

$$\tau_1 = \begin{pmatrix} 0 & 1 \\ 1 & 0 \end{pmatrix}, \quad \tau_2 = \begin{pmatrix} 0 & -i \\ i & 0 \end{pmatrix}, \quad \tau_3 = \begin{pmatrix} 1 & 0 \\ 0 & -1 \end{pmatrix}. \tag{391}$$

A generic isospin rotation parameterized by the "angles" $\theta_1$, $\theta_2$, $\theta_3$ is given by

$$U = \exp\left[ i\left( \frac{\theta_1 \tau_1}{2} + \frac{\theta_2 \tau_2}{2} + \frac{\theta_3 \tau_3}{2} \right) \right]. \tag{392}$$

This is analogous to the fact that the spin angular momentum operator is the generator of rotations in quantum mechanics. We will also see how this fact follows directly from group theory in the next section. Just like when using the Pauli matrices to describe real spins, its useful to combine the Pauli matrices into a vector,

$$\boldsymbol{\tau} = \begin{pmatrix} \tau_1 \\ \tau_2 \\ \tau_3 \end{pmatrix}. \tag{393}$$

However, this is not a vector in real space, but a vector in isospin space, and as such it is called an *isovector*. We can also combine the pion fields into an isovector by defining the Hermitian fields $\pi_1$, $\pi_2$, $\pi_3$ which are related to the physical $\pi^+$, $\pi^-$, $\pi^0$ by

$$\pi^\pm = \frac{\pi_1 \mp i\pi_2}{\sqrt{2}}, \qquad \pi^0 = \pi_3. \tag{394}$$

---

[30]That is to say that if we have two particles with $B = 2$ and $Q = 1$, the total baryon number of the combined state is $B = 4$ and the total charge is $Q = 2$, and so on.

[31]This is made more precise in the next section when we learn the Pauli matrices are the generators of $SU(2)$ in the fundamental representation.

These fields then transform as an isovector,

$$\boldsymbol{\pi} = \begin{pmatrix} \pi_1 \\ \pi_2 \\ \pi_3 \end{pmatrix}. \tag{395}$$

Just like the dot product of spatial vectors is rotationally invariant, the dot product of isovectors is isospin rotation-invariant, such as $\boldsymbol{\pi} \cdot \boldsymbol{\pi}$ or $\boldsymbol{\pi} \cdot \boldsymbol{\tau}$.

Given the isospinor $N$ for the nucleons and isovector $\boldsymbol{\pi}$ for the pions, we can write down an effective[32] Lagrangian that describes the long range interaction of nucleons and pions, encoded in the interaction term

$$\mathcal{L}_{\text{int}} = g_\pi \bar{N} \big[ i \gamma^\mu \gamma_5 \boldsymbol{\tau} \cdot (\partial_\mu \boldsymbol{\pi}) \big] N. \tag{396}$$

Despite its deceptive simplicity, this term contains a great deal of information. Let's start by just checking that it possesses all the symmetries that a sensible theory of nucleons should have: Lorentz invariance, isospin invariance, and parity conservation. It's easy to see the Lagrangian is Lorentz invariant since all the $\mu$'s are contracted. The isospin invariance is also manifest since we've already discussed how $\boldsymbol{\tau} \cdot \boldsymbol{\pi}$ is an isoscalar.[33]

The fact it is invariant under a parity transformation takes a little more work to see. Let's start by considering just the $\mu = 0$ term,

$$\mathcal{L}_{\text{int},0} = g_\pi \bar{N} \big[ i \gamma^0 \gamma_5 \boldsymbol{\tau} \cdot (\partial_0 \boldsymbol{\pi}) \big] N. \tag{397}$$

The factor $\partial_0 \boldsymbol{\pi}$ has odd parity since $\partial_0$ is even and the pion is intrinsically odd. The nucleons all have even parity, as does $\gamma^0$. On the other hand, $\gamma_5$ is a pseudoscalar and thus has negative parity. Together, the negative parity of the pion and $\gamma_5$ matrix cancel to give us an overall even parity, as a sensible Lagrangian should have.

Next, we can consider the $\mu \neq 0$ terms,

$$\mathcal{L}_{\text{int},i} = g_\pi \bar{N} \big[ i \gamma^i \gamma_5 \boldsymbol{\tau} \cdot (\partial_i \boldsymbol{\pi}) \big] N, \tag{398}$$

where $i = 1, 2, 3$ runs over the spatial components. Now, $\partial_i$ has negative parity so $\partial_i \boldsymbol{\pi}$ has a net positive parity since the pion is also odd. $\gamma_5$ still has negative parity, but the spatial components of $\gamma^\mu$ are a vector which means that they transform $\mathcal{P} : \gamma^i \to -\gamma^i$, and thus have negative parity. So, once again the net parity of these terms is positive, and all is well.

Now, let's take a few moments to unpack this Lagrangian. First of all, let's evaluate the matrix in isospin space,

$$\boldsymbol{\tau} \cdot (\partial_\mu \boldsymbol{\pi}) = (\partial_\mu \pi_1)\tau_1 + (\partial_\mu \pi_2)\tau_2 + (\partial_\mu \pi_3)\tau_3 = \begin{pmatrix} \partial_\mu \pi_3 & \partial_\mu(\pi_1 - i\pi_2) \\ \partial_\mu(\pi_1 + i\pi_2) & -\partial_\mu \pi_3 \end{pmatrix}. \tag{399}$$

We can now use (395) to write this in terms of the $\pi^+, \pi^-, \pi^0$:

$$\boldsymbol{\tau} \cdot (\partial_\mu \boldsymbol{\pi}) = \begin{pmatrix} \partial_\mu \pi^0 & \sqrt{2}\,\partial_\mu \pi^+ \\ \sqrt{2}\,\partial_\mu \pi^- & -\partial_\mu \pi^0 \end{pmatrix}. \tag{400}$$

The effective Lagrangian can then be written

$$\mathcal{L}_{\text{int}} = \begin{pmatrix} \bar{p} & \bar{n} \end{pmatrix} \left[ i g_\pi \gamma^\mu \gamma_5 \begin{pmatrix} \partial_\mu \pi^0 & \sqrt{2}\,\partial_\mu \pi^+ \\ \sqrt{2}\,\partial_\mu \pi^- & -\partial_\mu \pi^0 \end{pmatrix} \right] \begin{pmatrix} p \\ n \end{pmatrix}. \tag{401}$$

---

[32]Recall from the previous section that an effective theory is not renormalizable, but is useful for describing low energy physics

[33]By which we means it does not transform under isospin rotations

Carrying out the matrix multiplication we get four terms,

$$\mathcal{L}_{\text{int}} = ig_\pi\left[\bar{p}\gamma^\mu\gamma_5(\partial_\mu\pi^0)p + \sqrt{2}\,\bar{p}\gamma^\mu\gamma_5(\partial_\mu\pi^+)n + \sqrt{2}\,\bar{n}\gamma^\mu\gamma_5(\partial_\mu\pi^-)p - \bar{n}\gamma^\mu\gamma_5(\partial_\mu\pi^0)n\right]. \quad (402)$$

Recall that the proton and neutron are massive fermions, so $p$ and $n$ are each four-component Dirac spinors, and the gamma matrices act on these spinor components. If we strip away all of the details, (which are needed to ensure all the required symmetries are satisfied) these four terms correspond to four processes,

$$\mathcal{L}_{\text{int}} \sim \bar{p}\pi^0 p + \bar{p}\pi^+ n + \bar{n}\pi^- p + \bar{n}\pi^0 n. \quad (403)$$

Given these interactions, the Lagrangian encodes the longest range scattering channels between nucleons and pions. You can see this diagrammatically: its a nice exercise to draw all of the tree level diagrams and show that they correspond to the allowed pion-nucleon scattering processes.

In particular, if we take the non-relativistic limit of the tree diagram for a nucleon-nucleon interaction to be the long-distance potential experienced by the nucleons, we can Fourier transform it back to real space to arrive at the famous one-pion-exchange potential (OPEP) between nucleons 1 and 2,

$$V_{\text{OPEP}} = \frac{g_\pi^2 m_\pi^2}{12 m_n^2}(\boldsymbol{\tau}_1\cdot\boldsymbol{\tau}_2)\left\{\left[\boldsymbol{\sigma}_1\cdot\boldsymbol{\sigma}_2 + \underbrace{\frac{1}{2}\left(3(\boldsymbol{\sigma}_1\cdot\hat{\mathbf{r}})(\boldsymbol{\sigma}_2\cdot\hat{\mathbf{r}}) - \boldsymbol{\sigma}_1\cdot\boldsymbol{\sigma}_2\right)}_{S_{12}}\right]\right.$$
$$\left.\times\left(1 + \frac{3}{m_\pi r} + \frac{3}{m_\pi^2 r^2}\right)\frac{e^{-m_\pi r}}{r} - \frac{4\pi}{3}\boldsymbol{\sigma}_1\cdot\boldsymbol{\sigma}_2\,\delta^3(\mathbf{r})\right\}. \quad (404)$$

Although we won't go into any depth about this result, let's take note of a few things.

► We've assumed all of the pions have the same mass $m_\pi$ and the protons and neutrons have the same mass $m_n$

► The contribution we've labelled $S_{12}$ is a $J = 2$ tensor force, which mixes partial waves (this makes sense, since for example the deuteron is a mixture of $s$ and $d$ waves)

► Overall, this potential has the standard Yukawa form due to the overall $e^{-mr}/r$ factor

► The last delta-function term is a short-ranged interaction, which is generally not reliable

► Most importantly, this works experimentally!!

## 10.6 Implications of Isospin

Isospin invariance can be used to further our understanding of physical processes. For example, if we consider the scattering of hadrons or nuclei off of one another, we can decompose the scattering amplitude into separate isospin channels which do not mix with one another, with the relative strength of scattering in that channel given by the Clebsch-Gordan coefficients of the initial and final states.

Let's take $\pi^+ n$ scattering as an example: we have two possible processes, $\pi^+ + n \to \pi^+ + n$ or $\pi^+ + n \to \pi^0 + p$. The final state in the first process has $I = 3/2$, $I_3 = 1/2$ while the second has $I = 1/2$, $I_3 = 1/2$, corresponding to two isospin channels. In the initial state the pion has $I = 1$, $I_3 = 1$ and the neutron has $I = 1/2$, $I_3 = -1/2$. Using the notation $|I^\pi, I^n; I_3^\pi, I_3^n\rangle$ for

the combined state, we can decompose the state into its two isospin channels,

$$
\left|1,\tfrac{1}{2};1,-\tfrac{1}{2}\right\rangle = \left|\tfrac{3}{2},\tfrac{1}{2}\right\rangle \underbrace{\left\langle \tfrac{3}{2},\tfrac{1}{2}\middle|1,\tfrac{1}{2};1,-\tfrac{1}{2}\right\rangle}_{\text{CG coefficient}} + \left|\tfrac{1}{2},\tfrac{1}{2}\right\rangle \underbrace{\left\langle \tfrac{1}{2},\tfrac{1}{2}\middle|1,\tfrac{1}{2};1,-\tfrac{1}{2}\right\rangle}_{\text{CG coefficient}}
$$

$$
= \sqrt{\tfrac{1}{3}}\left|\tfrac{3}{2},\tfrac{1}{2}\right\rangle + \sqrt{\tfrac{2}{3}}\left|\tfrac{1}{2},\tfrac{1}{2}\right\rangle. \tag{405}
$$

So, even though there are ten pion-nucleon scattering channels, there are only two independent amplitudes. This means that if we measure two processes at various energies and angles we can determine the scattering amplitude in the $I = 3/2$ and $I = 1/2$ channels, from which we can predict the scattering rates for all pion-nucleon processes.

As a simple case, suppose we measure the differential scattering cross section for $\pi^+ p$ scattering, which has $I_3 = 3/2$ and thus can only occur in the $I = 3/2$ channel. The same is true for $\pi^- n$ process which also is entirely within the $I = 3/2$ channel. Since both processes have the same scattering channel, we can conclude

$$
\left.\frac{d\sigma}{d\Omega}\right|_{\pi^+ p} = \left.\frac{d\sigma}{d\Omega}\right|_{\pi^- n}. \tag{406}
$$

We can apply a similar treatment to the decay of hadrons. It turns out that the decay rate of a hadron is independent of its isospin projection ($I_3$), as a consequence of isospin invariance. However, the branching ratios (that is, the fraction of hadrons which decay via a given process) are fixed by the Clebsch-Gordan coefficients between the initial and final states. That is, if we have two processes where hadron $a$ decays into hadrons $b$ and $c$ or $b'$ and $c'$, i.e. $h_a \to h_b + h_c$ or $h_a \to h_{b'} + h_{c'}$, where $h_a$ has isospin $I^a$ and $I_3^a$, and similarly for $b$, $c$, $b'$ and $c'$, the relative probability for each process is

$$
\frac{p(a \to bc)}{p(a \to b'c')} = \left|\frac{\langle I^b, I^c; I_3^b, I_3^c | I^a, I_3^a\rangle}{\langle I^{b'}, I^{c'}; I_3^{b'}, I_3^{c'} | I^a, I_3^a\rangle}\right|^2, \tag{407}
$$

where $p(a \to bc)$ is the probability that $h_a$ decays into $h_b$ and $h_c$.

Let's work out an example to make this concrete. We could ask what the relative probabilities are for the two possible decays $\Delta^+ \to p\,\pi^0$ and $\Delta^+ \to n\pi^+$. The $\Delta^+$ has $I = 3/2$ and $I_3 = 1/2$, and we have worked out the isospins of the other particles in previous sections. Putting all of these pieces into the above formula, we find[34]

$$
\frac{p(\Delta^+ \to p\pi^0)}{p(\Delta^+ \to n\pi^+)} = \left|\frac{\left\langle 1,\tfrac{1}{2};0,\tfrac{1}{2}\middle|\tfrac{3}{2},\tfrac{1}{2}\right\rangle}{\left\langle 1,\tfrac{1}{2};1,-\tfrac{1}{2}\middle|\tfrac{3}{2},\tfrac{1}{2}\right\rangle}\right|^2 = \left|\frac{\sqrt{2/3}}{\sqrt{1/3}}\right|^2 = 2, \tag{408}
$$

from which we conclude that the $\Delta^+$ decays into a $p\,\pi^0$ 2/3 of the time, and into a $n\,\pi^+$ 1/3 of the time.

---

[34]To avoid having to deal with the funny table, the easiest way to look up Clebsch-Gordan coefficients is with the Mathematica command `ClebschGordan[{j_1,m_1},{j_2,m_2},{j,m}]` for the decomposition of $|j,m\rangle$ into $|j_1,m_1\rangle$ and $|j_2,m_2\rangle$. Notice that this convention for the ordering is different than what we are using!

# 11 Gauge Theories and the Standard Model

The standard model of particle physics is comprised of a few basic ingredients. As for particles, we have quarks (which are the constituents of hadrons) and leptons, which include electrons, muons, taus, and neutrinos. These particles all interact with one another via *gauge interactions*, the simplest of which (and the most familiar) is the electromagnetic interaction. In this section we will explain the basic idea behind gauge theories and what they have to do with particle physics. We'll start by gaining a new perspective on our old friend E&M by considering it as the simplest example of a gauge theory.

## 11.1 Electromagnetic Gauge Invariance

When we first came across E&M as freshmen, things were typically discussed in terms of the electric and magnetic fields, **E** and **B**. Later on, after we have grown up as physicists, we learn that life is made a good deal easier if we swap **E** and **B** for the scalar and vector potentials $\Phi$ and **A** via the definitions

$$
\begin{aligned}
\mathbf{E} &= -\nabla\Phi - \partial_t \mathbf{A}, \\
\mathbf{B} &= \nabla \times \mathbf{A}.
\end{aligned}
\tag{409}
$$

If we add relativity into the mix, we can combine the scalar and vector potentials into the four-potential (henceforth simply called the potential), $A_\mu = (\Phi, -\mathbf{A})$. However, its important to note that while specifying $A_\mu$ uniquely determines **E** and **B**, the converse is not true: for any configuration of **E** and **B** there are an infinite number of potentials which describe it. In fact, suppose we have some set of potentials $\Phi$ and **A**, which via (409) represent a configuration of **E** and **B**. Then, for *any* differentiable function of spacetime, $\Lambda(\mathbf{x}, t)$, we may perform the *gauge transformation*

$$
\begin{pmatrix} \Phi \\ \mathbf{A} \end{pmatrix} \rightarrow \begin{pmatrix} \Phi' \\ \mathbf{A}' \end{pmatrix} = \begin{pmatrix} \Phi \\ \mathbf{A} \end{pmatrix} + \begin{pmatrix} -\partial_t \Lambda \\ \nabla\Lambda \end{pmatrix},
\tag{410}
$$

under which the electric and magnetic fields are invariant,

$$
\begin{aligned}
\mathbf{E} &\rightarrow \mathbf{E}' = -\nabla\big(\Phi - \partial_t\Lambda\big) - \partial_t\big(\mathbf{A} + \nabla\Lambda\big) = -\nabla\Phi - \partial_t\mathbf{A} + \partial_t\nabla\Lambda - \partial_t\nabla\Lambda = \mathbf{E}, \\
\mathbf{B} &\rightarrow \mathbf{B}' = \nabla \times \big(\mathbf{A} + \nabla\Lambda\big)\nabla \times \mathbf{A} + \underbrace{\nabla \times (\nabla\Lambda)}_{0} = \mathbf{B}.
\end{aligned}
\tag{411}
$$

The point is that given a set of potentials $\Phi$ and **A**, we can alter them by any scalar field without changing the electric and magnetic fields. Things get dicey when we remember that its **E** and **B** that are the physical fields we measure in the lab, so this gauge redundancy leads us to interpret the potentials as just a cute trick that lets us compute more efficiently. In fact, since any choice of gauge *must* give us the same **E** and **B** fields, we're free to use this freedom to pick the gauge condition that makes our life easiest, and can rest assured we'll always get the right answer.

Another advantage of using the potentials is that it allows us to write down the Lagrangian or Hamiltonian for a charged particle interacting with the electromagnetic field,

$$
L = \frac{1}{2}m\dot{\mathbf{x}}^2 - q\big(\Phi - \dot{\mathbf{x}} \cdot \mathbf{A}\big).
\tag{412}
$$

As shown in the box in section 7.5, the Euler-Lagrange equation of motion we get from this Lagrangian is simply the Lorentz force law. Performing a Legendre transformation, the corresponding Hamiltonian is

$$
H = \frac{(\mathbf{p} - q\mathbf{A})^2}{2m} + q\Phi.
\tag{413}
$$

It turns out that there is no way to write such a Lagrangian or Hamiltonian using **E** and **B**, so if we want to use these formalisms we have to use potentials. However, recall that quantum mechanics is based off Lagrangian and Hamiltonians! This means that quantum theories must invariably be formulated using $\Phi$ and **A**. For example, using the Hamiltonian above, the Schrödinger equation for a charged particle in an electromagnetic field is

$$i\partial_t \psi = \left[ \frac{1}{2m}(-i\nabla - q\mathbf{A})^2 + q\Phi \right]\psi. \tag{414}$$

The cost of framing things in terms of potentials is the need to keep track of the extra unphysical information contained in them. Specifically, anything we calculate should be the same regardless of whether we use one potential or a gauge-transformed version of it. This constraint is called *gauge invariance*, or (somewhat misleadingly) gauge symmetry, and reflects the redundancy of our description of the **E** and **B** fields in terms of the potentials. If an observable quantity were not gauge invariant, i.e. depended on our arbitrary choice of $\Lambda$, things would be very bad: nothing would stop you and I from choosing two different $\Lambda$'s and getting different answers! Thus, for any consistent theory including the electromagnetic field, all physical observables must be gauge invariant.

In quantum mechanics, there is another kind of ambiguity. That is, if observable quantities generally go like $\psi^\star\psi$, the absolute phase of the wavefunction isn't physical. Remarkably, it turns out that this is intimately connected to the ambiguity of the potentials. Consider rotating the phase of the wavefunction as

$$\psi(\mathbf{x}, t) \rightarrow \psi'(\mathbf{x}, t) = e^{iq\Lambda(\mathbf{x}, t)}\psi(\mathbf{x}, t). \tag{415}$$

Note that this is not the same kind of phase rotation we've considered before. Previously we've rotated the phase of a field or wavefunction by a constant amount $\alpha$, which is called a *global $U(1)$ transformation*. Here, we have a space-time dependent field $\Lambda$ in the exponential, so we are rotating the phase of the wavefunction by a different amount at each point in space-time. This is called a *local $U(1)$ transformation* because we are locally twisting the phase by a different amount at each space-time point.

Since $\Lambda$ is dependent on **x** and $t$, the derivatives of $\psi'$ pick up an extra term relative to the derivatives of $\psi$. That is,

$$\begin{aligned} i\partial_t \psi' &= i\partial_t(e^{iq\Lambda}\psi) = e^{iq\Lambda}(i\partial_t\psi - i(\partial_t\Lambda)\psi), \\ -i\nabla\psi' &= -i\nabla(e^{iq\Lambda}\psi) = e^{iq\Lambda}(-i\nabla\psi + q(\nabla\Lambda)\psi). \end{aligned} \tag{416}$$

Keeping this in the back of our minds, let's go back to the Schrödinger equation (414) and consider how it changes when we gauge transform the potentials. We then have

$$i\partial_t \psi = \left[ \frac{1}{2m}(-i\nabla - q\mathbf{A} \textcolor{red}{- q\nabla\Lambda})^2 + q\Phi \textcolor{red}{- q\partial_t\Lambda} \right]\psi, \tag{417}$$

which does not appear to be gauge invariant. To help guide your eye, I've written the extra terms we picked up from the gauge transformation of the potentials in red. Now, the magic happens: let's simultaneously perform a local $U(1)$ transformation on the wave function, taking $\psi \rightarrow e^{iq\Lambda}\psi$. Keeping in mind the extra terms picked up from the derivatives, the gauge transformed Schrödinger equation (417) becomes

$$\textcolor{blue}{e^{iq\Lambda}}\left(i\partial_t \textcolor{red}{- q\partial_t\Lambda}\right)\psi = \textcolor{blue}{e^{iq\Lambda}}\left[ \frac{1}{2m}\left(-i\nabla \textcolor{blue}{+ q\nabla\Lambda} - q\mathbf{A} \textcolor{red}{- q\nabla\Lambda}\right)^2 + q\Phi \textcolor{red}{- q\partial_t\Lambda} \right]\psi, \tag{418}$$

where we've written the terms generated by the phase rotation in blue. It's now easy to see that the red and blue terms all cancel one another out, and that after the simultaneous transformation of the potentials *and* the wavefunction, we get back the Schrödinger equation (414)

that we started with. This means that in the quantum theory, a gauge transformation is the transformation

$$
\begin{aligned}
\psi &\to \psi' = e^{iq\Lambda}\psi\,, \\
\Phi &\to \Phi' = \Phi - \partial_t\Lambda\,, \\
\mathbf{A} &\to \mathbf{A}' = \mathbf{A} + \nabla\Lambda\,,
\end{aligned}
\tag{419}
$$

under which physics is invariant. Put another way, this means that if we have a solution to the Schrödinger equation $\psi$, we can locally twist its phase to $\psi'$, which is guaranteed to be another solution to the Schrödinger equation with a gauge-transformed version of the potentials.

## 11.2 Quantum Electrodynamics

So far, everything we've done has been non-relativistic quantum mechanics. Luckily, things carry over directly to relativistic field theory. Recall that we package the potentials together into a four-vector, $A_\mu = (\Phi, -\mathbf{A})$, and the $\mathbf{E}$ and $\mathbf{B}$ fields are the components of the field strength tensor $F_{\mu\nu} = \partial_\mu A_\nu - \partial_\nu A_\mu$. In this language, a gauge transformation (410) is written as

$$
A_\mu \to A'_\mu = A_\mu - \partial_\mu\Lambda\,.
\tag{420}
$$

Under the transformation, it's not hard to see that the field strength is invariant, as it should be since it is a physical observable,

$$
\begin{aligned}
F_{\mu\nu} \to F'_{\mu\nu} &= \partial_\mu(A_\nu - \partial_\nu\Lambda) - \partial_\nu(A_\mu - \partial_\mu\Lambda) \\
&= \partial_\mu A_\nu - \partial_\nu A_\mu + \partial_\mu\partial_\nu\Lambda - \partial_\mu\partial_\nu\Lambda \\
&= F_{\mu\nu}\,.
\end{aligned}
\tag{421}
$$

Also recall that the Lagrangian for the electromagnetic field is

$$
\mathcal{L}_{\text{Maxwell}} = -\frac{1}{4}F_{\mu\nu}F^{\mu\nu} - A_\mu J^\mu\,,
\tag{422}
$$

and the classical equations of motion $\partial_\mu F^{\mu\nu} = J^\nu$ give us the Maxwell equations, as we showed earlier. We can couple the electromagnetic field to matter via the $A_\mu J^\mu$ term: we just need to find the right form for the current. Let's consider a fermion field $\psi$, the free Lagrangian for which is

$$
\mathcal{L}_{\text{Dirac}} = \bar{\psi}(i\slashed{\partial} - m)\psi\,.
\tag{423}
$$

If the electromagnetic field is in the game, we want things to be gauge invariant. It turns out that under a gauge transformation, the Dirac field should transform just like the non-relativistic wavefunction, i.e. our theory should be invariant under the transformation

$$
\begin{aligned}
\psi &\to \psi' = e^{iq\Lambda}\psi\,, \\
A_\mu &\to A'_\mu = A_\mu - \partial_\mu\Lambda\,,
\end{aligned}
\tag{424}
$$

where $q$ is the charge of the fermion. As you can check, the extra terms generated from differentiating the gauge transformed Dirac field give us problems,

$$
\mathcal{L}_{\text{Dirac}} \to \bar{\psi}(i\slashed{\partial} - q\slashed{\partial}\Lambda - m)\psi\,.
\tag{425}
$$

Note that if $\psi \to e^{iq\Lambda}\psi$, then $\bar{\psi} \to e^{-iq\Lambda}\bar{\psi}$. Perhaps this isn't too surprising, since whenever we deal with electromagnetic fields we always need to shift the canonical momentum. In field theory language, there's a simple way to implement this: we just replace every derivative with the *gauge covariant derivative*,

$$
\mathcal{D}_\mu \equiv \partial_\mu + iqA_\mu\,.
\tag{426}
$$

So, replacing every $\partial$ we see with a $\mathcal{D}$, the Dirac Lagrangian becomes

$$\mathcal{L}_{\text{Dirac}} = \bar{\psi}(i\slashed{\mathcal{D}} - m)\psi \,. \tag{427}$$

Unpacking this, we have

$$\begin{aligned}\mathcal{L}_{\text{Dirac}} &= \bar{\psi}(i(\slashed{\partial} + iq\slashed{A}) - m)\psi \\ &= \bar{\psi}(i\slashed{\partial} - q\slashed{A} - m)\psi \,.\end{aligned} \tag{428}$$

Now, performing a gauge transformation (424), the transformation of $\slashed{A} \to \slashed{A} - \slashed{\partial}\Lambda$ gives us an extra term (in red),

$$\begin{aligned}\mathcal{L}_{\text{Dirac}} &\to \bar{\psi}(i\slashed{\partial} \textcolor{blue}{- q\slashed{\partial}\Lambda} - q(\slashed{A} \textcolor{red}{- \slashed{\partial}\Lambda}) - m)\psi \\ &= \bar{\psi}(i\slashed{\partial} - q\slashed{A} - m)\psi \,,\end{aligned} \tag{429}$$

so this version of the Dirac Lagrangian is now gauge invariant. Thus, the Lagrangian for a fermion with charge $q$ interacting with the electromagnetic field is simply $\mathcal{L}_{\text{Dirac}} + \mathcal{L}_{\text{Maxwell}}$, or

$$\mathcal{L}_{\text{QED}} = -\frac{1}{4}F_{\mu\nu}F^{\mu\nu} + \bar{\psi}(i\slashed{\partial} - q\slashed{A} - m)\psi \,. \tag{430}$$

This is the Lagrangian for *Quantum Electrodynamics* (QED), which is one of the most accurate theories of physics to date, and a part of the standard model. Now, remember that at the outset of this section we noted that the coupling between matter and the EM field is given by the term

$$\mathcal{L}_{\text{int}} = -A_\mu J^\mu \,, \tag{431}$$

which is gauge invariant. Reading off the term linear in $A_\mu$ from our Lagrangian, we identify the current $J^\mu$ to be

$$\mathcal{L}_{\text{int}} = -qA_\mu\bar{\psi}\gamma^\mu\psi \implies J^\mu = \bar{\psi}\gamma^\mu\psi \,, \tag{432}$$

and treat the charge $q$ as a coupling constant. However, we could conceivably have other kinds of currents in more complicated theories. In any case, it turns out that gauge invariance requires that any such current is conserved. To see this, note that under a gauge transformation the interaction term transforms as

$$\mathcal{L}_{\text{int}} \to \mathcal{L}'_{\text{int}} = \mathcal{L}_{\text{int}} + (\partial_\mu\Lambda)J^\mu \,. \tag{433}$$

At first sight, this non-gauge invariance may be troubling. But, remember that the action $S_{\text{int}} = \int \mathrm{d}^4x\, \mathcal{L}_{\text{int}}$ is what really matters, and it changes by

$$S_{\text{int}} \to S'_{\text{int}} = S_{\text{int}} + \int \mathrm{d}^4x\, J^\mu\,\partial_\mu\Lambda \,. \tag{434}$$

Integrating the last term by parts, we have[35]

$$S'_{\text{int}} = S_{\text{int}} - \int \mathrm{d}^4x\, \Lambda\,\partial_\mu J^\mu \,. \tag{435}$$

For this to be gauge invariant, we must have $S'_{\text{int}} = S_{\text{int}}$, which requires that the second term above must vanish for any arbitrary function $\Lambda$. This is true only if $\partial_\mu J^\mu = 0$, i.e. the current is conserved.

---

[35]As usual, the surface term vanishes. This is because the fields must vanish at spatial infinity for any physically realizable configuration. Otherwise, they would have an infinite energy. Alternatively, we can always pick a gauge function $\Lambda$ which vanishes at infinity.

Next, let's turn our entire discussion thus far on its head and reconsider gauge invariance from a new perspective. Instead of thinking of gauge invariance is a peculiar property of the theory, let's instead consider gauge invariance as the *basis* of our theory. Suppose we start with the Dirac Lagrangian and *require* that it is invariant under local $U(1)$ transformations. For this to be possible, we need to replace the partial derivative with a covariant derivative to cancel unwanted terms. But, to do so requires introducing a gauge field $A_\mu$ whose transformation cancels the transformation of the Dirac field. So, imposing local $U(1)$ invariance implies the existence of the electromagnetic field.

If the only place that $A_\mu$ appears in our theory is inside the covariant derivative, it is essentially just a background field that doesn't have any dynamics of its own. If we want the gauge field to be dynamical, we need to add a kinetic term for it, and such a term must be gauge invariant. For the theory to also be renormalizable, there is a unique kinetic term that we can write down: the Maxwell term, $\mathcal{L}_{\text{Maxwell}} = -\frac{1}{4}F_{\mu\nu}F^{\mu\nu}$. The factor of $-1/4$ is just a convention, but the contraction $F_{\mu\nu}F^{\mu\nu}$ is the only renormalizable and gauge invariant term that exists. Putting this together with the gauge invariant version of the Dirac Lagrangian (with $\partial \to \mathcal{D}$), we have

$$\mathcal{L} = \bar{\psi}(i\slashed{D} - m)\psi - \frac{1}{4}F_{\mu\nu}F^{\mu\nu}, \tag{436}$$

which is nothing other than the QED Lagrangian! In summary, imposing local $U(1)$ invariance on a Dirac field automatically gives us QED. This is remarkable! Requiring a single local symmetry is sufficient to give us the correct form for one of humanity's most successful theories, which in the classical limit recovers all of classical electrodynamics.[36]

In fact, the core of the standard model was derived by generalizations of this line of thinking to more complicated local symmetries. The first generalization was made by Yang and Mills. Like much of modern physics the initial motivation was misguided, but the underlying idea and mathematics was correct. Then, in 1968 Weinberg and Salam (informed by ideas from Glashow) used this idea to unify electromagnetism with the weak interaction. It was further generalized by Gell-Mann, Leutwyler, Fritzsch, and others in 1973 to describe the strong interactions using the theory now known as quantum chromodynamics (QCD). The history of the standard model is an interesting subject in itself: consider reading this recent essay by Steve Weinberg.

To understand these more complicated gauge theories requires a respectable knowledge of group theory. As such, We've included a *very* cursory introduction to the subject in the next section, after which we will dive into QCD.

## 11.3 A Quick and Dirty Group Theory Primer

Strictly speaking, a group $G$ is simply a set endowed with a multiplication operation, $\cdot$, which obeys a few properties:

▶ The group is closed under multiplication, so $g_1 \cdot g_2 \in G$ for all $g_1, g_2 \in G$

▶ Group multiplication is associative, so $g_1 \cdot (g_2 \cdot g_3) = (g_1 \cdot g_2) \cdot g_3$ for all $g_1, g_2, g_3 \in G$.

▶ There exists an identity element $e$ such that $e \cdot g = g \cdot e = g$ for all $g \in G$.

▶ For each group element $g$ there exists an inverse element $g^{-1}$ such that $g^{-1} \cdot g = g \cdot g^{-1} = e$.

---

[36]If this seems ad hoc, there is a wonderful geometrical interpretation. If you're familiar with general relativity or differential geometry, the basic idea is that the gauge field is a connection which we use to parallel transport the phase of the fermion field.

And that's it! Notice that we did *not* require that multiplication be commutative, so we need not have $g_1 \cdot g_2 = g_2 \cdot g_1$. However, for some groups the multiplication *is* commutative, in which case we say the group is *Abelian*. If the group multiplication is not commutative, the group is said to be *non-Abelian*.

The conditions above aren't too restrictive, so there are lots of groups that we can dream up that come in all different shapes and sizes. Let's consider a few examples of groups to get the basic idea.

▶ $\mathbb{Z}_2$: The group is comprised of two elements, $\mathbb{Z}_2 = \{1, -1\}$ with the group multiplication simply being normal scalar multiplication. This automatically tells us that the group multiplication is associative and commutative, and thus $\mathbb{Z}_2$ is an Abelian group. It's also easy to check the other properties hold: the group is closed under multiplication, the identity element is 1, and the inverse elements $(1)^{-1} = 1$ and $(-1)^{-1} = -1$ exist. Since it has a finite number of elements, it is said to be a *finite* or *discrete* group. At this point, you can forget that discrete groups exist, because we won't talk about them again for the rest of the course.[37]

▶ $U(1)$: Our old friend $U(1)$ is defined as the set of complex numbers with modulus one, which we can write as the set $\{e^{i\theta} | \theta \in \mathbb{R}\}$. Again, the group multiplication is just normal scalar multiplication, so the group is Abelian and the associativity axiom is satisfied. The identity element is $e^{i0} = 1$, and the inverse of an element $e^{i\theta}$ is $e^{-i\theta}$. In contrast to our previous example, there is a continuously infinite number of elements in this group. A continuous group is called a *Lie group* (pronounced like "Lee"), after the mathematician Sophus Lie, and it is these groups that will be our primary focus.

▶ $O(N)$: This is the group of $N \times N$ orthogonal matrices,[38] that is, the set of all matrices $O$ such that $O^T O = \mathbb{1}$. Since these are matrices, the group multiplication is now matrix multiplication, which means it is associative but *not* commutative. So, in general matrix groups are non-Abelian! It's also quick to check that the identity element is just the unit matrix $\mathbb{1} \in G$ and for any element $O \in G$ its transpose $O^T$ is also in $G$, but by definition $O^T = O^{-1}$ so every element has an inverse in $G$.

▶ $SU(N)$: This is the guy we really care about. $SU(N)$ is the the group of $N \times N$ special unitary matrices. Special means that for all $U \in G$, $\det U = 1$ and unitary means that $U^\dagger U = \mathbb{1}$. Group multiplication is again just matrix multiplication, so this is a Non-Abelian group. The identity element is the unit matrix and for every $U \in G$ its Hermitian conjugate $U^\dagger$ is also in $G$, and since $U^\dagger = U^{-1}$ every element has an inverse.

Having learned about a few different groups, from now on we're just going to talk about $SU(N)$. To start, we'd like to understand infinitesimal transformations. As we'll see later, it turns out that the magic of Lie groups is that understanding these transformations is enough to understand the whole group. An infinitesimal transformation is a transformation that is infinitesimally close to the identity operator, which means that we can expand it as

$$U(\varepsilon) = \mathbb{1} + i\varepsilon M + \dots . \tag{437}$$

Here $\varepsilon$ is an infinitesimally small (real) parameter, which you can think of as small rotation angle in some higher dimensional space. The factor of $i$ is just a convention, and the $M$ is

---

[37]However, they are important if you want to talk about crystals!

[38]Technically, we're being a little sloppy here by identifying the group with its fundamental representation. However, in this course we'll only discuss matrix groups in their fundamental representation, so we won't get into any trouble.

some $N \times N$ matrix. There could be other terms in this expansion, but they are all of order $\varepsilon^2$, which is taken to be doubly small and negligible compared to the linear term.

For $U$ to be an $SU(N)$ matrix, we must have $U^\dagger U = \mathbb{1}$. Putting (437) into this formula and keeping only first order terms, we have

$$
\begin{aligned}
(\mathbb{1} - i\varepsilon M^\dagger + \dots)(\mathbb{1} + i\varepsilon M + \dots) &= \mathbb{1}\,, \\
\mathbb{1} - i\varepsilon M^\dagger + i\varepsilon M + \mathcal{O}\left(\varepsilon^2\right) &= \mathbb{1}\,, \\
\mathbb{1} + i\varepsilon(M - M^\dagger) + \mathcal{O}\left(\varepsilon^2\right) &= \mathbb{1}\,.
\end{aligned}
\tag{438}
$$

For this to hold to linear order, the second term in the last line must vanish for any $\varepsilon$, which implies

$$
M = M^\dagger\,.
\tag{439}
$$

That is, $M$ must be Hermitian. We also know that for $U$ to be special, we need $\det U = 1$. Since $\varepsilon$ is very small, you can play with the formula for the determinant to show that

$$
\det(\mathbb{1} + i\varepsilon M) \approx 1 + i\varepsilon \operatorname{tr} M\,.
\tag{440}
$$

For $\det U = 1$ to hold, the second term must vanish, which means $M$ is traceless,

$$
\operatorname{tr} M = 0\,.
\tag{441}
$$

We've now learned that an infinitesimal $SU(N)$ transformation can generically be written as the identity plus a traceless Hermitian matrix. The set of all such matrices is called the *Lie Algebra* of the group, and is usually written in gothic font, like $\mathfrak{su}(N)$. We can write down a basis for the Lie Algebra, called the *generators* of the group. We denote them $T^a$ where $a$ is an index which tells us which generator we're talking about. It turns out that it takes $N^2 - 1$ many matrices to form a basis for the space of $N \times N$ traceless Hermitian matrices, so the index $a$ runs from 1 to $N^2 - 1$.

To avoid confusion, I'll emphasize that $a$ is *not* the matrix index which tells us which row and column we're talking about. If we wanted to include these indices we'd write the $i^{th}$ row and $j^{th}$ column of the generator $T^a$ as $(T^a)_{ij}$, but because things get messy we'll usually leave these indices implicit. For reasons that we'll see in the next section, we'll call the $a$ indices (which label the generators) the *color indices*.

If the generators $\{T^a\}$ form a basis for the Lie Algebra, then we can write any $M \in \mathfrak{su}(N)$ as a linear combination of the generators,

$$
\varepsilon M = \sum_{a=1}^{N^2-1} \varepsilon^a T^a \equiv \varepsilon^a T^a\,,
\tag{442}
$$

where the parameters $\varepsilon^a$ tell us how much of $M$ is in the direction of the $a^{th}$ generator. We'll also use the Einstein summation convention with the color indices, but we'll always write color indices upstairs.[39]

The reason the Lie algebra is called an algebra is...because it's an algebra! Technically speaking, an algebra is a vector space with some kind of multiplication defined. Formally, in the theory of Lie groups this multiplication isn't matrix multiplication (because the product of two generators may not be an element of the Lie algebra) but rather the commutator, $[T^a, T^b]$. If you walk over to the math department, they'll usually call this a *Lie bracket* instead. The reason the commutator is so important is that the algebra is *closed* with respect to it. That is,

---

[39] This is because there is no metric we need to worry about (or technically, the metric is just $g_{ab} = \delta_{ab}$ so upstairs and downstairs are the same)

the commutator of two elements of the Lie algebra is guaranteed to be another element of the Lie algebra. This means that for any two elements $M, M' \in \mathfrak{su}(N)$, we know that $[M, M']$ is also in $\mathfrak{su}(N)$ and thus can be written as a linear combination of the generators (which are a basis),

$$[M, M'] = c^a T^a. \tag{443}$$

But we can also write $M$ and $M'$ in terms of the generators and some expansion coefficients. If we strip away all the unimportant details, what really matters is the commutator of the generators themselves,

$$[T^a, T^b] = i f^{abc} T^c. \tag{444}$$

The coefficients $f^{abc}$ are called the *structure constants* of the group, and provide a convenient way to characterize the entire Lie Algebra.

In terms of the generators, a general infinitesimal $SU(N)$ transformation (437) is written

$$U(\varepsilon) = \mathbb{1} + i\varepsilon^a T^a + \dots. \tag{445}$$

Now, suppose we have two different infinitesimal transformations,

$$U(\varepsilon_1) = \mathbb{1} + i\varepsilon_1^a T^a, \qquad U(\varepsilon_2) = \mathbb{1} + i\varepsilon_2^a T^a, \tag{446}$$

and would like to act with one after the other (i.e. rotate by $\varepsilon_1$ and then $\varepsilon_2$). Because this is a group, the combined transformation is simply the product of the individual transformations, so $U(\varepsilon_1 + \varepsilon_2) = U(\varepsilon_1)U(\varepsilon_2)$. In terms of (446), to first order in $\varepsilon$ we have

$$U(\varepsilon_1 + \varepsilon_2) = U(\varepsilon_1)U(\varepsilon_2) = \mathbb{1} + i(\varepsilon_1^a + \varepsilon_2^a)T^a + \dots. \tag{447}$$

So it looks like multiplication in the group is addition in the algebra (what does this remind you of?).

Inspired by this observation, let's now get to the big idea of Lie groups. Suppose we want to make a finite transformation $U(\theta)$, characterized by some finite parameters $\theta$. Instead of rotating by $\theta$ all at once, we can first rotate by $\theta/2$, and then stop and rotate by $\theta/2$ a second time. Or, we could rotate by $\theta/5$ five times, or $\theta/42$ 42 times, etc. In general, we can chop the angle up into $N$ different pieces, and then act $N$ times with the smaller transformation.[40] In equations,

$$U(\theta) = \underbrace{U(\theta/N)U(\theta/N)\dots U(\theta/N)}_{N \text{ times}} = \left(U(\theta/N)\right)^N. \tag{448}$$

And there's nothing stopping us from making $N$ really really big, so we can take

$$U(\theta) = \lim_{N \to \infty} \left(U(\theta/N)\right)^N. \tag{449}$$

In this limit, $U(\theta/N)$ is now an infinitesimal transformation, and we can write it as

$$U(\theta/N) = \mathbb{1} + \frac{i\theta^a T^a}{N}. \tag{450}$$

Putting (450) into (449) we have

$$U(\theta) = \lim_{N \to \infty} \left(\mathbb{1} + \frac{i\theta^a T^a}{N}\right)^N. \tag{451}$$

---

[40]Please don't confuse the $N$ here with the $N$ in $SU(N)$. Unfortunately there is a finite number of letters!

Perhaps you remember the identity

$$e^x = \lim_{N \to \infty} \left(1 + \frac{x}{N}\right)^N. \tag{452}$$

Even though we're working with matrices rather than numbers, this identity still applies, and we find that we can write any finite transformation as the exponentiation of an element of the Lie Algebra,

$$U(\theta) = \exp(i\theta^a T^a). \tag{453}$$

This is an extremely useful fact! Recall that the exponential of a matrix is defined by its Taylor Series,

$$\exp(iM) \equiv \sum_{n=1}^{\infty} \left(\frac{iM}{n!}\right)^n = \mathbb{1} + iM - \frac{1}{2}M^2 - \frac{i}{6}M^3 + \dots. \tag{454}$$

Before moving on, it's nice to notice that this is self-consistent with our previous claims derived from infinitesimal transformations. First, if we have an infinitesimally small transformation, we can just keep the leading order term in the above Taylor series,

$$\exp(i\varepsilon^a T^a) \approx \mathbb{1} + i\varepsilon^a T^a + \mathcal{O}\left(\varepsilon^2\right). \tag{455}$$

It's also clear that for $U$ to be unitary, the generators must be Hermitian[41]

$$\mathbb{1} = U^\dagger U = e^{-i\theta^a (T^a)^\dagger} e^{i\theta^a T^a} = e^{i\theta^a (T^a - (T^a)^\dagger)} \implies T^a = (T^a)^\dagger. \tag{457}$$

We can also show that $\det U = 1$ implies $\operatorname{tr} T^a = 0$ by using the identity[42] $\operatorname{tr} \log A = \log \det A$. Applying this to $U = \exp(i\theta^a T^a)$, we have

$$\begin{aligned}
\log \det e^{i\theta^a T^a} &= \operatorname{tr} \log e^{i\theta^a T^a} \\
&= \operatorname{tr}(i\theta^a T^a) \\
&= i\theta^a \operatorname{tr} T^a.
\end{aligned} \tag{458}$$

Exponentiating both sides, we have

$$\begin{aligned}
e^{\log \det e^{i\theta^a T^a}} &= e^{i\theta^a \operatorname{tr} T^a}, \\
\det e^{i\theta^a T^a} &= e^{i\theta^a \operatorname{tr} T^a}.
\end{aligned} \tag{459}$$

The left hand side is $\det U$, and setting this equal to one requires that

$$e^{i\theta^a \operatorname{tr} T^a} = 1 \implies i\theta^a \operatorname{tr} T^a = 0 \implies \operatorname{tr} T^a = 0. \tag{460}$$

To make all of these rather abstract ideas clear, let's consider what is by now a very familiar example: the spin rotation group, $SU(2)$. Since $SU(2)$ is $SU(N)$ with $N = 2$, all of our previous discussion holds. The important task is to identify the generators of the group. We need $2^2 - 1 = 3$ traceless Hermitian $2 \times 2$ matrices which form a basis for the Lie Algebra. Of course, we already know what these matrices are: the Pauli matrices!

$$\sigma^1 = \begin{pmatrix} 0 & 1 \\ 1 & 0 \end{pmatrix}, \quad \sigma^2 = \begin{pmatrix} 0 & -i \\ i & 0 \end{pmatrix}, \quad \sigma^3 = \begin{pmatrix} 1 & 0 \\ 0 & -1 \end{pmatrix}. \tag{461}$$

---

[41]The product of the exponentials of two matrices is actually given by the Baker-Campbell-Hausdorff formula,

$$e^A e^B = e^{A+B+\cdots}, \tag{456}$$

where the dots are a bunch of terms which depend on the commutator $[A, B]$. In this particular case, $[i\theta^a T^a, i\theta^b T^b] == 0$ so that $e^A e^B = e^{A+B}$.

[42]This is a particularly useful identity which appears often in the path integral formulation of field theory.

Traditionally, we normalize the generators to instead be

$$S^a = \frac{\sigma^a}{2}, \tag{462}$$

which gives us the canonical normalization

$$\text{tr}\, S^a S^b = \frac{1}{2}\delta^{ab}. \tag{463}$$

Comparing the familiar angular momentum algebra

$$[S^a, S^b] = i\varepsilon^{abc} S^c, \tag{464}$$

to (444), we see that the structure constants of $SU(2)$ are simply $f^{abc} = \varepsilon^{abc}$. Reading off (453), we see a rotation of the spin quantization axis is generated by the spin angular momentum operators,

$$U(\theta) = e^{i\theta^a S^a}. \tag{465}$$

Since this is a rotation in three-dimensional space, it is helpful to write the parameters $\theta^a$ as $\theta^a = \theta \hat{n}^a$, where $\hat{n}^a$ are the components of a unit vector. We can then interpret the transformation

$$U(\theta) = e^{i\theta \hat{n}^a S^a} = e^{i\theta \hat{\mathbf{n}}\cdot\mathbf{S}}, \tag{466}$$

as a rotation about the $\hat{\mathbf{n}}$ axis by the angle $\theta$. Perhaps this is something you've already seen in your quantum mechanics class. We can write this in a simpler form by using some properties of the Pauli matrices. The two key properties are the commutator and anti-commutator,

$$[\sigma^a, \sigma^b] = 2i\varepsilon^{abc}\sigma^c, \qquad \{\sigma^a, \sigma^b\} = 2\delta^{ab}. \tag{467}$$

We can then write the product of two Pauli matrices as

$$\begin{aligned}
\sigma^a \sigma^b &= \frac{1}{2}\left(\sigma^a \sigma^b + \sigma^b \sigma^a + \sigma^a \sigma^b - \sigma^b \sigma^a\right) \\
&= \frac{1}{2}\left(\{\sigma^a, \sigma^b\} + [\sigma^a, \sigma^b]\right) \\
&= \frac{1}{2}\left(2\delta^{ab} + 2i\varepsilon^{abc}\sigma^c\right) \\
&= \delta^{ab} + i\varepsilon^{abc}\sigma^c.
\end{aligned} \tag{468}$$

Let's use this to evaluate $(\hat{\mathbf{n}}\cdot\boldsymbol{\sigma})^2$,

$$\begin{aligned}
(\hat{\mathbf{n}}\cdot\boldsymbol{\sigma})^2 &= (\hat{n}^a \sigma^a)(\hat{n}^b \sigma^b) \\
&= \hat{n}^a \hat{n}^b \sigma^a \sigma^b \\
&= \hat{n}^a \hat{n}^b (\delta^{ab} + i\varepsilon^{abc}\sigma^c) \\
&= \hat{n}^a \hat{n}^a \\
&= (\hat{\mathbf{n}}\cdot\hat{\mathbf{n}})\,\mathbb{1} \\
&= \mathbb{1}.
\end{aligned} \tag{469}$$

To get from the third to the fourth line, we noticed that the contraction of the symmetric $\hat{n}^a \hat{n}^b$ with the antisymmetric $\varepsilon^{abc}$ vanishes, and to get to the last line we used the fact that $\hat{\mathbf{n}}$ is a unit vector so $|\hat{\mathbf{n}}| = 1$.

With this in the back of our mind, let's turn to the rotation (466),

$$U(\theta) = e^{i\theta \hat{\mathbf{n}}\cdot\mathbf{S}} = e^{i\theta \hat{\mathbf{n}}\cdot\boldsymbol{\sigma}/2} = \sum_{n=1}^{\infty}\left(\frac{i\theta}{2n!}\right)^n (\hat{\mathbf{n}}\cdot\boldsymbol{\sigma})^n. \tag{470}$$

We can divide the infinite sum into two pieces: the terms with $n$ even, and the terms with $n$ odd,

$$U(\theta) = \sum_{n \text{ even}} \left( \frac{i\theta}{2n!} \right)^n \left( \underbrace{(\hat{\mathbf{n}} \cdot \boldsymbol{\sigma})^2}_{\mathbb{1}} \right)^{n/2} + \sum_{n \text{ odd}} \left( \frac{i\theta}{2n!} \right)^n \left( \underbrace{(\hat{\mathbf{n}} \cdot \boldsymbol{\sigma})^2}_{\mathbb{1}} \right)^{n/2} (\hat{\mathbf{n}} \cdot \boldsymbol{\sigma}) \tag{471}$$

$$= \underbrace{\left( \sum_{n \text{ even}} \left( \frac{i\theta}{2n!} \right)^n \right)}_{\cos\left(\frac{\theta}{2}\right)} \mathbb{1} + \underbrace{\left( \sum_{n \text{ odd}} \left( \frac{i\theta}{2n!} \right)^n \right)}_{i \sin\left(\frac{\theta}{2}\right)} (\hat{\mathbf{n}} \cdot \boldsymbol{\sigma}) \tag{472}$$

$$= \cos\left( \frac{\theta}{2} \right) \mathbb{1} + i \sin\left( \frac{\theta}{2} \right) (\hat{\mathbf{n}} \cdot \boldsymbol{\sigma}). \tag{473}$$

This is a useful representation which will be helpful on one of the homework problems. Having acquainted ourselves with the basics of Lie groups, let's get back to the physics!

## 11.4 Quantum Chromodynamics

We'll now consider the strong interactions through the lens of Quantum Chromodynamics (QCD). The fundamental particles in this theory are the *quarks*, which were introduced in the early 1960's to explain observed patterns of strange and non-strange hadrons. At the time, these quarks had three known flavors: up, down, and strange.[43] They are all spin $1/2$ fermions, and the up quark has charge $+2/3$ while the down and strange quarks have charge $-1/3$. One of the key physical ideas of the time was $SU(3)$ flavor symmetry—a generalization of isospin, which made qualitative and approximate quantitative predictions about the hadrons. The strange quark is also noticeably heavier than the up and down quarks, which means the $SU(3)$ flavor symmetry of rotating the quarks into each other has significantly more explicit breaking then isospin.[44] However, aside from this mass difference the quarks interact with one another in the same way. Other particles which interact via the strong interaction are made out of quarks, and come in two varieties. *Baryons* are made out of three quarks (and thus are fermions), and *mesons* are made out of a quark-antiquark pair (and are bosons).

Given these new particles, one generalizes the $SU(2)$ isospin symmetry to an $SU(3)$ symmetry, with the extra degree of freedom being the *strangeness* of the particle. One can then use the properties of the $SU(3)$ group (along with perturbation theory) to make predictions for the masses of other particles. At the time the quark model was developed, group theory was not yet a standard tool of particle physics, and not everyone was well-versed in its use. A major advantage of the quark model was that it effectively did the group theory for you, provided the quarks had the properties outlined above. In this capacity, the quark model was successful in accounting for many of the observations of the day.

However, at the time of the model's invention, it wasn't actually clear whether quarks were real particles, or just a useful mathematical device for working out the details of $SU(3)$ flavor. After all, no one had ever been able to observe an isolated quark! Today quarks are considered real particles, but they are *confined*: they exist as individual entities inside of hadrons, but cannot be pulled apart. One simple intuitive way to understand how this can come about is

---

[43]We now know there are in fact six flavors of quarks: the up, down, strange, charm, top, and bottom

[44]If we arrange the quark flavors into a column, the theory would have an $SU(3)$ symmetry if it was left invariant by a transformation of the form

$$\begin{pmatrix} u \\ d \\ s \end{pmatrix} \to U \begin{pmatrix} u \\ d \\ s \end{pmatrix},$$

where $U$ is an $SU(3)$ matrix.

for the theory to have the property that before you have enough energy to the quarks apart, you would have enough energy to pull a whole new hadron out of the vacuum.

It was also noticed that the quark model had a seemingly fatal flaw: it appeared to violate the spin-statistics theorem.[45] We can consider, for example, the $\Delta$ baryon. It is a low-energy excitation (in a simple model, it can be thought as a simultaneous flip of the spin and isospin of a nucleon) so we expect it to be in a spatial *s*-wave state (such that the energy from orbital angular momentum is minimized). At this point, recall that an *s*-wave configuration is spatially symmetric under particle interchange.

As we may have mentioned in the previous section, the $\Delta$ has isospin $I = 3/2$. Since each quark has isospin 1/2, this means that the $\Delta$'s three constituent quarks are in an isospin symmetric combination.[46] The $\Delta$ also has spin $s = 3/2$, and since the quarks are all spin 1/2, they are in a spin-symmetric configuration as well. Having exhausted all of the $\Delta$'s quantum numbers, it seems to be a fully symmetric state of three quarks. However, the quarks are fermions, and basic quantum mechanics tells us that they have to exist in an anti-symmetric state. This is obviously a major problem!

To fix this, we can postulate that the quark has some other property, let's call it *color*, and the $\Delta$ is antisymmetric with respect to it. In fact, Maryland's Wally Greenberg was the first to introduce something equivalent to color, but it didn't catch on with community in any significant way until Nambu re-formulated it in a more transparent fashion. The idea is that a quark (in addition to its flavor) comes in three colors: red, blue, and green. The strong interaction is taken to be completely symmetric with respect to color, i.e. if we arrange to colors as a column, QCD is invariant under transformations

$$\begin{pmatrix} r \\ b \\ g \end{pmatrix} \rightarrow U \begin{pmatrix} r \\ b \\ g \end{pmatrix}, \tag{474}$$

where $U$ is an $SU(3)$ matrix. We can understand confinement as the statement that all physical states are color neutral, or white. For example, baryons are configurations of three quarks that are antisymmetric with respect to color,

$$\text{Baryon} = r\,b\,g - b\,r\,g + b\,g\,r - g\,b\,r + g\,r\,b - r\,g\,b. \tag{475}$$

Similarly, a meson is the color-neutral quark-antiquark combination

$$\text{Meson} = r\bar{r} + b\bar{b} + g\bar{g}. \tag{476}$$

To translate these ideas into mathematics, we start by introducing the quark field, $q$, which has three different sets of indices. First of all, the quark is a fermion, so we represent it as a Dirac spinor, and thus it has an index $\alpha = 1, 2, 3, 4$ which specifies the four spinor components. The quark comes in one of three flavors, so it has a flavor index $f = 1, 2, 3$ which tells us whether it is up, down, or strange.[47] Finally, it has a color index $a = 1, 2, 3$ which tells us what color the quark is. So, if we wanted we could write the quark field as $q_{\alpha, f, a}$. As we're about to see, it is the color index which is central to QCD, so we'll typically suppress the flavor and spinor indices and just write $q_a$. Color indices will always be taken from the beginning of the roman alphabet $a, b, c, \ldots$ and written downstairs.

---

[45]This is the statement that integer spin particles are bosons and half-integer spin particles are fermions. Basically, it comes from the fact that if you try to quantize a spinor (half-integer spin) field with (bosonic) commutation relations, or a scalar/vector (integer spin) field with (fermionic) anti-commutation relations, then very bad things happen.

[46]$1/2 + 1/2 + 1/2 = 3/2$

[47]Again, we are neglecting the charm, top, and bottom quarks.

Having established notation, we can now state the core ideas of QCD. We mentioned that the strong interaction is invariant under a rotation of the color basis as in (474). The idea of a non-Abelian gauge theory such as QCD is to promote this global ($U$ is independent of spacetime) symmetry to a much stronger local symmetry, where $U(\mathbf{x}, t)$ is a spacetime-dependent $SU(3)$ matrix acting on the color indices. That is, we require that

$$q_a(\mathbf{x}, t) \rightarrow q'_a(\mathbf{x}, t) = U_{ab}(\mathbf{x}, t) q_b(\mathbf{x}, t), \tag{477}$$

is a symmetry of the theory. This is completely analogous to requiring local phase invariance in E&M, just now we are dealing with the non-Abelian group $SU(3)$ rather than the Abelian group $U(1)$. Since the quarks are fermions they should be described by the Dirac Lagrangian,

$$\mathcal{L} = \bar{q}(i\slashed{\partial} - m)q. \tag{478}$$

Let's see how this behaves under the transformation (477),

$$
\begin{aligned}
\bar{q}(i\slashed{\partial} - m)q &\rightarrow \bar{q}U^\dagger(i\slashed{\partial} - m)Uq \\
&= \bar{q}U^\dagger\big(Ui\slashed{\partial}q + {\color{blue}i(\slashed{\partial}U)q} - Umq\big) \\
&= \bar{q}\underbrace{U^\dagger U}_{\mathbb{1}} i\slashed{\partial}q + {\color{blue}i\gamma^\mu U^\dagger(\partial_\mu U)} - m\bar{q}\underbrace{U^\dagger U}_{\mathbb{1}}q \\
&= \bar{q}(i\slashed{\partial} + {\color{blue}i\gamma^\mu(U^\dagger\partial_\mu U)} - m)q.
\end{aligned} \tag{479}
$$

where the extra term generated by the derivative acting on the color matrix is written in blue. Just as in the Abelian case, this blue term means that the Dirac Lagrangian is not gauge-invariant. To get rid of this we can use the same trick we used in QED, which was to introduce a covariant derivative $\mathcal{D}_\mu$ that cancels the extra term. To do this, we need to introduce a *gauge field* $A_\mu$, which is analogous to the electromagnetic potential. The gauge covariant derivative is then

$$\mathcal{D}_\mu = \partial_\mu + igA_\mu, \tag{480}$$

where $g$ is a coupling constant. Replacing $\partial \rightarrow \mathcal{D}$ in the Dirac Lagrangian, we have

$$\mathcal{L} = \bar{q}(i\slashed{\mathcal{D}} - m)q = \bar{q}(i\slashed{\partial} - g\slashed{A} - m)q. \tag{481}$$

By requiring this Lagrangian be gauge invariant, we can deduce the transformation of $A_\mu$ under a color rotation. It's not hard to show the correct transformation is

$$A_\mu \rightarrow A'_\mu = UA_\mu U^\dagger + \frac{i}{g}(\partial_\mu U)U^\dagger. \tag{482}$$

We can check this is correct by considering the transformation of (481),

$$
\begin{aligned}
\bar{q}(i\slashed{\partial} - g\slashed{A} - m)q &\rightarrow \bar{q}(i\slashed{\partial} + {\color{blue}i(U^\dagger\slashed{\partial}U)} - gU^\dagger\slashed{A}'U - m)q \\
&= \bar{q}\left[i\slashed{\partial} + {\color{blue}i(U^\dagger\slashed{\partial}U)} - gU^\dagger\big(U\slashed{A}U^\dagger + {\color{red}\frac{i}{g}(\slashed{\partial}U)U^\dagger}\big)U - m\right]q \\
&= \bar{q}\left[i\slashed{\partial} + {\color{blue}i(U^\dagger\slashed{\partial}U)} - g\underbrace{U^\dagger U}_{\mathbb{1}}\slashed{A}\underbrace{U^\dagger U}_{\mathbb{1}} - {\color{red}g\frac{i}{g}U^\dagger(\slashed{\partial}U)\underbrace{U^\dagger U}_{\mathbb{1}}} - m\right]q \\
&= \bar{q}\left[i\slashed{\partial} + {\color{blue}i(U^\dagger\slashed{\partial}U)} - g\slashed{A} - {\color{red}iU^\dagger(\slashed{\partial}U)} - m\right]q \\
&= \bar{q}(i\slashed{\partial} - g\slashed{A} - m)q.
\end{aligned} \tag{483}
$$

So, everything does indeed work out and this version of the Dirac Lagrangian is gauge invariant. Let's now take a step back and pay a little more attention to this object $A_\mu$. First of all,

notice that it is a $3 \times 3$ matrix in color space! The simplest way to see this from the second term in the transformation (482), which is a product of $U$'s and is thus clearly a matrix.

The gauge field must also be Hermitian, since a term like $\bar{q}(g\slashed{A})q$ appears in the Lagrangian, and hence will also appear in the Hamiltonian, which is an observable quantity that must be represented by a Hermitian operator. It's easy enough to require $A_\mu$ be Hermitian, but it must stay Hermitian under gauge transformations, and the $i(\partial_\mu U)U^\dagger$ term in (482) looks like it could be a problem, in that it is not manifestly Hermitian. The key point is that $U$ is unitary, so $UU^\dagger = \mathbb{1}$. If we differentiate this, we obviously have $\partial_\mu \mathbb{1} = 0$, so $\partial_\mu(UU^\dagger) = 0$ as well. Using the product rule, this means

$$(\partial_\mu U)U^\dagger + U(\partial_\mu U^\dagger) = 0 \implies (\partial_\mu U)U^\dagger = -U(\partial_\mu U^\dagger). \tag{484}$$

Multiplying by $i$, we have

$$i(\partial_\mu U)U^\dagger = -iU(\partial_\mu U^\dagger). \tag{485}$$

The left-hand side is the term in the transformation we're worried about. Note that it's Hermitian conjugate is

$$\left(i(\partial_\mu U)U^\dagger\right)^\dagger = -iU(\partial_\mu U^\dagger), \tag{486}$$

which is precisely the right-hand side of (485). So, $i(\partial_\mu U)U^\dagger = [i(\partial_\mu U)U^\dagger]^\dagger$, which is to say the term is Hermitian, and we have no problems.

Further, $A_\mu$ must be traceless. The reason is that we are concerned with rotations in color space, and the trace just gives us an overall phase rotation that doesn't mix up the colors. So, in all we find that the gauge field $A_\mu$ is a $3 \times 3$ traceless Hermitian matrix.

But, recall from our group theory review that the set of all $3 \times 3$ traceless Hermitian matrices comprise the Lie algebra of the group $SU(3)$! This means that $A_\mu$ lives in the Lie Algebra, and thus can be written as a linear combination of the generators of $SU(3)$, which are called the *Gell-Mann matrices*, and are denoted by $\lambda^a$ with $a = 1, \dots 8$.[48] You can think of these as the $SU(3)$ version of the Pauli matrices. So, we can write

$$A_\mu = \sum_{a=1}^{8} A_\mu^a \lambda^a \equiv A_\mu^a \lambda^a. \tag{487}$$

In case you're curious, the Gell-Mann matrices are

$$\lambda^1 = \begin{pmatrix} 0 & 1 & 0 \\ 1 & 0 & 0 \\ 0 & 0 & 0 \end{pmatrix}, \qquad \lambda^2 = \begin{pmatrix} 0 & -i & 0 \\ i & 0 & 0 \\ 0 & 0 & 0 \end{pmatrix}, \qquad \lambda^3 = \begin{pmatrix} 1 & 0 & 0 \\ 0 & -1 & 0 \\ 0 & 0 & 0 \end{pmatrix},$$

$$\lambda^4 = \begin{pmatrix} 0 & 0 & 1 \\ 0 & 0 & 0 \\ 1 & 0 & 0 \end{pmatrix}, \qquad \lambda^5 = \begin{pmatrix} 0 & 0 & -i \\ 0 & 0 & 0 \\ i & 0 & 0 \end{pmatrix}, \qquad \lambda^6 = \begin{pmatrix} 0 & 0 & 0 \\ 0 & 0 & 1 \\ 0 & 1 & 0 \end{pmatrix},$$

$$\lambda^7 = \begin{pmatrix} 0 & 0 & 0 \\ 0 & 0 & -i \\ 0 & i & 0 \end{pmatrix}, \qquad \lambda^8 = \frac{1}{\sqrt{3}} \begin{pmatrix} 1 & 0 & 0 \\ 0 & 1 & 0 \\ 0 & 0 & -2 \end{pmatrix}.$$

These are chosen such that they have the standard normalization $\operatorname{tr} \lambda^a \lambda^a = 2$, just like the Pauli matrices. Now, the $A_\mu$ field is a matrix in color space, while the $A_\mu^a$, are coefficients specifying which matrix and are thus easier to work with. The spacetime index $\mu$ just tells us that all of these objects transform like vectors under Lorentz transformations, which is not our prime concern right now.

---

[48]Recall we have $N^2 - 1$ generators for $SU(N)$. With $N = 3$, this is $3^2 - 1 = 8$ generators.

Given the generators, we also know that we can write any $SU(3)$ color rotation as

$$U = \exp(i\theta^a \lambda^a), \tag{488}$$

and that they close under commutation, $[\lambda^a, \lambda^b] = if^{abc}\lambda^c$. The structure constants for $SU(3)$ are much more complicated than those of $SU(2)$. We won't write them all down, but for example $[\lambda^1, \lambda^2] = i\lambda^3$, $[\lambda^1, \lambda^4] = i\lambda^7$, and you can look up all of the rest.

So far, we've discussed just a single quark, but it's trivial to generalize our theory to all of the flavors. We simply reinstate the flavor index and sum over it,

$$\mathcal{L}_{\text{quark}} = \sum_f \bar{q}_f (i\slashed{D} - m_f) q_f, \tag{489}$$

where we've allowed the flavors to have different masses $m_f$. There's one final ingredient missing: the $A_\mu^a$ fields are not dynamical. That is, if we find the equations of motion for $A_\mu^a$, we just get

$$\sum_f \bar{q}_f \gamma^\mu \lambda^a q_f = 0. \tag{490}$$

This doesn't tell us anything about $A_\mu^a$, which is to say that it has no role in the dynamics of the theory. To give the gauge field dynamics we need to add a gauge invariant, renormalizable kinetic term (that is, it must include derivatives of $A_\mu^a$) to the Lagrangian. If we also want to maintain discrete symmetries, particularly time reversal, there is only one such term that exists:

$$\mathcal{L}_{\text{gluon}} = -\frac{1}{8} \text{tr} \, (\mathcal{D}_\mu A_\nu - \mathcal{D}_\nu A_\mu)(\mathcal{D}^\mu A^\nu - \mathcal{D}^\nu A^\mu), \tag{491}$$

where the trace is over the color indices. We can simplify this by expanding

$$
\begin{aligned}
\mathcal{D}_\mu A_\nu - \mathcal{D}_\nu A_\mu &= \partial_\mu A_\nu + igA_\mu A_\nu - \partial_\nu A_\mu - igA_\nu A_\mu \\
&= \partial_\mu A_\nu - \partial_\nu A_\mu + ig[A_\mu, A_\nu] \\
&= \sum_a (\partial_\mu A_\nu^a - \partial_\nu A_\mu^a)\lambda^a + ig \left[ \sum_b A_\mu^b \lambda^b, \sum_c A_\nu^c \lambda^c \right] \\
&= \sum_a (\partial_\mu A_\nu^a - \partial_\nu A_\mu^a)\lambda^a + ig \sum_{b,c} A_\mu^b A_\nu^c \underbrace{[\lambda^b, \lambda^c]}_{if^{bca}\lambda^a} \\
&= \sum_a (\partial_\mu A_\nu^a - \partial_\nu A_\mu^a)\lambda^a - g \sum_a A_\mu^b A_\nu^c f^{bca}\lambda^a \\
&= \sum_a \underbrace{\left( \partial_\mu A_\nu^a - \partial_\nu A_\mu^a - gf^{abc} A_\mu^b A_\nu^c \right)}_{\equiv F_{\mu\nu}^a} \lambda^a.
\end{aligned}
\tag{492}
$$

In the last line, we've defined the components of the *non-Abelian field strength*,

$$F_{\mu\nu}^a = \partial_\mu A_\nu^a - \partial_\nu A_\mu^a - gf^{abc} A_\mu^b A_\nu^c. \tag{493}$$

In terms of which we can rewrite the gauge field Lagrangian as simply

$$\mathcal{L}_{\text{gluon}} = -\frac{1}{4} F_{\mu\nu}^a F^{\mu\nu,a}, \tag{494}$$

which looks just like the Maxwell Lagrangian! Just like the electromagnetic potential field $A_\mu$ represents the photon in QED, the eight fields $A_\mu^a$ represent *gluons* in QCD, which are the massless gauge bosons that mediate the strong force, just as photons mediate the electromagnetic force. However, the extra terms in the gluon Lagrangian due to the non-Abelian nature

of the $SU(3)$ gauge symmetry give the gluons a much richer set of interactions. First off, note that since the gluon fields $A_\mu^a$ carry a color index, they possess color charge, in contrast to the photon in QED which does not carry electric charge. As a result, the gluons can have self-interactions.

To see this more concretely, let's consider some of the terms in the gluon Lagrangian and their corresponding diagrams. Denoting the gluon propagator as a curly line, we have a three gluon interaction,

$$-2g(\partial_\mu A_\nu^a - \partial_\nu A_\mu^a)(f^{abc}A^{\mu,b}A^{\nu,c}) = \qquad , \qquad (495)$$

as well as a four gluon interaction,

$$4g^2 f^{abc} f^{ade} A_\mu^b A_\nu^c A^{\mu,d} A^{\nu,e} = \qquad . \qquad (496)$$

This means that the gluons are self-interacting in a non-Abelian gauge theory, owing to the fact that they carry color charge and can therefore interact via gluon exchange. Recall that this is *not* the case in QED: the photon does not carry charge, and thus there are no direct photon-photon interactions. The existence of the gluon-gluon interactions are responsible for many of the rich features of QCD, and non-Abelian gauge theories in general.

# 12 Broken Symmetries

So far, these notes have mainly focused on the strong interaction, but the weak interaction is also of great interest if we wish to get a complete picture of fundamental physics. We arrived at QCD by promoting the global $SU(3)$ color symmetry to a local gauge symmetry, and one could reasonably hope that a similar procedure could apply to the weak interactions. After all, one of the failures of the Fermi theory (discussed briefly in section 9.6) was that it was non-renormalizable, and introducing gauge fields could rectify this. The second shortcoming of the Fermi theory was that it conserved parity, whereas since the 1950s the weak interaction had been known to violate parity: nature is left-handed. This means that any gauge field we introduce should couple only to left-handed currents, which for quarks would be

$$J_L^\mu = \bar{q}\big(i\gamma^\mu(1-\gamma^5)F\big)q\,, \tag{497}$$

where the $1-\gamma^5$ projects out only the left-handed components, and $F$ is some flavor dependent matrix. However, in a gauge theory this seemingly harmless requirement raises major issues: recall from section 11 that for the Lagrangian (including the source term $A_\mu J^\mu$) to be gauge invariant, the current must be conserved, $\partial_\mu J^\mu = 0$. So, a gauge theory of the weak interactions must have $\partial_\mu J_L^\mu = 0$. We can write the left-handed current (497) as the difference between a vector and axial current,

$$J_L^\mu = J_V^\mu - J_A^\mu\,, \tag{498}$$

where

$$J_V^\mu \sim \bar{\psi}(\gamma^\mu F)\psi\,, \qquad J_A^\mu \sim \bar{\psi}(\gamma^\mu \gamma^5 F)\psi\,. \tag{499}$$

We now face two issues: first, a vector current is only conserved if the masses of all of the particles involved are the same. As we've discussed, the masses of the up and down quark are not quite the same, which is problematic if we want quarks to participate in weak interactions (we can't play the usual game of saying the symmetry is only approximate, as a gauge symmetry is a redundancy, not a physical symmetry, and thus must be exact in a consistent theory). Secondly, if we want the axial current to be conserved the particles must be massless (we saw this in section 7). Neglecting any complications due to multiple flavors, a fermion field transforms under an axial rotation as

$$\psi \to e^{i\theta\gamma^5}\psi\,, \tag{500}$$

which, as is shown in one of the homework problems, implies that a mass term is not invariant under this transformation,

$$\bar{\psi}\psi \to \cos(2\theta)\bar{\psi}\psi + i\sin(2\theta)\bar{\psi}\gamma^5\psi \neq \bar{\psi}\psi\,. \tag{501}$$

Despite these apparent obstacles, it turns out that by introducing some new ideas we can still successfully formulate the weak interactions as a gauge theory. Namely, we can imagine that there is a symmetry of the fundamental theory which ensures that particles are massless, but which is *spontaneously broken* in the universe (and energy scales) that we observe. Understanding this notion of spontaneous symmetry breaking in field theories will occupy us for the rest of the section. We will begin by briefly considering spontaneous symmetry breaking in the more intuitive contexts of single particle physics and then condensed matter physics (where the ideas were initially developed, and remain a cornerstone of the field) before moving on to consider its role in fundamental physics.



SciPost Phys. Lect. Notes 34 (2021)

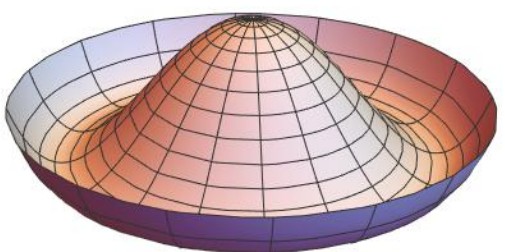

Figure 11: The Mexican Hat Potential

## 12.1 Symmetry Breaking in Single Particle Physics

Let's start by considering a simple classical mechanics problem: a particle confined to the $xy$ plane and subject to the potential

$$V(x,y) = \frac{V_0}{R^4}\left(x^2 + y^2 - R^2\right)^2, \tag{502}$$

where $R$ is a parameter with dimensions of length, and $V_0$ has dimensions of energy. This is called the *Mexican hat* or *wine bottle* potential, which is shown in Fig. 11 (for those unused to wine, it turns out that red wine typically comes in bottles whose bottoms are raised in the center so that sediment can settle in the rim), and will appear in several different contexts within this section. In this case, it is simply the potential experienced by a single particle in real space. It is easiest to work in polar coordinates, where $x = r\cos\theta$, $y = r\sin\theta$, and the Lagrangian can be written

$$\begin{aligned}
L &= \frac{1}{2}m(\dot{x}^2 + \dot{y}^2) - V(x,y) \\
&= \frac{1}{2}m(\dot{r}^2 + r^2\dot{\theta}^2) - \frac{V_0}{R^4}\left(r^2 - R^2\right)^2.
\end{aligned} \tag{503}$$

Notice that the system is rotationally invariant (i.e. $\frac{\partial L}{\partial \theta} = 0$) and in accordance with our discussion of Noether's theorem, the angular momentum $J = mr^2\dot{\theta}$ is conserved ($\dot{J} = 0$).

Now, let us consider the ground state of the classical theory. We don't need to perform any calculations to realize that the system's energy will be minimized if the particle sits at rest at some point along the rim (i.e. the minimum) of the potential at $r = R$. However, notice that this state is *not* rotationally invariant: the particle must "pick" one point along the rim, and that point will move under rotations. This is in fact the definition of spontaneous symmetry breaking: the ground state of the system does not respect one of the symmetries of the Lagrangian. Note that this phenomenon is not generic to any conceivable system: if we instead had the potential $V(r,\theta) = r^2$ the particle would reside at $r = 0$ in the ground state, maintaining the rotational variance of the Lagrangian.

> **Vibrational Modes**
>
> There is an interesting general result that holds as a consequence of symmetry breaking in classical single particle physics. Namely, symmetry breaking is associated with a zero-frequency vibrational mode (this is the baby version of Goldstone's theorem which we

discuss below). In classical mechanics, we can typically consider small deviations from the energy-minimizing configuration, and usually end up with oscillatory modes with a frequency

$$\omega = \sqrt{\frac{V''(\mathbf{r}_0)}{m}}, \tag{504}$$

where $\mathbf{r}_0$ is the position of the minimum (this is just a linearized restoring force). In the presence of a symmetry, these modes become degenerate: for example the rotational symmetry of a bowl-shaped potential means that climbing up the wall in the $x$ or $y$ (or any linear combination) direction both meet the same restoring force, and hence the oscillations have the same frequency. On the other hand, if the particle is in a Mexican hat potential, it can roll around the rim of the hat with no restoring force, corresponding to a zero-frequency mode. This is a useful picture to have in one's head when dealing with the more abstract realizations of symmetry breaking discussed in the remainder of this section.

If we consider this problem quantum mechanically, it turns out that the ground state has zero angular momentum: the wave function spreads itself evenly about the rim of the potential in a rotationally invariant manner, and thus there is no spontaneous symmetry breaking. However, if we consider a system with an infinite number of degrees of freedom (i.e. a field theory), the story can become more interesting, owing to the fact that the ground state can become infinitely degenerate.

## 12.2 Symmetry Breaking and Phase Transitions

At some point in your statistical mechanics class, you've probably met the Ising model, which describes a bunch of "spins" living on a lattice (let's say in two dimensions) governed by the Hamiltonian

$$H = -J \sum_{\langle i,j \rangle} \sigma_i^z \sigma_j^z . \tag{505}$$

This model represents a bunch of spins $\sigma$ sitting on a lattice (let's say in three dimensions) whose sites are labeled by $i$. Each spin can take only take one of two values: $\sigma_i^z = \pm 1$, i.e. it can point up or point down. To make things interesting, the spins can talk to their nearest neighbors through the interaction in the Hamiltonian, where the symbol $\langle i, j \rangle$ means that we should sum over all pairs of nearest neighbors (indicated as green links in Fig. 12). If $J > 0$ the energy of a pair will be minimized if $\sigma_i \sigma_j = 1$, i.e. neighboring spins want to line up and point in the same direction, while if $J < 0$ the spins will want to anti-align. Notice that this Hamiltonian has a $\mathbb{Z}_2$ symmetry, under which we can flip every spin on the lattice,

$$\sigma_i \rightarrow -\sigma_i \quad \forall i . \tag{506}$$

Now, let's consider the ground state of the system, let's say for $J > 0$. The energy of the system will be minimized if all of the spins point in the same direction, either all up or all down. That is, we have two degenerate ground states. However, if all of the spins point in the same direction, the ground state is no longer invariant under the $\mathbb{Z}_2$ transformation: flipping all of the spins turns one ground state into the other! So, the symmetry of the Hamiltonian is not a symmetry of the ground state, and thus we say the $\mathbb{Z}_2$ symmetry has been spontaneously broken.

If we consider this system at high temperatures, it is unlikely that we will find it in its ground state. This is simply because entropy now comes into play and there are many more configurations with random spin distributions than there are configurations with all of the



Figure 12: Spins on a lattice. The nearest-neighbor links around the central spin are indicated in green.

spins aligned. Such "random" states will respect the $\mathbb{Z}_2$ symmetry of the Hamiltonian, since on average there will be equal numbers of up and down spins or, put another way, the average spin of the system, or magnetization, $\langle \sigma_i \rangle \equiv m$ will be zero. This disordered high temperature state is qualitatively very different from the ordered ground state. In fact, they represent two distinct phases of matter which we can distinguish by their symmetry properties. At high temperatures the system respects the $\mathbb{Z}_2$ symmetry and the magnetization vanishes, so we call this the paramagnetic state, while at low temperatures the spins line up, breaking the $\mathbb{Z}_2$ symmetry and giving rise to a nonzero magnetization, in what is called the ferromagnetic state. This structure is generic, and is the foundation for the modern theory of phase transitions, as initially developed by Lev Landau in the 1950's.

In fact, its very easy to extend this to more sophisticated models. For example, if we allow the spins to now point in any direction in three-dimensional space, which amounts replacing the binary variables $\sigma_i^z$ on each site with unit vectors $\mathbf{S}_i$, we have the Heisenberg model,

$$H = -J \sum_{\langle i,j \rangle} \mathbf{S}_i \cdot \mathbf{S}_j . \tag{507}$$

This model has a continuous $O(3)$ rotational symmetry (under which $\mathbf{S}_i \rightarrow \mathbf{R}\,\mathbf{S}_i$ with $\mathbf{R}$ an orthogonal matrix), reflecting the spatial isotropy of the system. However, in the ground state all of the spins align, "choosing" a particular axis and breaking the rotational symmetry of the Hamiltonian. Now, there are an infinite number of different ground states, each corresponding to a different magnetization axis. This can be visualized using the Mexican hat potential, where we now interpret the radial direction as the magnitude of the magnetization (which is fixed in the ground state and determined by the parameters of the system) and the angular direction as the possible spatial orientations of the magnetization vector. The rim of the hat corresponds to the degenerate ground state manifold, each point along it representing a different possible ground state configuration. Note that now the Mexican hat lives in the space of possible magnetization vectors, *not* real space. The Mexican hat potential will inhabit an analogous space of possible field configurations in particle physics models, to which we will now turn.

## 12.3 Symmetry Breaking in Field Theories

In the previous section, we saw that simple models of magnets can have multiple degenerate ground states, and spontaneously break the symmetries of their Hamiltonian. The same can occur in models of fundamental physics, in which case the different ground states correspond to different *vacua*: all of which are equally valid but quantum mechanically disconnected vacuum states of our universe.

The notion of spontaneous symmetry breaking was first introduced to field theory in an effort to explain why the pion was anomalously light (the pion has a mass of $\sim 135$ MeV, which is five times lighter than other non-strange mesons). To explain this, it was conjectured that

the strong interactions enjoyed a second approximate symmetry (in addition to isospin) that was then spontaneously broken by our vacuum. To illustrate this idea, we will consider a toy model initially introduced by Gell-Mann and Levy, defined by the Lagrangian

$$\mathcal{L} = \frac{1}{2}\big(\partial_\mu \sigma \partial^\mu \sigma + \partial_\mu \boldsymbol{\pi} \cdot \partial^\mu \boldsymbol{\pi}\big) - \frac{\lambda}{4}\big(\sigma^2 + \boldsymbol{\pi} \cdot \boldsymbol{\pi} - f^2\big)^2, \tag{508}$$

where $\sigma$ is a (Lorentz) scalar field which also transforms as a scalar under isospin rotations and $\boldsymbol{\pi} = (\pi^x, \pi^y, \pi^z)$ are (Lorentz) pseudo-scalar fields which transform as a vector under isospin rotations. We also have the parameters $\lambda$ and $f$ and, to be explicit,

$$\partial_\mu \boldsymbol{\pi} \cdot \partial^\mu \boldsymbol{\pi} = \partial_\mu \pi^x \partial^\mu \pi^x + \partial_\mu \pi^y \partial^\mu \pi^y + \partial_\mu \pi^z \partial^\mu \pi^z. \tag{509}$$

This Lagrangian is invariant under $O(4)$ rotations of the $\sigma$ and $\boldsymbol{\pi}$ fields into one another. That is, we can mix up the definitions of the fields using orthogonal matrices, like

$$\begin{pmatrix} \sigma \\ \boldsymbol{\pi} \end{pmatrix} \to R \begin{pmatrix} \sigma \\ \boldsymbol{\pi} \end{pmatrix}. \tag{510}$$

In the vacuum state, the energy of the system will be minimized, including the potential term

$$V = \frac{\lambda}{4}\big(\sigma^2 + \boldsymbol{\pi} \cdot \boldsymbol{\pi} - f^2\big)^2. \tag{511}$$

This is of course minimized when $\sigma^2 + \boldsymbol{\pi} \cdot \boldsymbol{\pi} = f^2$. This condition specifies a three-dimensional surface in the four-dimensional space of possible field configurations, and can be thought of as the higher-dimensional analogue of the rim of the Mexican hat shown above. Any point along the surface represents a different vacuum state, and in "choosing" one to occupy, the $O(4)$ symmetry of the Lagrangian is spontaneously broken. Although any such point is equally valid, for the sake of concreteness it is useful to pick a particular vacuum, let's say along the $\sigma$ direction. That is, we can write

$$\sigma = f + \delta\sigma, \tag{512}$$

where $f$ is the vacuum value of the field, and the fluctuations around it, $\delta\sigma$, are what we see as the physical field. Substituting this decomposition back into the Lagrangian (508), we have

$$\mathcal{L} = \frac{1}{2}\big(\partial_\mu(\delta\sigma)\partial^\mu(\delta\sigma) + \partial_\mu \boldsymbol{\pi} \cdot \partial^\mu \boldsymbol{\pi}\big) - \frac{\lambda}{4}\big(2f\delta\sigma + (\delta\sigma)^2 + \boldsymbol{\pi} \cdot \boldsymbol{\pi}\big)^2. \tag{513}$$

If we expand the potential term, we can see the full set of interactions contained in the theory,

$$V = \lambda\left(f^2 \delta\sigma^2 + f\delta\sigma^3 + \frac{1}{4}\delta\sigma^4 + \frac{1}{4}\big(\boldsymbol{\pi} \cdot \boldsymbol{\pi}\big)^2 + f\delta\sigma\boldsymbol{\pi} \cdot \boldsymbol{\pi} + \frac{1}{2}\delta\sigma^2 \boldsymbol{\pi} \cdot \boldsymbol{\pi}\right). \tag{514}$$

The first term tells us that the $\delta\sigma$ field has a mass of $m_\sigma^2 = 2\lambda f^2$, and the remaining terms encode the allowed processes,

$$\lambda f\delta\sigma^3 \sim \tag{515}$$

$$\frac{\lambda}{4}\delta\sigma^4 \sim$$

$$\lambda f\delta\sigma\boldsymbol{\pi} \cdot \boldsymbol{\pi} \sim \tag{516}$$

$$\frac{\lambda}{2}\delta\sigma^2 \boldsymbol{\pi} \cdot \boldsymbol{\pi} \sim$$

Notably, there is no $\pi \cdot \pi$ term which would indicate that the pion is massive. That is, the spontaneous symmetry breaking has rendered the pion massless! This is actually a general result, known as *Goldstone's theorem*: whenever a continuous symmetry is spontaneously broken, there are massless particles in the spectrum (in the absence of long-ranged forces). The massless particles, in this case the pion, are called *Goldstone bosons*.

You may have noticed that we left out the four-pion interaction in (515). This is because there is no pion-pion scattering at zero momentum transfer in the theory with spontaneously broken symmetry. Recalling that the momentum-space propagator for a scalar field is $1/(q^2 - m^2)$ and that the mass of the $\sigma$ is $m_\sigma^2 2\lambda f^2$, we can evaluate the tree level diagrams,

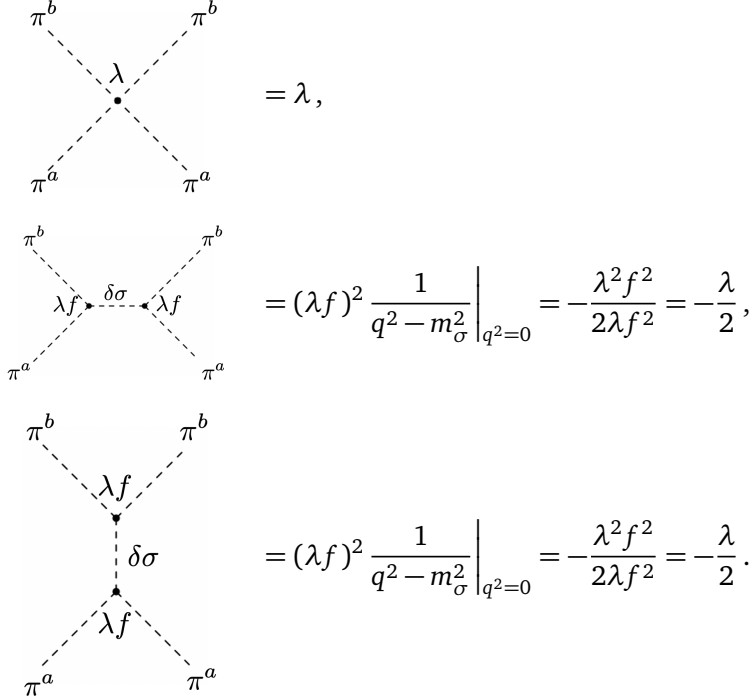

Summing these contributions, we see the matrix element for this process is zero,

$$|\mathcal{M}|^2 = \lambda - \frac{\lambda}{2} - \frac{\lambda}{2} = 0. \tag{517}$$

This is also a generic result: Goldstone bosons do not couple to themselves or other particles at zero momentum transfer.

Of course, in reality things are not so simple: the pion isn't massless! To reflect this, we should suppose that there is a small explicit breaking of the $O(4)$ symmetry: that is, that the symmetry is only approximate. Then, the approximate symmetry is spontaneously broken, as discussed above. This can be visualized as tipping the Mexican hat to favor a particular point along the rim. This can be implemented by adding a small symmetry-breaking term to the Lagrangian,

$$\delta\mathcal{L} = \Lambda^3 \sigma, \tag{518}$$

where $\Lambda$ is a (small) energy scale. In the presence of this term, the $O(4)$ symmetry is only approximate, and the pions are light, but not massless, and called *pseudo-Goldstone bosons*. This also means that pion-pion scattering is not exactly zero at zero momentum, but is also very small.

Although this is only a toy model, not an accurate depiction of reality, it does possess some of the gross features of QCD, in particular its pattern of symmetry breaking. Without the small

explicit symmetry breaking, the theory has an $O(4)$ symmetry under rotations of $\sigma$ and $\pi$. The group $O(4)$ has six generators (recall our brief discussion of group theory in the previous section), which correspond to six conserved currents. In this particular model, the currents are

$$J_\mu^a = \varepsilon^{abc}(\partial_\mu \pi^b)\pi^c , \tag{519}$$

$$J_{5,\mu}^a = \sigma \partial_\mu \pi^a - (\partial_\mu \sigma)\pi^a . \tag{520}$$

The first three are the standard conserved currents associated with isospin invariance (the indices $a, b, c = 1, 2, 3$ run over the components of the pion in isospin-space), while the second three are *axial currents* which arise from our ability to mix the $\sigma$ with the three pions. It is important to note that the axial currents have positive parity—which is the opposite of negative parity which one usually expects in a vector; recall that under parity $\mathbf{x} \to -\mathbf{x}$. These are often denoted as "axial vectors" or "pseudovectors". The magnetic field is an example of an axial vector.

Returning to our model, can write down the conserved charges associated with these currents,

$$Q^a = \int \mathrm{d}^3 x \, J_0^a ,$$

$$Q_5^a = \int \mathrm{d}^3 x \, J_{5,0}^a , \tag{521}$$

which form an algebra in that they close under commutation,

$$[Q^a, Q^b] = i\varepsilon^{abc}Q^c ,$$
$$[Q^a, Q_5^b] = i\varepsilon^{abc}Q_5^c ,$$
$$[Q_5^a, Q_5^b] = i\varepsilon^{abc}Q^c . \tag{522}$$

When the symmetry is spontaneously broken, the vacuum expectation value of $\sigma$ is nonzero, $\langle\sigma\rangle = f$, while $\langle\pi\rangle = 0$. Now, the vacuum is no longer invariant under an axial rotation which mixes $\sigma$ into $\pi$, since it will change the observables $\langle\sigma\rangle$ and $\langle\pi\rangle$. That is, in the symmetry-broken vacuum the axial rotatation symmetry is broken and the axial currents are not conserved. This corresponds to breaking three of the generators of the $O(4)$ group, which by Goldstone's theorem implies that we will have three Goldstone bosons: the pions. Adding back the small explicit breaking will give them a small mass, as explained earlier.

As any group theorist can tell you, $O(4)$ is isomorphic (that is, structurally identical) to $SU(2) \times SU(2)$. In fact, one can construct linear combinations of the conserved charges (521) such that they break into two groups which close among themselves. Defining the left- and right-handed (or *chiral*) charges to be

$$Q_L^a = \frac{1}{2}\left(Q^a - Q_5^a\right),$$

$$Q_R^a = \frac{1}{2}\left(Q^a + Q_5^a\right), \tag{523}$$

one can show that their commutation relations are simply

$$[Q_L^a, Q_L^b] = i\varepsilon^{abc}Q_L^c ,$$
$$[Q_R^a, Q_R^b] = i\varepsilon^{abc}Q_R^c ,$$
$$[Q_L^a, Q_R^b] = 0 . \tag{524}$$

The case is extremely similar in the actual theory of the strong interactions, QCD.In fact, this model has the same approximate symmetry as QCD in the limit that the up and down quarks are massless. Again, the up and down quarks are not actually massless, but they are very light—light enough that this is a very useful idealization.

To be concrete, let's consider QCD with just the up and down quarks (the others are irrelevant for this line of argument), with the Lagrangian

$$\mathcal{L} = \mathcal{L}_{\text{gluon}} + \bar{q}\left(i\slashed{D} - \frac{1}{2}(m_u + m_d) - \frac{1}{2}(m_u - m_d)\tau_3\right)q, \tag{525}$$

where $q = (u, d)$, so the second two terms are just a more symmetric way of writing $-\frac{1}{2}m_u\bar{u}u - \frac{1}{2}m_d\bar{d}d$. Under a (vector) isospin rotation through a small angle $\theta \ll 1$, the quark fields transform as

$$\begin{aligned} q &\to e^{i\boldsymbol{\theta}\cdot\boldsymbol{\tau}}q \approx q + i\boldsymbol{\theta}\cdot\boldsymbol{\tau}\,q, \\ \bar{q} &\to \bar{q}\,e^{-i\boldsymbol{\theta}\cdot\boldsymbol{\tau}} \approx \bar{q} - i\bar{q}\,\boldsymbol{\theta}\cdot\boldsymbol{\tau}. \end{aligned} \tag{526}$$

The last term of the Lagrangian (525) is not invariant under this rotation, and picks up an additional term

$$\delta\mathcal{L} = -\frac{1}{2}(m_u - m_d)\bar{q}(1 - i\boldsymbol{\theta}\cdot\boldsymbol{\tau})\tau_3(1 + i\boldsymbol{\theta}\cdot\boldsymbol{\tau})q \tag{527}$$

$$= -\frac{1}{2}(m_u - m_d)\bar{q}(i[\tau_3, \boldsymbol{\theta}\cdot\boldsymbol{\tau}])q + \mathcal{O}(\theta^2) \tag{528}$$

$$= +\frac{1}{2}(m_u - m_d)\bar{q}((\hat{\boldsymbol{z}} \times \boldsymbol{\theta})\cdot\boldsymbol{\tau})q. \tag{529}$$

We dropped terms of order $\theta^2$ in the second line, and used some $SU(2)$ identities to get to the third. If $m_u \approx m_d$ this extra term is small, and the vector rotation is an approximate symmetry of the theory. Of course, this is nothing but isospin invariance!

We can now consider axial rotations, under which the quarks transform (again for small $\theta$),

$$\begin{aligned} q &\to e^{i\boldsymbol{\theta}\cdot\boldsymbol{\tau}\gamma^5}q \approx q + i\,\boldsymbol{\theta}\cdot\boldsymbol{\tau}\gamma^5\,q, \\ \bar{q} &\to \bar{q}\,e^{-i\boldsymbol{\theta}\cdot\boldsymbol{\tau}} \approx \bar{q} - i\bar{q}\,\boldsymbol{\theta}\cdot\boldsymbol{\tau}\gamma^5. \end{aligned} \tag{530}$$

In this case, neither of the last two terms of (525) are invariant, and the axial rotation generates the extra terms (to first order in $\theta$)

$$\delta\mathcal{L} = -i(m_u + m_d)\bar{q}\boldsymbol{\theta}\cdot\boldsymbol{\tau}\gamma^5 q - i(m_u - m_d)\bar{q}\theta_3\gamma^5 q. \tag{531}$$

Given the first term, the axial rotation is only a symmetry if both the up and down quarks are massless. Invariance under axial rotations is known as *chiral symmetry*. In reality, the quarks are light compared to other hadronic scales, so chiral symmetry can be thought of as an approximate symmetry of QCD. This approximate symmetry is also spontaneously broken in our vacuum. The three broken generators correspond to three pseudo-Goldstone boson—the technical name for particles that in the absence of a small explicit symmetry breaking would be massless; these are, in fact, the pions seen in nature.

# 13   The Higgs Mechanism

We started the last section discussing the challenges of framing the weak interactions as a gauge theory. In particular, such a formulation would require the conservation of axial currents, which in turn requires that the particles involved are massless (as we just saw). Of course, this is a problem since not all particles which participate in weak interactions are massless. It is sensible to try to use the mechanism of spontaneous symmetry breaking to rectify this, however the situation is slightly more subtle in the case of gauge symmetries. For one, the symmetry must be exact (since gauge symmetries always are), and we run into the slightly confusing issue of what it actually means to "break" a gauge symmetry: after all, we introduced the gauge symmetry as a redundancy in our description, not a physical symmetry.

It turns out that the math behind spontaneous gauge symmetry breaking (otherwise known as the Higgs mechanism) in the electroweak sector of the standard model gets rather messy. To avoid complications, we will consider here a simpler model called the *Abelian Higgs* model which demonstrates many of the important concepts without excessive amounts of algebra. In particular, it shows how a gauge theory without massive fermions at the level of the underlying Lagrangian can have massive fermions in the physical spectrum. The model has a scalar field coupled to $U(1)$ gauge field, as opposed to the actual theory of the electroweak interactions which has the gauge group $SU(2) \times U(1)$. We'll start by first considering breaking a *global $U(1)$* symmetry before moving onto the Higgs mechanism in the gauged theory. Afterwards, we'll briefly note what is different in the more complicated case of electroweak symmetry breaking.

## 13.1   Global $U(1)$ Symmetry Breaking

Consider a complex scalar field, $H$, with a Mexican hat potential,

$$\mathcal{L} = \partial_\mu H^\star \partial^\mu H - \frac{\lambda^2}{2} \left( H^\star H - v^2 \right)^2 . \tag{532}$$

Instead of using the fields $H$ and $H^\star$, it will be convenient to parameterize the two degrees of freedom of the complex scalar field by two real fields $h$ and $\theta$, corresponding to the amplitude and phase:

$$H = h \, \mathrm{e}^{i\theta} . \tag{533}$$

The field $h$ represents fluctuations in the radial direction of the Mexican hat, while $\theta$ specifies the angular position along the rim. In terms of these fields, the derivatives of $H$ and $H^\star$ are

$$\begin{aligned}
\partial_\mu H^\star &= \mathrm{e}^{-i\theta} \left( \partial_\mu h - ih \partial_\mu \theta \right), \\
\partial^\mu H &= \mathrm{e}^{i\theta} \left( \partial^\mu h + ih \partial^\mu \theta \right).
\end{aligned} \tag{534}$$

and thus the kinetic term becomes

$$\partial_\mu H^\star \partial^\mu H = \partial_\mu h \, \partial^\mu h + h^2 \, \partial_\mu \theta \partial^\mu \theta , \tag{535}$$

and the Lagrangian is

$$\mathcal{L} = \partial_\mu h \, \partial^\mu h + h^2 \, \partial_\mu \theta \partial^\mu \theta - \frac{\lambda^2}{2} \left( h^2 - v^2 \right) . \tag{536}$$

In the vacuum state, the energy of the system will be minimized. To minimize the potential energy (the last term above), we must have $h = v$. That is, the field $h$ acquires a nonzero vacuum expectation value, breaking the global $U(1)$ symmetry since rotating $h \to \mathrm{e}^{i\alpha}h$ is not

a symmetry of the vacuum. We can then write the $h$ field as its vacuum value plus fluctuations around it, which become the physical field,

$$h = v + \delta h. \tag{537}$$

Plugging this expansion into the Lagrangian (536), we have

$$
\begin{aligned}
\mathcal{L} = \partial_\mu(\delta h)\partial^\mu(\delta h) + v^2\,\partial_\mu\theta\partial^\mu\theta + 2v\delta h\,\partial_\mu\theta\partial^\mu\theta + \delta h^2\,\partial_\mu\theta\partial^\mu\theta \\
- \frac{\lambda^2}{2}\big(4v^2\delta h^2 + 4v\delta h^3 + \delta h^4\big).
\end{aligned}
\tag{538}
$$

This is fairly complicated, but if we want to consider the behavior of the theory near the vacuum, we can assume $\delta h$ is small and only keep terms to second order in the fluctuations, leaving us with

$$\mathcal{L} = \partial_\mu(\delta h)\partial^\mu(\delta h) + v^2\,\partial_\mu\theta\partial^\mu\theta - 2\lambda^2 v^2\delta h^2 + \dots. \tag{539}$$

We see that there is no term quadratic in $\theta$, implying that the field is massless. That is, $\theta$ is the Goldstone boson associated with the broken $U(1)$ symmetry. On the other hand, there is a term quadratic in $\delta h$, indicating that it is massive. Recalling that the mass term for a real scalar field is $-\frac{1}{2}m^2 h^2$, we can read off the mass of $\delta h$ to be

$$m = \sqrt{2}\lambda v. \tag{540}$$

So, we see $\delta h$ is not a Goldstone boson, but rather a massive scalar field (sometimes called the amplitude Higgs mode). This is to be expected, since the broken $U(1)$ symmetry has only one generator and thus Goldstone's theorem only gives us only one massless particle. With this structure in mind, let us now consider the gauged version of the theory.

## 13.2 The Abelian Higgs Model

Let's now suppose the field $H$ is charged. We should then replace the derivatives with covariant derivatives, $(\mathcal{D}_\mu \equiv \partial_\mu + ieA_\mu)$ and add a dynamical term for the gauge field. With no symmetry breaking this model is usually called scalar QED, and has the Lagrangian

$$\mathcal{L} = \big(\mathcal{D}_\mu H\big)^\star \mathcal{D}^\mu H - m^2 H^\star H - \frac{1}{4}F_{\mu\nu}F^{\mu\nu}. \tag{541}$$

We can read off the interactions in this theory to be

$$ie\partial_\mu H^\star A_\mu H = \qquad , \tag{542}$$

$$e^2 A_\mu A^\mu H^\star H = \qquad , \tag{543}$$

where the dashed line is the $H$ propagator and the wavy line is the gauge field propagator. We can also see that it has a conserved Noether current

$$J^\mu = -\frac{ie}{2}\left(H^\star\partial^\mu H - (\partial^\mu H^\star)H\right). \tag{544}$$

We can now ask what happens when we replace the simple mass term for the $H$ field with a Mexican hat potential, changing the Lagrangian to

$$\mathcal{L} = \left(\mathcal{D}_\mu H\right)^\star \mathcal{D}^\mu H - \frac{\lambda^2}{2}\left(H^\star H - v^2\right) - \frac{1}{4}F_{\mu\nu}F^{\mu\nu}. \tag{545}$$

Let's notice two things immediately. First of all, the Mexican hat potential is (perturbatively) renormalizable, so adding it is a sensible thing to do. Also, you can check that the Noether current (544) is still conserved, which might not be what one would suspect for a theory with a spontaneously broken symmetry.

As in our pevious analysis of a theory with a broken global symmetry, let's write the scalar field as an amplitude and phase, $H = h\,e^{i\theta}$. Using the derivatives of $H$ calculated above, we can rewrite the Lagrangian as

$$\mathcal{L} = \partial_\mu h \partial^\mu h + h^2(\partial_\mu \theta + eA_\mu)(\partial^\mu \theta + eA^\mu) - \frac{\lambda^2}{2}\left(h^2 - v^2\right) - \frac{1}{4}F_{\mu\nu}F^{\mu\nu}. \tag{546}$$

Following the same argument as before, the vacuum wants to minimize its energy, and to satisfy the potential the amplitude field $h$ will be pinned to $v$, prompting us to expand $h = v + \delta h$. Plugging this expansion into the Lagrangian will give us something very messy and complicated, so we'll only keep terms to second order in $\delta h$ and $\partial_\mu \theta$, both of which should be small for states in proximity to the vacuum. We then have

$$\mathcal{L} = \partial_\mu(\delta h)\partial^\mu(\delta h) + v^2(\partial_\mu \theta + eA_\mu)(\partial^\mu \theta + eA^\mu) - 2\lambda^2 v^2 \delta h^2 - \frac{1}{4}F_{\mu\nu}F^{\mu\nu} + \dots, \tag{547}$$

where the three dots indicate terms beyond quadratic order in $\delta h$, $A_\mu$ or $\theta$, Notice that $\theta$ only appears in combination with $A_\mu$, leading us to change variables and define a new vector field

$$B_\mu = A_\mu + \frac{1}{e}\partial_\mu \theta. \tag{548}$$

Notice that I called this a vector field, not a gauge field. This is because under a gauge transformation, $A_\mu \to A_\mu - \partial_\mu \Lambda$ and $H \to e^{ie\Lambda}H$, which implies that the phase $\theta$ transforms as $\theta \to \theta + e\Lambda$. Together, this implies the transformation property of the new field $B_\mu$ is

$$B_\mu \to A_\mu - \partial_\mu \Lambda + \frac{1}{e}\partial_\mu(\theta + e\Lambda) = A_\mu + \frac{1}{e}\partial_\mu \theta = B_\mu. \tag{549}$$

That is, it is a gauge-invariant vector field! Notice also that $F_{\mu\nu}$ is invariant under this field redefinition, since

$$\begin{aligned}
\partial_\mu B_\nu - \partial_\nu B_\mu &= \partial_\mu\left(A_\nu + \frac{1}{e}\partial_\nu \theta\right) - \partial_\nu\left(A_\mu + \frac{1}{e}\partial_\mu \theta\right) \\
&= \partial_\mu A_\nu - \partial_\nu A_\mu + \frac{1}{e}\left(\partial_\mu \partial_\nu \theta - \partial_\nu \partial_\mu \theta\right) \\
&= \partial_\mu A_\nu - \partial_\nu A_\mu \\
&= F_{\mu\nu}.
\end{aligned} \tag{550}$$

In terms of this new vector field, the Lagrangian is

$$\mathcal{L} = \partial_\mu(\delta h)\partial^\mu(\delta h) - 2\lambda^2 v^2 \delta h^2 + e^2 v^2 B_\mu B^\mu - \frac{1}{4}F_{\mu\nu}F^{\mu\nu}. \tag{551}$$

We can now stop and take stock of the field content of our theory. We have a massive Higgs field $\delta h$ as in the previous example, but we also notice that two remarkable things have happened:

first of all, the Goldstone mode $\theta$ has completely disappeared! Secondly, the "photon" field $A_\mu$, now called $B_\mu$ has acquired a mass, given the presence of the third term which is quadratic in $B_\mu$. Given that our previous discussions of gauge theories led us to conclude that a photon mass is prohibited by gauge invariance, this is quite a surprise. It is also important to note that the Lagrangian is still gauge invariant, and has been throughout our discussion.

These two results (the vanishing of the Goldstone mode and the generation of mass for the gauge field) are intimately related, and collectively referred to as the *Higgs mechanism*[49] To see their relation, let's count degrees of freedom. A propagating massless photon has two polarizations (the two directions perpendicular to the direction of propagation) which correspond to two degrees of freedom, and the Goldstone mode $\theta$ is a real scalar field, which has one degree of freedom. Meanwhile, the massive vector field that we end up with has two transverse polarizations as well as a longitudinal polarization, corresponding to three degrees of freedom. The interpretation is now clear: the would-be Goldstone boson is absorbed into the photon field, giving it an extra polarization and a mass. Or, as Sidney Coleman famously said, the gauge field eats the Goldstone boson and becomes fat.

One implication of the gauge field acquiring a mass is that the interaction it mediates becomes finite-ranged. We can see this intuitively by appealing to our knowledge of the Yukawa force: the range of an interaction mediated by a particle of mass $m$ decays like $e^{-mr}$.

> ## Aside: The Meissner Effect
>
> One might ask if the Abelian Higgs model actually describes anything in the real world. As far as particle physics is concerned, the answer is currently no, but it turns out that the non-relativistic limit of the Abelian Higgs model is a realistic model of a superconductor called the Ginzburg-Landau model (which predates the Higgs mechanism by over ten years). More broadly, superconductivity is really best understood as the Higgs phase of electromagnetism (that is, we interpret the gauge field as the actual photon). One of the most striking properties of superconductors is the expulsion of magnetic flux, or *Meissner effect*. Essentially, this is the statement that magnetic fields cannot exist inside a superconductor (and the reason those superconducting levitation demonstrations work). This can easily be understood in terms of the Higgs mechanism: if the photon has a mass and decays like $e^{-mr}$ in the superconductor, magnetic fields will be exponentially suppressed in the bulk of the material (Why not the electric field too? Because superconductors are non-relativistic so the two are not treated equally!). In the field of superconductivity, the length scale over which the magnetic field can penetrate is called the *London penetration depth*, $\lambda = 1/m$.

### 13.3  Electroweak Symmetry Breaking

The Abelian Higgs model we've discussed so far is really just a toy model, and doesn't play any role in the standard model or particle physics in general. However, the Higgs mechanism is central to the electroweak sector of the standard model, as originally formulated by Weinberg and Salam in the late 1960s. In this model, the gauge symmetry acts only on the left-handed quarks, electrons, and neutrinos, but not on the components. As we've seen, this chiral decomposition implies that the quarks are massless. To implement the $SU(2) \times U(1)$ gauge symmetry, we introduce four gauge fields (three for the $SU(2)$ and one for the $U(1)$) which are massless as required by gauge invariance. We then add a scalar Higgs field which couples to the

---

[49]This idea is not just due to Higgs. In the context of particle physics it was independently developed by a number of people including Higgs, Kibble, Englert, and Polyakov. However, the idea was *actually* first developed by Philip Anderson several years earlier in the study of superconductivity. In light of this, the condensed matter community (and the historically accurate) tend to refer to this as the Anderson-Higgs mechanism.

gauge fields as well as to the fermions through a Yukawa-like interaction. The Higgs field lives in a Mexican hat potential, spontaneously "breaking" the symmetry and causing it to acquire a non-zero vacuum expectation value. Schematically, this gives us terms like $\langle h \rangle \bar{\psi} \psi$ for the fermions, which for $\langle h \rangle \neq 0$ corresponds to a mass term. In this way, the quarks and other particles acquire a mass via the Higgs mechanism.

The story with the gauge fields is a little more complicated, because the $SU(2) \times U(1)$ symmetry is not completely broken: there is a $U(1)$ symmetry that survives. Perhaps confusingly, this is *not* the same $U(1)$ symmetry that appears in the direct product of the original symmetry group (by which we mean that the local symmetry is mediated by a different gauge field). This residual symmetry is usually called $U(1)_{EM}$ because it is precisely the familiar $U(1)$ symmetry of electromagnetism. In light of this, we end up with one massless $U(1)$ gauge field – the photon – and the other three gauge fields acquire masses via the Higgs mechanism. These are the $W^{\pm}$ and $Z$ bosons which mediate the weak interaction.

---

### Aside: What does it mean to "break" a gauge symmetry?

It may be troubling that we are talking about "breaking" a gauge symmetry, considering we introduced gauge symmetries as unphysical redundancies which ease our description of massless vector fields. "Breaking" a gauge symmetry implies that two gauge equivalent states are no longer physically equivalent, i.e. our arbitrary choice of gauge now matters! Surely, this cannot occur in any consistent theory.

The phrase "spontaneously broken gauge symmetry" is actually an abuse of terminology, as the gauge symmetry is not actually broken. However, it is a well-motivated phrase because the situation at hand looks extremely similar to spontaneous symmetry breaking. To see this, let's consider the vacuum state of the Abelian Higgs model. We argued that the amplitude is fixed at $v$, but the phase is arbitrary, leading us to conclude the vacuum configuration of the Higgs field is

$$H = v e^{i\theta}, \tag{552}$$

for any phase angle $\theta$, At first glance, it looks like we have a continuum of degenerate ground states, each parameterized by the angle $\theta$, and signalling that a symmetry has been spontaneously broken.

However, we must recall that the Higgs field is not the only field in the game: we also have the gauge field, the vacuum configuration for which is a pure gauge, $A_{\mu} = \partial_{\mu} \omega$ for some scalar field $\omega$, such that the energy density $F_{\mu\nu} F^{\mu\nu} = 0$. The vacuum state of the system is then really

$$H = v e^{i\theta}, \quad A_{\mu} = \partial_{\mu} \omega. \tag{553}$$

Now we can consider making a gauge transformation, under which the vacuum state of each field transforms

$$H \to v e^{i(\theta + e\Lambda)}, \quad A_{\mu} \to \partial_{\mu}(\omega + e\Lambda). \tag{554}$$

This shows that rotating the phase of the Higgs field (which we thought above signalled the breaking of a symmetry) is in fact nothing more than a gauge transformation between two gauge-equivalent (and thus physically identical) vacuum states. So, despite appearances, gauge symmetry is not actually violated![a]

---

[a] Although the local symmetry is not broken (since such a thing would not make sense), there is a global subgroup that *is* broken. But, this is a technical detail of secondary importance to our discussion.

# 14 Reference Material

We did not provide references in the notes since most of the material is standard and well-known to people in the field—although the style and presentation were not standard.

Here we list some references to give the interested reader places to delve further into these issues, to look at the numerous issues that we left untouched, and perhaps to check on where some of the mathematical and scientific bodies were surreptitiously buried in these notes.

First we list three textbooks on nuclear and/or particle physics that complement these notes:

▶ *Introduction to Nuclear and Particle Physics*, A. Das and T. Ferbel, ISBN-13: 978-9812387448. This book is a survey written at the undergrad level and emphasizes experimental as well as theoretical physics and covers both nuclear and particle physics.

▶ *Introduction to Elementary Particles*, D. Griffiths, ISBN-13: 978-3527406012. This book is very well written, it is at the undergrad level and covers particle physics.

▶ *Foundations of Nuclear and Particle Physics* , T. W. Donnelly, J. A. Formaggio, B. R. Holstein, R. G. Milner and B. Surrow ISBN-13: 978-0521765114. This book is at a somewhat more advanced level. It emphasizes nuclear physics.

Quantum field theory is a cornerstone of modern physics and as we have seen in these notes the natural language for nuclear and particle physics. There is no shortage of books about field theory for those who wish to learn at a serious level. However, with so many options it can seem overwhelming. To give some guidance, we have included some suggestions should you wish to develop a more detailed understanding of the field theoretic topics introduced in these notes (in rough order of increasing sophistication).

▶ *Quantum Field Theory in a Nutshell* by A. Zee, ISBN-13: 978-0691140346. Equal parts a proper QFT text and a popular science book, this book makes no pretense of rigor or seriousness, and is an unabashedly fun and intuitive treatment of the subject. The second half of the book covers a wide variety of applications of QFT, many of which you won't find in other introductory books. The only point of caution is that Zee uses the path integral approach from the beginning, which is different from the canonical formalism developed in this course. But, in terms of both content and philosophy, this book is the most natural continuation of our class.

▶ *D. Tong's QFT Notes.* If you want to learn QFT in a serious way, this is a great introduction. Particularly attractive is the cost: these notes are free! However, they cover only the material for a one semester course, so they're missing coverage of some of the fancier topics we previewed, such as renormalization and non-Abelian gauge theories. Tong also has excellent notes on other topics; in particular his lectures on gauge theory are a superb (albeit somewhat advanced) reference.

▶ *Quantum Field Theory* by L. Ryder, ISBN-13: 978-0521478144. This book is a little old, but the explanations are simple and clear. It also de-emphasizes perturbative methods, which you may or may not like.

▶ *Quantum Field Theory and the Standard Model* by M. Schwartz, ISBN-13: 978-1107034730. This is a comprehensive and well-written introduction to QFT. However, it's not as easy reading as the previous books listed.



▶ *An Introduction to Quantum Field Theory* by M. Peskin and D. Schroeder, ISBN-13: 978-0813350196. The classic introductory quantum field theory reference.

▶ *Condensed Matter Field Theory* by A. Altland and B. Simons, ISBN-13: 978-0521769754. Field theory is an important tool in condensed matter physics as well as particle theory, as was hinted at by our discussion of the Ising and Heisenberg models of symmetry breaking in ferromagnetism. We include this book not just for the sake of variety, but also because it is extremely well-written and from a modern perspective.

SciPost Phys. Lect. Notes 34 (2021)

## 15 Problems

### 15.1 Problems for Sections 2 and 3

1. To the extent that the semi-emperical mass formula is accurate, the most stable nucleus with a fixed number of nucleons $A$ will have $f_p$ fixed at a value that maximizes the binding energy.

   (a) Using the semi-emperical mass formula, derive an expression for $f_p(A)$, the optimal ratio of protons to total nucleons as a function of $A$.

   (b) From your expression, estimate the approximate number of protons in the most stable nucleus of $A = 208$.

   (c) In reality, the most stable nucleus with $A = 208$ is $^{208}$Pb which has $Z = 82$. How close is the estimate from the semi-empirical mass formula?

2. Suppose one wishes to estimate the total binding energy per nucleon of the most stable nucleus as a function of $A$; one can insert the solution of 1a into the semi-empirical mass formula and divide by $A$.

   (a) Do this and plot the binding energy per nucleon of the most stable nucleus for fixed $A$ as a function of $A$. This is easy if you use Mathematica or Matlab.

   (b) From the plot, crudely estimate the value of $A$ with the highest binding energy per nucleon and the value of the binding energy at that $A$.

   (c) In reality, the nucleus with the highest binding energy is $^{56}Fe$ with a binding energy per nucleon of about 8.8 MeV. How close is your estimate to this value? How close is your estimate for the value of $A$ that achieves this value?

3. We've seen that electron scattering allows us to determine the form-factor, which is the Fourier transform of the charge density with respect to the momentum transfer, $\mathbf{q}$ (in the center of mass frame). Assuming the charge distribution is spherically symmetric, it is a function of $q^2$ (where $q = |\mathbf{q}|$):

$$g_E(q^2) = \int d^3r \, e^{-i\mathbf{q}\cdot\mathbf{r}} \, \rho(r),$$

   where $\rho$ is given in units of the fundamental charge $e$.

   (a) Show that the total charge $Z$ is given by $Z = g_E(0)$

   (b) Show that

$$g_E(q^2) = 2\pi \int_0^\infty dr \int_0^\pi d\theta \sin\theta \, r^2 e^{-iqr\cos\theta} \rho(r) = 4\pi \int_0^\infty dr \, r \frac{\sin qr}{q} \, \rho(r).$$

   (c) Show that $\langle r^2 \rangle$, defined as $\langle r^2 \rangle \equiv Z^{-1} \int d^3r \, r^2 \rho(r)$ is given by

$$\langle r^2 \rangle = -\frac{6}{Z} \left. \frac{dg_E(q^2)}{dq^2} \right|_{q^2=0}.$$

   The square root of $\langle r^2 \rangle$ is called the charge radius.

4. Suppose that a charge of $Ze$ is uniformly distributed over a sphere of radius $R$, and the distribution is taken to be static.

**(a)** Compute the form factor $g_E(q^2)$

**(b)** Verify explicitly that $g_E(0) = Z$ for this distribution

**(c)** Show from the definition of $\langle r^2 \rangle$ that $\langle r^2 \rangle = \frac{3}{5} R^2$

**(d)** Verify explicitly that $\langle r^2 \rangle = -\frac{6}{Z} \frac{dg_E}{dq^2}\big|_{q^2=0}$ for this distribution

**(e)** Consider nonrelativistic electron scattering off of the static charge distribution given in this problem. Suppose that the initial momentum of the electron is $b/R$, where $b$ is a constant and we use units where $\hbar = 1$. Find an expression for the ratio of the differential cross section for this process to the differential cross section for electron scattering off of a point charge of magnitude $Z$ as a function of the scattering angle $\theta$. *Hint: express the momentum transfer in terms of b,R, and $\theta$.*

5. We briefly discussed the Fermi gas model for nuclear matter.

**(a)** We showed that the density for nuclear matter in this model is given by $\rho_{NM} = \frac{2k_F^3}{3\pi^2}$. Assuming the density of matter is .16 fm$^{-3}$, estimate the value of the Fermi momentum in MeV

**(b)** Show that the expression for the average kinetic energy of a nucleon in nuclear matter in the model is given by $KE/A = \frac{3}{5} \frac{k_F^2}{2M_N}$ where $M_N$ is the mass of the nucleon (in a model as crude as this it does not make sense to distinguish between protons and neutrons)

**(c)** Estimate $KE/A$ in MeV

**(d)** The Fermi gas model implicitly assumes a (constant) potential well that binds the nucleons. Estimate the depth of that well in MeV using your results from the previous parts of this problem and the empirical fact that the binding energy per nucleon is approximately 16 MeV.

## 15.2 Problems for Sections 4, 5, and 6

6. The metric tensor, $g_{\mu\nu}$ is a four-tensor, which means that if one transforms into another inertial frame (denoted with a prime), $g'^{\mu\nu} = \Lambda^\mu{}_\alpha \Lambda^\nu{}_\beta g^{\alpha\beta}$, where $\Lambda$ is the matrix for the Lorentz transformation. Using the fact that the product of two Lorentz vectors $A^\mu B_\mu = g^{\mu\nu} A_\mu B_\nu$ is the same in all frames, show that for an arbitrary Lorentz transformation $g'^{\mu\nu} = g^{\mu\nu}$. That is, show that component-by-component the values of the matrix elements of the metric tensor are the same in both frames.

7. In our discussion of the Klein-Gordon equation we implicitly assumed that we coupled the Klein-Gordon field to a static source localized at the nucleon. In this problem we will consider a more general situation: the field coupled to an arbitrary Lorentz scalar source $J_s$, that may depend on spacetime. The Klein-Gordon equation is then $(\partial_\mu \partial^\mu + m^2)\phi(\mathbf{x}, t) = J_s(\mathbf{x}, t)$.

**(a)** Suppose that I can find a function of spacetime, $G(\mathbf{x}, t)$ that satisfies

$$\left(\partial_\mu \partial^\mu + m^2\right) G(\mathbf{x}, t) = -\delta^3(\mathbf{x})\delta(t).$$

Show that the the Klein-Gordon equation with a source is automatically solved if

$$\phi(\mathbf{x}, t) = -\int dt' d^3 x' \, G(\mathbf{x} - \mathbf{x}', t - t') J_s(\mathbf{x}', t'),$$

where the integrals are taken over all spacetime.

*$G(\mathbf{x}, t)$ is called the Green's function of the Klein-Gordon equation. It reduces the sourced equation into a simple integral. In the context of quantum field theory it is referred to as the propagator, as it acts to propagate the field from a source at a spacetime point $(\mathbf{x}', t')$ to the point $(\mathbf{x}, t)$.*

**(b)** The propagator is easy to find in momentum space. Defining $(\mathbf{x}, t) \equiv x$, we can Fourier transform $G(x)$ to four-momentum space, that is

$$G(x) = \int \frac{\mathrm{d}^4 q}{(2\pi)^4} \, \mathrm{e}^{-iq_\mu x^\mu} \, S(q),$$

where $S(q)$ is the momentum-space propagator. Show that $(\partial_\mu \partial^\mu + m^2)G(x) = -\delta^4(x)$ implies that $(q_\mu q^\mu - m^2)S(q) = 1$.

*It turns out that due to boundary conditions at infinity (which are ultimately tied to causality) there is an ambiguity in just taking $S(q) = (q_\mu q^\mu - m^2)^{-1}$. Feynman showed that the correct way to do this within the context of field theory is to instead take $S(q) = (q_\mu q^\mu - m^2 + i\varepsilon)^{-1}$, where $\varepsilon$ is an infinitesimally small positive constant.*

**(c)** Consider the case where the source is static: $J_s(\mathbf{x}, t) = J_s(\mathbf{x}, 0)$, and accordingly $\phi(\mathbf{x}, t) = \phi(\mathbf{x}, 0)$. From parts (a) and (b), show that the static solution to the Klein-Gordon equation is

$$\int \mathrm{d}^3 x \, \mathrm{e}^{i\mathbf{q}\cdot\mathbf{x}} \, \phi(\mathbf{x}, 0) = \frac{\int \mathrm{d}^3 x' \, \mathrm{e}^{i\mathbf{q}\cdot\mathbf{x}'} \, J_s(\mathbf{x}', 0)}{\mathbf{q} \cdot \mathbf{q} + m^2}.$$

*Hint: you essentially just need to show that the static limit forces the zeroeth component of $q$ to be zero.*

**(d)** Suppose that the static source can be well approximated as a point which we will take to be the origin: $J_s(\mathbf{x}', 0) = 4\pi g \, \delta^3(\mathbf{x}')$ (the factor of $4\pi$ is conventional). Show that

$$\frac{\int \mathrm{d}^3 x' \, \mathrm{e}^{i\mathbf{q}\cdot\mathbf{x}'} \, J_s(\mathbf{x}', 0)}{\mathbf{q} \cdot \mathbf{q} + m^2} = \frac{4\pi g}{\mathbf{q} \cdot \mathbf{q} + m^2}.$$

From your result in (c) this implies $\int \mathrm{d}^3 x \, \mathrm{e}^{i\mathbf{q}\cdot\mathbf{x}} \, \phi(\mathbf{x}, 0) = \frac{4\pi g}{\mathbf{q}\cdot\mathbf{q} + m^2}$.

**(e)** In the notes we asserted that for a point charge at the origin, $\phi(\mathbf{x}, 0) = g \, \mathrm{e}^{-mr}/r$. For this solution compute $\int \mathrm{d}^3 x \, \mathrm{e}^{i\mathbf{q}\cdot\mathbf{x}} \, \phi(\mathbf{x}, 0)$ to verify the equality above.

**8.** This problem concerns the properties of the Dirac matrices.

**(a)** The fundamental property of the Dirac matrices is that $\gamma^\mu \gamma^\nu + \gamma^\nu \gamma^\mu = 2g^{\mu\nu}\mathbb{1}$ where $\mathbb{1}$ is the identity matrix. The Dirac matrices are components of a four-vector which means if one transforms to another inertial frame (denoted with a prime) the new components are given by $\gamma'^\mu = \Lambda^\mu{}_\alpha \gamma^\alpha$ where $\Lambda$ is the Lorentz transformation matrix. Show that if $\gamma^\mu \gamma^\nu + \gamma^\nu \gamma^\mu = 2g^{\mu\nu}\mathbb{1}$ then $\gamma'^\mu \gamma'^\nu + \gamma'^\nu \gamma'^\mu = 2g^{\mu\nu}\mathbb{1}$, i.e. that the defining characteristic of the Dirac matrices holds in every frame.

**(b)** Show explicitly for all four gamma matrices that $\operatorname{tr} \gamma^\mu = 0$.

**(c)** Use $\gamma^\mu \gamma^\nu + \gamma^\nu \gamma^\mu = 2g^{\mu\nu}\mathbb{1}$ to show that $\operatorname{tr}(\gamma^\mu \gamma^\nu) = 4g^{\mu\nu}$ for all $\mu, \nu$.

**(d)** Show that $\operatorname{tr}(\gamma^\mu \gamma^\nu \gamma^\rho) = 0$ for all $\mu, \nu, \rho$.

**(e)** For an arbitrary four-vector $A$, use $\gamma^\mu \gamma^\nu + \gamma^\nu \gamma^\mu = 2g^{\mu\nu}\mathbb{1}$ to show that $\frac{1}{4} \operatorname{tr}(A_\mu \gamma^\mu \gamma^\nu) = A^\nu$.

## 15.3 Problems for Section 8

**9.** In the notes we used a mode composition for $\phi$, the free Klein-Gordon field:

$$\phi(\mathbf{x}, t) = \int \frac{d^3k}{(2\pi)^3} \frac{1}{\sqrt{2\omega_\mathbf{k}}} \left( a_\mathbf{k}^\dagger e^{-ik_\mu x^\mu} + a_\mathbf{k} e^{ik_\mu x^\mu} \right).$$

We stated that if $[a_\mathbf{k}, a_{\mathbf{k}'}^\dagger] = (2\pi)^3 \delta^3(\mathbf{k} - \mathbf{k}')$ then $[\phi(\mathbf{x}, t), \Pi(\mathbf{x}', t)] = i\,\delta^3(\mathbf{x} - \mathbf{x}')$ where $\Pi = \dot{\phi}$ and $\omega_\mathbf{k} = \sqrt{\mathbf{k}^2 + m^2}$. Show that this is true.

**10.** We wrote the Hamiltonian density for the free Klein-Gordon theory as

$$\mathcal{H} = \frac{1}{2}\left( \Pi^2 + \nabla\phi \cdot \nabla\phi + m^2\phi^2 \right)$$

and stated that in terms of the creation and annihilation operators this Hamiltonian is written

$$\mathcal{H} = \int \frac{d^3k}{(2\pi)^3} \frac{\omega_\mathbf{k}}{2}\left( a_\mathbf{k}^\dagger a_\mathbf{k} + a_\mathbf{k} a_\mathbf{k}^\dagger \right).$$

Show that this is true.

**11.** In the notes we stated the general formula for a decay. Consider the following theory, with two types of scalars $\phi$ and $\sigma$ with masses of $m$ and $m/3$ respectively, governed by the Lagrangian

$$\mathcal{L} = \frac{1}{2}\left( \partial_\mu\phi\,\partial^\mu\phi + \partial_\mu + m^2\phi^2 + \partial_\mu\sigma\,\partial^\mu\sigma + \frac{m^2}{9}\sigma^2 \right) + g\sigma^2\phi.$$

(This theory is mathematically sick, but is ok in perturbation theory). The purpose of this problem is to calculate the total decay rate $\Gamma$ of a $\phi$ particle into two $\sigma$ at lowest order in the coupling $g$. The invariant matrix element is $|\mathcal{M}|^2 = g^2$. Your task is to determine the kinematic factor and thus determine the decay rate.

**12.** This problem concerns the mass dimensions of boson and fermion fields in spacetime of dimension $d$.

**(a)** Show that the dimension of a boson field is $\frac{d}{2} - 1$

**(b)** Show that the dimension of a fermion field is $\frac{d-1}{2}$

**13.** In this problem several Lagrangian densities will be given. Your task is to determine which of these are renormalizable in perturbation theory in various space-time dimensions by considering the mass dimensions of the operators in them. In this problem fermion fields will be represented by $\psi$ and boson fields by $\phi$. Coefficients with either Latin or Greek letters represent constants of the theory, with appropriate dimension.

**(a)** $\mathcal{L} = \frac{1}{2}(\partial_\mu\phi\,\partial^\mu\phi + m^2\phi^2) + \frac{\lambda^4}{4!}\phi^4$

 (i) Is this renormalizable in $d = 2$? (one spatial dimension plus time)
 (ii) Is this renormalizable in $d = 3$? (two spatial dimensions plus time)
 (iii) Is this renormalizable in $d = 4$? (three spatial dimensions plus time)
 (iv) Is this renormalizable in $d = 5$? (four spatial dimensions plus time)

**(b)** $\mathcal{L} = \frac{1}{2}(\partial_\mu\phi\,\partial^\mu\phi + m^2\phi^2) + \frac{\lambda^4}{4!}\phi^4 + \frac{\kappa^6}{6!}\phi^6$

 (i) Is this renormalizable in $d = 2$? (one spatial dimension plus time)

(ii) Is this renormalizable in $d = 3$? (two spatial dimensions plus time)

(iii) Is this renormalizable in $d = 4$? (three spatial dimensions plus time)

(iv) Is this renormalizable in $d = 5$? (four spatial dimensions plus time)

(c) $\mathcal{L} = \frac{1}{2}(\partial_\mu \phi \, \partial^\mu \phi + m^2 \phi^2) + \bar{\psi}(i\gamma^\mu \partial_\mu - m)\psi + g\phi\bar{\psi}\psi$

(i) Is this renormalizable in $d = 2$? (one spatial dimension plus time)

(ii) Is this renormalizable in $d = 3$? (two spatial dimensions plus time)

(iii) Is this renormalizable in $d = 4$? (three spatial dimensions plus time)

(iv) Is this renormalizable in $d = 5$? (four spatial dimensions plus time)

(d) $\mathcal{L} = \frac{1}{2}(\partial_\mu \phi \, \partial^\mu \phi + m^2 \phi^2) + \bar{\psi}(i\gamma^\mu \partial_\mu - m)\psi + g\phi\bar{\psi}\psi + \frac{1}{2}h\,\phi^2\bar{\psi}\psi$

(i) Is this renormalizable in $d = 2$? (one spatial dimension plus time)

(ii) Is this renormalizable in $d = 3$? (two spatial dimensions plus time)

(iii) Is this renormalizable in $d = 4$? (three spatial dimensions plus time)

(iv) Is this renormalizable in $d = 5$? (four spatial dimensions plus time)

## 15.4 Problems for Section 9

**14.** The Fermi theory of beta decay has propagating electron, neutrinos, protons and neutrons denoted ($e$, $\nu$, $p$, and $n$) represented by Dirac fields with standard kinetic terms in the Lagrangian density. The key to the theory is the interaction term in the Lagrangian which is given by $\mathcal{L}_I = G(\bar{p}\gamma^\mu n + \bar{n}\gamma^\mu p)(\bar{e}\gamma_\mu \nu + \bar{\nu}\gamma_\mu e)$ where $G$ is an experimentally determined coupling constant. Subsequently, the electroweak theory of Weinberg and Salam supplanted it; but the Fermi theory is useful in many phenomenological applications.

(a) Show that the this theory is not renormalizable. This means the theory cannot be regarded as fundamental, but when used at tree level it remains a useful phenomenological description for many processes.

(b) Explain why this interaction allows for a neutron to decay into a proton emitting an electron and an antineutrino.

(c) Show that this theory is not invariant under any of the following transformations:

(i) $p \to e^{i\theta_p} p$

(ii) $n \to e^{i\theta_n} n$

(iii) $e \to e^{i\theta_e} e$

(iv) $\nu \to e^{i\theta_\nu} \nu$

(d) Explain why (c) means that there is no conservation law associated with proton number, neutron number, electron number or neutrino number

(e) Show that despite (c) there are still two independent $U(1)$ transformations which leave the Lagrangian invariant

(f) Write down the conserved currents associated with these (it turns out one is the baryon current and one is the lepton current).

**15.** This problem concerns the isospin properties of the Fermi theory from the preceding problem.

(a) Show explicitly that $\mathcal{L}_I$ from the previous problem can be written $\mathcal{L}_I = G\bar{N}\gamma^\mu \tau_x N(\bar{e}\gamma_\mu \nu + \bar{\nu}\gamma_\mu e)$ where $N$ is a nuclear isospinor whose components are $p$ and $n$, and $\tau_x$ is the first Pauli matrix in isospin space.

(b) From (a) show that the Fermi theory is not invariant under general isospin transformations.

**(c)** Show that if a hadron or nuclear state decays by a weak interaction in the form given by the Fermi interaction, the isospin of the hadrons/nuclei in the final state $I_f$ (where the square of the isospin is $I_f(I_f + 1)$) is equal to $I_i + 1$ or $I_i - 1$ where $I_i$ is the isospin of the initial state.

16. The next set of problems involves reactions among hadrons (assuming isospin is perfect). Here is a table of some hadrons of interest:

| Hadron | Mass (MeV) | Spin | Parity | Isospin |
|---|---|---|---|---|
| $\pi$ | 137 | 0 | − | 1 |
| $\rho$ | 775 | 1 | − | 1 |
| $\omega$ | 783 | 1 | − | 0 |
| $f_0$ (or $\sigma$) | 500 | 0 | + | 0 |
| $f_0(980)$ | 980 | 0 | + | 0 |
| $\eta$ | 548 | 0 | − | 0 |
| $a_0$ | 980 | 0 | + | 1 |
| $a_1$ | 1260 | 1 | + | 1 |
| $N$ | 939 | 1/2 | + | 1/2 |
| $\Delta$ | 1232 | 3/2 | + | 3/2 |
| $N^\star(1440)$ | 1440 | 1/2 | + | 1/2 |
| $N^\star(1520)$ | 1520 | 3/2 | - | 1/2 |
| $N^\star(1535)$ | 1535 | 1/2 | - | 1/2 |

Consider the following possible decays. State whether the decay is allowed or forbidden by energy/momentum conservation (from the masses), parity, spin,isospin, and, if identical particles, Bose/Fermi symmetry. If the decay is allowed, you need to indicate why it is allowed and if forbidden why it is forbidden.

**(a)** $\omega \to \pi^+ \pi^-$

**(b)** $\rho^0 \to \pi^+ \pi^-$

**(c)** $\rho^0 \to \pi^0 \pi^0$

**(d)** $a_1^0 \to \pi^+ \pi^-$

**(e)** $a_1^0 \to \rho^0 \pi^0$

**(f)** $a_1^+ \to \rho^+ \pi^0$

**(g)** $\rho^0 \to \eta \pi^0$

**(h)** $\Delta^+ \to n \pi^+$

**(i)** $\Delta^+ \to p \eta$

**(j)** $N^\star(1535)^+ \to p \eta$

17. This problem considers the implications of isospin on nucleon-nucleon scattering.

**(a)** Show that the differential cross-section at fixed energy and angle for proton-proton scattering is the same as that of neutron-neutron scattering.

**(b)** Is the differential cross-section for proton-neutron scattering equal to that of proton-proton and neutron-neutron scattering? Why or why not?

18. This problem considers decays into various final states.

**(a)** The $\Delta$ decays into a pion plus a nucleon. What fraction of the time does the $\Delta^0$ decay into $p\pi^+$, and what fraction to $n\pi^0$?

**(b)** The $N^\star(1520)^+$ decays into a single pion and a nucleon approximately 60% of the time. Among those cases when a $N^\star(1520)^+$ decays into a pion and a nucleon, what fraction of the time does it does it decay into $n\pi^+$ and what fraction into $p\pi^0$?

**(c)** The $f_0(980)$ decays into two pions. What fraction of the time does it decay into $\pi^+\pi^-$ and what fraction $\pi^0\pi^0$?

## 15.5 Problems for Section 10

**19.** The QCD Lagrangian can be written as $\mathcal{L} = \mathcal{L}_{\text{quarks}} + \mathcal{L}_{\text{gluons}}$ where the quark part of the Lagrangian, $\mathcal{L}_{\text{quarks}} = \sum_f \bar{q}_f\big(i\gamma^\mu \mathcal{D}_\mu - m_f\big)q_f$ implicitly includes gluons through the covariant derivative, and the sum over $f$ runs over the flavors $(u,d,s,c,t,b)$ where $m_f$ is the mass for each flavor of quark. The sums over the Dirac and color indices are implicit in this expression. One can rewrite this as a light quark $(u,d)$ part, $\mathcal{L}_{ud} = \bar{q}_{ud}\big(i\gamma^\mu \mathcal{D}_\mu - \underline{m}\big)q_{ud}$ where $q_{ud}$ is an isospinor containing an isospin index (up or down) and $\underline{m}$ is a matrix in isospin space, plus a similar part for the heavier quarks.

**(a)** Show that $\underline{m} = \frac{1}{2}(m_u+m_d)\mathbb{1} + \frac{1}{2}(m_u-m_d)\tau_3$ where $\mathbb{1}$ and $\tau^3$ are matrices in isospin space.

**(b)** Show that if $m_u = m_d$ then the QCD Lagrangian is invariant under all transformations of the form $q_{ud} \to \exp\big(\frac{i}{2}\sum_a \theta_a \tau_a\big)q_{ud}$ where $\theta_a$ are arbitrary parameters and $\tau_a$ are the Pauli matrices in isospin space.

**(c)** Show that this implies that when $m_u = m_d$, there is a conserved isospin current in QCD associated with the symmetry in (b): $J^a_\mu = \bar{q}_{ud}\big(\gamma_\mu \frac{\tau^a}{2}\big)q_{ud}$ where $a = 1,2,3$ is the isospin index.

**20.** We stated that the Lagrangian for for electrodynamics is of the form $\mathcal{L} = -\frac{1}{4}F_{\mu\nu}F^{\mu\nu} - A_\mu J^\mu$, where $J^\mu$ is a conserved current. We noted that while $\mathcal{L}$ is not invariant, the action (which controls the physics) is.

**(a)** Show that if a term $-\frac{1}{2}m^2 A^\mu A_\mu$ were added to the Lagrangian, the theory would no longer be gauge invariant.

**(b)** The photon in the original theory is massless. Show that adding the term above to the theory would give the photon a mass. The easiest way to do this is to show that the equations of motion for the modified theory are $(\partial_\mu \partial^\mu + m^2)A_\nu = J_\nu$, which is the equation of motion for a massive boson. This implies that gauge invariance ensures the photon is massless.

**(c)** In the notes we showed that gauge invariance requires a conserved current. Explain why the results in the previous parts mean the converse need not be true, i.e. the current can be conserved even if the theory is not gauge invariant.

**21.** Physical observables in gauge theories are associated with expectation values of gauge invariant operators. Indicate which of the following are possible physical observables (where $e$ is the magnitude of the charge of the electron and $\psi$ is the electron field) in QED and why:

**(a)** $\langle \bar{\psi}\psi \rangle$

**(b)** $\langle \bar{\psi}eA_\mu \gamma^\mu \psi \rangle$

**(c)** $\langle \bar{\psi}\gamma^\mu \partial_\mu \psi \rangle$

**(d)** $\langle \bar{\psi}\partial_\mu \partial^\mu \psi \rangle$

**(e)** $\langle \bar{\psi}(\partial_\mu + eA_\mu)(\partial_\mu + eA^\mu)\psi \rangle$

**(f)** $\langle F_{\mu\nu}\bar{\psi}\gamma^\mu\gamma^\nu\psi \rangle$

**(g)** $\langle F_{\alpha\beta}\bar{\psi}(\gamma^\mu\partial_\mu)(\gamma^\nu\partial_\nu)\psi \rangle$

**22.** Under an axial rotation $\psi \to \psi' = e^{i\theta\gamma^5}\psi$ where $\theta$ is a parameter. The purpose of this problem is to show that under this rotation $\bar{\psi}\psi \to \bar{\psi}\psi \cos\theta + i\bar{\psi}\gamma^5\psi \sin\theta$.

**(a)** First show that $(\gamma^5)^2 = \mathbb{1}$ where $\mathbb{1}$ is the identity matrix.

**(b)** Show that $e^{i\theta\gamma^5} = \cos\theta + i\gamma^5 \sin\theta$.

**(c)** Show that under an axial rotation $\bar{\psi} \to \bar{\psi}\, e^{i\theta\gamma^5}$.

**(d)** Use your previous results to show $\bar{\psi}\psi \to \bar{\psi}\, e^{2i\theta\gamma^5}\psi = \bar{\psi}\psi \cos\theta + i\bar{\psi}\gamma^5\psi \sin\theta$.