# Peer review of "A Somewhat Random Walk Through Nuclear and Particle Physics"

_SciPost Physics Lecture Notes, doi:SciPost Phys. Lect. Notes 34 (2021)_

## Round 2 · Referee Report · Anonymous · 2021-5-24

Report

I have gone through the lecture notes entitled "A Somewhat Random Walk Through Nuclear and Particle Physics" by T.D. Cohen and N.R. Poniatowski. Overall, in my opinion, they provide a good introduction to many fundamental concepts in particle and nuclear physics. Given the level of the course and a relatively large number of the topics discussed, some of them are explained very briefly, whereas some others in more detail. While the structure and the content of the notes can be decided only by the authors, I would like to encourage them to extend the discussion on different nuclear models in section 4 and, in particular, provide more details on the shell model.

Further, I have several questions/comments I would like the authors to address.

1-On p. 48, it is stated that the Dirac sea picture is "not in line with the modern understanding of physics via quantum field theory". Please elaborate on why this is the case.

2-Below eq. (250) it is stated that the amplitudes $a(\mathbf{k},t)$ and $a^\dagger(\mathbf{k})$ ($t$ is missing) are time-dependent. However, when going from eq. (250) to eq. (254), the time derivative is not applied to $a$ and $a^\dagger$. Please clarify this passage.

3-On p. 79, Fermi's golden rule is mentioned, but it is not explained anywhere in the text. I would invite the authors to add a red box with an explanation of this rule.

4-I am wondering whether "An Introduction to Quantum Field Theory" by M.E. Peskin and D.V. Shroeder, that became a standard textbook for a QFT course, has its place in the list of recommendations in section 14.

5-Finally, there is a number of minor corrections (mostly typos) to be implemented, see "Requested changes".

Requested changes

1-in eq. (13): $r \to r'$
2-in eq. (61), line 2: $\bar{\mathbf{v}}^T \to \mathbf{v}^T$
3-in eq. (150): $\dot{r}_j \to \dot{x}_j$
4-in eq. (173): $1/(2m)$ is missing in the last two equalities
5-in the in-line equation in the line below eq. (178): the last equality is wrong and redundant; please remove
6-in eq. (186): $q_\mu \to q_\nu$
7-in eq. (223): in the middle equality, one $\partial_\alpha\phi$ in the first term is redundant; please remove
8-in the line below eq. (269): $n_{\mathbf{k}_m} \to n_{\mathbf{k}_\alpha}$
9-in the line preceding eq. (305): $\{d_{\mathbf{p},s},d_{\mathbf{p},s}\}=1 \to \{d_{\mathbf{p},s},d_{\mathbf{p},s}^\dagger\}=1$
10-in eq. (305): $\{d_{\mathbf{p},s},d_{\mathbf{p},s}\} \to \{d_{\mathbf{p},s},d_{\mathbf{p},s}^\dagger\}$
11-in the paragraph before eq. (331): please remove $N$ from the four-fermion interaction
12-please improve the quality of Figs. 3 and 4 - the tick labels as well as the names of the nuclei are barely readable
13-please refer to figures in the text (I see the references only to figs. 1, 2 and 3, but not to the others)
14-in the line below eq. (304): Fermi sea $\to$ Dirac sea
15-in the criteria of renormalizability on p. 87 (italic text): less $\to$ less or equal
16-there are many typos of the type "were were", "added added", "is is", "the the", "when when", "that that", "in in", "and and"; please use the search in the source file to fix them
17-in many instances, "its" is used instead of "it is (it's)"; please review and correct accordingly
18-two lines below eq. (336): $m_\phi^2 \to m_\phi$

  • validity: -
  • significance: -
  • originality: -
  • clarity: -
  • formatting: -
  • grammar: -

Author:  Nicholas Poniatowski  on 2021-08-27  [id 1713]

(in reply to Report 1 on 2021-05-24)

We thank the reviewer for their careful reading of the manuscript, as well as their constructive suggestions and detailed identification of typos throughout the notes. We also thank them for their positive impression of the notes, and apologize for the delay in re-submitting the revised version of them.

The intent of these notes (and the course they are based upon) is to sample a number of various topics relevant to modern nuclear and particle physics. So, although we agree with the reviewer that the physics of nuclear models, the shell model included, is an important subject, to do justice to these models would require a full semester course. We believe that the short account we provided in the present version of the manuscript presents the most important aspects of the shell model in a self-contained manner.

Below, we respond briefly to each of the requests made by the reviewer, and how they have been incorporated into the revised manuscript, which we believe is now suitable for publication in SciPost Physics Lecture Notes.

  1. We have added a footnote explaining how the Dirac sea picture contrasts from the modern understanding of quantum field theory, and why we still deemed it a useful analogy to discuss in the notes.
  2. The time-dependence of a_k was a typo, which we thank the reviewer for pointing out to us. We had already Fourier-transformed in the time domain, so that the amplitudes a_k are time-independent.
  3. We added a brief footnote explaining Fermi’s Golden rule, and pointing the unfamiliar reader to standard quantum mechanics textbooks. In our experience, Fermi’s Golden Rule is rather unintuitive to students who are not already familiar with it. Given this fact, and the relatively long derivation required to develop it, we believed that this concept is not well-suited to a short explanatory box, and rather referred the unfamiliar reader to the literature.
  4. We have added a reference to this canonical text in the reference list, and thank the reviewer for suggesting we include it.
  5. We sincerely thank the reviewer for their extremely careful reading of the manuscript, and for pointing out these numerous typos. All of them have been corrected in the revised version of the manuscript. We have also re-made Figures 3 and 4 to make them more readable, and referred to all of the Figures in the text.

---

## Round 3 · Author Response

We thank the reviewer for their time and careful reading of the manuscript. We have revised the manuscript, according to the suggestions of the reviewer, and believe that the improved version is now suitable for publication in SciPost Physics Lecture Notes.

---

## Round 3 · List of Changes

- All of the typos pointed out by the reviewer have been corrected
- Fig's 3 and 4 have been re-made in higher quality
- All figures are now referred to in the text
- Several footnotes have been added, according to the reviewer's suggestions.

---

## Editorial Decision

published